# Pointwise Generalization in Deep Neural Networks

## Abstract

We address the fundamental question of why deep neural networks generalize by establishing a pointwise generalization theory for fully connected networks. For each trained model, we characterize the hypothesis via a pointwise Riemannian Dimension, derived from the eigenvalues of the *learned feature representations* across layers. This approach establishes a principled framework for deriving tight, hypothesis-dependent generalization bounds that accurately characterize the rich, nonlinear regime, systematically upgrading over approaches based on model size, products of norms, and infinite-width linearizations, yielding guarantees that are orders of magnitude tighter in both theory and experiment. Analytically, we identify the structural properties and mathematical principles that explain the tractability of deep networks. Empirically, the pointwise Riemannian Dimension exhibits substantial feature compression, decreases with increased over-parameterization, and captures the implicit bias of optimizers. Taken together, our results indicate that deep networks are mathematically tractable in practical regimes and that their generalization is sharply explained by pointwise, spectrum-aware complexity.

## 1 Introduction

Deep learning has ushered in a new era of AI, delivering striking generalization across scientific tasks. Yet, a fundamental paradox remains: while classical theory predicts severe overfitting for massive models, practice exhibits strong generalization. This gap has fueled a prevailing view that neural networks are opaque "black boxes" resistant to principled explanation (Goodfellow et al., 2016). We narrow this gap by addressing the generalization problem for the canonical fully connected Deep Neural Network (DNN). We demonstrate that, under minimal and verifiable spectral conditions on the *learned feature representations*, deep neural networks fall into a tractable regime with tight generalization guaranties. Methodologically, our characterization leverages a pointwise generalization paradigm that fundamentally transcends classical uniform-convergence approaches and pure weight-space compressions, reshaping the theoretical foundation for representation learning. To our knowledge, this offers one of the first fully rigorous treatments that establishes the tractability of deep neural networks by contemporary machine-learning standards.

We study standard fully connected (feed-forward) networks on a dataset $X = [x_1, \ldots, x_n] \in \mathbb{R}^{d_0 \times n}$, where each column is one input example. The network has widths $d_1, \ldots, d_L$, and weight matrices $W_l \in \mathbb{R}^{d_l \times d_{l-1}}$ for $l = 1, \ldots, L$. We define the *feature matrix* at layer $l$ by the recursion

$$F_l(W, X) := \sigma_l\big(W_l \, F_{l-1}(W, X)\big) \in \mathbb{R}^{d_l \times n}, \qquad l = 1, \ldots, L, \tag{1}$$

where $F_0 := X$ and the nonlinear activation $\sigma_l$ acts columnwise. Each *column* of $F_l$ is the feature vector of one data point at layer $l$; each *row* of $F_l$ is the activation of one neuron across the dataset.

Our focus is the *generalization gap*—the difference between test and training loss at the learned weights $W$. Informally—up to universal constants, mild logarithmic factors, and reasonable simplification (made precise in Theorem 4 with discussion on the feasibility of this simplification)—we prove that this gap is controlled by the *effective dimension* of the learned features: uniformly over every $W \in \mathbb{R}^{\sum_l d_l \cdot d_{l-1}}$,

$$\mathcal{L}_{\text{test}}(W) - \mathcal{L}_{\text{train}}(W) \lesssim \sqrt{\frac{1}{n} \sum_{l=1}^{L} (d_l + d_{l-1}) \, d_{\text{eff}}\Big(F_{l-1}(W, X) \, F_{l-1}(W, X)^\top\Big)}. \tag{2}$$

Here $d_{\text{eff}}(\cdot)$ denotes the (layerwise) *effective dimension*—a smoothed, spectrum-aware notion of rank—of the feature Gram matrix $F_{l-1}(W, X)F_{l-1}(W, X)^\top$, i.e., the number of meaningful directions the feature data actually occupies at that layer. Intuitively, each layer contributes a term proportional to its size $(d_l + d_{l-1})$ multiplied by how many directions its features $F_{l-1}(W, X)$ truly use, $d_{\text{eff}}$. When features are correlated, low rank, or exhibit a rapidly decaying spectrum (a few large eigenvalues dominating many small ones), $d_{\text{eff}}$ is small, so the bound remains tight even for very wide/deep networks. Such "feature compression" phenomena is widely observed in modern deep learning (Huh et al., 2021; Wang et al., 2025; Parker et al., 2023). Strikingly, in our experiments, increasing overparameterization often induces pronounced *feature-rank compression*: the bound (2) decreases as model size grows (Section 5); for example, in ResNet trained on CIFAR-10, a majority of layers compress to (near-)zero effective rank.

Inequality (2) yields a strong *uniform, hypothesis- and data-dependent* guarantee, which we term *pointwise generalization*. It tracks how features evolve across layers of the trained model and explains overparameterization in practice. Under minimal spectral conditions on the learned feature representations, our theory renders fully connected networks tractable. The spectrum-aware effective-dimension notion we adopt is standard and minimax-sharp in linear and kernel settings (Even & Massoulié, 2021). In contrast, existing bounds either (i) rely on infinite–width linearizations (the NTK line of work, e.g., Jacot et al. (2018)), (ii) blow up exponentially with products of norms (e.g., Bartlett et al. (2017); Neyshabur et al. (2018)), or (iii) scale with model size (e.g., VC dimension (Bartlett et al., 2019)). Our bounds avoid these pathologies, delivering tight, pointwise guarantees via unified principles and systematic methodologies. By directly addressing the nonlinear, feature-learning regime—emphasized in Bartlett et al. (2021); Zhang et al. (2021); Nagarajan & Kolter (2019); Wilson (2025)—we show that generalization in deep neural networks is mathematically *tractable*.

**Contributions:** The paper is organized into three parts: (i) pointwise generalization framework; (ii) structural principles of deep networks; and (iii) empirical validation. Related work, further explanations, discussions and details, and all proofs are deferred to the appendices. We summarize the main novelties in each part below.

**A Tight Pointwise Generalization Framework.** We develop a pointwise framework that analyzes the *trained* hypothesis and yields generalization bounds with (qualified) matching upper and lower rates via a finite-scale notion of *pointwise dimension*. This fundamentally upgrades generic chaining and all covering–number approaches by assigning each hypothesis its *own complexity* that directly controls its error. The bounds can also be read as an optimally tuned PAC–Bayes objective specialized to deterministic predictors. This framework reframes generalization as a study of pointwise geometry at finite scale, clarifying why nonlinear models generalize without uniform convergence.

**Structural Principles and Tight Bounds for Neural Networks.** We develop a *non-perturbative* approach that uses exact telescoping decompositions (rather than Taylor linearizations) to preserve the finite-scale geometry of deep networks. This yields our first structural principle: *cross-layer correlations factor through the feature matrices and approximately preserve a pointwise linear structure*. We then show that bounding the pointwise dimension reduces to the gold standard of *effective dimension* on local charts, and we extend this to a global statement by constructing an *ellipsoidal* covering over the set of subspaces (Grassmannian). This extension—novel beyond the classical differential–geometric/Lie–algebraic treatments—establishes our second structural principle: *the complexity of the global atlas (covering reference eigenspaces) remains commensurate with that of the local charts*. Building on these principles, we introduce *Riemannian Dimension*—a spectrum-aware, pointwise effective complexity—that governs generalization at the trained model and yields tight, analyzable bounds. We review each step and argue that the resulting bounds are tight in a qualified sense; moreover, they *exponentially sharpen* spectral–norm bounds (see Appendix F.5.1).

**Empirical Findings and Evidences.** The experiments are designed to systematically examine three central questions in modern deep learning: (i) why does overparameterization often improve generalization? (ii) how does feature learning evolve during training? and (iii) what implicit regularization is encoded by the baseline optimizer? Across the experimental results, we observe that (i) the overparameterization impressively leads to decreasing Riemannian Dimension; (ii) feature learning compresses the effective ranks of learned features during the training; and (iii) SGD with momentum implicitly regularizes the Riemannian Dimension.

## 2 A POINTWISE FRAMEWORK OF GENERALIZATION

Let $\mathcal{F}$ be a hypothesis class, $z$ be random data drawn from an unknown distribution $\mathbb{P}$ (e.g., input-label pair $z = (x, y)$), and $\ell(f; z)$ be real-valued loss function. Denote by $\mathbb{P}_n$ the empirical distribution supported on an i.i.d. sample $S = \{z_i\}_{i=1}^n \sim \mathbb{P}^n$. Our goal is to control the *generalization gap* $(\mathbb{P} - \mathbb{P}_n)\ell(f; z)$ in the following manner: for $\delta \in (0, 1)$, with probability at least $1 - \delta$, uniformly over every $f \in \mathcal{F}$,

$$(\mathbb{P} - \mathbb{P}_n)\ell(f; z) := \mathbb{E}_{z \sim \mathbb{P}}\big[\ell(f; z)\big] - \frac{1}{n}\sum_{i=1}^n \ell(f; z_i) \leq C\sqrt{\frac{d(f) + \log\frac{1}{\delta}}{n}}, \tag{3}$$

where $d(f)$ is a hypothesis-dependent complexity measure that aims to characterize the intrinsic complexity of every *trained* hypothesis $f$, in contrast to class-wide, uniformly defined complexity measures. Further details are provided in the appendix.

In the spirit of (3), we introduce the central notion of this section—the *pointwise dimension*: a finite-scale analogue of ideas from fractal geometry (Falconer, 1997) and a pointwise counterpart distilled from generic chaining (Fernique, 1975). Throughout, "metric" $\varrho$ means a *pseudometric*: all metric axioms hold except that $\varrho(f_1, f_2) = 0$ need not imply $f_1 = f_2$.

**Definition 1 (Pointwise Dimension)** *Given a function class $\mathcal{F}$, a metric $\varrho$ on $\mathcal{F}$, and a prior $\pi$ over $\mathcal{F}$, the local dimension at $f$ with scale $\varepsilon$ is defined as the log inverse density of the $\varepsilon-$ ball $B_\varrho(f, \varepsilon) = \{f' \in \mathcal{F} : \varrho(f, f') \leq \varepsilon\}$ centered at $f$:*

$$\log\frac{1}{\pi(B_\varrho(f, \varepsilon))}. \tag{4}$$

We define the loss-induced empirical $L_2(\mathbb{P}_n)$ metric $\varrho_{n,\ell}$ as $\varrho_{n,\ell}(f_1, f_2) = \sqrt{\frac{1}{n}\sum_{i=1}^n (\ell(f_1; z_i) - \ell(f_2; z_i))^2}$. Equipped with this data–dependent metric, we now state the following unified pointwise dimension generalization upper bound.

**Theorem 1 (Pointwise Dimension Generalization Bound)** *Let $\ell(f; z) \in [0, 1]$. There exists an absolute constant $C > 0$ such that for any prior $\pi$ on $\mathcal{F}$ and any $\delta \in (0, 1)$, with probability at least $1 - \delta$, uniformly over every $f \in \mathcal{F}$*

$$(\mathbb{P} - \mathbb{P}_n)\ell(f; z) \leq C\left(\inf_{\alpha > 0}\left\{\alpha + \frac{1}{\sqrt{n}}\int_\alpha^1 \sqrt{\log\frac{1}{\pi(B_{\varrho_{n,\ell}}(f, \varepsilon))}}d\varepsilon\right\} + \sqrt{\frac{\log\frac{\log(2n)}{\delta}}{n}}\right).$$

The concept of pointwise dimension and the unified generalization bound in Theorem 1 strengthen several established methodologies such as PAC-Bayesian analysis, Kolmogorov complexity, chaining, and generic chaining. We elaborate on this unified strengthening in the next two paragraphs.

**Theorem 1 sharpens best PAC–Bayes optimization.** By the monotonicity of the pointwise dimension in $\varepsilon$, a direct relaxation of Theorem 1 yields the one–shot bound

$$(\mathbb{P} - \mathbb{P}_n)\,\ell(f; z) \leq C\left(\inf_{\alpha > 0}\left\{\underbrace{\alpha}_{\text{bias (approximate } f)} + \underbrace{\sqrt{\frac{\log\frac{1}{\pi(B_{\varrho_{n,\ell}}(f, \alpha))}}{n}}}_{\text{variance (PAC–Bayes term)}}\right\} + \sqrt{\frac{\log\big(\log(2n)/\delta\big)}{n}}\right). \tag{5}$$

Intuitively, the pointwise dimension uses prior mass over a *ball* around $f$, so it applies to general (uncountable) classes, overcoming limitations of hypothesis–by–hypothesis bounds such as Occam/description–length and Kolmogorov complexity (Lotfi et al., 2022). Additionally, our perspective brings the best possible PAC-Bayesian mechanism: generalization is recast as a bias–variance tradeoff optimized over a user–chosen posterior, applies to *deterministic* hypotheses, and shows that the pointwise dimension optimally governs the complexity (see Section C.3 for this perspective). This clarifies and strengthens earlier PAC–Bayes approaches that adopt linear-in-parameter Gaussian approximations, i.e., they linearize $f$ in the weights and thereby ignore its nonlinearity, to obtain computable, non-vacuous generalization bounds (e.g., (Hinton & Van Camp, 1993; Dziugaite & Roy, 2017); see the second part of Section C.1 for details)).

**Theorem 1 upgrades generic chaining to a pointwise form.** The theorem extends generic chaining (notably the majorizing measure integral (Fernique, 1975; Talagrand, 1987)) to *pointwise* bounds, and is therefore strictly stronger than entropy–integral bounds based on *uniform* covering numbers (e.g., Dudley's integral), whose integrand takes a supremum over the entire class $\mathcal{F}$ rather than localizing at the realized hypothesis; see Section 3 of Block et al. (2021) and Section 4.1 of Chen et al. (2024). In particular, (34) in Appendix C.4 shows that

$$\inf_{\pi} \sup_{f \in \mathcal{F}} \frac{1}{\pi(B_{\varrho}(f, \varepsilon))} \tag{6}$$

is (up to absolute constants) equivalent to the *canonical covering number* of $\mathcal{F}$ with metric $\varrho$ at scale $\varepsilon$. Consequently, Theorem 1 goes beyond classical covering analyses by (i) recasting covering-number complexity as the inverse-prior-density objective (6), and (ii) localizing this complexity *pointwise* in $f$. We view this "prior-density + localization" perspective as a paradigm shift for future statistical complexity analysis. The multiscale integral is particularly valuable: it applies to rich classes where the pointwise dimension can grow as $O\big(d(f)\,\varepsilon^{-2}\big)$ yet still yields a $\sqrt{d(f)/n}$ rate; by contrast, the one–shot relaxation (5) typically requires growth no worse than $O\big(d(f)\log(1/\varepsilon)\big)$ to achieve the same rate.

Finally, the integral upper bound in Theorem 1 is *tight* in the following qualified worst-case sense: no uniform improvement valid simultaneously for all hypotheses and all priors is possible. This is witnessed by a matching lower bound.

**Theorem 2 (Worst-Case Lower Bound)** *Let $\ell(f; z) \in [0, 1]$. There exists absolute constants $c, c' > 0$ so that*

$$\mathbb{E}\left[ \sup_{\pi \in \Delta(\mathcal{F}), f \in \mathcal{F}} \left( (\mathbb{P} - \mathbb{P}_n)\ell(f; z) - \frac{c}{\sqrt{n \log n}} \int_0^1 \sqrt{\log \frac{1}{\pi(B_{\varrho_{n,\ell}}(f, \varepsilon))}} d\varepsilon \right) + \frac{c' \sup_{\mathcal{F}} \mathbb{E}[\ell(f; z)]}{\sqrt{n \log n}} \right] \geq 0,$$

*where notation $\mathbb{E}$ means taking expectation over sample.*

The lower bound certifies the *worst-case tightness* of our pointwise-dimension upper bound in Theorem 1 (noting that fixing $\alpha = 0$ relative to Theorem 1 only increase the lower bound), analogous to *minimax optimality* in frequentist statistics (Wald, 1945). This worst-case tightness does not preclude sharper guarantees for a fixed hypothesis $f$. However, a strictly *pointwise* lower bound—one that conditions on the realized hypothesis $f$ without the outer $\sup_{f \in \mathcal{F}}$—is generally unattainable, because any admissible prior $\pi$ must be chosen independently of $f$ (a "no free lunch" constraint).

# 3 DEEP NEURAL NETWORKS AND RIEMANNIAN DIMENSION

In this section we develop a systematical pointwise dimension analysis for deep neural networks. Section 3.1 formalizes the standard fully connected architecture and notation. Section 3.2 introduces a non-perturbative calculus (avoiding infinitesimal Taylor expansions) to analyze finite-scale behavior—the scale at which generalization is actually governed, and which is intrinsically captured by the pointwise-dimension framework. Section 3.3 introduces a hierarchical covering scheme—our key technical innovation—that overcomes the well-known linear/kernel bottleneck in classical statistical learning and enables a principled treatment of genuinely nonlinear models.

## 3.1 NEURAL NETWORK SETUP

We consider fully connected (feed-forward) networks that map an input $x \in \mathbb{R}^{d_0}$ to an output $f_L(W, x) \in \mathbb{R}^{d_L}$. The architecture is specified by widths $d_0, \ldots, d_L$ and weight matrices $W = \{W_1, \ldots, W_L\}$ with $W_l \in \mathbb{R}^{d_l \times d_{l-1}}$ for $l = 1, \ldots, L$. Let $\sigma_1, \ldots, \sigma_L$ be nonlinear activations (e.g., ReLU), acting componentwise on column vectors, and each $\sigma_l : \mathbb{R}^{d_l} \to \mathbb{R}^{d_l}$ is assumed 1-Lipschitz. The network's forward map is the composition

$$f_L(W, x) := \sigma_L\Big( W_L \, \sigma_{L-1}\big( W_{L-1} \cdots \sigma_1(W_1 x) \big) \Big).$$

Let $X = [x_1, \ldots, x_n] \in \mathbb{R}^{d_0 \times n}$ collect the $n$ training inputs as columns. For each layer $l \in \{1, \ldots, L\}$, define the depth-$l$ map and the corresponding *feature matrix*

$$f_l(W, x) := \sigma_l\Big( W_l \, \sigma_{l-1}\big( W_{l-1} \cdots \sigma_1(W_1 x) \big) \Big), \quad F_l(W, X) := \big[ f_l(W, x_1) \cdots f_l(W, x_n) \big] \in \mathbb{R}^{d_l \times n}.$$

Equivalently (full, non-recursive form consist with (1)),

$$F_l(W, X) \;=\; \sigma_l\Big(W_l\,\sigma_{l-1}\big(W_{l-1}\,\cdots\,\sigma_1(W_1 X)\big)\Big),$$

where for a matrix $A = [a_1, \ldots, a_n]$ we write $\sigma_l(A) := [\,\sigma_l(a_1), \ldots, \sigma_l(a_n)\,]$. Thus $F_L(W, X)$ collects the network outputs on the dataset $X$.

We denote $\|\cdot\|_{\mathsf{F}}$ for the Frobenius norm, $\|\cdot\|_{\mathrm{op}}$ for the spectral norm, and $\|\cdot\|_2$ for the Euclidean norm on vectors. We abbreviate norm balls by $B_{\mathsf{F}}(R)$, $B_{\mathrm{op}}(R)$, and $B_2(R)$ (all centered at 0; radius being $R$). The empirical $L_2(\mathbb{P}_n)$ distance between two hypotheses $W, W'$ is (a $1/\sqrt{n}$ scaling is used to keep consistency with Section 2)

$$\varrho_n(W, W') \;:=\; \sqrt{\big\|F_L(W, X) - F_L(W', X)\big\|_{\mathsf{F}}^2 / n}\,.$$

The function-level empirical metric and generalization statements in Section 2 for the loss $x \mapsto \ell(f_L(W, x), y)$ at data–label pairs $z = (x, y)$ specialize, on the dataset $X$, to the metric $\varrho_n$ defined above. We assume the loss $\ell(\cdot, y)$ is $\beta$-Lipschitz in its first argument with respect to $f_L(W, x)$, which bridges the metric $\varrho_{n,\ell}$ studied in Section 2 to $\varrho_n$ defined on the weight space.

### 3.2 Non-Perturbative Expansion and Layer-wise Correlation

Throughout, our finite-scale analysis relies on *non-perturbative* expansions. Borrowing terminology from theoretical physics, "non-perturbative" here means we avoid Taylor/derivative expansions and instead use exact, telescoping algebraic identities that hold at finite scale. For example,

$$W_2'W_1' - W_2 W_1 = W_2'(W_1' - W_1) + (W_2' - W_2)W_1, \qquad \Sigma'^{-1} - \Sigma^{-1} = \Sigma'^{-1}(\Sigma - \Sigma')\Sigma^{-1},$$

with analogous decompositions used throughout. This viewpoint preserves the full finite-scale geometry of deep networks, rather than linearizing around an infinitesimal neighborhood.

To present our non-perturbative expansion for DNN, we define *local Lipchitz constant* $M_{l \to L}(W, \varepsilon)$, which characterizes the sensitivity of the layer $L$ output, $F_L$, to variations in layer $l$'s output, within a neighborhood around $F_l$. Formally, we assume that for every $W' \in B_{\varrho_n}(W, \varepsilon)$

$$\|F_L(F_l(W', X), \{W_i'\}_{i=l+1}^L) - F_L(F_l(W, X), \{W_i'\}_{i=l+1}^L)\|_{\mathsf{F}} \le M_{l \to L}(W, \varepsilon)\|F_l(W', X) - F_l(W, X)\|_{\mathsf{F}}.$$

Local Lipschitz constants are typically much smaller than products of spectral norms and can be computed by formal–verification toolchains (Shi et al., 2022). In our bounds these constants appear only inside *logarithmic factors*, so they do not affect the leading rates. For completeness, we discuss them carefully in Appendix F.5.1. We propose a telescoping decomposition to replace conventional Taylor expansion, where in each summand the only difference lie in $W_l'$ and $W_l$.

$$F_L(W', X) - F_L(W, X)$$

$$= \sum_{l=1}^L [\underbrace{\sigma_L(W_L' \cdots W_{l+1}'}_{\text{controled by } M_{l \to L}} \underbrace{\sigma_l}_{\text{by } 1}(W_l' \underbrace{F_{l-1}(W, X)}_{\text{learned feature}})) - \sigma_L(W_L' \cdots W_{l+1}' \sigma_l(W_l \underbrace{F_{l-1}(W, X)}_{\text{learned feature}}))]. \quad (7)$$

Note that this is a *non-perturbative* expansion that holds unconditionally and does not rely on infinitesimal approximation, and crucially keeps the *learned* feature $F_{l-1}(W, X)$ at the *trained* weight $W$. From this decomposition and applying basic inequalities, we have the following key lemma.

**Lemma 1 (Non-Perturbative Feature Expansion)** *For all $W' \in B_{\varrho_n}(W, \varepsilon)$,*

$$\|F(W', X) - F(W, X)\|_{\mathsf{F}}^2 \le \sum_{l=1}^L L \cdot M_{l \to L}[W, \varepsilon]^2 \cdot \|(W_l' - W_l)F_{l-1}(W, X)\|_{\mathsf{F}}^2. \quad (8)$$

*The lemma captures the first structural principle of fully connected DNN:* cross-layer correlations mostly pass through the feature matrices, preserving an approximate pointwise linear structure.

Since enlarging the metric only shrinks metric balls and hence *increases* the pointwise dimension (4) we analyze in Section 2 (formalized as Lemma 19; metric domination lemma), it suffices to analyze pointwise dimension under the *pointwise ellipsoidal metric* that appears on the right-hand

side of Lemma 1. Concretely, $F_{l-1}(W, X) F_{l-1}(W, X)^\top$, the feature Gram matrix from layer $l-1$, faithfully encodes the spectral information induced by the network–data geometry at layer $l$. Working with the corresponding pointwise ellipsoidal metric yields sharp, *pointwise, spectrum-aware* bounds with the desired properties for deep networks, and underpins our tractability results (with the structural principles and technical innovations to developed in the next subsection).

### 3.3 HIERARCHICAL COVERING FROM LOCAL CHART TO GLOBAL ATLAS

Lemma 1 suggests that the following *pointwise ellipsoidal metric* dominates $n \cdot \varrho_n$ at every $W$ (here, NP stands for "non-perturbative"):

$$G_{\mathrm{NP}}(W) = \mathrm{blockdiag}\left(\cdots, LM_{l \to L}^2(W, \varepsilon) \cdot F_{l-1}(W, X) F_{l-1}^\top(W, X) \otimes I_{d_l}, \cdots\right)$$

$$\varrho_{G_{\mathrm{NP}}(W)}(W, W') = \mathrm{vec}(W' - W)^\top G_{\mathrm{NP}}(W) \mathrm{vec}(W' - W). \tag{9}$$

We are therefore interested in bounding the enlarged pointwise dimension under the pointwise ellipsoidal metric $\varrho_{G_{\mathrm{NP}}(W)}$:

$$\log \frac{1}{\pi(B_{\varrho_n}(f(W, \cdot), \varepsilon))} \leq \log \frac{1}{\pi(B_{\varrho_{G_{\mathrm{NP}}(W)}}(W, \sqrt{n}\varepsilon))}.$$

This section offers a deep dive past classical effective dimension, shifting to hierarchical covering and a global geometric analysis.

#### 3.3.1 GOLDEN STANDARD: EFFECTIVE DIMENSION

Classical studies of static ellipsoidal metrics suggest that if $\pi$ is chosen to be uniformly constrained on the top-$r$ eigenspace of a PSD matrix $G(W)$, and the vectorized weights $W \in \mathbb{R}^p$ are restricted to the Euclidean ball $B_2(R) := \{w \in \mathbb{R}^p : \|w\|_2 \leq R\}$, then one can achieve a tight effective dimension as follows: define the *effective rank*

$$r_{\mathrm{eff}}(G(W), R, \varepsilon) := \max\{k : \lambda_k(G(W)) R^2 \geq n\varepsilon^2/2\}, \tag{10}$$

where the eigenvalues $\{\lambda_k(G(W))\}$ are ordered nonincreasingly; and define the spectrum-aware *effective dimension*

$$d_{\mathrm{eff}}(G(W), R, \varepsilon) := \frac{1}{2} \sum_{k=1}^{r_{\mathrm{eff}}(G(W), R, \varepsilon)} \log\left(\frac{8R^2 \lambda_k(G(W))}{n\varepsilon^2}\right). \tag{11}$$

This definition serves as a gold standard for static ellipsoidal metrics and is asymptotically tight, as established by the covering number of the unit ball with ellipsoids in Dumer et al. (2004). For brevity, we write $r$ for $r_{\mathrm{eff}}(G(W), R, \varepsilon)$, and denote by $\mathcal{V} \subseteq \mathbb{R}^p$ the $r$-dimensional subspace corresponding to the top-$r_{\mathrm{eff}}$ eigenspace of $G(W)$.

#### 3.3.2 KEY CHALLENGE: PRIOR INDEPENDENCE FROM $W$.

However, the main challenge is that the prior $\pi$ must be chosen independently of the training data. This means that the construction of $\pi$ cannot rely on knowledge of the learned weights $W$, including their top-$r_{\mathrm{eff}}$ eigenspace, yet still capture the underlying geometric structure. The next lemma extends classical results on static ellipsoidal metrics by showing that a uniform prior over a reference subspace $\bar{\mathcal{V}}$ suffices to bound the pointwise dimension for all $W$ whose top-$r$ eigenspace of $G(W)$ can be approximated by $\bar{\mathcal{V}}$.

**Lemma 2 (Pointwise Dimension via Reference Subspace)** *Consider the weight space $B_2(R) \subset \mathbb{R}^p$ for vectorized weights, and a pointwise ellipsoidal metric defined via PSD $G(W)$. Let $\bar{\mathcal{V}} \subseteq \mathbb{R}^p$ be a fixed $r$-dimensional subspace. Define the prior $\pi_{\bar{\mathcal{V}}} = \mathrm{Unif}(B_2(1.58R) \cap \bar{\mathcal{V}})$. Then, uniformly over all $(W, \varepsilon)$ such that the top-$r$ eigenspace $\mathcal{V}$ of $G(W)$ can be approximated by $\bar{\mathcal{V}}$ to precision*

$$\varrho_{\mathrm{proj}, G(W)}(\mathcal{V}, \bar{\mathcal{V}}) := \left\| G(W)^{1/2}(\mathcal{P}_\mathcal{V} - \mathcal{P}_{\bar{\mathcal{V}}}) \right\|_{\mathrm{op}} \leq \frac{\sqrt{n}\varepsilon}{4R}, \tag{12}$$

*we have*

$$\log \frac{1}{\pi_{\bar{\mathcal{V}}}(B_{\varrho_{G(W)}}(W, \sqrt{n}\varepsilon))} \leq \frac{1}{2} \sum_{k=1}^{r_{\mathrm{eff}}(G(W), R, \varepsilon)} \log\left(\frac{40R^2 \lambda_k(G(W))}{n\varepsilon^2}\right) = d_{\mathrm{eff}}(G(W), \sqrt{5}R, \varepsilon).$$

In (12), $\mathcal{P}_{\mathcal{V}}$ denotes the orthogonal projector onto the subspace $\mathcal{V}$, and $\varrho_{\mathrm{proj},G(W)}$ thus defines an ellipsoidal projection metric between subspaces. Further details are provided in the appendix.

### 3.3.3 Hierarchical covering (mixture prior over subspaces).

We introduce a hierarchical covering framework that pushes learning beyond classical linear and kernel paradigms, providing a principled toolkit for genuinely nonlinear models—one of the central innovations of this work. It operates on two levels: a bottom-level local-chart covering that captures spectrum-aware behavior within a fixed subspace, and a top-level global geometric analysis over the Grassmannian. (i) For each reference subspace $\bar{\mathcal{V}}$, placing a uniform prior on $\bar{\mathcal{V}}$ yields a tight pointwise-dimension bound for all "local" weights W whose top$-r$ eigenspace of $G(W)$ is well approximated by $\bar{\mathcal{V}}$ (see Lemma 2). (ii) At the top level, we place a prior over reference subspaces $\bar{\mathcal{V}}$ and average the local priors, producing a data-independent prior and the final bound.

By combining these two levels of priors, we obtain a pointwise dimension bound using a prior $\pi$ that is completely blind to the choice of $W$. To formalize this, we introduce a top-level distribution $\mu$ over the Grassmannian

$$\mathrm{Gr}(p,r) := \{r\text{–dimensional linear subspaces of } \mathbb{R}^p\}$$

the collection of all $r$-dimensional subspaces, and define $\pi(W) = \sum_{\mathcal{V}} \mu(\mathcal{V}) \pi_{\mathcal{V}}(W)$. We refer to this two-stage construction as the hierarchical covering argument. Under the resulting prior $\pi$, the following bound holds uniformly over all (vectorized) $W \in B_2(R)$, the pointwise dimension $\log \frac{1}{\pi(B_{\varrho_{G(W)}}(W,\sqrt{n}\varepsilon))}$ is bounded by two parts:

$$\underbrace{\log \frac{1}{\mu(B_{\varrho_{\mathrm{proj},G(W)}}(\mathcal{V},\sqrt{n\varepsilon/4R}))}}_{\text{covering Grassmannian (global atlas)}} + \underbrace{\sup_{\bar{\mathcal{V}} \in B_{\varrho_{\mathrm{proj},G(W)}}(\mathcal{V},\sqrt{n\varepsilon/4R})} \log \frac{1}{\pi_{\bar{\mathcal{V}}}(B_{\varrho_{G(W)}}(W,\sqrt{n}\varepsilon))}}_{\text{covering local charts}}, \quad (13)$$

In differential–geometric terms, our argument has two components.

- *Local (chart) analysis:* fixing a reference subspace $\bar{\mathcal{V}}$, we use effective dimension as the gold standard to determine the metric entropy of the corresponding local chart.
- *Global (atlas) covering:* we cover the Grassmannian by such reference subspaces, i.e., we bound the metric entropy of the global atlas and account for the cost of transitioning across local charts.

Lemma 2 controls the local part, while the following new result (Lemma 3) on the *ellipsoidal* covering of the Grassmannian controls the global part.[1]

**Lemma 3 (Ellipsoidal Covering of the Grassmannian manifold)** *Consider the Grassmannian* $\mathrm{Gr}(d,r)$. *For uniform prior* $\mu = \mathrm{Unif}(\mathrm{Gr}(d,r))$, *we have that for every* $\mathcal{V} \in \mathrm{Gr}(d,r)$, *every* $\varepsilon > 0$ *and every PSD matrix* $\Sigma$ *with eigenvalues* $\lambda_1 \geq \cdots \geq \lambda_d$, *we have the pointwise dimension bound*

$$\log \frac{1}{\mu(B_{\varrho_{\mathrm{proj},\Sigma}}(\mathcal{V},\varepsilon))} \leq \frac{d-r}{2} \sum_{k=1}^{r} \log \frac{C \max\{\lambda_k,\varepsilon^2\}}{\varepsilon^2} + \frac{r}{2} \sum_{k=1}^{d-r} \log \frac{C \max\{\lambda_k,\varepsilon^2\}}{\varepsilon^2},$$

*where* $C > 0$ *is an absolute constant.*

The result above is mathematically significant in its own right. It extends the classical metric-entropy (covering number) theory for the Grassmannian—where log covering number $\asymp r(d-r)\log(C/\varepsilon)$ under the *isotropic* projection metric— to an *ellipsoidal* (anisotropic) metric that captures feature–

---

[1]Since the effective rank $r$ of $\bar{\mathcal{V}}$ can take any value in $\{1, \ldots, p\}$, the top-level Grassmannian covering must range over all $\mathrm{Gr}(p,r)$. This adds only a negligible $O(\log p)$ overhead to the global-level cost. Accordingly, we construct a data-independent prior in three prior hierarchy: ("global-$r$") choose the rank $r$ (paying the $\log p$ overhead), ("global-$\bar{\mathcal{V}}$") choose a reference subspace $\bar{\mathcal{V}} \in \mathrm{Gr}(p,r)$, and ("local") sample within $\bar{\mathcal{V}}$ using the local chart prior; see Figure 3 in the Appendix for an illustration. For conceptual clarity, Lemma 3 focuses on the Grassmannian covering cost at a fixed rank $r$; and we defer the layer–specific specialization (to each $d_{l-1} \times d_{l-1}$ feature–matrix block) to the calculation in (15).

and model–induced geometry. This generalization translates the traditional differential-geometric and Lie-algebraic treatments (see Appendix E) and, we believe, illustrates a two–way exchange: deep mathematical structure is essential to understanding generalization in modern neural networks, and, conversely, generalization theory can motivate new questions and results in pure mathematics.

Leveraging the block–decomposable structure in (9), the $l$–th block is

$$G_l(W) = A_l(W) \otimes I_{d_l}, \text{ where } A_l(W) = LM_{l \to L}^2(W, \varepsilon) \cdot F_{l-1}(W, X)F_{l-1}(W, X)^\top \in \mathbb{R}^{d_{l-1} \times d_{l-1}}.$$

Since the Kronecker factor is $I_{d_l}$, the spectrum of $G_l$ consists of the eigenvalues of the feature Gram matrix $F_{l-1}F_{l-1}^\top$ (scaled by $LM_{l \to L}^2$), each repeated $d_l$ times. Consequently, the *local–chart* (within–subspace) covering cost at layer $l$ scales as

$$d_l \cdot d_{\text{eff}}\Big(LM_{l \to L}^2(W, \varepsilon) \cdot F_{l-1}(W, X)F_{l-1}(W, X)^\top\Big), \tag{14}$$

while the *atlas* (subspace–selection) cost is the Grassmannian term over $\text{Gr}\big(d_{l-1}, r_{\text{eff}}[W, l]\big)$, where $r_{\text{eff}}[W, l]$ is the effective rank of $A_l(W) \in \mathbb{R}^{d_{l-1} \times d_{l-1}}$. By Lemma 3 (and the footnote preceding it), the *global-atlas* (choosing-subspace) covering cost at layer $\ell$ scales as

$$d_{l-1} \cdot d_{\text{eff}}\Big(LM_{l \to L}^2(W, \varepsilon) \cdot F_{l-1}(W, X)F_{l-1}(W, X)^\top\Big) + \log(d_{l-1}). \tag{15}$$

Together, (14) and (15) yield a clean layerwise decomposition: the width $d_l$ multiplies the spectral complexity of incoming features (local charts), whereas the input dimension $d_{l-1}$ governs the Grassmannian covering (global atlas). This complementary, seemingly magical "duality" underlies the calculation below.

**Theorem 3 (Riemannian Dimension for DNN)** *Consider the weight space $B_{\mathsf{F}}(R)$, and a point-wise ellipsoidal metric defined via the ellipsoidal metric $G_{\text{NP}}(W)$ defined in (9). Define the point-wise Riemannian Dimension*

$$d_{\mathsf{R}}(W, \varepsilon) = \sum_{l=1}^{L} \Big( \underbrace{d_l \cdot d_{\text{eff}}(A_l(W))}_{\text{covering local charts}} + \underbrace{d_{l-1} \cdot d_{\text{eff}}(A_l(W))}_{\text{covering global atlas}} + \underbrace{\log(d_{l-1})}_{\text{covering discrete } r_{\text{eff}}} + \log n \Big),$$

*where $A_l(W)$ is the the feature matrix $LM_{l \to L}^2(W, \varepsilon) \cdot F_{l-1}(W, X)F_{l-1}^\top(W, X)$; and $d_{\text{eff}}(A_l(W))$ is abbreviation of $d_{\text{eff}}(A_l(W), C \max\{\|W\|_{\mathsf{F}}, R/2^n\}, \varepsilon)$ with $C > 0$ an absolute constant. Then we have the pointwise dimension bound: there exists a prior $\pi$ such that uniformly over all $W \in B_{\mathsf{F}}(R)$,*

$$\log \frac{1}{\pi(B_{\varrho_n}(f(W, \cdot), \varepsilon))} \leq d_{\mathsf{R}}(W, \varepsilon).$$

This concludes our program for fully connected networks: we establish *Riemannian Dimension* as a principled complexity measure that explains—and sharply bounds—generalization. We summarize the *second structural principle of fully connected DNN*: The complexity of the *global atlas* (covering the space of reference top eigenspaces) remains commensurate with the layerwise, spectrum–aware complexity of covering the *local charts*. On closer inspection, the effect hinges on the block–decomposable structure in (9). This structure is intrinsic to layered neural networks and typically absent in generic nonlinear models, which helps explain why DNN are particularly amenable to sharp generalization analysis.

## 4 GENERALIZATION BOUNDS FOR DNN

We are now ready to state our generalization bound for fully connected DNN here. Combining Theorem 3 and Theorem 1, we establish the following theorem.

**Theorem 4 (Generalization Bound for DNN)** *Let the loss $\ell(f(W, x), y)$ be bounded in $[0, 1]$ and $\beta$–Lipschitz with respect to $f(W, x)$, for every $\delta \in (0, 1)$, with probability at least $1 - \delta$, uniformly over all $W \in B_{\mathsf{F}}(R)$,*

$$(\mathbb{P} - \mathbb{P}_n)\ell(f(W, x), y) \leq C_1 \left( \inf_{\alpha > 0} \left\{ \alpha + \frac{\beta}{\sqrt{n}} \int_{\alpha}^1 \sqrt{d_{\mathsf{R}}(W, \varepsilon)} d\varepsilon \right\} + \sqrt{\frac{\log \frac{\log(2n)}{\delta}}{n}} \right),$$

*where the Riemannian Dimension is defined by*

$$d_{\mathbf{R}}(W, \varepsilon) = \sum_{l=1}^{L} \left( (d_l + d_{l-1}) \underbrace{\sum_{k=1}^{r_{\mathrm{eff}}[W,l]} \log \frac{8C_2^2 \lambda_k(F_{l-1}F_{l-1}^\top)}{n\varepsilon^2}}_{\text{spectrum of inner layers } 1:l-1} \right.$$

$$\left. + (d_l + d_{l-1}) r_{\mathrm{eff}}[W,l] \cdot \underbrace{\log\left(M_{l\to L}^2(W, \varepsilon) L \max\{\|W\|_{\mathsf{F}}^2, R^2/4^n\}\right) + \log(d_{l-1}n)}_{\text{spectrum of outer layers } l+1:L} \right), \qquad (16)$$

*where $F_{l-1}$ is learned feature $F_l(W, X)$; and the effective rank $r_{\mathrm{eff}}[W,l]$ is the abbreviation of $r_{\mathrm{eff}}(LM_{l\to L}^2(W, \varepsilon)F_{l-1}F_{l-1}^\top, C_2 \max\{\|W\|_{\mathsf{F}}, R/2^n\}, \varepsilon)$, where $C_1, C_2 > 0$ are absolute constants.*

**Interpreting** (16) **to the informal rate** (2). Although $r_{\mathrm{eff}}[W,l]$ incorporates local Lipschitz factors—specifically, the effective rank is computed for $LM_{l\to L}^2(W, \varepsilon) F_{l-1}F_{l-1}^\top$ rather than $F_{l-1}F_{l-1}^\top$ alone—when $F_{l-1}F_{l-1}^\top$ exhibits rapidly decaying eigenvalues this dependence is strongly suppressed; it disappears entirely under strict low rank (as also observed in our experiments). Consequently, under mild low-rank or spectral-decay conditions, the bound aligns with the informal rate (2). In (16), the first and second parts correspond to the inner and outer layers, respectively. For each layer. For each layer $l$, the first ("log–eigenvalues") term in (16) quantifies the contribution of the inner layers $1:l-1$ via the feature Gram $F_{l-1}F_{l-1}^\top$, while the second ("log–Lipschitz") term captures the influence of the outer layers $l+1:L$ through $M_{l\to L}$—making explicit how the outer layers enter the bound and restoring inner/outer symmetry. Together, these terms provide a complete layerwise account of the effective dimension in the informal rate (2).

**Tightness of each step and resulting bounds.** We conclude by reviewing our comprehensive theory for generalization in fully connected networks and justifying the tightness of the resulting bounds. **First**, in Section 2 we develop a framework based on *pointwise dimension*. The upper and lower bounds match in a qualified (non-uniform) sense (see remarks after Theorem 1), and the framework has a profound connection to finite-scale geometry—evidence that this is the right organizing principle. **Second**, Section 3 introduces a *non-perturbative* expansion. Lemma 1 applies Cauchy–Schwarz layerwise (treating each layer as a block). While there may be room to improve depth dependence, the telescoping decomposition (7) is an exact *equality*, so the expansion is generally sharp (and fully avoid linearization). **Third**, the hierarchical covering argument shows that the resulting *Riemannian Dimension* bound matches the gold standard of *effective dimension*. Thus our pointwise, spectrum-aware bounds achieve the optimal form dictated by static ellipsoid theory.

**Comparison with Norm Bounds, VC, and NTK.** Our framework yields *exponentially* tighter rates than norm–product bounds, refines VC–type statements into hypothesis– and data–dependent guarantees, and replaces infinitesimal linearization with a finite-scale, non-perturbative analysis that holds simultaneously for every trained hypothesis. For space, we defer further explanations to Appendix F.1 and the recovery of representative norm bounds and comparisons to Appendix F.5.1.

## 5 EXPERIMENTS

We evaluate the proposed Riemannian Dimension (RD) on two standard settings: (i) width sweeps for fully connected networks (FCNs) on MNIST (LeCun et al., 1998); and (ii) depth sweeps for ResNets on CIFAR-10 (Krizhevsky, 2009). FCNs use a 9-hidden-layer architecture with shared hidden width $h \in \{2^6, 2^7, \ldots, 2^{12}\}$; ResNets follow the canonical three-stage, basic-block designs (ResNet-20/32/44/56/74/110) (He et al., 2016). We organize results around three questions: (Q1) why overparameterization can *improve* generalization; (Q2) how feature learning compresses intrinsic dimension over training; and (Q3) whether baseline optimizers exhibit low-RD implicit bias. Full setup and additional plots are deferred to the appendix.

**RD Explains the Blessing of Overparameterization.** We compare RD against classical capacity surrogates (spectral-norm bounds (Bartlett et al., 2017) and VC-dimension proxies (Bartlett et al., 2019)). Final-epoch metrics of FCNs on MNIST and ResNets on CIFAR-10 are reported in Table 1 and Table 2, respectively. In these Tables, the train error quickly collapses to zero for sufficiently large models, confirming their expressive capacity. Consistently, the generalization can continue to

Table 1: Final-epoch FCN results on MNIST. "Gen" is test minus train error. Spectral-norm column reports the spectrally normalized margin bound of Bartlett et al. (2017). VC uses the near-tight proxy of Bartlett et al. (2019) (reported as $P\,L\log P$). RD is our Riemannian Dimension.

| Model | Train | Gen | Spectral Norm | #Params | VC dim | RD |
|---|---|---|---|---|---|---|
| Width-$2^6$ | 0.0002 | 0.0205 | $3.146 \times 10^{15}$ | $5.961 \times 10^6$ | $9.299 \times 10^8$ | $6.433 \times 10^7$ |
| Width-$2^7$ | 0.0002 | 0.0187 | $2.695 \times 10^{15}$ | $6.167 \times 10^6$ | $9.641 \times 10^8$ | $6.097 \times 10^7$ |
| Width-$2^8$ | 0.0000 | 0.0191 | $2.093 \times 10^{15}$ | $6.726 \times 10^6$ | $1.057 \times 10^9$ | $5.589 \times 10^7$ |
| Width-$2^9$ | 0.0000 | 0.0186 | $2.401 \times 10^{15}$ | $8.434 \times 10^6$ | $1.345 \times 10^9$ | $5.316 \times 10^7$ |
| Width-$2^{10}$ | 0.0000 | 0.0215 | $4.816 \times 10^{15}$ | $1.421 \times 10^7$ | $2.340 \times 10^9$ | $5.266 \times 10^7$ |
| Width-$2^{11}$ | 0.0000 | 0.0160 | $1.001 \times 10^{16}$ | $3.520 \times 10^7$ | $6.116 \times 10^9$ | $4.972 \times 10^7$ |
| Width-$2^{12}$ | 0.0000 | 0.0210 | $1.466 \times 10^{16}$ | $1.149 \times 10^8$ | $2.133 \times 10^{10}$ | $4.803 \times 10^7$ |

Table 2: Final-Epoch Metrics of ResNets on CIFAR-10

| Model | Train Error | Gen Gap | # Parameters | VC dimension | R-D |
|---|---|---|---|---|---|
| ResNet-20 | 0.0016 | 0.0752 | $2.690 \times 10^5$ | $6.727 \times 10^7$ | $8.801 \times 10^6$ |
| ResNet-32 | 0.0003 | 0.0695 | $4.630 \times 10^5$ | $1.933 \times 10^8$ | $9.992 \times 10^6$ |
| ResNet-44 | 0.0001 | 0.0627 | $6.570 \times 10^5$ | $3.872 \times 10^8$ | $6.339 \times 10^6$ |
| ResNet-56 | 0.0000 | 0.0637 | $8.510 \times 10^5$ | $6.507 \times 10^8$ | $5.200 \times 10^6$ |
| ResNet-74 | 0.0000 | 0.0615 | $1.142 \times 10^6$ | $1.179 \times 10^9$ | $3.237 \times 10^6$ |
| ResNet-110 | 0.0000 | 0.0576 | $1.724 \times 10^6$ | $2.723 \times 10^9$ | $2.583 \times 10^6$ |

be improved as parameters increase, especially on ResNets (Table 2). This phenomenon means the overfitting does not appear and reflects a paradoxical truth of deep learning: over-parameterization is not a curse, but can benefit the generalization. However, classical complexity measures—e.g., the spectral norm and the VC dimension, often scale exponentially as the parameter count grows. Notably, the spectral norm is about $10^6$ times larger than the VC dimension and seems to be a worse complexity measure (see Table 1). The two measures therefore struggle to explain the generalization of modern overparameterized networks. In contrast, our Riemannian Dimension exhibits a consistent downward trend as model size grows—both under width scaling (last column of Table 1) and depth scaling (last column of Table 2), and it is about $10^3$ times smaller than the VC dimension, suggesting that the effective dimension—not raw parameter count—is the true indicator of generalization in deep learning. In summary, increased parameterization is associated with reduced effective model complexity, and Riemannian Dimension faithfully characterizes this phenomenon.

**Feature Learning Compresses Effective Rank.** We track the effective ranks of layerwise feature Gram matrices $F_{l-1}F_{l-1}^\top$ (scaled by $L\|W\|_{\mathsf{F}}^2 \prod_{i>l}\|W_i\|_{\mathrm{op}}^2$ per our theory). Across both FCN-width and ResNet-depth sweeps, effective ranks drop sharply after a brief transient and compress more with larger width/depth. On the largest FCN, the total effective rank shrinks by up to $\sim 300\times$; for the deepest ResNets, most layers compress near zero. This progressive, spectrum-aware compression explains why RD falls with capacity while test error improves. (Appendix: layerwise trajectories and ablations; we use the conservative spectral-product proxy for local Lipschitz terms.)

**SGD Finds Low-RD Solutions.** Finally, we examine optimizer bias. With standard SGD+momentum, RD consistently *decreases* by orders of magnitude during training (after an early transient), while VC-style proxies remain essentially unchanged. Thus, beyond driving the loss to zero, the optimizer preferentially selects low-RD interpolating solutions, aligning optimization dynamics with our complexity measure. (Appendix: training-time RD curves and robustness to optimizer hyperparameters.)

# 6 CONCLUSION

We establish a principled foundation for generalization in deep neural networks. Key innovations include a pointwise generalization framework, a non-perturbative calculus, and a hierarchical covering theory. Empirical validations confirm our predictions in deep learning practice.

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

CONTENTS

Table 3: Final-epoch Effective Ranks for FCNs on MNIST, where Width$-2^\star$ means $h = 2^\star$, and where for the form A/B, A represents the effective rank and B represents the original dimension, and where Layer-1 means the input layer.

| Metric | Width-$2^6$ | Width-$2^7$ | Width-$2^8$ | Width-$2^9$ | Width-$2^{10}$ | Width-$2^{11}$ | Width-$2^{12}$ |
|---|---|---|---|---|---|---|---|
| Layer-1 | 713/763 | 712/763 | 710/763 | 710/763 | 707/763 | 707/763 | 704/763 |
| Layer-2 | 2048/2048 | 2044/2048 | 2042/2048 | 2048/2048 | 2047/2048 | 2048/2048 | 2048/2048 |
| Layer-3 | 2048/2048 | 2045/2048 | 2037/2048 | 2019/2048 | 1925/2048 | 1460/2048 | 1009/2048 |
| Layer-4 | 61/64 | 97/128 | 92/256 | 85/512 | 79/1024 | 79/2048 | 59/4096 |
| Layer-5 | 23/64 | 43/128 | 34/256 | 33/512 | 28/1024 | 26/2048 | 22/4096 |
| Layer-6 | 20/64 | 24/128 | 20/256 | 21/512 | 19/1024 | 18/2048 | 15/4096 |
| Layer-7 | 15/64 | 18/128 | 17/256 | 15/512 | 15/1024 | 14/2048 | 13/4096 |
| Layer-8 | 15/64 | 14/128 | 15/256 | 11/512 | 13/1024 | 13/2048 | 12/4096 |
| Layer-9 | 14/64 | 14/128 | 15/256 | 13/512 | 13/1024 | 12/2048 | 12/4096 |
| Layer-10 | 13/64 | 13/128 | 12/256 | 14/512 | 12/1024 | 13/2048 | 14/4096 |
| Total | 4970 | 5024 | 4994 | 4969 | 4858 | 4390 | 3908 |

Table 4: Final-epoch Effective Ranks for ResNets on CIFAR-10, where for the form A/B, A represents the effective rank and B represents the original dimension, and where Layer-0% means the input layer.

| Metric | ResNet-20 | ResNet-32 | ResNet-44 | ResNet-56 | ResNet-74 | ResNet-110 |
|---|---|---|---|---|---|---|
| Layer-0% | 384/3072 | 384/3072 | 17/3072 | 0/3072 | 0/3072 | 0/3072 |
| Layer-25% | 2048/16384 | 2048/16384 | 7/16384 | 1/16384 | 0/16384 | 0/16384 |
| Layer-50% | 1024/8192 | 1024/8192 | 1024/8192 | 227/8192 | 0/8192 | 0/8192 |
| Layer-75% | 512/4096 | 512/4096 | 512/4096 | 512/4096 | 58/4096 | 0/4096 |
| Layer-100% | 8/64 | 8/64 | 8/64 | 8/64 | 8/64 | 8/64 |
| Total | 23432 | 37768 | 27564 | 16294 | 11401 | 6925 |

# A  FURTHER EXPLANATIONS OF EXPERIMENTAL RESULTS

## A.1  FEATURE LEARNING COMPRESSES EFFECTIVE RANK

We investigate the dynamics of feature learning by monitoring the effective rank of the feature Gram matrices $F_{l-1}F_{l-1}^\top$, with the normalization $\cdot L\|W\|_F^2 \prod_{i>l}\|W_i\|_{\mathrm{op}}^2$ dictated by our theory. Here, replacing the local Lipschitz constant $M_{l\to L}(W, \varepsilon)$ by the spectral-norm product $\prod_{i>l}\|W_i\|_{\mathrm{op}}$ is conservative: state-of-the-art formal-verification toolchains (Shi et al., 2022) can compute local Lipschitz constants much more sharply—with well-developed packages and rigorous numerical guarantees—than this crude product bound, and could therefore further strengthen all our empirical results (an active research area). On the other hand, this relaxation—dropping the $\varepsilon-$dependence when making the conservative substitution—can be justified rigorously (see Appendix F.5.2, especially Step 4 in the proof of Corollary 1), and we adopt this simplification in our experiments. We report our empirical results in Tables 3, 4 and Figure 1.

Experimental results reveal some clear patterns: (1) As training proceeds, the effective ranks of feature grams decreases sharply after a short transient; refer to Figure 1. (2) Increased parameter counts, both under width scaling (FCNs) and depth scaling (ResNets), foster compressing effective ranks of feature grams in both the rate and the degree; refer to Figure 1. (3) On the largest FCN, the degree of effective rank compression can reach as much as $1/300$, which explains why the Riemannian Dimension can achieve such a significant improvement over the VC dimension; refer to Table 3. While on the largest ResNet, the effective ranks of the vast majority of layers compress to zero, which explains why deeper networks can, paradoxically, exhibit a smaller Riemannian Dimension; refer to Table 4. These experimental results indicates that feature learning steadily reduces the intrinsic dimensionality of features over training and aim to learn a lower-dimensional feature manifold, and the overparameterization intensifies this reduction.

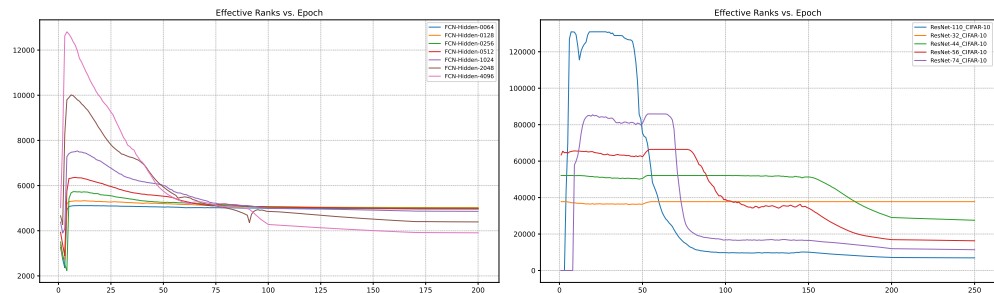

Figure 1: Effective Rank evolutions of FCNs on MNIST (left) and ResNets on CIFAR-10 (right) across the training

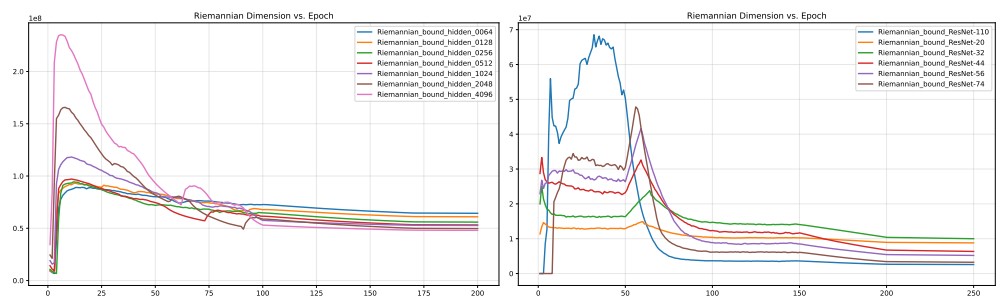

Figure 2: Riemannian Dimension evolutions of FCNs on MNIST (left) and ResNets on CIFAR-10 (right) across the training

## A.2 SGD FINDS LOW RIEMANNIAN DIMENSION POINT

Related literature has shown that various norms are implicit bias of optimizers, but typically limited to linear models (Vardi, 2023). This section studies whether SGD with momentum, in modern deep learning, implicit regularized Riemannian Dimension across training dynamics. We examine whether this optimizer preferentially converge to solutions with lower Riemannian Dimension point, and the experimental results are presented in Figure 2.

Empirical results show a repeatable pattern across the architectures: SGD with momentum drives the networks toward solutions with lower intrinsic Riemannian Dimension complexity, after an early transient; refer to Figure 2. Notably, Riemannian Dimension drops by orders of magnitude, whereas VC dimension remains essentially unchanged. The alignment between optimization dynamics and complexity control supports the view that SGD with momentum implicitly regularizes the Riemannian Dimension. Therefore, optimization is not merely as a mechanism for convergence; it is a primary driver of generalization through its systematic preference for low-complexity solutions. Riemannian Dimension provides a practical and theoretically grounded lens through which the implicit bias of optimizes in machine learning can be quantitatively assessed.

## A.3 EXPERIMENTAL SETUP

We introduce detailed experimental setups. We evaluate our Riemannian Dimension bound on two standard architectures—Fully Connected Networks (FCNs) and ResNets, using two benchmark datasets—MNIST (LeCun et al., 1998) and CIFAR-10 (Krizhevsky, 2009), respectively. The architecture of FCNs: we consider a 9-hidden-layer FCN in which the first two hidden layers have width $2^{11}$ and the remaining seven hidden layers share a common width $h$, with $h \in \{2^6, 2^7, 2^8, 2^9, 2^{10}, 2^{11}, 2^{12}\}$. The output layer is a linear classifier mapping to 10 logits, and we use ReLU as the activation and use PyTorch's default initialization (Kaiming uniform for ReLU). Increasing $h$ monotonically enlarges both layer widths and the total parameter count, yielding a clean capacity sweep at fixed depth. The architecture of ResNets: we adopt the canonical ResNet architectures, ResNet-20, ResNet-32, ResNet-44, ResNet-56, ResNet-74, and ResNet-110, which

differ only in the number of residual blocks per stage while maintaining the same overall architecture (three-stage, basic-block design) as introduced by (He et al., 2016). Following the practice of (He et al., 2016), we apply BatchNorm and ReLU after each convolution, with shortcut connections added as needed, and a global average pooling layer precedes the final linear classifier. These ResNet architectures provides a clean capacity sweep via depth.

We adopt standard training pipelines widely used in the benchmarks. (1) The training Protocol of FCNs is: SGD with momentum optimizer where momentum $= 0.9$, learning rate $= 0.01$, and weight decay $= 5 \times 10^{-4}$; 200 epochs and 128 batch size; a step decay at epochs $\{100, 170\}$, where the learning rate is scaled by $\times 0.1$. (2) The training Protocol of ResNets is: SGD with momentum optimizer where momentum $= 0.9$, learning rate $= 0.1$, and weight decay $= 5 \times 10^{-4}$; 250 epochs and 128 batch size; a step decay at epochs $\{50, 150, 200\}$, where the learning rate is scaled by $\times 0.1$; Following practical training conditions, we apply standard data augmentation on CIFAR-10: random horizontal flips and 4-pixel random crops with zero-padding.

In the experiments of FCNs and ResNets, to enable layerwise analysis of the evolving feature representations and support our computation of Riemannian Dimension, we register forward hooks on all nonlinearity layers. For layers followed by pooling, we replace the last recorded ReLU activation with the corresponding pooled output. We also pre-register the input hook to capture the feature matrix of the data. These hooks ensures precise extraction of nonlinearity activations at each depth throughout training. We set the hyper-parameter $\varepsilon$ via a one–dimensional ternary-search procedure: at the end of each training stage we perform a 500-step ternary search for FCNs and a 50-step ternary search for ResNets over the admissible interval $\left[\sqrt{1/n}, \ \max_{l=1,\ldots,L} \sqrt{\frac{2L\lambda_{\max}(F_{l-1}F_{l-1}^{\top}) \cdot \|W\|_F^2 \prod_{i>l} \|W_i\|_{\text{op}}^2}{n}}\right]$. The search selects the value that minimizes our one-shot Riemannian Dimension-based generalization bound (5). We note that tighter bounds could be achieved with more refined optimization procedures on $\varepsilon$. For FCNs, we compute full feature gram matrices. While for ResNets, the feature matrix $F$ is formed by flattening each activation map into a vector of dimension $d = C \cdot H \cdot W$, where $C, H, W$ are the channel, height, and width of the feature map respectively. To align with our theory, we simplify ResNets to fully connected (feed-forward) networks when computing our bound; we apply the same simplification to the associated VC-dimension and parameter-count calculations to maintain consistency. To avoid out-of-memory in computing full feature gram matrices in high-dimensional convolutional layers, we use the standard Gaussian sketching approximation, where each feature gram matrix uses a Gaussian sketch with parameter $r = \min(8192, \lfloor d/8 \rfloor)$ (Woodruff et al., 2014). By standard subspace-embedding guarantees, such Gaussian sketches preserve Gram quadratic forms—and hence the spectra—of the feature matrices with high probability, introducing only negligible distortion and leaving our conclusions unchanged (Woodruff et al., 2014).

# B   RELATED WORKS

Given the breadth of work on generalization and its empirical proxies, the mathematical grounding of our approach, and its conceptual relevance to vision and language practice, we streamline the exposition by concentrating on the most relevant prior results.

**Theoretical Generalization Bounds for DNN.** A significant lineage of research anchors generalization bounds to various norms of network weights (e.g., path (Neyshabur et al., 2015a), spectral (Neyshabur et al., 2018; Bartlett et al., 2017; Arora et al., 2018), Frobenius (Neyshabur et al., 2015b; Golowich et al., 2020)). While offering conceptual insights, these bounds, often derived from globally uniform complexity measures like covering numbers or Rademacher complexity, frequently suffer from exponential dependencies on depth or layer norms, rendering them vacuous for practical, deep architectures. Compelling empirical evidence (Farhang et al., 2022; Razin & Cohen, 2020) further suggests that norm-based bounds alone are insufficient to fully elucidate the generalization phenomenon in deep learning. The kernel perspective (Belkin et al., 2018), epitomized by NTK theory (Jacot et al., 2018; Arora et al., 2019; Golikov et al., 2022), yields sharp guarantees by linearizing a network around its initialization—effectively casting training as kernel ridge regression with a fixed kernel. Within this linear/lazy regime, precise calculations explain both double descent (Belkin et al., 2019) and benign overfitting (Bartlett et al., 2020), and an eigenspace-projection viewpoint provides dimension-reduction and feature-compression insights (Bartlett et al., 2017). In-

vestigations beyond the lazy regime exist, but most analyses either study the two-layer infinite-width (mean-field) limit (e.g., (Mei et al., 2018; Chizat & Bach, 2018)) or remain in a neighborhood of initialization (Woodworth et al., 2020). While insightful, these settings are idealized and struggle to capture the behavior of finite, deep networks (see Chapter 6 of (Misiakiewicz & Montanari, 2023)). More broadly, linear, lazy, or infinite-width approximations fail to reflect the feature learning that arises when parameters move far from initialization and representations evolve. This omission is widely viewed as a central bottleneck in current theory; indeed, the rich, representation-learning regime is often argued to be the key phenomenon distinguishing modern deep learning from long-standing frameworks (see, e.g., Wilson (2025)). Building on these directions, we establish—to our knowledge—the first pointwise generalization bounds for nonlinear DNN that are comparable in sharpness to prior linearization results and, crucially, remain valid in the practical feature-learning regime.

**Other Theoretical Perspectives of Generalization.** A growing line of work connects generalization to geometric notions of fractal dimension (Birdal et al., 2021; Dupuis et al., 2023; Simsekli et al., 2020; Andreeva et al., 2024; Camuto et al., 2021), typically through Hausdorff– or Minkowski–type dimensions of optimization trajectories or invariant measures. However, these fractal dimensions are globally uniform, infinitesimal-scale ($\varepsilon \to 0$) notions of complexity. In contrast, our theory is built on a pointwise, finite-scale notion of geometric dimension. Section C.1 is precisely devoted to this distinction: we move from globally uniform to pointwise dimension and show that generalization is governed by the finite-scale pointwise dimension rather than its asymptotic limit. Several PAC–Bayesian approaches operate directly in parameter space $W$, endowing the weights with an explicit stochastic model and directly computing the KL divergence between a hand–designed prior and a posterior over $W$ (Hinton & Van Camp, 1993; Dziugaite & Roy, 2017; Lotfi et al., 2022; 2024); e.g., Gaussian distribution in Dziugaite & Roy (2017). These parameter-space bounds are valuable for certifying that certain trained weight configurations admit non-vacuous PAC-Bayes guarantees, but they largely treat the network as a black box and do not directly capture how architecture and feature geometry control generalization. Alternative theoretical frameworks include algorithmic stability analyses, which are used primarily for one-hidden-layer networks and connected to the NTK/lazy-training viewpoint (Richards & Kuzborskij, 2021; Lei et al., 2022); and VC-dimension methods (Bartlett et al., 2019), which has been discussed in Section F.1.

**Pointwise and Non-Perturbative Foundations.** Our use of "pointwise" draws inspiration from several threads that emphasize hypothesis-specific complexity: the asymptotic pointwise dimension in fractal geometry (Falconer, 1997), PAC-Bayes analyses that tailor complexity to the chosen random posterior (McAllester, 1998; Alquier, 2024), and the Fernique–Talagrand integral in the majorizing-measure formulation of generic chaining (Fernique, 1975; Talagrand, 1987; Block et al., 2021). The synthesis of PAC-Bayes bounds with generic chaining dates back to Audibert & Bousquet (2003); Audibert & Bousquert (2007), and mutual information based bounds have also been combined with chaining (Russo & Zou, 2016; Xu & Raginsky, 2017; Asadi et al., 2018; Liu, 2025). To the best of our knowledge, this paper is the first work to establish a sharp pointwise bound for deterministic hypotheses in an uncountable class via localization to metric balls, explicitly connecting the result to pointwise dimension. A generic conversion from classical (subset-homogeneous) uniform convergence to pointwise generalization bounds, established in Xu & Zeevi (2020; 2025), serves as a guiding principle and plays a central role in our proof of Theorem 1. The adjective "non-perturbative," borrowed from physics (nLab contributors, 2025a) and central to the study of strongly correlated systems (nLab contributors, 2025b), underscores that our theory remains valid far beyond infinitesimal neighborhoods of initialization—an essential property for deeply nonlinear, feature-learning DNN.

**Connections to Differential Geometry and Lie Algebra.** From a geometric perspective, Hausdorff dimension provides an asymptotic, covering-based notion of capacity (fundamental in geometric measure theory (Simon, 2018)), while differential and Riemannian geometry (Jost, 2008) develop the use of local charts and global atlases to analyze non-Euclidean manifolds. Our results motivate viewing generalization as a finite-scale problem in geometric analysis. The Grassmannian and families of orthogonal subspaces are traditionally studied via Lie groups; using differential-geometric tools, Szarek (1997); Pajor (1998) established finite-scale isotropic metric-entropy characterizations,

which motivate our hierarchical covering viewpoint from local charts to a global atlas and our ellipsoidal entropy framework.

**Empirical Indicators of Generalization.** Complementing theory, much research has focused on empirical indicators that explain the generalization of deep learning. Phenomena like *Neural Collapse* (Papyan et al., 2020; Parker et al., 2023; Kothapalli, 2022) reveals the emergence of low-rank geometric structures in last-layer features. Studies on *Intrinsic Dimension* (Li et al., 2018; Huh et al., 2021) similarly suggest that deeper models exhibit an inductive bias toward low-rank last-layer feature representations. A line of work focuses on *Dynamic NTK variants* (Atanasov et al., 2021; Baratin et al., 2021; Fort et al., 2020; Kopitkov & Indelman, 2020) or related feature-gradient kernels (Radhakrishnan et al., 2024), where the kernels evolve along optimization trajectories, has empirically shown that the dynamic kernel evolution is linked to generalization behaviour. Other probes, examining Fisher information (Karakida et al., 2019; Jastrzebski et al., 2021), Hessian spectral properties (Ghorbani et al., 2019; Rahaman et al., 2019), and output-input Jacobians (Novak et al., 2018), offer another lens. Collectively, existing empirical probes offer valuable, though often partial, insights—typically from a specific layer perspective, or through a constructed similarity analysis—without a unifying formalism and a theory foundation. Our proposed empirical indicator, rooted in a mathematically sharp theory, resonates with their goals (our theory is in fact supported by many of their experiments) while advancing them. It provides a principled, formal measure for characterizing the generalization of neural networks.

**Feature Compression in Deep Models for Vision and Language.** Across vision and language, deep networks exhibit a robust layer–wise compression of representations. In computer vision, Ansuini et al. (2019) measure intrinsic dimensionality across convolutional layers and find early expansion followed by sharp reduction, with lower late–stage dimensionality correlating with stronger generalization; Feng et al. (2022) likewise show that feature matrices in CNNs and vision transformers become progressively low–rank with depth, at fixed width, indicating active compression of task–relevant information. Parallel trends appear in NLP: Cai et al. (2021) demonstrate that contextual embeddings (e.g., BERT) occupy narrow, anisotropic cones despite high nominal dimension, and Razzhigaev et al. (2024) document a two–phase training trajectory—initial expansion, then sustained compression. A complementary line grounded in the Information Bottleneck (Tishby & Zaslavsky, 2015) interprets these findings as the selective removal of task–irrelevant variability: Shwartz-Ziv & Tishby (2017) observe that networks spend most of training compressing internal features toward a prediction–compression trade–off, while Patel & Shwartz-Ziv (2024) show gradient descent reduces the local rank of intermediate activations. Balzano et al. (2025) provide a complementary tutorial on low-rank structures arising during the training and adaptation of large models, emphasizing how gradient-descent dynamics and implicit regularization generate low-rank representations. Taken together, these phenomena motivate our investigation: compression is not merely qualitative, but admits precise, hypothesis–specific complexity that governs generalization.

## C  FURTHER EXPLANATIONS AND PROOFS FOR POINTWISE GENERALIZATION FRAMEWORK (SECTION 2)

In section C.1, we bring new understandings on the nature of generalization, providing further explanations of Section 2. The rest of this section is mainly devoted to a full proof of Theorem 1 (the integral upper bound). Conceptually, the pointwise–dimension principle already follows from elementary PAC–Bayes arguments—see Lemma 6 and the subsequent remark in Appendix C.3. We present the full derivation to make explicit structural properties (e.g., unified blueprint, subset homogeneity, population–empirical isomorphism) that a rigorous proof requires.

### C.1  NECESSITY OF FINITE-SCALE POINTWISE GEOMETRY AND STRUCTURAL ANALYSIS

The transition from uniform convergence to the "prior-density + localization" (pointwise dimension) perspective offers a fundamental tightening over standard covering number approaches. However, translating this theoretical advantage into a practical framework for deep learning requires addressing two distinct challenges. First, we will distinguish the *geometric nature* of generalization from classical infinitesimal geometry: relevance lies not in the limit $\varepsilon \to 0$, but motivates a new program

of finite-scale geometric analysis. Second, we must overcome the *computational intractability* of evaluating the pointwise dimension directly, which necessitates a dedicated structural analysis for deep neural networks.

**Asymptotic vs. Finite-Scale Dimension.** Although powerful in mathematics, standard differential-geometric tools (e.g., pointwise metrics and subspace angles) have not been systematically used in generalization theory, largely because they define dimension in infinitesimal notions. For instance, the *asymptotic pointwise dimension*—central to fractal and Riemannian geometry (Falconer, 1997; Jost, 2008) and used to characterize Hausdorff and packing dimensions (e.g., Theorem 3 of Lutz (2016))—is defined via a limit:

$$\lim_{\varepsilon \to 0} \frac{\log \pi(B_\varrho(f, \varepsilon))}{\log \varepsilon}.$$

We argue that generalization is distinct from, and in some ways more challenging than, infinitesimal geometry: the nature of generalization in deep models lies in reducing geometric dimension at a *finite scale* of precision for each hypothesis. Crucially, the pointwise dimension $\log \frac{1}{\pi(B_\varrho(f, \varepsilon))}$ is monotonic: it naturally decreases as the resolution $\varepsilon$ increases. Therefore, a finite-scale analysis reveals significant dimension reduction that infinitesimal analysis misses. In our one-shot bound (5), the objective is to identify the optimal finite scale $\varepsilon^\star$ where the trade-off between precision and pointwise complexity is minimized. At this scale, the effective dimension can be orders of magnitude smaller than the asymptotic dimension, explaining the tractability of overparameterized models. To the best of our knowledge, this distinction is novel; prior uses of geometric dimension in generalization (e.g., Birdal et al. (2021)) have largely emphasized globally uniform and infinitesimal notions. And the Neural Tangent Kernel (NTK) (Jacot et al., 2018) and Gaussian-process (Lee et al., 2018) viewpoints are valid only in an infinitesimal neighborhood of initialization (equivalently, in the infinite-width regime). A precise account of deep-model generalization thus calls for a shift from infinitesimal calculus to finite-scale, pointwise geometry.

**Computation and the Necessity of Structural Analysis.** Although tight, Theorem 1—like many abstract bounds (PAC-Bayes, mutual-information, generic chaining)—is generally not computationally tractable on its own; practical use requires adapting it to the function class at hand and introducing suitable relaxations. If we denote an effective dimension by $d(f) = \log \frac{1}{\pi(B_{\varrho_{n,\ell}}(f, \varepsilon^\star))}$ ($\varepsilon^\star$ tuned in the one-shot bound (5)), a brute-force Monte Carlo estimator using i.i.d. draws $f' \sim \pi$ would require on the order of $e^{d(f)}$ samples to obtain a single hit $f' \in B_{\varrho_{n,\ell}}(f, \varepsilon)$ with constant probability. For high-dimensional deep networks, where $d(f)$ should be moderate to large, this is computationally prohibitive.

This intractability helps explain why much of the PAC–Bayes literature pivoted to *inherent linearization* via closed-form calculations under Gaussian priors/posteriors on the parameter space: by relaxing the search over general posteriors on the nonlinear hypothesis class $\mathcal{F}$ to Gaussians over weights $W \in \mathbb{R}^p$, one obtains tractable objectives (Hinton & Van Camp, 1993; Dziugaite & Roy, 2017); see Sections 3 and 6 in Dziugaite & Roy (2017) for these objectives. However, this strategy implicitly imposes a uniform linearization that discards the distinctive *pointwise* geometry of deep networks, effectively flattening a curved manifold. To retain the sharpness of pointwise dimension without incurring the simulation barrier, we therefore avoid black-box sampling and instead develop explicit *structural principles* of deep networks that allow analytic control of the pointwise dimension—yielding generalization guarantees that are both theoretically rigorous and practically computable.

## C.2 THE "UNIFORM POINTWISE CONVERGENCE" PRINCIPLE

In this section, we present a unified blueprint for establishing pointwise generalization bounds. We state necessary and sufficient conditions for pointwise generalization and show that, when applied carefully, the resulting pointwise bounds are no harder to obtain than classical uniform-convergence guarantees.

We begin by citing a general principle for converting *subset-homogeneous* uniform convergence guarantees—i.e., bounds in which the same pointwise complexity applies for every fixed subset $\mathcal{H} \subseteq \mathcal{F}$—into pointwise generalization bounds. This conversion, introduced by the name "uniform

localized convergence" principle in (Xu & Zeevi, 2020) (short conference version) and Xu & Zeevi (2025) (full journal version), provides a direct mechanism for obtaining the type of pointwise generalization bounds central to our work. We state this result as "uniform pointwise convergence" principle.

**Lemma 4 ("Uniform Pointwise Convergence" Principle) (Proposition 1 in Xu & Zeevi (2020; 2025)).** *For a function class $\mathcal{F}$ and functional $d : \mathcal{F} \to [0, R]$, assume there is a function $\psi(r; \delta)$, which is non-decreasing with respect to $r$, non-increasing with respect to $\delta$, and satisfies that $\forall \delta \in (0, 1)$, $\forall r \in [0, R]$, with probability at least $1 - \delta$,*

$$\sup_{f \in \mathcal{F} : d(f) \leq r} (\mathbb{P} - \mathbb{P}_n) \ell(f; z) \leq \psi(r; \delta). \tag{17}$$

*Then, given any $\delta \in (0, 1)$ and $r_0 \in (0, R]$, with probability at least $1 - \delta$, uniformly over all $f \in \mathcal{F}$,*

$$(\mathbb{P} - \mathbb{P}_n) \ell(f; z) \leq \psi \left( \max\{2d(f), r_0\}; \delta \left( \log_2 \frac{2R}{r_0} \right)^{-1} \right). \tag{18}$$

This lemma provides a succinct proof that serves as a unifying principle to sharpen classical localization, building on Section 2 of Xu & Zeevi (2025). A key advantage of this framework is its level of abstraction: it establishes subset homogeneity as the necessary and sufficient condition for pointwise generalization when the complexity functional $d(\cdot)$ is data–independent, and likewise when $d(\cdot)$ is swap–invariant and depends on both the observed sample $S = \{z_i\}_{i=1}^n$ and an i.i.d. ghost sample $S' = \{z_i'\}_{i=1}^n$. It also provides a clean treatment of data–dependent functionals and their induced (random) sublevel sets $\{f \in \mathcal{F} : d(f) \leq r)$, as outlined before Section 4 of Xu & Zeevi (2025). Crucially, this approach circumvents the circularities that often arise when combining symmetrization with localization or offset arguments.

### C.2.1 NECESSARY AND SUFFICIENT CONDITIONS FOR POINTWISE GENERALIZATION

We leverage this "uniform pointwise convergence" principle to streamline the derivation of our bounds. Let $d(\cdot)$ denote a pointwise complexity functional, which we categorize into data-independent forms and data-dependent forms. Let $\psi(\cdot; \delta)$ be a non-decreasing function (typically $\psi(r; \delta) \asymp \sqrt{(r + \log(1/\delta))/n}$). We provide a clean characterization of pointwise generalization.

**Necessary Condition: Subset Homogeneity.** A valid pointwise generalization guarantee (i.e., (3)) necessitates *subset homogeneity*. That is, if the pointwise inequality

$$(\mathbb{P} - \mathbb{P}_n) \ell(f; z) \leq \psi(d(f); \delta) \tag{19}$$

holds with probability at least $1 - \delta$, then (19) must imply that for every fixed (i.e., data-independent) subset $\mathcal{H} \subseteq \mathcal{F}$,

$$\sup_{f \in \mathcal{H}} (\mathbb{P} - \mathbb{P}_n) \ell(f; z) \leq \sup_{f \in \mathcal{H}} \psi(d(f); \delta).$$

Crucially, the complexity evaluation $d(f)$ must *not* depend on the chosen subset $\mathcal{H}$. For instance, for the pointwise dimension $\log \frac{1}{\pi(B_\varrho(f, \varepsilon))}$, the prior $\pi$ (in particular, its support) should be independent of $\mathcal{H}$. This contrasts with classical empirical-process techniques—e.g., naive uses of Rademacher complexity and generic chaining—where the *pre-specified* index sets dictate the proxy $d(\cdot)$ via the chosen Rademacher expectation, admissible tree construction, or prior.

Subset homogeneity is thus the primary eligibility check for any candidate pointwise complexity functional. In Appendix C.4, we complete this check by establishing that the pointwise dimension is *ambiently equivalent*: using a prior $\pi \in \Delta(\mathcal{F})$ or its restriction $\pi \in \Delta(\mathcal{H})$ produces complexities that agree in order (up to absolute constants).

**Sufficient Condition: Subset Homogeneity + Data-Independent (or Symmetrized) $d(\cdot)$.** Assuming the following subset-homogeneity uniform convergence condition: for every fixed (i.e., data-independent) subset $\mathcal{H} \subseteq \mathcal{F}$ and $\delta \in (0, 1)$, with probability at least $1 - \delta$,

$$\sup_{f \in \mathcal{H}} (\mathbb{P} - \mathbb{P}_n) \ell(f; z) \leq \sup_{f \in \mathcal{H}} \psi(d(f); \delta). \tag{20}$$

By taking the sublevel set

$$\mathcal{H} = \{f \in \mathcal{F} : d(f) \leq r\},$$

the condition (20) (applied to this fixed sublevel set) directly implies the surrogate conditions (17) in Lemma 4, and hence the pointwise bound (18). Thus, subset homogeneity is a necessary and sufficient condition for a data-independent $d(\cdot)$ to imply a pointwise generalization bound.

Likewise, in Appendix C.5, we show that when the complexity $d(\cdot)$ may depend on both the observed sample $S = \{z_i\}_{i=1}^n$ and an i.i.d. ghost sample $S' = \{z_i'\}_{i=1}^n$, *provided it is swap–invariant in* $(S, S')$ *(i.e., invariant under any exchange* $z_i \leftrightarrow z_i'$*)*, subset homogeneity suffices to yield a pointwise generalization bound via a final swap–symmetrization argument. This establishes Theorem 5: a pointwise generalization result in which the complexity is evaluated using both the observed sample $S$ and the ghost sample $S'$.

**Toward pointwise bounds using only observed sample.** If one seeks bounds that are fully computable from the observed data $\{z_i\}_{i=1}^n$ alone (e.g., Theorem 1), without ghost sample or sample splitting, the analysis is more involved. A practical route is two–step: (i) first derive a symmetrized pointwise bound using a complexity functional based on $(S, S')$ (which is already valid and sharp); (ii) then prove an isomorphism between the $L_2(\mathbb{P}_S)$– and $L_2(\mathbb{P}_{S'})$–induced pointwise complexities (following Appendix A.4 of Xu & Zeevi (2025)) so as to replace population or ghost–dependent terms by empirical ones, yielding a fully data–dependent bound. Theorem 1 will be fully proved in Appendix C.6.

## C.3 THE PAC-BAYES OPTIMIZATION PROBLEM

We illustrate why pointwise dimension is a natural consequence of *best* PAC-Bayes optimization.

**Lemma 5 (PAC–Bayes Bound (Catoni, 2003); see also Theorem 2.1 in Alquier (2024))** *Let* $\pi$ *be a prior on a hypothesis class $\mathcal{F}$ independent to the data, and let $\ell \colon \mathcal{F} \times \mathcal{Z} \to [0, 1]$ be a bounded loss. Fix confidence $\delta \in (0, 1)$ and sample size $n$. Then for every $\eta > 0$, with probability at least $1 - \delta$ over $n$ i.i.d. draws $z_1, \ldots, z_n \sim \mathbb{P}$, for* every *distribution $\mu$ on $\mathcal{F}$ simultaneously,*

$$(\mathbb{P} - \mathbb{P}_n)\langle \mu, \ell(f; z)\rangle \leq \inf_{\eta > 0}\left\{\frac{\mathrm{KL}(\mu, \pi) + \log\frac{1}{\delta}}{\eta n} + \frac{\eta}{8}\right\} = \sqrt{\frac{\mathrm{KL}(\mu, \pi) + \log\frac{1}{\delta}}{8n}}.$$

We now use the PAC-Bayes bound (which holds uniformly for every random posterior $\mu$) to approximate a deterministic hypothesis $f$. On the event that the above PAC-Bayes bound holds, with probability at least $1 - \delta$, we have that uniformly over every random $\mu \in \Delta(\mathcal{F})$ every deterministic $f \in \mathcal{F}$, for every $\eta > 0$, the following uniform "deterministic hypothesis" bound holds:

$$
\begin{aligned}
&(\mathbb{P} - \mathbb{P}_n)\ell(f; z) \\
={}&\langle \mu, (\mathbb{P} - \mathbb{P}_n)\ell(\cdot; z)\rangle + \langle \mu, (\mathbb{P}_n - \mathbb{P})[\ell(\cdot; z) - \ell(f; z)]\rangle \\
\leq{}&\frac{\eta}{8} + \frac{\mathrm{KL}(\mu, \pi) + \log\frac{1}{\delta}}{\eta n} + \langle \mu, \frac{1}{n}\sum_{i=1}^n |\ell(\cdot; z) - \ell(f; z)|\rangle + \langle \mu, \mathbb{E}|\ell(\cdot; z) - \ell(f; z)|\rangle \\
={}&\frac{\eta}{8} + \frac{\mathrm{KL}(\mu, \pi) + \log\frac{1}{\delta}}{\eta n} + \langle \mu, \tilde{\varrho}(\cdot, f)\rangle,
\end{aligned}
\tag{21}
$$

where the metric $\tilde{\varrho}$ is defined as the sum of loss-induced $L_1(\mathbb{P}_n)$ metric and $L_1(\mathbb{P})$ metric:

$$\tilde{\varrho}(f', f) = \frac{1}{n}\sum_{i=1}^n |\ell(f'; z) - \ell(f; z)| + \mathbb{E}|\ell(f'; z) - \ell(f; z)|. \tag{22}$$

In (21), the inequality uses the PAC-Bayes bound (Lemma 5) to bound the first term, which we term the "variance" term, and use absolute values to bound the second term, which we term the "bias" term.

Motivated by the above bias-variance optimization (21) via PAC-Bayes, for a given prior $\pi$, metric $\varrho$, and confidence $\delta \in (0, 1)$ we define the *PAC-Bayes optimization objective*

$$V(\mu, \eta, f, \varrho) := \underbrace{\frac{\eta}{8} + \frac{\mathrm{KL}(\mu, \pi) + \log\frac{1}{\delta}}{\eta n}}_{\text{Variance}} + \underbrace{\langle \mu, \varrho(\cdot, f) \rangle}_{\text{Bias}}, \tag{23}$$

where $\eta > 0$, $n$ is the sample size, $\mu$ is a posterior over hypotheses. Here, the "Variance" term arises from a PAC-Bayes bound (Lemma 5) applied to $\mu$, and the "Bias" term $\langle \mu, \varrho(\cdot, f) \rangle :=$ $\mathbb{E}_{h \sim \mu}\big[\varrho(h, f)\big]$ measures how well the randomized $\mu$ approximates the target $f$.

**Optimizing the Posterior $\mu$ for the Objective** (23)   The intuitive analysis (21) explains how the PAC-Bayesian optimization objective naturally bounds the generalization gap. We now minimize the posterior $\mu$ in (23). It is straightforward that (23) is minimized by the Gibbs posterior. To obtain a closed-form characterization of the optimized value, we proceed in two steps: (i) derive an explicit pointwise-dimension upper bound by taking $\mu$ to be the $\pi$-normalized distribution on the metric ball $B_\varrho(f, \varepsilon)$ (Lemma 6), and (ii) show that this choice is near-optimal (Lemma 7).

### C.3.1   POINTWISE DIMENSION BOUND VIA METRIC BALL

Given any prior $\pi$ on $\mathcal{F}$ and any $f \in \mathcal{F}$, take $\mu$ to be the $\pi-$normalized distribution on the metric ball $B_\varrho(f, \varepsilon)$, i.e.,

$$\mu(A) = \frac{\pi(A \cap B_\varrho(f, \varepsilon))}{\pi(B_\varrho(f, \varepsilon))} \quad \text{for all measurable } A \subseteq \mathcal{F}. \tag{24}$$

This simple choice is essentially optimal in that it yields the same analytical upper bound as the Gibbs posterior that minimizes the bound (later presented in Lemma 7).

**Lemma 6 (Pointwise Dimension and Pointwise Generalization Upper Bound)** *For          the PAC–Bayes objective* (23)*, let $\mu$ be $\pi-$normalized on $B_\varrho(f, \varepsilon)$, i.e.*

$$\frac{d\mu}{d\pi}(h) = \begin{cases} \dfrac{1}{\pi\big(B_\varrho(f, \varepsilon)\big)}, & h \in B_\varrho(f, \varepsilon), \\ 0, & h \notin B_\varrho(f, \varepsilon). \end{cases}$$

*Then, with $\eta^\star = \sqrt{8\big(\mathrm{KL}(\mu, \pi) + \log(1/\delta)\big)/n}$,*

$$V\big(\mu, \eta^\star, f, \varrho\big) \le \sqrt{\frac{\mathrm{KL}(\mu, \pi) + \log(1/\delta)}{2n}} + \varepsilon = \sqrt{\frac{\log\frac{1}{\pi(B_\varrho(f, \varepsilon))} + \log(1/\delta)}{2n}} + \varepsilon. \tag{25}$$

*Combining the upper bound* (25) *with* (21) *yields the pointwise generalization bound: for every $\delta \in (0, 1)$, with probability at least $1 - \delta$, uniformly over every $f \in \mathcal{F}$,*

$$(\mathbb{P} - \mathbb{P}_n)\ell(f; z) \le \inf_{\varepsilon > 0} \left\{ \sqrt{\frac{\log\frac{1}{\pi(B_{\tilde{\varrho}}(f, \varepsilon))} + \log(1/\delta)}{2n}} + \varepsilon \right\},$$

*where $\tilde{\varrho}$ is the mixed $L_1(\mathbb{P}_n) + L_1(\mathbb{P})$ metric defined by $\tilde{\varrho}(f', f) = \frac{1}{n}\sum_{i=1}^n |\ell(f'; z) - \ell(f; z)| +$ $\mathbb{E}|\ell(f'; z) - \ell(f; z)|$.*

**Remark (why this intuition matters).**   Since the mixed $L_2-$metric dominates the sum of empirical and population $L_1-$metrics, consider

$$\bar{\varrho}(f', f) := \left( \frac{1}{n}\sum_{i=1}^n \big(\ell(f'; z_i) - \ell(f; z_i)\big)^2 + \mathbb{E}\big[\big(\ell(f'; Z) - \ell(f; Z)\big)^2\big] \right)^{1/2}. \tag{26}$$

By Lemma 19, pointwise dimension is monotone in the underlying metric; hence replacing $\tilde{\varrho}$ by the larger metric $\sqrt{2}\bar{\varrho}$ yields a valid pointwise generalization bound. For a trained predictor $\hat{f}$, this

means we may estimate the bound using the observed sample $S = \{z_i\}_{i=1}^n$ together with an i.i.d. ghost sample $S' = \{z_i'\}_{i=1}^n$ to evaluate balls in the mixed metric (26).

Theorem 1 then *sharpens* this picture: it turns the one–shot PAC–Bayes bound into a chaining–integral and removes the need for the ghost sample by working solely with the empirical $L_2(\mathbb{P}_n)$–metric. The core spirit of Theorem 1 remains the same with the PAC–Bayes bias–variance optimization; the practical differences are (i) integral vs. one–shot control and (ii) using $S$ alone instead of $(S, S')$.

**Proof of Lemma 6:** For the choice (24),

$$\mathrm{KL}(\mu, \pi) = \int_{\mathcal{F}} \log\Big(\frac{d\mu}{d\pi}(h)\Big)\, \mu(dh) = \int_{B_\varrho(f,\varepsilon)} \log\Big(\frac{1}{\pi(B_\varrho(f,\varepsilon))}\Big)\, \mu(dh) = \log\frac{1}{\pi(B_\varrho(f,\varepsilon))}. \quad (27)$$

Moreover, by construction,

$$\langle \mu, \varrho(\cdot, f)\rangle = \int_{B_\varrho(f,\varepsilon)} \varrho(h, f)\, \mu(dh) \le \varepsilon.$$

Plugging (27) into (23) and minimizing $\frac{\eta}{8} + \frac{\mathrm{KL}(\mu,\pi) + \log(1/\delta)}{\eta n}$ over $\eta > 0$ gives $\sqrt{(\mathrm{KL}(\mu,\pi) + \log(1/\delta))/(2n)}$, which together with the bias bound $\langle \mu, \varrho(\cdot, f)\rangle \le \varepsilon$ yields the claimed bound (25).

$\square$

### C.3.2 LOWER BOUND AND OPTIMALITY OF PAC-BAYES OPTIMIZATION

The following lemma indicates that the uniform-ball posterior is optimal up to the min–max gap: the lower bound $\min\{a, \varepsilon\}$ and the upper bound $\max\{a, \varepsilon\}$ bracket the optimum, coincide when $a = \varepsilon$, and have the same order whenever $a$ and $\varepsilon$ are comparable.

**Lemma 7 (Optimality of Pointwise Dimension in PAC-Bayes Optimization)** *For the PAC–Bayes optimization objective $V(\mu, \eta, f, \varrho)$ defined in* (23)*, we have that for every $f \in \mathcal{F}$, $\eta > 0$, and $\varepsilon > 0$,*

$$\inf_\mu V(\mu, \eta, f, \varrho) \ge \frac{\eta}{8} + \frac{\log\frac{1}{\delta}}{\eta n} + \min\Big\{\frac{1}{\eta n} \log \frac{1}{\pi(B_\varrho(f,\varepsilon))}, \varepsilon\Big\} - \frac{\log 2}{\eta n}. \quad (28)$$

*Consequently, for every $f \in \mathcal{F}$, $\eta > 0$, and $\varepsilon > 0$,*

$$\frac{\eta}{8} + \frac{\log\frac{1}{\delta}}{\eta n} + \min\Big\{\frac{\log\frac{1}{\pi(B_\varrho(f,\varepsilon))}}{\eta n}, \varepsilon\Big\} - \frac{\log 2}{\eta n} \le \inf_\mu V(\mu, \eta, f, \varrho) \le \frac{\eta}{8} + \frac{\log\frac{1}{\pi(B_\varrho(f,\varepsilon))} + \log\frac{1}{\delta}}{\eta n} + \varepsilon. \quad (29)$$

**Proof of Lemma 7** The upper bound in (29) is already proved in Lemma 6, so we only need to prove the lower bound (28). The Donsker–Varadhan variational identity states that for any measurable $h$,

$$-\log \int e^h\, d\pi = \inf_\mu \Big\{\mathrm{KL}(\mu, \pi) - \int h\, d\mu\Big\}.$$

Apply it with $h = -\eta n \varrho(\cdot, f)$ to obtain

$$-\log \int e^{-\eta n \varrho(\cdot, f)}\, d\pi = \inf_\mu \Big\{\mathrm{KL}(\mu, \pi) + \int \eta n \varrho(\cdot, f)\, d\mu\Big\},$$

which implies that

$$\frac{\eta}{8} + \frac{\log\frac{1}{\delta}}{\eta n} - \frac{1}{\eta n} \log \int e^{-\eta n \varrho(\cdot, f)}\, d\pi = \inf_\mu \Big\{\frac{\eta}{8} + \frac{\mathrm{KL}(\mu, \pi) + \log\frac{1}{\delta}}{\eta n} + \langle \mu, \varrho(\cdot, f)\rangle\Big\}. \quad (30)$$

By splitting the dual integral,

$$\int e^{-\eta n \varrho(\cdot, f)}\, d\pi = \int_{B_\varrho(f,\varepsilon)} e^{-\eta n \varrho(\cdot, f)}\, d\pi + \int_{B_\varrho(f,\varepsilon)^c} e^{-\eta n \varrho(\cdot, f)}\, d\pi$$

$$\le \pi(B_\varrho(f,\varepsilon)) + e^{-\eta n \varepsilon}(1 - \pi(B_\varrho(f,\varepsilon)))$$

$$\le \pi(B_\varrho(f,\varepsilon)) + e^{-\eta n \varepsilon},$$

where $B_\varrho(f, \varepsilon)^c$ is complement of $B_\varrho(f, \varepsilon)$; and we have used $e^{-\eta n \varrho(\cdot, f)} \leq 1$ on $B_\varrho(f, \varepsilon)$ and $e^{-\eta n \varrho(\cdot, f)} \leq e^{-\eta n \varepsilon}$ on $B_\varrho(f, \varepsilon)^c$. Hence

$$\inf_\mu V(\mu, \eta, f, \varrho) \geq \frac{\eta}{8} + \frac{\log \frac{1}{\delta}}{\eta n} - \frac{1}{\eta n} \log \Big( \pi(B_\varrho(f, \varepsilon)) + e^{-\eta n \varepsilon} \Big). \tag{31}$$

The simplified form (28) follows from $a + b \leq 2 \max\{a, b\}$ or equivalently $-\log(a + b) \geq -\log 2 + \min\{-\log a, -\log b\}$ on (31). Combining (23), (27) and (28) yields the sandwich (29).

$\square$

## C.4 Subset Homogeneity of Pointwise Dimension

We show that, for any $f \in \mathcal{H} \subseteq \mathcal{F}$, the pointwise–dimension functional defined with a prior $\pi$ is unchanged in order (up to absolute constants) whether $\pi$ is supported on $\mathcal{H}$ or on the ambient class $\mathcal{F}$. Hence one may take $\pi \in \Delta(\mathcal{F})$ without restricting it to any particular subset, which suffices to meet the subset–homogeneity condition in Appendix 2.

**Lemma 8 (Ambient Equivalence of Pointwise Dimension)** *Let $(\mathcal{F}, \varrho)$ be a metric space and let $\mathcal{H} \subseteq \mathcal{F}$ be a subset. Consider a nearest-point selector $p : \mathcal{F} \to f$ satisfying $\varrho(f, p(f)) = \min_{h \in \mathcal{H}} \varrho(f, h)$ for all $f \in \mathcal{F}$, and the pushforward measure induced by the nearest-point selector:*

$$\pi_\mathcal{H}(h) := \int \pi(f) \mathbb{1}\{p(f) = h\} df.$$

*Then for every $\varepsilon > 0$ we have*

$$\pi_\mathcal{H}(B_\varrho(f, 2\varepsilon)) \geq \pi(B_\varrho(f, \varepsilon)), \quad \log \frac{1}{\pi_\mathcal{H}(B_\varrho(f, 2\varepsilon))} \leq \log \frac{1}{\pi(B_\varrho(f, \varepsilon))}.$$

*Consequently, for $a > 0$, $b > 0$, $\mu \in \Delta(\mathcal{F})$, $f \in \mathcal{H}$, define the majorizing measure integral*

$$I(\pi, f, \varrho, r) := \inf_{0 \leq \alpha \leq \sqrt{r}} \left\{ \alpha + \frac{1}{\sqrt{n}} \int_\alpha^{\sqrt{r}} \sqrt{\log \frac{1}{\pi(B_\varrho(f, \varepsilon))}} \, d\varepsilon \right\}$$

*Then we have*

$$\frac{1}{2} \inf_{\mu \in \Delta(\mathcal{H})} \sup_{f \in \mathcal{H}} I(\mu, f, \varrho, 4r) \leq \inf_{\pi \in \Delta(\mathcal{F})} \sup_{f \in \mathcal{H}} I(\pi, f, \varrho, r) \leq \inf_{\mu \in \Delta(\mathcal{H})} \sup_{f \in \mathcal{H}} I(\mu, f, \varrho, r). \tag{32}$$

**Proof of Lemma 8:**   The upper bound in (32) is immediate since $\Delta(\mathcal{H}) \subset \Delta(\mathcal{F})$: taking $\mu$ supported on $\mathcal{H}$ gives $\inf_{\pi \in \Delta(\mathcal{F})} \sup_{f \in \mathcal{H}} I(\pi, f, \varrho, r) \leq \inf_{\mu \in \Delta(\mathcal{H})} \sup_{f \in \mathcal{H}} I(\mu, f, \varrho, r)$.

For the lower bound in (32), take $\pi_\mathcal{H}$ to be the pushforward induced by the nearest-point selector. For any $f \in \mathcal{H}$ and $\varepsilon > 0$, if $f' \in B_\varrho(f, \varepsilon)$ then

$$\varrho\big(p(f'), f\big) \leq \varrho\big(p(f'), f'\big) + \varrho(f', f) = \min_{f \in \mathcal{H}} \varrho(f', f) + \varrho(f', f) \leq 2\varepsilon,$$

hence $p(f') \in B_\varrho(f, 2\varepsilon)$ and

$$\pi_\mathcal{H}\big(B_\varrho(f, 2\varepsilon)\big) \geq \pi\big(B_\varrho(f, \varepsilon)\big), \quad \log \frac{1}{\pi_\mathcal{H}(B_\varrho(f, 2\varepsilon))} \leq \log \frac{1}{\pi(B_\varrho(f, \varepsilon))}. \tag{33}$$

Therefore,

$$I(\pi, f, \varrho, r) = \inf_{0 \leq \alpha \leq \sqrt{r}} \left\{ \alpha + \frac{1}{\sqrt{n}} \int_\alpha^{\sqrt{r}} \sqrt{\log \frac{1}{\pi(B_\varrho(f, \varepsilon))}} \, d\varepsilon \right\}$$

$$\geq \inf_{0 \leq \alpha \leq \sqrt{r}} \left\{ \alpha + \frac{1}{\sqrt{n}} \int_\alpha^{\sqrt{r}} \sqrt{\log \frac{1}{\pi_\mathcal{H}(B_\varrho(f, 2\varepsilon))}} \, d\varepsilon \right\}$$

$$= \frac{1}{2} \inf_{0 \leq \alpha \leq \sqrt{r}} \left\{ 2\alpha + \frac{1}{\sqrt{n}} \int_{2\alpha}^{2\sqrt{r}} \sqrt{\log \frac{1}{\pi_\mathcal{H}(B_\varrho(f, \varepsilon))}} \, d\varepsilon \right\}$$

$$= \frac{1}{2} I(\pi_\mathcal{H}, f, \varrho, 4r),$$

where the first inequality is by (33); the second equality is by the change of variables. Taking $\sup_{f \in \mathcal{H}}$ and then $\inf_{\pi \in \Delta(\mathcal{F})}, \inf_{\mu \in \Delta(\mathcal{H})}$ yields the desired lower bound.

$\square$

**Relationship to Fractional Covering Number**    Additionally, note that the minimax quantity

$$\mathrm{N}'(\mathcal{H}, \varrho, \varepsilon) := \inf_{\pi \in \Delta(\mathcal{F})} \sup_{f \in \mathcal{H}} \frac{1}{\pi\big(B_\varrho(f, \varepsilon)\big)}$$

is the *fractional covering number*; see Section 3 of Block et al. (2021) for its role in chaining; see also Chen et al. (2024) for connections to information-theoretic lower bounds (e.g., Fano's method, the Yang–Barron method, and local packing). In particular, with $\mathrm{N}(\mathcal{H}, \varrho, \varepsilon)$ denoting the (internal) covering number from Definition 5, we have the order equivalence (Lemma 8 in Block et al. (2021); Lemma 14 in Chen et al. (2024))

$$\log \mathrm{N}(\mathcal{H}, \varrho, 2\varepsilon) \ \leq \ \log \mathrm{N}'(\mathcal{H}, \varrho, \varepsilon) = \inf_{\pi \in \Delta(\mathcal{F})} \sup_{f \in \mathcal{H}} \log \frac{1}{\pi\big(B_\varrho(f, \varepsilon)\big)} \ \leq \ \log \mathrm{N}(\mathcal{H}, \varrho, \varepsilon). \quad (34)$$

The covering number in Definition 5 does not depend on the ambient set $\mathcal{F}$, which in turn suggests that the pointwise dimension enjoys favorable ambient–equivalence properties.

**Collapsing the Distinction between Chaining and Generic Chaining.**    A simple illustration of the strength of our pointwise blueprint is the multi–dimensional setting. Let $(d^{(1)}, \ldots, d^{(k)}) : \mathcal{F} \to (0, R]^k$ be coordinatewise nondecreasing complexities and let $\psi(\cdot; \delta)$ be monotone. Our blueprint makes no essential distinction between the two uniform forms

$$\text{(sup–inside)} \quad \sup_{f \in \mathcal{H}} (\mathbb{P} - \mathbb{P}_n)\, \ell(f; z) \ \leq \ \psi\Big( \sup_{f \in \mathcal{H}} d_n^{(1)}(f), \ldots, \sup_{f \in \mathcal{H}} d_n^{(k)}(f); \ \delta \Big),$$

$$\text{(sup–outside)} \quad \sup_{f \in \mathcal{H}} (\mathbb{P} - \mathbb{P}_n)\, \ell(f; z) \ \leq \ \sup_{f \in \mathcal{H}} \psi\Big( d_n^{(1)}(f), \ldots, d_n^{(k)}(f); \ \delta \Big),$$

in the sense that *either* one leads to the same pointwise conclusion after peeling.

More precisely, fix a base scale $r_0 \in (0, R]$. Then with probability at least $1 - \delta$, for every $f \in \mathcal{F}$,

$$(\mathbb{P} - \mathbb{P}_n)\, \ell(f; z) \ \leq \ \psi\left( \Big( \cdots, \max\big\{ 2d^{(j)}(f), r_0 \big\}, \cdots \Big); \ \delta\Big( \log_2 \frac{2R}{r_0} \Big)^{-k} \right). \quad (35)$$

The most straightforward proof uses essentially the same peeling argument as in Lemma 4, with the only change that we use a grid of size $(\log_2(2R/r_0))^k$ (partition each coordinate into $\log_2(2R/r_0)$ dyadic scales); see the short proof of Proposition 1 in Xu & Zeevi (2025). Alternatively, this can proved by applying Lemma 4 for $k$ times, where at each step we remove one dimension functional and divided confidence by $\log_2(2R/r_0)$. Moreover, the multi–dimensional pointwise bound (35) shows that its right–hand side, viewed as a *scalar* complexity, yields an equally tight pointwise bound. Hence the multi–dimensional formulation does not improve the best-achievable rates beyond a suitably defined one–dimensional complexity (as in generic chaining).

Conceptually, this shows that the apparent gap between classical chaining (entropy integral; sup–inside), generic chaining (majorizing measures; sup–outside), and our pointwise generic–chaining bound (Theorem 1) disappears within the blueprint: each is just a subset–homogeneous uniform statement that implies the same pointwise bound up to absolute constants and minor logarithms.

### C.5    Pointwise Generalization Bound via Ghost Sample

In this section, we prove an easier variant of Theorem 1 that permits *swap-invariant* randomized priors depending on both the observed sample and its ghost counterpart. This setting subsumes—and strengthens—the conditional mutual information (CMI) framework of Steinke & Zakynthinou (2020).

Let $S = (z_1, \ldots, z_n)$ and $S' = (z_1', \ldots, z_n')$ be two i.i.d. samples drawn from $\mathbb{P}^n$, independent of each other. For each index $i \in \{1, \ldots, n\}$, define the coordinate–swap map

$$\tau_i(S, S') := \big( (z_1, \ldots, z_{i-1}, z_i', z_{i+1}, \ldots, z_n), (z_1', \ldots, z_{i-1}', z_i, z_{i+1}', \ldots, z_n') \big).$$

A randomized, data-dependent prior is a mapping $\pi_{(\cdot,\cdot)} : \mathcal{Z}^{2n} \to \Delta(\mathcal{F})$; we write $\pi_{(S,S')} \in \Delta(\mathcal{F})$ for the realized prior over $\mathcal{F}$ (a distribution on $\mathcal{F}$ that may depend on $(S, S')$). We say that $\pi$ is *swap–invariant* on $(S, S')$, if

$$\pi_{(S,S')} = \pi_{\tau_i(S,S')} \qquad \text{for all } i = 1, \ldots, n \text{ and for } \mathbb{P}^{2n}\text{-a.e. } (S, S').$$

Equivalently, $\pi$ depends only on the unordered multiset $\{\{(z_i, z_i')\}_{i=1}^n\}$ and not on which element of each pair is designated as "observed" versus "ghost."

**Connection to CMI.** This notion covers the conditional–mutual–information (CMI) framework of Steinke & Zakynthinou (2020). In the CMI setup, one draws paired data $Z = ((Z_i^{(0)}, Z_i^{(1)}))_{i=1}^n \overset{\text{i.i.d.}}{\sim} (\mathbb{P} \times \mathbb{P})^n$ and an independent selector $U \in \{0, 1\}^n$. The training and ghost sets are $S_U = (Z_1^{(U_1)}, \ldots, Z_n^{(U_n)})$ and $S_{\bar{U}} = (Z_1^{(1-U_1)}, \ldots, Z_n^{(1-U_n)})$. Any prior $\pi$ that is a function of $Z$ only (independent of $U$) is swap–invariant, since flipping $U_i$ implements $\tau_i$. Conversely, swap–invariance for all $i$ is equivalent to invariance under all coordinatewise flips of $U$, hence independence from $U$.

Throughout, let $S = \{z_i\}_{i=1}^n$ be the observed sample and $S' = \{z_i'\}_{i=1}^n$ an i.i.d. ghost sample, independent of $S$. We write $\mathbb{P}_S$ for the empirical measure $\mathbb{P}_n$ based on $S$, and $\varrho_{S,\ell}$ for the metric $\varrho_{n,\ell}$ from the main paper. Let $\mathbb{P}_{S'}$ denote the empirical measure based on $S'$. For any integrable function $g : \mathcal{Z} \to \mathbb{R}$ (we write $g(z)$ when convenient; e.g., $g(z) = \ell(f; z)$), define the empirical averaging operators

$$\mathbb{P}_S g := \frac{1}{n} \sum_{i=1}^n g(z_i), \qquad \mathbb{P}_{S'} g := \frac{1}{n} \sum_{i=1}^n g(z_i').$$

We use the shorthand $(\mathbb{P}_S \pm \mathbb{P}_{S'}) g := \mathbb{P}_S g \pm \mathbb{P}_{S'} g$ for the sum/difference of the two sample–average operators, and the same notation when $\mathbb{P}_S \pm \mathbb{P}_{S'}$ appear inside norms or distances.

**Theorem 5 (Pointwise Generalization via Ghost Sample)** *Let $\ell(f; z) \in [0, 1]$. There exists an absolute constant $C > 0$ such that for any swap-invariant prior $\pi_{(\cdot,\cdot)}$ on $(S, S')$, and any $\delta \in (0, 1)$, with probability at least $1 - \delta$ over $(S, S')$, uniformly in $f \in \mathcal{F}$,*

$$(\mathbb{P}_{S'} - \mathbb{P}_S) \, \ell(f; z)$$

$$\leq C \left( \inf_{\alpha \geq 0} \left\{ \alpha + \frac{1}{\sqrt{n}} \int_\alpha^{\sqrt{2(\mathbb{P}_S + \mathbb{P}_{S'}) \ell(f;z)^2}} \sqrt{\log \frac{1}{\pi_{(S,S')}(B_{\varrho_{(S,S'),\ell}}(f, \varepsilon))}} \, d\varepsilon \right\} + \sqrt{\frac{\log(\log(2n)/\delta)}{n}} \right),$$

*where $\varrho_{(S,S'),\ell}(f_1, f_2) = \sqrt{(\mathbb{P}_S + \mathbb{P}_{S'})(\ell(f_1; z) - \ell(f_2; z))^2}$.*

**Proof of Theorem 5:** The proof of the upper bound in Theorem 5 consists of three steps: 1. Subset-Homogeneous Uniform Convergence; 2. Generic Conversion to Pointwise Generalization Bound; 3. High-Probability Symmetrization.

**Step 1: Subset-Homogeneous Uniform Convergence.** Let $S = \{z_i\}_{i=1}^n$ be the observed sample, and $S' = \{z_i'\}_{i=1}^n$ be an i.i.d. ghost sample. We consider the symmetrized loss

$$\tilde{\ell}(f; (z, z')) = \ell(f; z') - \ell(f; z). \tag{36}$$

Since $\ell(f; z)$ is uniformly bounded in $[0, 1]$, $\tilde{\ell}(f; (z, z'))$ is uniformly bounded in $[-1, 1]$. We adopt the notation

$$\varrho_{S,\ell}(f_1, f_2) = \varrho_{n,\ell}(f_1, f_2) = \sqrt{\mathbb{P}_S(\ell(f_1; z) - \ell(f_2; z))^2}.$$

from the main paper. Furthermore, we define the loss-induced $L_2$ metrics $\varrho_{S',\ell}$ and $\varrho_{(S,S'),\ell}$ by

$$\varrho_{S',\ell}(f_1, f_2) = \sqrt{\mathbb{P}_{S'}(\ell(f_1; z) - \ell(f_2; z))^2},$$
$$\varrho_{(S,S'),\ell}(f_1, f_2) = \sqrt{(\mathbb{P}_S + \mathbb{P}_{S'})(\ell(f_1; z) - \ell(f_2; z))^2}.$$

By Minkowski's inequality (see, e.g., Wikipedia contributors (2025c)) and $\sqrt{a} + \sqrt{b} \le \sqrt{2(a+b)}$, we have

$$\sqrt{\frac{1}{n}\sum_{i=1}^{n}(\tilde{\ell}(f_1;(z_i,z_i'))-\tilde{\ell}(f_2;(z_i,z_i')))^2} \le \varrho_{S,\ell}(f_1,f_2) + \varrho_{S',\ell}(f_1,f_2) \le \sqrt{2}\varrho_{(S,S'),\ell}(f_1,f_2).$$

(37)

For every fixed $R \in [0,2]$, we define

$$\mathcal{F}_R = \{f \in \mathcal{F} : (\mathbb{P}_S + \mathbb{P}_{S'})\ell(f;z)^2 \le R\}.$$

The goal of setting $R$ is to further localize the integrand upper limit as in Theorem 5.

Now applying the truncated integral bound (Lemma 14) to the empirical Rademacher complexity: let $\{\xi_i\}_{i=1}^{n}$ be i.i.d. Rademacher variables, then conditioned on $(S,S')$, given any subset $\mathcal{H} \subseteq \mathcal{F}_R$, we have that for all $\delta \in (0,1)$, with probability at least $1 - \delta$ (the randomness all comes from $\{\xi_i\}_{i=1}^{n}$),

$$\sup_{f\in\mathcal{H}} \frac{1}{n}\sum_{i=1}^{n} \xi_i \tilde{\ell}(f;(z_i,z_i)) \le \mathbb{E}_\xi \left[\sup_{f\in\mathcal{H}} \frac{1}{n}\sum_{i=1}^{n} \xi_i \tilde{\ell}(f;(z_i,z_i))\right] + \sqrt{\frac{2\log\frac{1}{\delta}}{n}}$$

$$\le C_0 \inf_{\alpha\ge 0}\left\{\alpha + \frac{1}{\sqrt{n}} \inf_{\mu\in\Delta(\mathcal{H})} \sup_{f\in\mathcal{H}} \int_\alpha^{2\sqrt{2R}} \sqrt{\log\frac{1}{\mu(B_{\tilde{\varrho}}(f,\varepsilon))}}d\varepsilon\right\} + \sqrt{\frac{2\log\frac{1}{\delta}}{n}},$$

where $C_0 > 0$ is an absolute constant, and $\tilde{\varrho}(f_1,f_2) := \sqrt{\frac{1}{n}\sum_{i=1}^{n}(\tilde{\ell}(f_1;(z_i,z_i'))-\tilde{\ell}(f_2;(z_i,z_i')))^2}$. Here, the first inequality is by Mcdiarmid's inequality (Lemma 16); and the second inequality is by Lemma 14; and the integral is capped at $2\sqrt{2R}$ because

$$\sup_{f_1\in\mathcal{H},f_2\in\mathcal{H}} \tilde{\varrho}(f_1,f_2) \le \sup_{f_1\in\mathcal{H},f_2\in\mathcal{H}} \sqrt{2}\varrho_{(S,S'),\ell}(f_1,f_2) \le \sup_{f\in\mathcal{H}} 2\sqrt{2}\sqrt{(\mathbb{P}_S + \mathbb{P}_{S'})\ell(f;z)^2} \le 2\sqrt{2R},$$

where the first inequality is due to (37) and the second inequality is due to Minkowski's inequality (Wikipedia contributors, 2025c). By the ambient–equivalence of the pointwise–dimension functional (Lemma 8), we have (note that we take the support of $\pi$ to be $\mathcal{F}$ rather than $\mathcal{H}$ or $\mathcal{F}_R$)

$$\inf_{\alpha\ge 0}\left\{\alpha + \frac{1}{\sqrt{n}} \inf_{\mu\in\Delta(\mathcal{H})} \sup_{f\in\mathcal{H}} \int_\alpha^{2\sqrt{2R}} \sqrt{\log\frac{1}{\mu(B_{\tilde{\varrho}}(f,\varepsilon))}}d\varepsilon\right\}$$

$$\le \inf_{\alpha\ge 0} 2\left\{\alpha + \frac{1}{\sqrt{n}} \inf_{\pi\in\Delta(\mathcal{F})} \sup_{f\in\mathcal{H}} \int_\alpha^{\sqrt{2R}} \sqrt{\log\frac{1}{\pi(B_{\tilde{\varrho}}(f,\varepsilon))}}d\varepsilon\right\}.$$

By (37) and the fact that pointwise dimension is monotone in the underlying metric (Lemma 19), we have that for any $\pi \in \Delta(\mathcal{F})$,

$$\int_\alpha^{\sqrt{2R}} \sqrt{\log\frac{1}{\pi(B_{\tilde{\varrho}}(f,\varepsilon))}}d\varepsilon \le \int_\alpha^{\sqrt{2R}} \sqrt{\log\frac{1}{\pi(B_{\sqrt{2}\varrho_{(S,S'),\ell}}(f,\varepsilon))}}d\varepsilon = \sqrt{2}\int_{\alpha/\sqrt{2}}^{\sqrt{R}} \sqrt{\log\frac{1}{\pi(B_{\varrho_{(S,S'),\ell}}(f,\varepsilon))}}d\varepsilon,$$

where the equality follows by a change of variables. Combining the above three inequalities, we prove the following *subset-homogeneous* uniform convergence argument when choosing an arbitrary $\pi \in \Delta(\mathcal{F})$: conditioned on $(S,S')$, for any $\mathcal{H} \subseteq \mathcal{F}$ and $\delta \in (0,1)$, with probability at least $1 - \delta$ (the randomness all comes from $\{\xi_i\}_{i=1}^{n}$),

$$\sup_{f\in\mathcal{H}} \frac{1}{n}\sum_{i=1}^{n} \xi_i(\ell(f;z_i')-\ell(f;z_i)) \le \sup_{f\in\mathcal{H}} C_1 \inf_{\alpha\ge 0}\left\{\alpha + \frac{1}{\sqrt{n}}\int_\alpha^{\sqrt{R}} \sqrt{\log\frac{1}{\pi(B_{\varrho_{(S,S'),\ell}}(f,\varepsilon))}}d\varepsilon\right\} + \sqrt{\frac{2\log\frac{1}{\delta}}{n}},$$

(38)

where $C_1 = 2\sqrt{2}C_0 > 0$ is an absolute constant.

Conditioned on $(S, S')$, for fixed $\pi_{(S,S')} \in \Delta(\mathcal{F})$ that is independent with $\{\xi_i\}_{i=1}^n$, define the pointwise complexity

$$d_{S,S'}(f) := \left( \inf_{\alpha \geq 0} \left\{ \sqrt{n}\, \alpha + \int_\alpha^{\sqrt{R}} \sqrt{\log \frac{1}{\pi_{(S,S')}\big(B_{\varrho_{(S,S'),\ell}}(f,\varepsilon)\big)}} \, d\varepsilon \right\} \right)^2. \tag{39}$$

Then, by (38), conditioned on $(S, S')$, for any $\mathcal{H} \subseteq \mathcal{F}_R$ and any $\delta \in (0,1)$, with probability at least $1 - \delta$ (the randomness all comes from $\{\xi_i\}_{i=1}^n$),

$$\sup_{f \in \mathcal{H}} \frac{1}{n} \sum_{i=1}^n \xi_i(\ell(f; z_i') - \ell(f; z_i)) \leq \sup_{f \in \mathcal{H}} \left( C_1 \sqrt{\frac{d_{S,S'}(f)}{n}} + \sqrt{\frac{2 \log \frac{2}{\delta}}{n}} \right). \tag{40}$$

As discussed in Appendix C.2.1, this condition is both necessary and sufficient to establish pointwise convergence when the complexity functional is the $\{\xi_i\}_{i=1}^n$-independent $d_{S,S'}(\cdot)$ when conditioned on $(S, S')$.

**Step 2: Generic Conversion to Pointwise Generalization Bound.** All the analysis in this step is condition on $(S, S')$, thus all the randomness discussed here comes from $\{\xi_i\}_{i=1}^n$. Choosing $\alpha = \sqrt{R}$ in (39) yields $d_{S,S'}(f) \leq (\sqrt{Rn})^2 \leq 2n$ for all $f$, so Lemma 4 applies with the upper bound of $d(f)$ being $2n$. For every $r \in [0, 2n]$, we take the subset

$$\mathcal{H} = \{f \in \mathcal{F}_R : d_{S,S'}(f) \leq r\},$$

which, by (40), implies that $\forall \delta \in (0,1)$ and $\forall r \in [0, 32n]$, with probability at least $1 - \delta$

$$\sup_{f : d_{S,S'}(f) \leq r} \frac{1}{n} \sum_{i=1}^n \xi_i(\ell(f; z_i') - \ell(f; z_i)) \leq C_1 \sqrt{\frac{r}{n}} + \sqrt{\frac{2 \log \frac{2}{\delta}}{n}}, \tag{41}$$

where and $C_1$ is an absolute constant. The inequality (41) is precisely the condition (17) in the generic conversion provided in Lemma 4 (here, the expectation (equal to 0) and the empirical average are taken for $\{\xi_i\}_{i=1}^n$). Thus applying Lemma 4 we have the pointwise generalization bound: conditioned on $(S, S')$, for any $\delta \in (0,1)$, by taking $r_0 = 1/n$, with probability at least $1 - \delta$, uniformly over all $f \in \mathcal{F}_R$,

$$\begin{aligned}
\frac{1}{n} \sum_{i=1}^n \xi_i(\ell(f; z_i') - \ell(f; z_i)) &\leq \frac{C_1}{\sqrt{n}} \sqrt{\max\left\{ 2 d_{S,S'}(f), \frac{1}{n} \right\}} + \sqrt{\frac{2 \log \frac{2 \log_2(4n^2)}{\delta}}{n}} \\
&\leq C_1 \sqrt{\frac{2 d_{S,S'}(f)}{n}} + \frac{C_1}{n} + \sqrt{\frac{2 \log \frac{4 \log_2(2n)}{\delta}}{n}}. \\
&\leq C_2 \left( \sqrt{\frac{d_{S,S'}(f)}{n}} + \sqrt{\frac{\log \frac{\log(2n)}{\delta}}{n}} \right),
\end{aligned} \tag{42}$$

where $C_2 > 0$ is an absolute constant, where the second inequality is because there exists $C_2 \geq \sqrt{2} C_1$ such that for all positive integer $n$,

$$\frac{C_1}{n} + \sqrt{\frac{2 \left( \log \frac{1}{\delta} + \log(\log 2 + \log n) + \log \frac{4}{\log 2} \right)}{n}} \leq C_2 \sqrt{\frac{\log \frac{1}{\delta} + \log(\log 2 + \log n)}{n}}.$$

Thus we prove the pointwise generalization bound (42) for the complexity functional $d_{S,S'}(\cdot)$ defined in (39), under the randomness of $\{\xi_i\}_{i=1}^n$.

Now we again apply the "uniform pointwise generalization" principle to further localize $R$ in (42) around the data–dependent quantity

$$d(f) := (\mathbb{P}_S + \mathbb{P}_{S'})\ell(f; z)^2 \in [0, 2].$$

Applying Lemma 4 with $R_0 = 1/n$ (spirit: taking $R_k = 2^k R_0$, and then using an union bound over these dyadic grid $R_k$ to (42) uniformly over all $k = 1, \cdots, \lceil \log_2(2n) \rceil$), we have that for all $\delta \in (0, 1)$, with probability at least $1 - \delta$, for all $f \in \mathcal{F}$,

$$\frac{1}{n} \sum_{i=1}^{n} \xi_i(\ell(f; z_i') - \ell(f; z_i))$$

$$\leq C_2 \left( \inf_{\alpha \geq 0} \left\{ \alpha + \frac{1}{\sqrt{n}} \int_{\alpha}^{\sqrt{\max\{2(\mathbb{P}_S + \mathbb{P}_{S'})\ell(f;z)^2, R_0\}}} \sqrt{\frac{1}{\pi_{(S,S')}(B_{\varrho_{(S,S'),\ell}}(f, \varepsilon))}} d\varepsilon \right\} + \sqrt{\frac{\log \frac{\log(2n)\lceil \log_2(4n) \rceil}{\delta}}{n}} \right).$$

If the maximum in $\max\{2(\mathbb{P}_S + \mathbb{P}_{S'})\ell(f; z)^2, R_0\}$ is attained at $R_0 = 1/n$, then the upper limit in the integral equals to $4\sqrt{1/n}$. In this case we may choose $\alpha = 4\sqrt{1/n}$, so that the complexity measure term vanishes. The remaining contribution is then of order $O(1/\sqrt{n})$, which can be absorbed into the absolute constant and the $\sqrt{\log(2n)/n}$ term. Thus we prove the following pointwise generalization bound: there exists an absolute constant $C_3 > 0$ such that for all $\pi \in \Delta(\mathcal{F})$, conditioned on $(S, S')$, for any $\delta \in (0, 1)$, with probability at least $1 - \delta$, uniformly over all $f \in \mathcal{F}$,

$$\frac{1}{n} \sum_{i=1}^{n} \xi_i(\ell(f; z_i') - \ell(f; z_i))$$

$$\leq C_3 \left( \inf_{\alpha \geq 0} \left\{ \alpha + \frac{1}{\sqrt{n}} \int_{\alpha}^{\sqrt{2(\mathbb{P}_S + \mathbb{P}_{S'})\ell(f;z)^2}} \sqrt{\frac{1}{\pi_{(S,S')}(B_{\varrho_{(S,S'),\ell}}(f, \varepsilon))}} d\varepsilon \right\} + \sqrt{\frac{\log \frac{\log(2n)}{\delta}}{n}} \right). \tag{43}$$

**Step 3: High-Probability Symmetrization.** Recall that $S = \{z_i\}_{i=1}^{n}$ and $S' = \{z_i'\}_{i=1}^{n}$ are i.i.d. samples, independent of each other, and $\{\xi_i\}_{i=1}^{n}$ are i.i.d. Rademacher signs, independent of $(S, S')$. The mixed (ghost) metric

$$\varrho_{(S,S'),\ell}(f_1, f_2) = \sqrt{(\mathbb{P}_S + \mathbb{P}_{S'})\big(\ell(f_1; z) - \ell(f_2; z)\big)^2},$$

is swap-invariant to the pair $(z_i, z_i')$ for each $i = 1, \cdots, n$. By the definition of swap-invariant prior before Theorem 5, the prior $\pi_{(S,S')} \in \Delta(\mathcal{F})$ is also swap-invariant to the pair $(z_i, z_i')$.

Denote the functionals

$$X(f; S, S'; \delta)$$

$$:= \frac{1}{n} \sum_{i=1}^{n} (\ell(f; z_i') - \ell(f; z_i))$$

$$- C_3 \left( \inf_{\alpha \geq 0} \left\{ \alpha + \frac{1}{\sqrt{n}} \int_{\alpha}^{\sqrt{2(\mathbb{P}_S + \mathbb{P}_{S'})\ell(f;z)^2}} \sqrt{\frac{1}{\pi_{(S,S')}(B_{\varrho_{(S,S'),\ell}}(f, \varepsilon))}} d\varepsilon \right\} + \sqrt{\frac{\log \frac{\log(2n)}{\delta}}{n}} \right),$$

and

$$Y(f; S, S', \{\xi_i\}_{i=1}^{n}; \delta)$$

$$:= \frac{1}{n} \sum_{i=1}^{n} \xi_i(\ell(f; z_i') - \ell(f; z_i))$$

$$- C_3 \left( \inf_{\alpha \geq 0} \left\{ \alpha + \frac{1}{\sqrt{n}} \int_{\alpha}^{\sqrt{2(\mathbb{P}_S + \mathbb{P}_{S'})\ell(f;z)^2}} \sqrt{\frac{1}{\pi_{(S,S')}(B_{\varrho_{(S,S'),\ell}}(f, \varepsilon))}} d\varepsilon \right\} + \sqrt{\frac{\log \frac{\log(2n)}{\delta}}{n}} \right).$$

**Symmetry argument.** We write $\stackrel{d}{=}$ to denote equality in distribution (i.e., the random variables have the same law, equivalently the same cumulative distribution function). For each $i \in \{1, \ldots, n\}$,

let $\tau_i(S, S')$ be the pair obtained by swapping $z_i$ and $z_i'$. Since $\varrho_{(S,S'),\ell}$ and $\pi_{(S,S')}$ are invariant under $(S, S') \mapsto \tau_i(S, S')$ and $\tau_i(S, S') \stackrel{d}{=} (S, S')$, we have, for all $t \in \mathbb{R}$,

$$
\begin{aligned}
&\Pr\big(Y(f; S, S', \{\xi_i\}_{i=1}^n; \delta) \leq t\big) \\
&= \tfrac{1}{2} \Pr\big(Y(f; S, S', \{\xi_i\}_{i=1}^n; \delta) \leq t | \xi_i = 1\big) + \tfrac{1}{2} \Pr\big(Y(f; S, S', \{\xi_i\}_{i=1}^n; \delta) \leq t | \xi_i = -1\big) \\
&= \Pr\big(Y(f; S, S', \{\xi_i\}_{i=1}^n; \delta) \leq t | \xi_i = 1\big),
\end{aligned}
$$

i.e., $Y(f; S, S', \{\xi_i\}_{i=1}^n; \delta) \stackrel{d}{=} Y(f; S, S', \{\xi_1, \cdots, \xi_{i-1}, 1, \xi_{i+1}, \cdots, \xi_n\}; \delta)$. In the second equality, we have used the fact: conditioning on $(S, S')$, the transformation

$$
(S, S', \{\xi_j\}_{j=1}^n) \longmapsto \big(\tau_i(S, S'), \{\xi_1, \ldots, \xi_{i-1}, -\xi_i, \xi_{i+1}, \ldots, \xi_n\}\big)
$$

leaves the value of $Y$ unchanged (by the swap–invariance of $\varrho_{(S,S'),\ell}$ and $\pi_{(S,S')}$) and preserves the joint law of $(S, S', \{\xi_j\}_{j=1}^n)$, because $(z_i, z_i')$ are i.i.d. and $\xi_i$ is a symmetric Rademacher sign; hence the two conditional distributions coincide, and

$$
\Pr\big(Y(f; S, S', \{\xi_i\}_{i=1}^n; \delta) \leq t \mid \xi_i = 1\big) = \Pr\big(Y(f; S, S', \{\xi_i\}_{i=1}^n; \delta) \leq t \mid \xi_i = -1\big),
$$

so

$$
\Pr\big(Y(f; S, S', \{\xi_i\}_{i=1}^n; \delta) \leq t\big) = \Pr\big(Y(f; S, S', \{\xi_i\}_{i=1}^n; \delta) \leq t \mid \xi_i = 1\big).
$$

Iterate over all indices $i = 1, \cdots, n$, we obtain that

$$
Y(f; S, S', \{\xi_i\}_{i=1}^n; \delta) \stackrel{d}{=} Y(f; S, S', \{1, \cdots, 1\}; \delta) = X(f; S, S'; \delta).
$$

By the conclusion (43) in Step 2, for any $\delta \in (0, 1)$,

$$
\Pr_\xi \left( Y(f; S, S', \{\xi_i\}_{i=1}^n; \delta) \leq 0 \text{ for all } f \in \mathcal{F} \,\Big|\, S, S' \right) \geq 1 - \delta.
$$

By equality in distribution between $Y$ and $X$ (the symmetry argument above), this implies

$$
\Pr_\xi \left( X(f; S, S'; \delta) \leq 0 \text{ for all } f \in \mathcal{F} \,\Big|\, S, S' \right) \geq 1 - \delta \quad \text{for all } (S, S').
$$

Let

$$
A(S, S') := \Big\{ X(f; S, S'; \delta) \leq 0 \text{ for all } f \in \mathcal{F} \Big\}.
$$

Note that $A(S, S')$ (and hence its indicator $\mathbb{1}_A(S, S')$) depends only on $(S, S')$ and is independent of the Rademacher signs $\{\xi_i\}_{i=1}^n$. Using the tower property of conditional expectation, we obtain

$$
\begin{aligned}
\Pr_{S,S'} \big(A(S, S')\big) &= \mathbb{E}_{S,S'}\big[\mathbb{1}_A(S, S')\big] \\
&= \mathbb{E}_{S,S'}\Big[\mathbb{E}_\xi \big[\mathbb{1}_A(S, S') \mid S, S'\big]\Big] \\
&= \mathbb{E}_{S,S'}\Big[\Pr_\xi \big(A(S, S') \mid S, S'\big)\Big] \\
&\geq \mathbb{E}_{S,S'}[1 - \delta] = 1 - \delta,
\end{aligned}
$$

where the inequality uses the conditional bound above.

Hence, with probability at least $1 - \delta$ over the draw of $(S, S')$, we have, uniformly over all $f \in \mathcal{F}$,

$$
\frac{1}{n} \sum_{i=1}^n \big(\ell(f; z_i') - \ell(f; z_i)\big)
$$

$$
\leq C_3 \left( \inf_{\alpha \geq 0} \left\{ \alpha + \frac{1}{\sqrt{n}} \int_\alpha^{\sqrt{2(\mathbb{P}_S + \mathbb{P}_{S'})\ell(f;z)^2}} \sqrt{\log \frac{1}{\pi_{(S,S')}(B_{\varrho_{(S,S'),\ell}}(f, \varepsilon))}} \, d\varepsilon \right\} + \sqrt{\frac{\log(\log(2n)/\delta)}{n}} \right),
$$

where $\varrho_{(S,S'),\ell}(f_1, f_2) = \sqrt{(\mathbb{P}_S + \mathbb{P}_{S'})\big(\ell(f_1; z) - \ell(f_2; z)\big)^2}$, and $C_3 > 0$ is an absolute constant.

$\square$

## C.6 PROOF OF THEOREM 1

Theorem 5 have established a pointwise generalization bound through both the observed sample $S$ and the ghost sample $S'$. In this section we build on Theorem 5 to a prove pointwise generalization bound that only depends on the observed sample $S$. As outlined in Appendix C.2.1, the key is to estalish pointwise isomorphism between the $L_2(\mathbb{P}_S)$ and $L_2(\mathbb{P}_{S'})$ metrics.

**From Ghost Sample to Observed Sample.** Recall that we use $\mathbb{P}_S$ to be the same notation as $\mathbb{P}_n$ in the main paper; $\varrho_{S,\ell}$ to be the same notation as the $\varrho_{n,\ell}$ metric in the main paper; and $\mathbb{P}_{S'}$ to be the empirical distribution and sample average operator actioned on the ghost sample $S'$, similar to how $\mathbb{P}_S$ actioned on $S$ (see the comments before Theorem 5).

Let $\mathcal{G} \subseteq \{g : \mathcal{Z} \to [0, M]\}$, let $\varrho$ be a semi-metric on $\mathcal{G}$, and let $\mu \in \Delta(\mathcal{G})$ be a prior. For $g \in \mathcal{G}$ and $r \geq 0$ define

$$I(\mu, g, \varrho, r) := \inf_{0 \leq \alpha \leq \sqrt{r}} \left\{ \alpha + \frac{1}{\sqrt{n}} \int_\alpha^{\sqrt{r}} \sqrt{\log \frac{1}{\mu(B_\varrho(g, \varepsilon))}} \, d\varepsilon \right\}. \tag{44}$$

Under this definition, Theorem 5 implies that: there exists absolute constant $C_1 > 0$ such that, every fixed prior $\pi \in \Delta(\mathcal{F})$ independent of $(S, S')$ (such fixed prior clearly satisfies the condition in Theorem 5) and every $\delta \in (0, 1)$, uniformly over all $f \in \mathcal{F}$, we have

$$(\mathbb{P}_{S'} - \mathbb{P}_S)\ell(f; z) \leq C_1 \left( I\left(\pi, f, \varrho_{(S,S'),\ell}, 2(\mathbb{P}_S + \mathbb{P}_{S'})\ell(f; z)^2\right) + \sqrt{\frac{\log(\log(2n)/\delta)}{n}} \right). \tag{45}$$

We have the following lemma that converts Theorem 5 to Theorem 1.

**Lemma 9 (Ghost to Observed Conversion)** *For every fixed prior* $\pi \in \Delta(\mathcal{F})$ *independent of* $(S, S')$ *and every* $\delta \in (0, 1)$, *if with probability at least* $1 - \delta$, *uniformly for all* $f \in \mathcal{F}$,

$$I\left(\pi, f, \varrho_{(S,S'),\ell}, 2(\mathbb{P}_S + \mathbb{P}_{S'})\ell(f; z)^2\right) \leq C_2 \left( I\left(\pi, f, \varrho_{S,\ell}, 1\right) + \sqrt{\frac{\log(\log(2n)/\delta)}{n}} \right), \tag{46}$$

*where* $C_2 > 0$ *is an absolute constant, then there exists an absolute constant* $C > 0$ *such that for all* $\delta \in (0, 1)$, *with probability at least* $1 - \delta$, *uniformly over all* $f \in \mathcal{F}$,

$$(\mathbb{P} - \mathbb{P}_n)\ell(f; z) \leq C \left( I(\pi, f, \varrho_{S,\ell}, 1) + \sqrt{\frac{\log(\log(2n)/\delta)}{n}} \right),$$

*and this is exactly Theorem 1.*

**Proof of Lemma 9:** Setting the confidence parameter to $\delta/2$ in (45) and to $\delta/2$ in (46), a union bound implies that both inequalities hold simultaneously with probability at least $1 - \delta$. Combining them yields Lemma 9.

$\square$

We now verify condition (46) in Lemma 9, thereby completing the proof of Theorem 1. Condition (46) asserts a *pointwise isomorphism*: uniformly over all $f \in \mathcal{F}$, an $L_2(\mathbb{P}_{S'})$–induced quantity is controlled (up to absolute constants) by its fully $L_2(\mathbb{P}_S)$–induced counterpart.

Such isomorphisms are a longstanding theme in empirical process theory. Our proof adopts a fixed-point (localized Rademacher / generic-chaining) approach to transfer bounds from $\mathbb{P}_S + \mathbb{P}_{S'}$ to purely $\mathbb{P}_S$, yielding the desired *pointwise*, uniform comparison over $\mathcal{F}$. This route follows Section 4 in Bartlett et al. (2005) and Appendix A.4 of Xu & Zeevi (2025), but is developed here in the new context of pointwise–dimension functionals.

For classical *class-wide* isomorphisms (where the deviation depends on a global complexity of the class rather than a pointwise functional), see Klartag & Mendelson (2005) for bounded classes; Mendelson et al. (2007); Mendelson (2010) for sub-Gaussian and heavy-tailed regimes; Mendelson (2015) for weak small-ball conditions in unbounded settings; and Mendelson (2021) for a unified synthesis of bounded and small-ball analyses. These works are now cornerstones of the field. Our contribution refines this line by replacing global complexity with a *purely pointwise* complexity in the isomorphism comparison.

**Basic properties of the truncated pointwise integral.** We first state some basic properties of the truncated pointwise integtal: it is sub-root in $r$ and Lipchitz in $g$.

**Lemma 10 (Basic properties of the truncated pointwise integral)** *For the truncated integral* $I(\mu, g, \varrho, r)$ *defined in* (44)*, the following hold.*

(i) *(Sub-root and fixed point) For each fixed* $g \in \mathcal{G}$*, the map* $r \mapsto I(\mu, g, \varrho, r)$ *is a* sub-root *function. Consequently, by Definition 3.1 and Lemma 3.2 in Bartlett et al. (2005), there exists a unique* $r^\star \in (0, \infty)$ *such that* $I(\mu, g, \varrho, r^\star) = r^\star$*, and for all* $r > 0$*,*

$$r \geq I(\mu, g, \varrho, r) \quad \Longleftrightarrow \quad r \geq r^\star.$$

(ii) *(Lipschitz shift in* $g$*) For every* $g_1 \in \mathcal{G}$*,* $g_2 \in \mathcal{G}$*,* $r \geq 0$*,*

$$I(\mu, g_2, \varrho, r) \leq I(\mu, g_1, \varrho, r) + \varrho(g_1, g_2).$$

**Proof of Lemma 10:** Fix $g \in \mathcal{G}$ and abbreviate

$$h_g(\varepsilon) := \sqrt{\log \frac{1}{\mu(B_\varrho(g, \varepsilon))}}, \qquad F_g(u) := \int_0^u h_g(\varepsilon) \, d\varepsilon, \quad u \geq 0.$$

Since $\varepsilon \mapsto B_\varrho(g, \varepsilon)$ is increasing, $\mu(B_\varrho(g, \varepsilon))$ is nondecreasing, hence $\varepsilon \mapsto h_g(\varepsilon)$ is nonincreasing; therefore $F_g$ is concave, nondecreasing and $F_g(0) = 0$.

**(i) Sub-root.** Write, for $0 \leq \alpha \leq \sqrt{r}$,

$$\Phi_g(r, \alpha) := \alpha + \frac{1}{\sqrt{n}} \big( F_g(\sqrt{r}) - F_g(\alpha) \big) \quad \text{so that} \quad I(\mu, g, \varrho, r) = \inf_{0 \leq \alpha \leq \sqrt{r}} \Phi_g(r, \alpha).$$

Nonnegativity is immediate. Monotonicity in $r$ holds because $F_g$ is nondecreasing. To prove the sub-root property, consider for fixed $\alpha$ the function

$$r \longmapsto \frac{\Phi_g(r, \alpha)}{\sqrt{r}} = \frac{\alpha}{\sqrt{r}} + \frac{1}{\sqrt{n}} \frac{F_g(\sqrt{r})}{\sqrt{r}} - \frac{1}{\sqrt{n}} \frac{F_g(\alpha)}{\sqrt{r}}.$$

The first and third terms are of the form $c/\sqrt{r}$ and are thus nonincreasing in $r$. For the middle term, set $u = \sqrt{r}$; by concavity of $F_g$ and $F_g(0) = 0$, the map $u \mapsto F_g(u)/u$ is nonincreasing on $(0, \infty)$. Hence $r \mapsto F_g(\sqrt{r})/\sqrt{r}$ is nonincreasing. Therefore, for every fixed $\alpha$, $r \mapsto \Phi_g(r, \alpha)/\sqrt{r}$ is nonincreasing. Taking the infimum over $\alpha$ preserves this property: $r \mapsto I(\mu, g, \varrho, r)/\sqrt{r}$ is nonincreasing. Thus $I(\mu, g, \varrho, \cdot)$ is sub-root. The fixed-point and characterization then follow from Lemma 3.2 in Bartlett et al. (2005).

**(ii) Shift in** $g$**.** Assume $\varrho(g_1, g_2) = \beta$. By the triangle inequality, for all $\varepsilon \geq 0$,

$$B_\varrho(g_1, \varepsilon) \subseteq B_\varrho(g_2, \varepsilon + \beta).$$

Hence $\mu(B_\varrho(g_2, \varepsilon + \beta)) \geq \mu(B_\varrho(g_1, \varepsilon))$ and therefore

$$\sqrt{\log \frac{1}{\mu(B_\varrho(g_2, \varepsilon + \beta))}} \leq \sqrt{\log \frac{1}{\mu(B_\varrho(g_1, \varepsilon))}}.$$

Using this, a change of variables $u = \varepsilon + \beta$, and the constraint $0 \le \alpha \le \sqrt{r}$, we obtain

$$
\begin{aligned}
I(\mu, g_2, \varrho, r) &= \inf_{0 \le \alpha \le \sqrt{r}} \left\{ \alpha + \frac{1}{\sqrt{n}} \int_\alpha^{\sqrt{r}} \sqrt{\log \frac{1}{\mu(B_\varrho(g_2, \varepsilon))}} \, d\varepsilon \right\} \\
&\le \inf_{0 \le \alpha \le \sqrt{r}} \left\{ \alpha + \frac{1}{\sqrt{n}} \int_\alpha^{\sqrt{r}} \sqrt{\log \frac{1}{\mu(B_\varrho(g_2, \varepsilon + \beta))}} \, d\varepsilon \right\} \\
&= \inf_{0 \le \alpha \le \sqrt{r}} \left\{ \alpha + \frac{1}{\sqrt{n}} \int_{\alpha + \beta}^{\sqrt{r} + \beta} \sqrt{\log \frac{1}{\mu(B_\varrho(g_2, u))}} \, du \right\} \\
&\le \inf_{0 \le \alpha \le \sqrt{r}} \left\{ \alpha + \beta + \frac{1}{\sqrt{n}} \int_\alpha^{\sqrt{r}} \sqrt{\log \frac{1}{\mu(B_\varrho(g_2, u))}} \, du \right\} \\
&\le \inf_{0 \le \alpha \le \sqrt{r}} \left\{ \alpha + \beta + \frac{1}{\sqrt{n}} \int_\alpha^{\sqrt{r}} \sqrt{\log \frac{1}{\mu(B_\varrho(g_1, u))}} \, du \right\} \\
&= I(\mu, g_1, \varrho, r) + \beta,
\end{aligned}
$$

which proves the claim.

$\square$

**Pointwise Isomorphism via Fixed Point Analysis.** Define $\bar{\varrho}_{(S,S'),\ell}$ to be the quadratic-loss-induced $L_2$ metric over the product space $\mathcal{F} \times \mathcal{F}$, given by

$$
\bar{\varrho}_{(S,S'),\ell}((f_1', f_2'), (f_1, f_2)) = \left( (\mathbb{P}_S + \mathbb{P}_{S'})[(\ell(f_1', z) - \ell(f_2', z))^2 - (\ell(f_1, z) - \ell(f_2, z))^2] \right)^{1/2}.
$$

By Theorem 5, there exists an absolute constant $C_1 > 0$ such that given a fixed, data-independent prior $\mu \in \Delta(\mathcal{F} \times \mathcal{F})$, for every $\delta \in (0, 1)$, with probability at least $1 - \delta$, uniformly over all $f_1 \in \mathcal{F}, f_2 \in \mathcal{F}$,

$$
\left| (\mathbb{P}_{S'} - \mathbb{P}_S)(\ell(f_1; z) - \ell(f_2; z))^2 \right|
$$
$$
\le C_1 \left( \inf_{\alpha \ge 0} \left\{ \alpha + \frac{1}{\sqrt{n}} \int_\alpha^{\sqrt{2(\mathbb{P}_S + \mathbb{P}_{S'})(\ell(f_1;z) - \ell(f_2;z))^4}} \sqrt{\log \frac{1}{\mu(B_{\varrho_{(S,S')}}((f_1, f_2), \varepsilon)\})}} d\varepsilon \right\} + \sqrt{\frac{\log(\log(2n)/\delta)}{n}} \right)
$$
$$
\le C_1 \left( I(\mu, (f_1, f_2), \bar{\varrho}_{(S,S'),\ell}, 2(\mathbb{P}_S + \mathbb{P}_{S'})(\ell(f_1; z) - \ell(f_2; z))^2) + \sqrt{\frac{\log(\log(2n)/\delta)}{n}} \right), \quad (47)
$$

where the first inequality applies Theorem 5 twice—once with $g(z) = (\ell(f_1; z) - \ell(f_2; z))^2$ at confidence level $\delta/2$ and once with $g(z) = -(\ell(f_1; z) - \ell(f_2; z))^2$ at confidence level $\delta/2$—and then takes a union bound; the second inequality uses the uniform bound $|\ell| \le 1$, which implies $|\ell(f_1; z) - \ell(f_2; z)| \le 1$ and hence $(\ell(f_1; z) - \ell(f_2; z))^4 \le (\ell(f_1; z) - \ell(f_2; z))^2$ pointwise, yielding the $L_4$–$L_2$ comparison.

Given a fixed, data-independent $\pi \in \Delta(\mathcal{F})$, take $\mu$ to be the independent product measure $\pi \otimes \pi$. By Minkowski's inequality (Wikipedia contributors, 2025c) and $\ell(f; z) \in [0, 1]$ we have that for all $f_1 \in \mathcal{F}, f_2 \in \mathcal{F}$,

$$
\bar{\varrho}_{(S,S'),\ell}((f_1', f_2'), (f_1, f_2)) \le 2\varrho_{(S,S'),\ell}(f_1', f_1) + 2\varrho_{(S,S'),\ell}(f_2', f_2).
$$

Then we have the decomposition

$$
\begin{aligned}
&\log \frac{1}{\mu(B_{\bar{\varrho}_{(S,S'),\ell}}((f_1, f_2), \varepsilon)} \\
&\le \log \frac{1}{\pi \otimes \pi(f_1' \in \mathcal{F}, f_2' \in \mathcal{F} : \varrho_{(S,S'),\ell}(f_1', f_1) \le \frac{\varepsilon}{4}, \varrho_{(S,S'),\ell}(f_2', f_2) \le \frac{\varepsilon}{4})} \\
&= \log \frac{1}{\pi(B_{\varrho_{(S,S'),\ell}}(f_1, \varepsilon/4)} + \log \frac{1}{\pi(B_{\varrho_{(S,S'),\ell}}(f_2, \varepsilon/4)}. \quad (48)
\end{aligned}
$$

Combining (47) and (48), we obtain that for all $\delta \in (0,1)$, with probability at least $1 - \delta$, for all $f_1 \in \mathcal{F}, f_2 \in \mathcal{F}$,

$$\varrho_{(S,S'),\ell}^2(f_1, f_2) - 2\varrho_{S,\ell}^2(f_1, f_2)$$

$$= (\mathbb{P}_{S'} - \mathbb{P}_S)(\ell(f_1; z) - \ell(f_2; z))^2$$

$$\leq C_2 \bigg( I\left(\pi, f_1, \varrho_{(S,S'),\ell}, (\mathbb{P}_S + \mathbb{P}_{S'})(\ell(f_1; z) - \ell(f_2; z))^2/8\right) \tag{49}$$

$$+ I\left(\pi, f_2, \varrho_{(S,S'),\ell}, (\mathbb{P}_S + \mathbb{P}_{S'})(\ell(f_1; z) - \ell(f_2; z))^2/8\right) + \sqrt{\frac{\log(\log(2n)/\delta)}{n}} \bigg), \tag{50}$$

where $C_2 = 4C_1 > 0$ are absolute constants.

By Lemma 10,

$$\psi_{S,S'}(r; f) := \sup_{f' \in \mathcal{F}: \varrho_{(S,S'),\ell}^2(f',f) \leq r} C_2 \left( I(\pi, f, \varrho_{(S,S'),\ell}, r/8) + I(\pi, f', \varrho_{(S,S'),\ell}, r/8) + \sqrt{\frac{\log(\log(2n)/\delta)}{n}} \right)$$

is a sub-root function, so there exists a unique fixed point $r_{S,S'}^\star$ such that

$$r_{S,S'}^\star(f) = \psi_{S,S'}(r_{S,S'}^\star(f); f).$$

By the definition of sub-root function, for $r \geq 4r_{S,S'}^\star(f)$, we have that

$$\sup_{(\mathbb{P}_S + \mathbb{P}_{S'})(\ell(f';z) - \ell(f;z))^2 \leq r} (\varrho_{(S,S'),\ell}(f_1, f_2) - 2\varrho_{S,\ell}^2(f_1, f_2)) \leq \psi_{S,S'}(r; f)$$

$$\leq \sqrt{\frac{r}{r_{S,S'}^\star(f)}} \psi_{S,S'}(r_{S,S'}^\star(f); f) = \sqrt{r}\sqrt{r_{S,S'}^\star(f)} \leq \frac{1}{2}r, \tag{51}$$

where the first inequality is due to (50); the second inequality is by the definition of sub-root function; the equality is by the definition of fixed point; and the last inequality is by $r \geq 4r_{S,S'}^\star(f)$. Combining (47) and (51), we have the following: with probability at least $1 - \delta$, for all $f \in \mathcal{F}, f' \in \mathcal{F}$ such that $(\mathbb{P}_S + \mathbb{P}_{S'})(\ell(f'; z) - \ell(f; z))^2 \geq 4r_{S,S'}^\star(f)$,

$$\left| (\mathbb{P}_S - \mathbb{P}_{S'})(\ell(f'; z) - \ell(f; z))^2 \right| \leq \frac{1}{2}(\mathbb{P}_S + \mathbb{P}_{S'})(\ell(f'; z) - \ell(f; z))^2,$$

which implies that with probability at least $1 - \delta$, whenever $(\mathbb{P}_S + \mathbb{P}_{S'})(\ell(f'; z) - \ell(f; z))^2 \geq 4r_{S,S'}^\star(f)$,

$$\frac{4}{3}\mathbb{P}_S(\ell(f'; z) - \ell(f; z))^2 \leq (\mathbb{P}_S + \mathbb{P}_{S'})(\ell(f'; z) - \ell(f; z))^2 \leq 4\mathbb{P}_S(\ell(f'; z) - \ell(f; z))^2. \tag{52}$$

Therefore, with probability at least $1 - \delta$, for all $f \in \mathcal{F}$ and $r \geq 4r_{S,S'}^\star(f)$,

$$\psi_S(r; f) := \sup_{f' \in \mathcal{F}: 2\varrho_{(S,S'),\ell}^2(f',f) \leq 3r/2} C_2 \left( I(\pi, f, 2\varrho_{S,\ell}, r/8) + I(\pi, f', 2\varrho_{S,\ell}, r/8) + \sqrt{\frac{\log(\log(2n)/\delta)}{n}} \right)$$

$$\tag{53}$$

is a surrogate function of $\psi_{S,S'}(r; f)$: with probability at least $1 - \delta$,

$$\psi_S(r; f) \geq \psi_{S,S'}(r; f), \quad \forall r \geq r_{S,S'}^\star(f), \forall f \in \mathcal{F}.$$

Here replacing $\varrho_{(S,S'),\ell}$ by $2\varrho_S$ inside the integral is by the metric monotonicity of pointwise dimension (Lemma 19) and the right hand side of (52); and replacing the constraint $\varrho_{(S,S'),\ell}(f'f) \leq r$ by the new constraint $2\varrho_{S,\ell}(f'f) \leq 3r/2$ outside the parentheses is due to the left hand side of (52) and its implication: with probability at least $1 - \delta$,

$$\{f' \in \mathcal{F} : \varrho_{(S,S'),\ell}(f'f) \leq r\} \subseteq \{f' \in \mathcal{F} : 2\varrho_{S,\ell}(f'f) \leq 3r/2\}, \quad \forall r \geq r_{S,S'}^\star(f), \forall f \in \mathcal{F}.$$

This means that with probability at least $1 - \delta$, for all $f \in \mathcal{F}$,

$$\psi_S(r_{S,S'}^\star(f); f) \geq \psi_{S,S'}(r_{S,S'}^\star(f); f) = r_{S,S'}^\star(f).$$

Define the fixed point of $\psi_S(f; r)$ to be $r_S^\star(f)$. By the above inequality and the fact that sub-root function has an unique fixed point (Lemma 10), we must have

$$r_S^\star(f) \geq r_{S,S'}^\star(f). \tag{54}$$

This implies that

1. For all $f' \in \mathcal{F}, f \in \mathcal{F}$ such that $2\varrho_{S,\ell}^2(f', f) \geq 3r_S^\star(f)/2$, by (52) and (54), we have that

$$(\mathbb{P}_S + \mathbb{P}_{S'})(\ell(f'; z) - \ell(f; z))^2 \leq 4\mathbb{P}_S(\ell(f'; z) - \ell(f; z))^2.$$

2. For all $f' \in \mathcal{F}, f \in \mathcal{F}$ such that $2/3 \cdot 2\varrho_{S,\ell}^2(f', f) < r_S^\star(f)$, we have that

$$\frac{2}{3} \cdot 2\varrho_{S,\ell}^2(f', f) \leq \psi\left(\frac{2}{3} \cdot 2\varrho_{S,\ell}^2(f', f); f\right)$$

$$\leq C_2\left(I(\pi, f, 2\varrho_{S,\ell}, r/8) + I(\pi, f', 2\varrho_{S,\ell}, r/8) + \sqrt{\frac{\log(\log(2n))}{n}}\right),$$

where the first inequality is a simple consequence of the definition of fixed point: when $r \leq r_S^\star(f), r \leq \psi_S(r; f)$; and the second inequality is be the definition of $\psi_S(r; f)$ in (53).

Together, we obtain that with probability at least $1 - \delta$, uniformly over all $f \in \mathcal{F}$ and $f \in \mathcal{F}$,

$$(\mathbb{P}_S + \mathbb{P}_{S'})(\ell(f'; z) - \ell(f; z))^2 - 4\mathbb{P}_S(\ell(f'; z) - \ell(f; z))^2$$

$$\leq C_2\left(2I(\pi, f, 2\varrho_{S,\ell}, \varrho_{S,\ell}(f', f)/4) + 2\varrho_{S,\ell}(f', f) + \sqrt{\frac{\log(\log(2n)/\delta)}{n}}\right)$$

By definition (53), we have

$$\psi_{(S,S'),\ell}(\varrho_{(S,S'),\ell}(f', f); f) \leq C_2\left(2I(\pi, f, 2\varrho_{S,\ell}, \varrho_{S,\ell}(f', f)/4) + 2\varrho_{S,\ell}(f', f) + \sqrt{\frac{\log(\log(2n)/\delta)}{n}}\right).$$

This is an inequality in the form of

$$r - 2r' \leq C_2(a(r'/8) + 2\sqrt{r'}),$$

where

$$a(r) = 2I(\pi, f, 2\varrho_{S,\ell}, r) + \sqrt{\frac{\log(\log(2n)/\delta)}{n}}.$$

Solving the above inequality we have that there exists absolute constant $C_3 > 0, C4 > 0$ such that with probability at least $1 - \delta$, uniformly over $f \in \mathcal{F}$ and $f' \in \mathcal{F}$,

$$\varrho_{(S,S'),\ell}(f_1, f_2) \leq C_3\varrho_{S,\ell}(f_1, f_2) + C_4\left(2I(\pi, f, 2\varrho_{S,\ell}, \varrho_{S,\ell}(f_1, f_2)/4) + \sqrt{\frac{\log(\log(2n)/\delta)}{n}}\right)$$

By the Lipchitz property in Lemma 10, we prove that there exists absolute constant $C_5 > 0$ such that

$$I(\pi, f, \varrho_{(S,S'),\ell}, 2(\mathbb{P}_S + \mathbb{P}_{S'})\ell(f; z)^2) \leq C_5\left(I(\pi, f, \varrho_{S,\ell}, 1) + \sqrt{\frac{\log(\log(2n)/\delta)}{n}}\right)$$

This is exactly the condition (46) in Lemma 9, which enables us to prove Theorem 1 from Theorem 5.

$$\square$$

### C.7 PROOF OF THEOREM 2

We use the classical result that the expected uniform convergence is lower bounded by Gaussian complexity of the centered class, up to a $\sqrt{\log n}$ factor, see Definition 2 and Lemma 15 in the auxiliary lemma part for this classical result. To be specific, by Lemma 15 we have that

$$\mathbb{E}_z\left[\sup_{f \in \mathcal{F}}(\mathbb{P} - \mathbb{P}_n)\ell(f; z)\right] \geq \frac{c_1}{\sqrt{\log n}}\mathbb{E}_{g,z}\left[\sup_{f \in \mathcal{F}}\frac{1}{n}\sum_{i=1}^n g_i(\ell(f; z_i) - \mathbb{E}_z[\ell(f; z)])\right]$$

$$\geq \frac{c_1}{\sqrt{\log n}}\mathbb{E}_{g,z}\left[\sup_{f \in \mathcal{F}}\frac{1}{n}\sum_{i=1}^n g_i\ell(f; z_i) - \left|\frac{1}{n}\sum_{i=1}^n g_i\right| \cdot \sup_{\mathcal{F}}\mathbb{E}[\ell(f; z)]\right]$$

$$= \frac{c_1}{\sqrt{\log n}}\mathbb{E}_{g,z}\left[\sup_{f \in \mathcal{F}}\frac{1}{n}\sum_{i=1}^n g_i\ell(f; z_i)\right] - \frac{c_1}{\sqrt{\log n}}\sqrt{\frac{2}{\pi n}}\sup_{\mathcal{F}}\mathbb{E}[\ell(f; z)], \quad (55)$$

where $c_1 > 0$ is an absolute constant, and the equality use the fact that $\mathbb{E}[|Y|] = \sqrt{\frac{2}{\pi n}}$ for $Y \sim N(0, 1/n)$.

Now applying Lemma 12 to lower bounding the Gaussian process $\frac{1}{n} \sum_{i=1}^{n} g_i \ell(f; z_i)$ by the integral, we have for any $\{z_i\}_{i=1}^{n}$,

$$\mathbb{E}_g \left[ \sup_{f \in \mathcal{F}} \frac{1}{n} \sum_{i=1}^{n} g_i \ell(f; z_i) \right] \geq c_2 \inf_{\pi} \sup_{f \in \mathcal{F}} \int_0^1 \sqrt{\log \frac{1}{\pi(B_{\varrho_{n,\ell}}(f, \varepsilon))}} d\varepsilon,$$

taking expectation on both side yields

$$\mathbb{E}_{g,z} \left[ \sup_{f \in \mathcal{F}} \frac{1}{n} \sum_{i=1}^{n} g_i \ell(f; z_i) \right] \geq c_2 \mathbb{E} \inf_{\pi} \sup_{f \in \mathcal{F}} \int_0^1 \sqrt{\log \frac{1}{\pi(B_{\varrho_{n,\ell}}(f, \varepsilon))}} d\varepsilon. \tag{56}$$

Combining (55) and (56), we have that there exist absolute constants $c, c' > 0$ such that

$$\mathbb{E} \left[ \sup_{f \in \mathcal{F}} (\mathbb{P} - \mathbb{P}_n) \ell(f; z) \right] \geq \frac{c}{\sqrt{n \log n}} \mathbb{E} \inf_{\pi} \sup_{f \in \mathcal{F}} \int_0^1 \sqrt{\log \frac{1}{\pi(B_{\varrho_{n,\ell}}(f, \varepsilon))}} d\varepsilon - \frac{c' \sup_{\mathcal{F}} \mathbb{E}[\ell(f; z)]}{\sqrt{n \log n}}.$$

This inequality implies the following result

$$\mathbb{E} \left[ \sup_{\pi \in \Delta(\mathcal{F}), f \in \mathcal{F}} \left( (\mathbb{P} - \mathbb{P}_n) \ell(f; z) - \frac{c}{\sqrt{n \log n}} \int_0^1 \sqrt{\log \frac{1}{\pi(B_{\varrho_{n,\ell}}(f, \varepsilon))}} d\varepsilon \right) + \frac{c' \sup_{\mathcal{F}} \mathbb{E}[\ell(f; z)]}{\sqrt{n \log n}} \right] \geq 0,$$

where we have used the facts that $-\inf_x h(x) = \sup_x(-h(x))$ and $\sup_x h_1(x) - \sup_x h_2(x) \leq \sup_x(h_1(x) - h_2(x))$.

$\square$

## C.8 Background on Gaussian and Empirical Processes

It is now well understood that the supremum of Gaussian process can be tightly characterized by the majorizing measure integral via matching upper and lower bounds up to absolute constants (Fernique, 1975; Talagrand, 1987); the goal of this section is to extend this characterization to (1) bounded empirical processes and (2) a truncated form of integral.

**Background on Gaussian Processes.** We begin by recalling several key results from a series of seminal papers by Talagrand, Fernique, and others, which introduces the majorizing-measure formulation of the generic chaining framework (Fernique, 1975; Talagrand, 1987). Note that generic chaining have several equivament formulations (Talagrand, 2005), and the one closest to our purpose is through majorizing measure.

A *centered Gaussian random variable* $X$ is a real-valued measurable function on the outcome space such that the law of $X$ has density

$$(2\pi\sigma^2)^{-1/2} \exp\left(-\frac{x^2}{2\sigma^2}\right).$$

The law of $X$ is thus determined by $\sigma = (\mathbb{E}[X^2])^{1/2}$. If $\sigma = 1$, $X$ is called *standard normal*.

A *Gaussian process* is a family $\{X_t\}_{t \in T}$ of random variables indexed by some set $T$, such that every finite linear combination $\sum_{j=1}^{k} \alpha_j X_{t_j}$ is Gaussian. On the index set $T$, consider the semi-metric $\varrho$ given by

$$\varrho(u, v) = \sqrt{\mathbb{E}[(X_u - X_v)^2]}. \tag{57}$$

Gaussian processes are thus a very rigid class of stochastic processes, with exceptionally nice properties that have been fully developed in the literature.

Fernique (1975) proved the following integral upper bound.

**Lemma 11 (Upper Bound of Gaussian Processes via Majorizing Measure, Fernique (1975))**
*Given a Gaussian process $(X_t)_{t \in T}$ with its metric $\varrho$ defined by (57), we have*

$$\mathbb{E}\left[\sup_{t \in T} X_t\right] \leq C \inf_{\pi \in \Delta(T)} \sup_{t \in T} \int_0^\infty \sqrt{\log \frac{1}{\pi(B_\varrho(t, \varepsilon))}} d\varepsilon,$$

*where $C > 0$ is an absolute constant.*

A prior $\pi$ that makes the right hand side in Lemma 11 finite is called a *majorizing measure*. Fernique conjectured as early as 1974 that the existence of majorizing measures might characterize the boundedness of Gaussian processes. He proved a number of important partial results, and his determination eventually motivated the Talagrand to attack the problem in 1987. Talagrand (1987) proved that the integral in Lemma 11 is tight up to absolute constants; the upper bound in Lemma 11 is thus called the Fernique-Talagrand (majorizing measure) integral.

**Lemma 12 (Lower Bound of Gaussian Processes via Majorzing Measure, Talagrand (1987))**
*Given a Gaussian process $(X_t)_{t \in T}$ with its metric $\varrho$ defined by (57), we have*

$$\mathbb{E}\left[\sup_{t \in T} X_t\right] \geq c \inf_{\pi \in \Delta(T)} \sup_{t \in T} \int_0^\infty \sqrt{\log \frac{1}{\pi(B_{\varrho_2}(t, \varepsilon))}} d\varepsilon,$$

*where $c > 0$ is an absolute constant.*

Thus the Fernique-Talagrand integral gives a complete characterization to the supremum of Gaussian process.

**Background on Empirical Processes.** We now give several results on upper and lower bounding empirical process by Rademacher and Gaussian complexities Giné & Zinn (1984); Bartlett & Mendelson (2002).

**Definition 2 (Rademacher and Gaussian complexities)** *For a function class $\mathcal{F}$ that consists of mappings from $\mathcal{Z}$ to $\mathbb{R}$, define the Rademacher complexity of $\mathcal{F}$ as*

$$R_n(\mathcal{F}) := \mathbb{E}_{z,\xi}\left[\sup_{f \in \mathcal{F}} \frac{1}{n} \sum_{i=1}^n \xi_i f(z_i)\right],$$

*where $\{\xi_i\}_{i=1}^n$ are i.i.d. Rademacher variables; and define the Gaussian complexity of $\mathcal{F}$ as*

$$G_n(\mathcal{F}) := \mathbb{E}_{z,g}\left[\sup_{f \in \mathcal{F}} \frac{1}{n} \sum_{i=1}^n g_i f(z_i)\right],$$

*where $\{g_i\}_{i=1}^n$ are i.i.d. standard Gaussian variables.*

It is well-known that Rademacher and Gaussian complexities are upper bounds of empirical processes (see, e.g., Lemma 7.4 in Van Handel (2014)):

**Lemma 13 (Upper Bounds with Rademacher and Gaussian Complexities)** *For any function class $\mathcal{F}$ that consists of mappings from $\mathcal{Z}$ to $\mathbb{R}$, we have*

$$\mathbb{E}\left[\sup_{f \in \mathcal{F}} (\mathbb{P} - \mathbb{P}_n) f(z)\right] \leq 2R_n(\mathcal{F}) \leq \sqrt{2\pi} G_n(\mathcal{F}),$$

*where $R_n(\mathcal{F})$ and $G_n(\mathcal{F})$ are (expected) Rademacher and Gaussian complexities defined in Definition 2.*

We state a truncated form of the Fernique-Talagrand integral, adapted from Theorem 3 of Block et al. (2021), and use it in the proof of Theorem 1. Up to absolute constants, this truncated form is equivalent to the classical (nontruncated) Fernique-Talagrand integral; throughout, we interpret

both forms as placing the $\inf_\pi$ and $\sup_{f \in \mathcal{F}}$ outside the integral.[2] The truncated variant is often more convenient for deriving tighter relaxations—for example, when fixing a particular prior $\pi$ rather than taking $\inf_\pi$, as used in Theorem 1.

**Lemma 14 (Truncated integral bound)** *Given a function class $\mathcal{F}$ that consists of mappings from $\mathcal{Z}$ to $[0, 1]$. Define the empirical $L_2(\mathbb{P}_n)$ pseudometric*

$$\rho_n(f_1, f_2) := \sqrt{\frac{1}{n} \sum_{i=1}^{n} (f_1(z_i) - f_2(z_i))^2}.$$

*There exists an absolute constant $C > 0$ such that*

$$\mathbb{E}_\xi \left[ \sup_{f \in \mathcal{F}} \frac{1}{n} \sum_{i=1}^{n} \xi_i f(z_i) \right] \le C \inf_{\alpha \ge 0} \left\{ \alpha + \frac{1}{\sqrt{n}} \inf_{\pi \in \Delta(\mathcal{F})} \sup_{f \in \mathcal{F}} \int_\alpha^1 \sqrt{\log \frac{1}{\pi(B_{\rho_n}(f, \varepsilon))}} \, d\varepsilon \right\},$$

*where $\{\xi_i\}_{i=1}^{n}$ are i.i.d. Rademacher variables, and the left hand side of the above inequality is called the empirical Rademacher complexity.*

**Remarks.** (i) Because $f \in [0, 1]$, the diameter of $\mathcal{F}$ with $\rho_n$ is bounded by 1, which justifies truncating the integral at 1 and adding the small–scale term $\alpha$. (ii) An analogous bound holds for Gaussian processes; we state the Rademacher version since it directly controls empirical processes via symmetrization and is what we need for Theorem 1. (iii) The proof of Lemma 14 is a straightforward adaptation of Theorem 3 in Block et al. (2021), specializing their sequential argument to the classical i.i.d. setting (with only minor notational changes).

The following result illustrate that Gaussian and Rademacher complexities can also be used to lower bounding empirical processes.

**Lemma 15 (Lower Bounds with Rademacher and Gaussian Complexities)** *For any function class $\mathcal{F}$ that consists of mappings from $\mathcal{Z}$ to $\mathbb{R}$, defined its centered class $\tilde{\mathcal{F}}$ as $\{f - \mathbb{E}[f(z)] : f \in \mathcal{F}\}$. We have*

$$\mathbb{E} \left[ \sup_{f \in \mathcal{F}} (\mathbb{P} - \mathbb{P}_n) f(z) \right] \ge \frac{1}{2} R_n(\tilde{\mathcal{F}}) \ge \frac{c}{\sqrt{\log n}} G_n(\tilde{\mathcal{F}}),$$

*where $c > 0$ is an absolute constant.*

**Proof of Lemma 15:** Both the fact that uniform convergence admit a lower bound in terms of the Rademacher complexity of the centered class, and the result that Rademacher complexity itself is bounded below by Gaussian complexity up to a factor of $\sqrt{\log n}$, are classical and admit simple proofs. For a full proof of the first inequality, see Theorem 14.3 in Rinaldo & Yan (2016); for a reference and proof sketch of the second inequality, see Problem 7.1 in Van Handel (2014).

$\square$

**Basic Concentration Inequalities.** We state Mcdiarmid's inequality, Hoeffding's inquality, and Bernstein's inequality.

**Lemma 16 (McDiarmid's inequality (bounded differences), McDiarmid (1998))** *Let $Z_1, \ldots, Z_n$ be independent random variables with $Z_i \in \mathcal{Z}_i$. Let $h : \mathcal{Z}_1 \times \cdots \times \mathcal{Z}_n \to \mathbb{R}$ be a measurable function satisfying the bounded difference property: there are constants $c_1, \ldots, c_n \ge 0$ such that for all $i \in \{1, \cdots, n\}$ and all $Z_1 \in \mathcal{Z}_1, \cdots, Z_n \in \mathcal{Z}_n$,*

$$\sup_{Z_i' \in \mathcal{Z}_i} \left| h(Z_1, \cdots, Z_{i-1}, Z_i, Z_{i+1}, \cdots, Z_n) - h(Z_1, \cdots, Z_{i-1}, Z_i', Z_{i+1}, \cdots, Z_n) \right| \le c_i.$$

---

[2]Sketch: for the $\gamma_2$ functional, one may cap the chaining diameter at 1 at any scale $\alpha \in (0, 1]$, absorbing finer scales into an additive $\alpha$ term. By the standard equivalences among the $\gamma_2$ functional, admissible trees, and the Fernique–Talagrand integral (see §6.2 of Talagrand (2014)), the truncated and nontruncated forms are equivalent up to absolute constants.

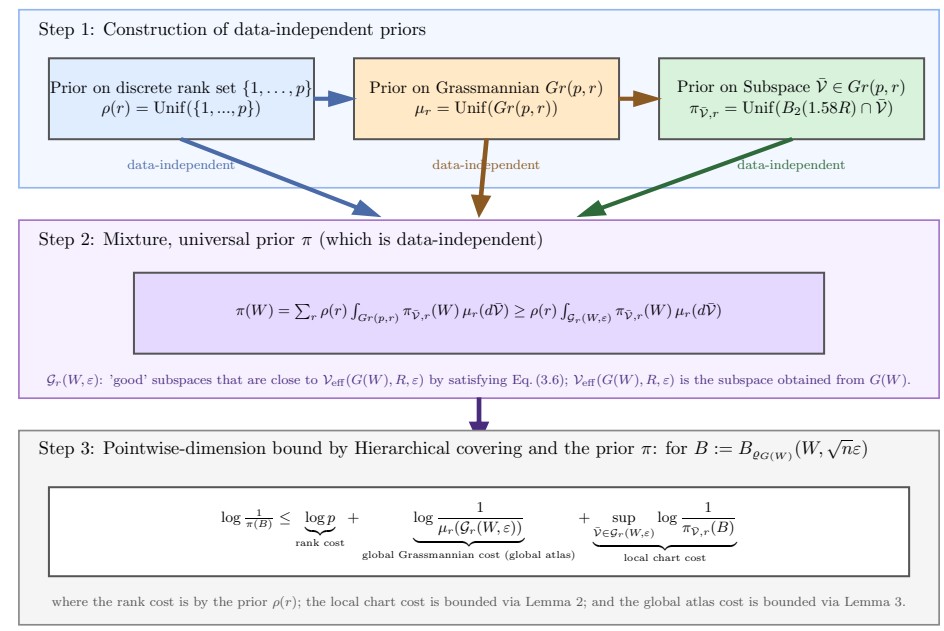

Figure 3: Hierarchical construction of the data-independent prior $\pi$ and its role in the pointwise-dimension bound (one single-layer case).

*Then for every $t \geq 0$,*

$$\Pr\big(h(Z_1, \cdots, Z_n) - \mathbb{E}[h(Z_1, \cdots, Z_n)] \geq t\big) \leq \exp\left(-\frac{2t^2}{\sum_{i=1}^n c_i^2}\right).$$

**Lemma 17 (Hoeffding's inequality, Chapter 2 in Vershynin (2018))** *Let $Z_1, \cdots, Z_n$ be independent random variables with $a_i \leq Z_i \leq b_i$ almost surely. Then for every $t \geq 0$,*

$$\Pr\left(\sum_{i=1}^n Z_i - \mathbb{E}[Z] \geq t\right) \leq \exp\left(-\frac{2t^2}{\sum_{i=1}^n (b_i - a_i)^2}\right).$$

**Lemma 18 (Bernstein's inequality, Chapter 2 in Vershynin (2018))** *Let $Z_1, \cdots, Z_n$ be independent mean–zero random variables with $|Z_i| \leq M$ almost surely. Then for every $t \geq 0$,*

$$\Pr\left(\sum_{i=1}^n Z_i \geq t\right) \leq \exp\left(-\frac{\frac{1}{2} t^2}{\sum_{i=1}^n \mathbb{E}[Z_i^2] + \frac{1}{3} M t}\right).$$

# D  FURTHER EXPLANATIONS AND PROOFS FOR DEEP NEURAL NETWORKS AND RIEMANNIAN DIMENSION (SECTION 3)

## D.1  ILLUSTRATIVE FIGURES

For intuition, we illustrate the construction of the prior $\pi$ in the single-layer case—via the schematic in Figure 3. From a top-down view, the prior $\pi$ can be generated by first sampling the effective rank $r$, then a subspace $\bar{\mathcal{V}}$ on the Grassmannian, and finally a weight $W$ inside that subspace. The general $L$-layer setting is then obtained by applying the same construction independently to each layer and taking a product measure, which is enabled by the layer-wise decomposable structure of neural networks (a consequence of our non-perturbative analysis).

## D.2 Proof of Lemma 1 (Non-Perturbative Feature Expansion)

We start with the telescoping decomposition presented in the main paper, which serves as a non-perturvative replacement of conventional Taylor expansion, where in each summand the only difference lie in $W_l'$ and $W_l$.

$$F_L(W', X) - F_L(W, X)$$

$$= \sum_{l=1}^{L} [\sigma_L(\underbrace{W_L' \cdots W_{l+1}'}_{\text{controled by } M_{l \to L}} \underbrace{\sigma_l}_{\text{by 1}}(W_l' \underbrace{F_{l-1}(W, X)}_{\text{learned feature}})) - \sigma_L(W_L' \cdots W_{l+1}' \sigma_l(W_l \underbrace{F_{l-1}(W, X)}_{\text{learned feature}}))],$$

Applying Cauchy-Schwartz inequality to the above identity, we have

$$\|F(W', X) - F(W, X)\|_{\mathsf{F}}^2 \tag{58}$$

$$\leq \sum_{l=1}^{L} L \|\sigma_L(W_L' \cdots W_{l+1}' \sigma_l(W_l' F_{l-1}(W, X))) - \sigma_L(W_L' \cdots W_{l+1}' \sigma_l(W_l F_{l-1}(W, X)))\|_{\mathsf{F}}^2 \tag{59}$$

By the definition of local Lipschitz constant in Section 3, for all $W' \in B_{\varrho_n}(W, \varepsilon)$,

$$\|\sigma_L(W_L' \cdots W_{l+1}' \sigma_l(W_l' F_{l-1}(W, X))) - \sigma_L(W_L' \cdots W_{l+1}' \sigma_l(W_l F_{l-1}(W, X)))\|_{\mathsf{F}}$$

$$\leq M_{l \to L}[W, \varepsilon] \|\sigma_l(W_l' F_{l-1}(W, X)) - \sigma_l(W_l F_{l-1}(W, X))\|_{\mathsf{F}}. \tag{60}$$

Because the activation function $\sigma_l$ is $1-$Lipschitz for each column, we have

$$\|\sigma_l(W_l' F_{l-1}(W, X)) - \sigma_l(W_l F_{l-1}(W, X))\|_{\mathsf{F}} \leq \|(W_l' - W_l) F_{t-1}(W, X)\|_{\mathsf{F}}. \tag{61}$$

Combining (58) (60) and (61), we prove that

$$\|F(W', X) - F(W, X)\|_{\mathsf{F}}^2 \leq \sum_{l=1}^{L} L \cdot M_{l \to L}[W, \varepsilon]^2 \cdot \|(W_l' - W_l) F_{l-1}(W, X)\|_{\mathsf{F}}^2.$$

$\square$

## D.3 Metric Domination Lemma

Our non-perturbative expansion facilitates bounding the pointwise dimension of complex geometries via metric comparison. By constructing a simpler, dominating metric (i.e., one that is pointwise larger), we establish that the pointwise dimension of the original geometry is upper bounded by that of this new, more structured geometry. This "enlargement" for analytical tractability, a concept with roots in comparison geometry and majorization principles, is operationalized in Lemma 19.

**Lemma 19 (Metric Domination Lemma)** *For two metrics $\varrho_1, \varrho_2$ defined on $\mathbb{R}^p$, if $\varrho_1(W', W) \leq \varrho_2(W', W)$ for all $W' \in B_{\varrho_2}(W, \varepsilon)$, then for any prior $\pi \in \Delta(\mathbb{R}^p)$ and any $\varepsilon > 0$, we have*

$$\log \frac{1}{\pi(B_{\varrho_1}(W, \varepsilon))} \leq \log \frac{1}{\pi(B_{\varrho_2}(W, \varepsilon))}.$$

**Proof of Lemma 19:** Because $\varrho_1(W', W) \leq \varrho_2(W', W)$ for all $W' \in B_{\varrho_2}(W, \varepsilon)$, we have that

$$B_{\varrho_1}(W, \varepsilon) \supseteq B_{\varrho_2}(W, \varepsilon).$$

So for any prior $\pi$ on $\mathbb{R}^p$, monotonicity of measures gives

$$\pi(B_{\varrho_1}(W, \varepsilon)) \geq \pi(B_{\varrho_2}(W, \varepsilon)),$$

this implies

$$\log \frac{1}{\pi(B_{\varrho_1}(W, \varepsilon))} \leq \log \frac{1}{\pi(B_{\varrho_2}(W, \varepsilon))}.$$

$\square$

We then state an extension of the metric domination lemma, which turns pointwise dimension in a high-dimensional space into a lower-dimensional subspace.

**Lemma 20 (Subspace Metric Domination Lemma)** *Given a metric $\varrho_1$ defined on $\mathbb{R}^p$ a subspace $\mathcal{V} \subseteq \mathbb{R}^p$, and a metric $\varrho_2$ defined on $\mathcal{V}$. Define the orthogonal projector to subspace $\mathcal{V}$ as $\mathcal{P}_{\mathcal{V}}(W) := \arg\min_{\tilde{W} \in \mathcal{V}} \|\tilde{W} - W\|_2$. If there exists $\varepsilon_1 \in (0, \varepsilon)$ such that for every $W' \in \mathcal{V}$,*

$$(\varrho_1(W', W))^2 \leq (\varrho_2(W', \mathcal{P}_{\mathcal{V}}(W)))^2 + \varepsilon_1^2, \tag{62}$$

*then for any prior $\pi \in \Delta(\mathcal{V})$, we have*

$$\log \frac{1}{\pi(B_{\varrho_1}(W, \varepsilon))} \leq \log \frac{1}{\pi(B_{\varrho_2}(\mathcal{P}_{\mathcal{V}}(W), \sqrt{\varepsilon^2 - \varepsilon_1^2}))}. \tag{63}$$

**Proof of Lemma 20:** By the condition (62), we know

$$B_{\varrho_1}(W, \varepsilon) \supseteq B_{\varrho_1}(W, \varepsilon) \cap \mathcal{V} \supseteq B_{\varrho_2}(\mathcal{P}_{\mathcal{V}}(W), \sqrt{\varepsilon^2 - \varepsilon_1^2}),$$

and this gives the desired conclusion (63) in Lemma 20.

$\square$

### D.4 POINTWISE DIMENSION BOUND WITH REFERENCE SUBSPACE

**Set Up of Reference Effective Subspace**   Consider the weight space $B_2(R) \subset \mathbb{R}^p$ for vectorized weights $W$, where $B_2(R) := \{w \in \mathbb{R}^p : \|w\|_2 \leq R\}$. Given any fixed $p \times p$ PSD matrix $G(W)$, order the eigenvalues $\lambda_1(G(W)), \cdots, \lambda_p(G(W))$ nonincreasingly. For notational convenience, we suppress the dependence on $G(W)$ and write simply $\lambda_k$ when no confusion can arise. We denote $\mathcal{V}_{\text{eff}}(G(W)), R, \varepsilon)$ to be the *effective subspace*—the true top-$r_{\text{eff}}$ eigenspace—of $G(W)$. For notiaional convenience, we use $r_{\text{eff}}$ as the abbreviation of $r_{\text{eff}}(G(W), R, \varepsilon)$, and $\mathcal{V}$ as an abbreviation of $\mathcal{V}_{\text{eff}}(G(W)), R, \varepsilon)$ when no confusion can arise.

Assume there is another $r-$dimensional subspace $\bar{\mathcal{V}}$. We will show that if $\bar{\mathcal{V}}$ approximates $\mathcal{V}$, then using a prior supported on $\bar{\mathcal{V}}$ still yields a valid effective-dimension bound. This observation underpins the hierarchical covering argument in Theorem 3. For a self-contained introduction to subspaces (collectively known as the Grassmannian) and their frame parameterizations (the Stiefel manifold); see Section E.1, where we translate algebraic and differential-geometric insights into machine learning terminology.

**Motivation of Approximate Effective Subspace.**   We can view the orthogonal projector to a subspace as a matrix (see the definition via the Stiefel parameterization in (87)), which is consistent with the earlier operator notation characterized by $\ell_2$–distance in Lemma 20. Now define the projected metric $\varrho_{G(W)}^{\bar{\mathcal{V}}}$ as

$$\varrho_{G(W)}^{\bar{\mathcal{V}}}(W_1, W_2) = \sqrt{(\mathcal{P}_{\bar{\mathcal{V}}}(W_1) - \mathcal{P}_{\bar{\mathcal{V}}}(W_1))^\top G(W)(\mathcal{P}_{\bar{\mathcal{V}}}(W_2) - \mathcal{P}_{\bar{\mathcal{V}}}(W_2))} = \sqrt{(W_1 - W_2)^\top \mathcal{P}_{\bar{\mathcal{V}}}^\top G(W) \mathcal{P}_{\bar{\mathcal{V}}}(W_1 - W_2)}.$$

By the subspace metric dominance lemma (Lemma 20), if $\mathcal{P}_{\bar{\mathcal{V}}}^\top G(W) \mathcal{P}_{\bar{\mathcal{V}}}$ approximates $G(W)$, we can use prior over $\bar{\mathcal{V}}$ to bound the pointwise dimension and achieve dimension reduction.

We will require the following approximation error condition:

$$\varrho_{\text{proj}, G(W)}(\mathcal{V}, \bar{\mathcal{V}}) = \|G(W)^{\frac{1}{2}}(\mathcal{P}_{\mathcal{V}} - \mathcal{P}_{\bar{\mathcal{V}}})\|_{\text{op}} \leq \frac{\sqrt{n}\varepsilon}{4R}.$$

In Section E, we systematically study the ellipsoidal covering of Grassmannian, and establish that we can *always* find $\bar{\mathcal{V}}$ that approximates $\mathcal{V}$ to the desired precision, with an additional covering cost of the Grassmannain bound in the Riemannian Dimension. This generalizes the canonical projection metric between subspaces into ellipsoidal set-up.

**Effective Dimension Bound for Approximate Effective Subspace.**   We now present the lemma that establish effective dimension bound using prior supported on approximate effective subspace $\bar{\mathcal{V}}$ (not necessarily the true effective subspace $\mathcal{V}_{\text{eff}}(G(W)), R, \varepsilon)$). We state the main result of this subsection (Lemma 2 in the main paper).

Consider the weight space $B_2(R) \subset \mathbb{R}^p$ for vectorized weights, and a pointwise ellipsoidal metric defined via PSD $G(W)$. Let $\mathcal{V} \subseteq \mathbb{R}^p$ be a fixed $r$-dimensional subspace. Define the prior $\pi_{\bar{\mathcal{V}}} = \text{Unif}(B_2(1.58R) \cap \bar{\mathcal{V}})$. Then, uniformly over all $(W, \varepsilon)$ such that the top-$r$ eigenspace $\mathcal{V}$ of $G(W)$ can be approximated by $\bar{\mathcal{V}}$ to precision

$$\varrho_{\text{proj}, G(W)}(\mathcal{V}, \bar{\mathcal{V}}) := \left\| G(W)^{1/2} \left( \mathcal{P}_{\mathcal{V}} - \mathcal{P}_{\bar{\mathcal{V}}} \right) \right\|_{\text{op}} \leq \frac{\sqrt{n}\varepsilon}{4R}, \tag{64}$$

we have

$$\log \frac{1}{\pi_{\bar{\mathcal{V}}}(B_{\varrho_{G(W)}}(W, \sqrt{n}\varepsilon))} \leq \frac{1}{2} \sum_{k=1}^{r_{\text{eff}}(G(W), R, \varepsilon)} \log \left( \frac{40R^2 \lambda_k(G(W))}{n\varepsilon^2} \right) = d_{\text{eff}}(G(W), \sqrt{5}R, \varepsilon).$$

**Proof of Lemma 2:** Given a fixed PSD matrix $G(W)$ with eigenvalues $\lambda_1 \geq \cdots \geq \lambda_p$, denote $r_{\text{eff}} = r_{\text{eff}}(G(W), R, \varepsilon)$, and the projected metric $\varrho_{G(W)}^{\bar{\mathcal{V}}}$ on $\bar{\mathcal{V}}$:

$$\varrho_{G(W)}^{\bar{\mathcal{V}}}(W_1, W_2) = \sqrt{(W_1 - W_2)^\top \mathcal{P}_{\bar{\mathcal{V}}}^\top G(W) \mathcal{P}_{\bar{\mathcal{V}}}(W_1 - W_2)}.$$

Since $\mathcal{V}$ is the top-$r_{\text{eff}}$ eigenspace of $G(W)$, by the elementary property of eigendecomposition we have that

$$\begin{aligned} G(W) =& \mathcal{P}_{\mathcal{V}}^\top G(W) \mathcal{P}_{\mathcal{V}} + \mathcal{P}_{\mathcal{V}_\perp}^\top G(W) \mathcal{P}_{\mathcal{V}_\perp} \\ \preceq& \mathcal{P}_{\mathcal{V}}^\top G(W) \mathcal{P}_{\mathcal{V}} + \lambda_{r_{\text{eff}}+1} \cdot \mathcal{P}_{\mathcal{V}_\perp}^\top \mathcal{P}_{\mathcal{V}_\perp}, \end{aligned} \tag{65}$$

where $\mathcal{V}_\perp$ is orthogonal complement of $\mathcal{V}$. It is also straightforward to see

$$\mathcal{P}_{\mathcal{V}}^\top G(W) \mathcal{P}_{\mathcal{V}} \preceq 2\mathcal{P}_{\bar{\mathcal{V}}}^\top G(W) \mathcal{P}_{\bar{\mathcal{V}}} + 2(\mathcal{P}_{\mathcal{V}} - \mathcal{P}_{\bar{\mathcal{V}}})^\top G(W)(\mathcal{P}_{\mathcal{V}} - \mathcal{P}_{\bar{\mathcal{V}}}). \tag{66}$$

Combining (65) and (66), we have the fundamental loewner order inequality

$$G(W) \preceq 2\mathcal{P}_{\bar{\mathcal{V}}}^\top G(W) \mathcal{P}_{\bar{\mathcal{V}}} + 2(\mathcal{P}_{\mathcal{V}} - \mathcal{P}_{\bar{\mathcal{V}}})^\top G(W)(\mathcal{P}_{\mathcal{V}} - \mathcal{P}_{\bar{\mathcal{V}}}) + \lambda_{r_{\text{eff}}+1} \cdot \mathcal{P}_{\mathcal{V}_\perp}^\top \mathcal{P}_{\mathcal{V}_\perp}. \tag{67}$$

In order to apply the subspace metric domination lemma (Lemma 20), we hope to bound $\|W' - W\|_2^2$ and apply that bound to the two last reminder terms in the right hand side of (67).

To bound $\|W' - W\|_2^2$, we firstly state the following lemma on the eigenvalue of $\mathcal{P}_{\bar{\mathcal{V}}}^\top G(W) \mathcal{P}_{\bar{\mathcal{V}}}$, whose proof is deferred until after the current proof.

**Lemma 21 (Eigenvalue Bound for Projected Metric Tensor)** *Assume $\mathcal{V}$ is the top-$r$ eigenspace of a PSD matrix $\Sigma$ with eigenvalues $\lambda_1 \geq \cdots \geq \lambda_p$, then for a $r-$dimensional subspace $\bar{\mathcal{V}}$ we have that for $k = 1, 2, \cdots, r$,*

$$\lambda_k \geq \lambda_k(\mathcal{P}_{\bar{\mathcal{V}}}^\top \Sigma \mathcal{P}_{\bar{\mathcal{V}}}) \geq \lambda_k/2 - \|\Sigma^{\frac{1}{2}}(\mathcal{P}_{\mathcal{V}} - \mathcal{P}_{\bar{\mathcal{V}}})\|_{\text{op}}^2.$$

For every $W' \in B_{\varrho_{G(W)}^{\bar{\mathcal{V}}}}(\mathcal{P}_{\bar{\mathcal{V}}}(W), \sqrt{n}\varepsilon/4)$, we have $\forall k = 1, \cdots, r_{\text{eff}}$,

$$\begin{aligned} \|W' - \mathcal{P}_{\bar{\mathcal{V}}}(W)\|_2^2 \leq& \frac{(W' - \mathcal{P}_{\bar{\mathcal{V}}}(W))^\top \mathcal{P}_{\bar{\mathcal{V}}}^\top G(W) \mathcal{P}_{\bar{\mathcal{V}}}(W' - \mathcal{P}_{\bar{\mathcal{V}}}(W))}{\lambda_{r_{\text{eff}}}(\mathcal{P}_{\bar{\mathcal{V}}}^\top G(W) \mathcal{P}_{\bar{\mathcal{V}}})} \leq \frac{n\varepsilon^2}{16\lambda_{r_{\text{eff}}}(\mathcal{P}_{\bar{\mathcal{V}}}^\top G(W) \mathcal{P}_{\bar{\mathcal{V}}})} \\ \leq& \frac{n\varepsilon^2}{8\lambda_{r_{\text{eff}}} - 16\|G(W)^{\frac{1}{2}}(\mathcal{P}_{\mathcal{V}} - \mathcal{P}_{\bar{\mathcal{V}}})\|_{\text{op}}^2} \leq \frac{1}{3}R^2, \end{aligned} \tag{68}$$

where the first inequality holds because if $A$ is a symmetric positive definite matrix, then for all vectors $x$, we have $x^\top A x \geq \lambda_{\min}(A)\|x\|_2^2$; the second inequality used the condition of $W' \in B_{\varrho_{G(W)}^{\bar{\mathcal{V}}}}(\mathcal{P}_{\bar{\mathcal{V}}}(W), \sqrt{n}\varepsilon/4)$; the third inequality uses Lemma 21; and the last inequality uses $\lambda_{r_{\text{eff}}} \geq \frac{n\varepsilon^2}{2R^2}$ (by definition (10) of effective rank) and the approximation error condition (64). On the other hand, we have that $\|W\|_2^2 \leq R^2$, so that for every $W' \in B_{\varrho_{G(W)}^{\bar{\mathcal{V}}}}(\mathcal{P}_{\bar{\mathcal{V}}}(W), \sqrt{n}\varepsilon/4)$

$$\|W' - W\|_2^2 = \|W' - \mathcal{P}_{\bar{\mathcal{V}}}(W)\|_2^2 + \|\mathcal{P}_{\bar{\mathcal{V}}_\perp}(W)\|_2^2 \leq \frac{4}{3}R^2,$$

combined with (68).

From the fundamental loewner order inequality (67), we establish the desired metric domination condition: for all $W' \in B_{\varrho^{\bar{\mathcal{V}}}_{G(W)}}(\mathcal{P}_{\bar{\mathcal{V}}}(W), \sqrt{n}\varepsilon/4)$ and $W \in B_2(R)$,

$$(W' - W)^\top G(W)(W' - W)$$
$$\leq (W' - W)^\top (2\mathcal{P}_{\bar{\mathcal{V}}}^\top G(W)\mathcal{P}_{\bar{\mathcal{V}}})(W' - W) + (2\|G(W)^{\frac{1}{2}}(\mathcal{P}_{\mathcal{V}} - \mathcal{P}_{\bar{\mathcal{V}}})\|_{op}^2 + \lambda_{r_{\text{eff}}+1})\|W' - W\|_2^2$$
$$\leq 2\varrho^{\bar{\mathcal{V}}}_{G(W)}(W', \mathcal{P}_{\bar{\mathcal{V}}}(W))^2 + \frac{5n\varepsilon^2}{6},$$

where the first inequality holds because of the loewner order inequality (67) and the property of operator norm: $x^\top Ax \leq \|A\|_{op} \cdot \|x\|_2^2$ (one could also apply Lemma 21 to validate $\|\mathcal{P}_{\mathcal{V}_\perp}^\top \mathcal{P}_{\mathcal{V}_\perp}\|_{op} \leq 1$); and the last inequality uses the fact $\lambda_{r_{\text{eff}}+1} < \frac{n\varepsilon^2}{2R^2}$ (by definition 10 of effective rank) and the approximation error condition (64). Now we can apply the subspace metric domination lemma (Lemma 20) and obtain: for any $\pi \in \Delta(\bar{\mathcal{V}})$,

$$\log \frac{1}{\pi\left(B_{\varrho_{G(W)}}(W, \sqrt{n}\varepsilon)\right)} \leq \log \frac{1}{\pi\left(B_{\sqrt{2}\varrho^{\bar{\mathcal{V}}}_{G(W)}}(\mathcal{P}_{\bar{\mathcal{V}}}(W), \sqrt{n}\varepsilon/\sqrt{6})\right)} \leq \log \frac{1}{\pi\left(B_{\varrho^{\bar{\mathcal{V}}}_{G(W)}}(\mathcal{P}_{\bar{\mathcal{V}}}(W), \sqrt{n}\varepsilon/4)\right)}. \tag{69}$$

In particular, we choose $\pi$ to be the uniform prior over $\bar{\mathcal{V}}$:

$$\pi_{\bar{\mathcal{V}}} = \text{Unif}(B_2(1.58R) \cap \bar{\mathcal{V}}).$$

Then we aim to prove that $B_{\varrho^{\bar{\mathcal{V}}}_{G(W)}}(\mathcal{P}_{\bar{\mathcal{V}}}(W), \sqrt{n}\varepsilon/4) \subseteq \bar{\mathcal{V}} \cap B_2(1.58R)$. This is true because: 1) for every $W' \in B_{\varrho^{\bar{\mathcal{V}}}_{G(W)}}(\mathcal{P}_{\bar{\mathcal{V}}}(W), \sqrt{n}\varepsilon/4)$, (68) suggests $\|W' - \mathcal{P}_{\bar{\mathcal{V}}}(W)\|_2^2 \leq \frac{1}{3}R^2$, and 2) for very $W \in B_2(R)$, we have $\|\mathcal{P}_{\bar{\mathcal{V}}}(W)\|_2 \leq \|W\|_2 \leq R$. Combining this and the above inequality we have

$$\|W'\|_2 \leq \|W' - \mathcal{P}_{\bar{\mathcal{V}}}(W)\|_2 + \|\mathcal{P}_{\bar{\mathcal{V}}}(W)\|_2 \leq (\sqrt{1/3} + 1)R < 1.58R.$$

This proves that $B_{\varrho^{\bar{\mathcal{V}}}_{G(W)}}(\mathcal{P}_{\bar{\mathcal{V}}}(W), \sqrt{n}\varepsilon/4) \subseteq \bar{\mathcal{V}} \cap B_2(1.58R)$, so we have

$$\log \frac{1}{\pi_{\bar{\mathcal{V}}}(B_{\varrho^{\bar{\mathcal{V}}}_{G(W)}}(\mathcal{P}_{\bar{\mathcal{V}}}(W), \sqrt{n}\varepsilon/4)} = \frac{\text{Vol}(\bar{\mathcal{V}} \cap B_2(1.58R))}{\text{Vol}(B_{\varrho^{\bar{\mathcal{V}}}_{G(W)}}(\mathcal{P}_{\bar{\mathcal{V}}}(W), \sqrt{n}\varepsilon/4))}. \tag{70}$$

By the change–of–variables theorem in multivariate calculus (Wikipedia contributors, 2025a), the linear map $T = G(W)^{\frac{1}{2}}$ implies the volume formula for ellipsoid $E = B_{\varrho^{\bar{\mathcal{V}}}_{G(W)}}(\mathcal{P}_{\bar{\mathcal{V}}}(W), \sqrt{n}\varepsilon/4)$ with dimension $r_{\text{eff}}$, eigenvalues $\{\lambda_k(\mathcal{P}_{\bar{\mathcal{V}}}^\top G(W)\mathcal{P}_{\bar{\mathcal{V}}})\}_{k=1}^{r_{\text{eff}}}$ and radius $\sqrt{n}\varepsilon/4$

$$\text{Vol}(E) = |\det T|^{-1}\text{Vol}(T(E)) = (\det G(W))^{-1/2}\text{Vol}(B_2(\sqrt{n}\varepsilon/4)) = \left(\prod_{k=1}^{r_{\text{eff}}} \lambda_k\right)^{-1/2}\text{Vol}(B_2(\sqrt{n}\varepsilon/4)),$$

Also by the change-of-variable theorem, we have that the volume of $r_{\text{eff}}-$dimensional isotropic ball $\mathcal{V} \cap B_2(2R)$ is

$$\text{Vol}(\bar{\mathcal{V}} \cap B_2(1.58R)) = \left(\frac{1.58R}{\sqrt{n}\varepsilon/4}\right)^{r_{\text{eff}}}\text{Vol}(B_2(\sqrt{n}\varepsilon/4)).$$

Hence, applying (69) (70) and combining it with the two above volume equalities, we have

$$\log \frac{1}{\pi_{\bar{\mathcal{V}}}\left(B_{\varrho_{G(W)}}(W, \sqrt{n}\varepsilon)\right)} \leq \log \frac{1}{\pi_{\bar{\mathcal{V}}}(B_{\varrho^{\bar{\mathcal{V}}}_{G(W)}}(\mathcal{P}_{\bar{\mathcal{V}}}(W), \sqrt{n}\varepsilon/4))} = \log \frac{\text{Vol}(\bar{\mathcal{V}} \cap B_2(1.58R))}{\text{Vol}(B_{\varrho^{\bar{\mathcal{V}}}_{G(W)}}(\mathcal{P}_{\bar{\mathcal{V}}}(W), \sqrt{n}\varepsilon/4))}$$
$$\leq \frac{1}{2}\log \frac{(1.58R)^{2r_{\text{eff}}}\prod_{k=1}^{r_{\text{eff}}} \lambda_k}{(\sqrt{n}\varepsilon/4)^{2r_{\text{eff}}}} \leq \frac{1}{2}\sum_{k=1}^{r_{\text{eff}}} \log \frac{40R^2\lambda_k}{n\varepsilon^2}$$
$$= d_{\text{eff}}(G(W), \sqrt{5}R, \varepsilon).$$

Finally, since the prior construction $\pi_{\bar{\mathcal{V}}} = \mathrm{Unif}(B_2(1.58R) \cap \bar{\mathcal{V}})$ only depends on $\bar{\mathcal{V}}$ rather than $W$ and $\varepsilon$, we have that uniformly over all $(W, \varepsilon) \in B_2(R) \times [0, \infty)$ such that $\bar{\mathcal{V}}$ approximates $\mathcal{V}_{\mathrm{eff}}(G(W), R, \varepsilon)$ to the precision (64),

$$\log \frac{1}{\pi_{\bar{\mathcal{V}}}(B_{\varrho_{G(W)}}(W, \sqrt{n}\varepsilon))} \leq d_{\mathrm{eff}}(G(W), \sqrt{5}R, \varepsilon),$$

which is the claimed bound. $\qquad\square$

**Proof of Lemma 21:** The Courant–Fischer–Weyl max-min characterization (Wikipedia contributors, 2025b) states that for any Hermitian (i.e. symmetric for real matrices studying here) matrix,

$$\lambda_k(\Sigma) = \max_{\substack{\mathcal{S} \subseteq \mathbb{R}^p \\ \dim \mathcal{S} = k}} \min_{\substack{W \in \mathcal{S} \\ W \neq 0}} \frac{W^\top \Sigma W}{\|W\|_2^2},$$

and we have that for any $r-$dimensional subspace $\bar{\mathcal{V}}$,

$$\lambda_k(\mathcal{P}_{\bar{\mathcal{V}}}^\top \Sigma \mathcal{P}_{\bar{\mathcal{V}}}) = \max_{\substack{\mathcal{S} \subseteq \bar{\mathcal{V}} \\ \dim \mathcal{S} = k}} \min_{\substack{W \in \mathcal{S} \\ W \neq 0}} \frac{W^\top \mathcal{P}_{\bar{\mathcal{V}}}^\top \Sigma \mathcal{P}_{\bar{\mathcal{V}}} W}{\|W\|_2^2},$$

so we have $\lambda_k(\mathcal{P}_{\bar{\mathcal{V}}}^\top \Sigma \mathcal{P}_{\bar{\mathcal{V}}}) \leq \lambda_k$ for $k = 1, 2, \cdots, r$.

Moreover, by the elementary property of eigendecomposition we have $\lambda_k = \lambda_k(\mathcal{P}_{\mathcal{V}}^\top \Sigma \mathcal{P}_{\mathcal{V}})$, and by the Courant–Fischer–Weyl max-min characterization we know that,

$$\begin{aligned}
\lambda_k(\mathcal{P}_{\mathcal{V}}^\top \Sigma \mathcal{P}_{\mathcal{V}}) &= \max_{\substack{\mathcal{S} \subseteq \mathbb{R}^p \\ \dim \mathcal{S} = k}} \min_{\substack{W \in \mathcal{S} \\ W \neq 0}} \frac{W^\top (\mathcal{P}_{\mathcal{V}}^\top \Sigma \mathcal{P}_{\mathcal{V}}) W}{\|W\|_2^2} \\
&\leq \max_{\substack{\mathcal{S} \subseteq \mathbb{R}^p \\ \dim \mathcal{S} = k}} \min_{\substack{W \in \mathcal{S} \\ W \neq 0}} \frac{W^\top (2\mathcal{P}_{\bar{\mathcal{V}}}^\top \Sigma \mathcal{P}_{\bar{\mathcal{V}}}) W + \|\mathcal{P}_{\mathcal{V}}^\top \Sigma \mathcal{P}_{\mathcal{V}} - 2\mathcal{P}_{\bar{\mathcal{V}}}^\top \Sigma \mathcal{P}_{\bar{\mathcal{V}}}\|_{\mathrm{op}} \|W\|_2^2}{\|W\|_2^2} \\
&= 2\lambda_k(\mathcal{P}_{\bar{\mathcal{V}}}^\top \Sigma \mathcal{P}_{\bar{\mathcal{V}}}) + \|\mathcal{P}_{\mathcal{V}}^\top \Sigma \mathcal{P}_{\mathcal{V}} - 2\mathcal{P}_{\bar{\mathcal{V}}}^\top \Sigma \mathcal{P}_{\bar{\mathcal{V}}}\|_{\mathrm{op}} \\
&\leq 2\lambda_k(\mathcal{P}_{\bar{\mathcal{V}}}^\top \Sigma \mathcal{P}_{\bar{\mathcal{V}}}) + 2\|(\mathcal{P}_{\mathcal{V}} - \mathcal{P}_{\bar{\mathcal{V}}})^\top \Sigma (\mathcal{P}_{\mathcal{V}} - \mathcal{P}_{\bar{\mathcal{V}}})\|_{\mathrm{op}},
\end{aligned}$$

where the first inequality is because for every fixed $S$ and $W$ we have $W^\top (\mathcal{P}_{\mathcal{V}}^\top \Sigma \mathcal{P}_{\mathcal{V}}) W \leq W^\top (2\mathcal{P}_{\bar{\mathcal{V}}}^\top \Sigma \mathcal{P}_{\bar{\mathcal{V}}}) W + \|\mathcal{P}_{\mathcal{V}}^\top \Sigma \mathcal{P}_{\mathcal{V}} - 2\mathcal{P}_{\bar{\mathcal{V}}}^\top \Sigma \mathcal{P}_{\bar{\mathcal{V}}}\|_{\mathrm{op}} \|W\|_2^2$; and the last inequality is due to (66). Therefore we have

$$\lambda_k(\mathcal{P}_{\bar{\mathcal{V}}}^\top \Sigma \mathcal{P}_{\bar{\mathcal{V}}}) \geq \lambda_k/2 - \|\Sigma^{\frac{1}{2}}(\mathcal{P}_{\mathcal{V}} - \mathcal{P}_{\bar{\mathcal{V}}})\|_{\mathrm{op}}^2.$$

$\qquad\square$

## D.5 Proof of Riemannian Dimension Bound for DNN (Theorem 3)

In the language of Riemannian geometry (Jost, 2008), we regard a pointwise PSD, matrix-valued function $G(W)$ as a (possibly degenerate) *metric tensor*; such a $G(W)$ endows the parameter space $\mathbb{R}^{\sum_{l=1}^L d_{l-1}d_l}$ with a (semi-)Riemannian manifold structure. The pointwise ellipsoidal metric in (9) belongs to the following family of block-decomposable metric tensors.

**Definition 3 (Metric Tensor of NN-surrogate Type)** *A metric tensor $G(W)$ (pointwise PSD-valued function of size $\sum_{l=1}^L d_{l-1}d_l \times \sum_{l=1}^L d_{l-1}d_l$) is of "NN-surrogate" type if $G(W)$ is in the form*

$$G(W) = \mathrm{blockdiag}(A_1(W) \otimes I_{d_1}, \cdots, A_l(W) \otimes I_{d_l}, \cdots, A_L(W) \otimes I_{d_L})$$

*where $A_l(W) \in \mathbb{R}^{d_{l-1} \times d_{l-1}}$.*

By Lemma 1, the non-perturbative feature expansion gives rise to the metric tensor $G_{\mathrm{NP}}(W)$ defined in (9); $G_{\mathrm{NP}}(W)$ belongs to the "NN-surrogate" class. We first record some elementary decomposition properties for this family of NN-surrogate metric tensors, and then prove Theorem 3.

### D.5.1 Decomposition Properties of NN-surrogate Metric Tensor

The NN-surrogate metric tensor $G(W)$ in Definition 3 has decomposition properties described by the next lemma.

**Lemma 22 (Decomposition Properties of NN-surrogate Metric Tensor)** *Given a NN-surrogate metric tensor $G(W)$ defined in Definition 3, for every $W$, we have the following decomposition properties: First, the effective rank and dimension decompose to*

$$r_{\text{eff}}(G(W), R, \varepsilon) = \sum_{l=1}^{L} d_l \cdot r_{\text{eff}}(A_l(W), R, \varepsilon);$$

$$d_{\text{eff}}(G(W), R, \varepsilon) = \sum_{l=1}^{L} d_l \cdot d_{\text{eff}}(A_l(W), R, \varepsilon).$$

*Second, denote $\mathcal{V}_{\text{eff}}(A_l(W), R, \varepsilon)$ the effective subspace (i.e., the top-$r_{\text{eff}}(A_l(W), R, \varepsilon)$ eigenspace) of $A_l(W)$. Then the effective subspace of $G(W)$ is*

$$\mathcal{V}_{\text{eff}}(G(W), R, \varepsilon) = \mathcal{V}_{\text{eff}}(A_1(W), R, \varepsilon)^{d_1} \times \cdots \times \mathcal{V}_{\text{eff}}(A_L(W), R, \varepsilon)^{d_L}.$$

**Proof of Lemma 22.** It is straightforward to see that, first, the effective rank of the fixed matrix $G(W)$ is

$$
\begin{aligned}
& r_{\text{eff}}(G(W), R, \varepsilon) \\
&= \max\{k : 2\lambda_k(G(W))R^2 \geq n\varepsilon^2\} \\
&= \sum_{l=1}^{L} \max\{k : 2\lambda_k(A_l(W) \otimes I_{d_l})R^2 \geq n\varepsilon^2\} \\
&= \sum_{l=1}^{L} d_l \max\{k : 2\lambda_k(A_l(W))R^2 \geq n\varepsilon^2\} \\
&= \sum_{l=1}^{L} d_l \cdot r_{\text{eff}}(A_l(W), R, \varepsilon);
\end{aligned}
$$

and the effective dimension of the fixed matrix $G(W)$ is

$$
\begin{aligned}
& d_{\text{eff}}(G(W), R, \varepsilon) \\
&= \frac{1}{2} \sum_{k=1}^{r_{\text{eff}}(G(W),R,\varepsilon)} \log\left(\frac{8R^2\lambda_k(G(W))}{n\varepsilon^2}\right) \\
&= \sum_{l=1}^{L} \frac{1}{2} \sum_{k=1}^{r_{\text{eff}}(A_l(W)\otimes I_{d_l},R,\varepsilon)} \log\left(\frac{8R^2\lambda_k(A_l(W) \otimes I_{d_l})}{n\varepsilon^2}\right) \\
&= \sum_{l=1}^{L} d_l \cdot \frac{1}{2} \sum_{k=1}^{r_{\text{eff}}(A_l(W),R,\varepsilon)} \log\left(\frac{8R^2\lambda_k(A_l(W))}{n\varepsilon^2}\right) \\
&= \sum_{l=1}^{L} d_l \cdot d_{\text{eff}}(A_l(W), R, \varepsilon).
\end{aligned}
$$

Second, as the effective subspace of the matrix tensor product $A_l(W) \otimes I_{d_l}$ is subspace tensor product $\mathcal{V}_{\text{eff}}(A_l(W), R, \varepsilon)^{d_l}$, the effective subspace for NN-surrogate metric tensor $G(W) = \text{blockdiag}(\cdots; A_l(W) \otimes I_{d_l}; \cdots)$ is

$$\mathcal{V}_{\text{eff}}(G(W), R, \varepsilon) := \mathcal{V}_{\text{eff}}(A_1(W), R, \varepsilon)^{d_1} \times \cdots \times \mathcal{V}_{\text{eff}}(A_L(W), R, \varepsilon)^{d_L}.$$

$\square$

### D.5.2 PROOF OF THEOREM 3

We firstly prove the following result, which is almost Theorem 3, with the only difference being that the radius in the effective dimension depends on the global radius $R$ rather than the pointwise Frobenious norm $\|W\|_{\mathsf{F}}$. Extending this result to Theorem 3 can be achieved via a simple application of the "uniform pointwise convergence" principle (Xu & Zeevi, 2025) illustrated in Lemma 4.

**Lemma 23 (Riemannian Dimension for NN-surrogate Metric Tensor—Global Radius Version)**
*Consider the NN-surrogate metric tensor in Definition 3, and the weight space $B_{\mathsf{F}}(R)$. Then we have that the pointwise dimension is bounded by the pointwise Riemannian Dimension as the following: there exists a prior $\pi$ such that uniformly over all $W \in B_{\mathsf{F}}(R)$,*

$$\log \frac{1}{\pi(B_{\varrho_{G(W)}}(W, \sqrt{n}\varepsilon))} \leq \sum_{l=1}^{L} \Big( \underbrace{d_l \cdot d_{\mathrm{eff}}(A_l(W), CR, \varepsilon)}_{\text{"must pay" cost at each } W} + \underbrace{d_{l-1} \cdot d_{\mathrm{eff}}(A_l(W), CR, \varepsilon)}_{\text{covering cost of Grassmannian}} + \underbrace{\log(d_{l-1})}_{\text{covering cost of } r_{\mathrm{eff}} \in [d_{l-1}]} \Big),$$

*where $C > 0$ is an absolute constant.*

**Proof of Lemma 23:** The proof has two key steps: 1. Hierarchical covering argument, and 2. Bound covering Cost of the Grassmannian. A crucial lemma about the ellipsoidal covering of the Grassmannian, which is new even in the pure mathematics context, is deferred to Section E.

**Step 1: Hierarchical Covering.** As explained the main paper, the major difficulty is that the prior measure $\pi_{\mathcal{V}}$ it constructed, is defined over the effective subspace $\mathcal{V}$, which itself encodes information of the point $W$ and $\varepsilon > 0$. The goal of our proof is to construct a "universal" prior $\pi$ that does not depend on $\mathcal{V}$. This is achieved via a hierarchical covering argument (13), which we make rigorous below.

The key idea of hierarchical covering is as follows: Firstly, for all $W$, we search for subspace $\bar{\mathcal{V}}$ that approximates the true effective subspace (top-$r_{\mathrm{eff}}$ eigenspace) $\mathcal{V}_{\mathrm{eff}}(G(W), R, \varepsilon)$ to the precision required by (64):

$$\|G(W)^{\frac{1}{2}}(\mathcal{P}_{\mathcal{V}} - \mathcal{P}_{\bar{\mathcal{V}}})\|_{\mathrm{op}} \leq \frac{\sqrt{n}\varepsilon}{4R}, \tag{71}$$

where $G(W)^{\frac{1}{2}}$ is the unique square root of PSD matrix $G(W)$ (see, e.g, (Wikipedia contributors, 2025d)). Then by Lemma 2 (Pointwise Dimension Bound for Nonlinear Manifold with Approximate Effective Subspace), for every $(W, \varepsilon) \in B_2(R) \times [0, \infty)$ such that $\bar{\mathcal{V}}$ approximates $\mathcal{V}_{\mathrm{eff}}(G(W), R, \varepsilon)$ to the precision (71), the prior $\pi_{\bar{\mathcal{V}}} = \mathrm{Unif}(B_2(1.58R) \cap \bar{\mathcal{V}})$ satisfies

$$\log \frac{1}{\pi_{\bar{\mathcal{V}}}(B_{\varrho_{G(W)}}(W, \sqrt{n}\varepsilon))} \leq d_{\mathrm{eff}}(G(W), \sqrt{5}R, \varepsilon) = \sum_{l=1}^{L} d_l \cdot d_{\mathrm{eff}}(A_l(W), \sqrt{5}R, \varepsilon), \tag{72}$$

where the first inequality is by Lemma 2 (see definition (11) of effective dimension); and the last equality is by the decomposition property of NN-surrogate metric tensor (Lemma 22).

Secondly, we put a prior $\mu$ over all possible subspaces $\mathcal{V}$ and construct the "universal" prior

$$\pi(W) = \sum_{\mathcal{V}} \mu(\mathcal{V}) \times \pi_{\mathcal{V}}(W), \tag{73}$$

which implies that uniformly over all $W \in B_{\mathsf{F}}(R)$,

$$\log \frac{1}{\pi(B_{\varrho_{G(W)}}(W, \sqrt{n}\varepsilon))}$$

$$= \log \frac{1}{\sum_{\mathcal{V}} \mu(\mathcal{V}) \pi_{\mathcal{V}}(B_{\varrho_{G(W)}}(W, \sqrt{n}\varepsilon))}$$

$$\leq \log \frac{1}{\mu(\bar{\mathcal{V}} : \bar{\mathcal{V}} \text{ satisfies (71)}) \inf_{\bar{\mathcal{V}} \text{ satisfies (71)}} \pi_{\bar{\mathcal{V}}}(B_{\varrho_{G(W)}}(W, \sqrt{n}\varepsilon))}$$

$$\leq \underbrace{\log \frac{1}{\mu(\bar{\mathcal{V}} : \bar{\mathcal{V}} \text{ satisfies (71)}}}_{\text{covering cost of the Grassmannian}} + \sum_{l=1}^{L} d_l \cdot d_{\mathrm{eff}}(A_l(W), \sqrt{5}R, \varepsilon), \tag{74}$$

where the first equality is by definition (73) of the "universal" prior $\pi$; the first inequality is straightforward; and the last inequality is by (72) (the result of the "must pay" part in the hierarchical covering) and the equivalence between $B_2(R)$ and $B_{\mathsf{F}}(R)$.

The above hierarchical covering argument successfully gives a valid Riemannian Dimension, with the cost of the additional covering cost given by the subspace prior $\mu$. This explains our basic proof idea. The remaining proof executes this basic proof idea.

**Step 2: Bounding Covering Cost of the Grassmannian.**    Section E provides a systematical study to the ellipsoidal metric entropy of Grassmannian manifold, which we detail the conclusion below.

Define
$$\mathrm{Gr}(d, r) := \{ r\text{--dimensional linear subspaces of } \mathbb{R}^d \}$$
as the *Grassmannian manifold*.

Given a $d \times d$ PSD $\Sigma$, define the anisometric projection metric between two subspaces by (labeled as Definition 4 in Section E)
$$\varrho_{\mathrm{proj}, \Sigma}(\mathcal{V}, \bar{\mathcal{V}}) = \|\Sigma^{\frac{1}{2}}(\mathcal{P}_{\mathcal{V}} - \mathcal{P}_{\bar{\mathcal{V}}})\|_{\mathrm{op}}, \tag{75}$$
where $\Sigma^{\frac{1}{2}}$ is the square root of the PSD matrix $\Sigma$ (see, e.g., (Wikipedia contributors, 2025d)).

Lemma 3 states that (note that we use $\varepsilon_1$ and $C_0$ here instead of $\varepsilon$ and $C$ in the original statement of Lemma 3), given a Grassmannian $\mathrm{Gr}(d, r)$, for uniform prior $\mu = \mathrm{Unif}(\mathrm{Gr}(d, r))$, we have that for every $\mathcal{V} \in \mathrm{Gr}(d, r)$, every $\varepsilon_1 > 0$ and PSD matrix $\Sigma \in \mathbb{R}^{d \times d}$ with eigenvalues $\lambda_1 \geq \cdots \lambda_d \geq 0$, we have the pointwise dimension bound
$$\log \frac{1}{\mu(B_{\varrho_{\mathrm{proj}, \Sigma}}(\mathcal{V}, \varepsilon_1))} \leq \frac{d - r}{2} \sum_{k=1}^{r} \log \frac{C_0 \max\{\lambda_k, \varepsilon_1^2\}}{\varepsilon_1^2} + \frac{r}{2} \sum_{k=1}^{d-r} \log \frac{C_0 \max\{\lambda_k, \varepsilon_1^2\}}{\varepsilon_1^2}, \tag{76}$$
where $C_0 > 0$ is an absolute constant. We will use the result (76) and (74) to prove Theorem 3.

For a particular layer $l$, $d_{l-1} \times d_{l-1}$ PSD matrix $A_l(W)$, and a fixed rank $r_l$ denote $\mathrm{Gr}(d_{l-1}, r_l)$ as a Grassmannian (the collection of all $r_l$-dimensional in $\mathbb{R}^{d_{l-1}}$). By (76) we have that there exists a prior $\mu_l$ over $\mathrm{Gr}(d_{l-1}, r_l)$ such that for every $(W, \varepsilon_1)$ such that $r_{\mathrm{eff}}(A_l(W), R, \varepsilon_1) = r_l$, and $\lambda_{r_l+1}(A_l(W)) \leq c\varepsilon_1^2 \leq \lambda_{r_l}(A_l(W))$ where $c \geq 1$ can be any absolute constants no smaller than 1 (later we will specialize to $c = 8$),
$$\log \frac{1}{\mu_l(\bar{\mathcal{V}} : \varrho_{\mathrm{proj}, A_l(W)}(\mathcal{V}_{\mathrm{eff}}(A_l(W), R, \varepsilon), \bar{\mathcal{V}}) \leq \varepsilon_1)} \leq \frac{d_{l-1}}{2} \sum_{k=1}^{r_l} \log \frac{C_1 \lambda_k(A_l(W))}{\varepsilon_1^2}, \tag{77}$$
where $C_1 = c \max\{C_0, 1\} \geq 1$ is an absolute constant depending only on the absolute constant $c$ (later we take $c = 8$ so $C_1 = 8 \max\{C_0, 1\}$ is indeed an absolute constant). (77) is because: 1) all eigenvalues with index at least $r_l + 1$ (each no larger than $c\varepsilon_1^2$) contribute only through the second term in (76). Their cumulative effect is at most
$$\mathbb{1}\{d_{l-1} - r_l > r_l\} \cdot \frac{r_l}{2} \sum_{k=r_l+1}^{d_{l-1}-r_l} \log \frac{C_0 c \varepsilon_1^2}{\varepsilon_1^2} = \frac{r_l \max\{d_{l-1} - 2r_l, 0\}}{2} \log C_0 c \leq \frac{r_l(d_{l-1} - r_l)}{2} \log C_0 c$$
unaffected to the spectrum, and we absorb this into the absolute constant $C_1$. And 2) all eigenvalues with index at most $r_l$'s contribution leads to at most
$$\frac{d_{l-1} - r_l}{2} \sum_{k=1}^{r_l} \log \frac{C_0 \lambda_k(A_l(W))}{\varepsilon_1^2} + \frac{r_l}{2} \sum_{k=1}^{\max\{r_l, d_{l-1}-r_l\}} \log \frac{C_0 \lambda_k(A_l(W))}{\varepsilon_1^2} \leq \frac{d_{l-1}}{2} \sum_{k=1}^{r_l} \log \frac{\max\{C_0, 1\} \lambda_k(A_l(W))}{\varepsilon_1^2}.$$
Summing up the contributions two parts of the spectrum together, we get the right hand side of (77).

By the subspace decomposition property in Lemma 22, we have that for $\bar{\mathcal{V}} = (\cdots, \underbrace{\bar{\mathcal{V}}_l, \cdots, \bar{\mathcal{V}}_l}_{\text{repeat } d_l \text{ times}}, \cdots)$,
$$\varrho_{\mathrm{proj}, G(W)}(\mathcal{V}_{\mathrm{eff}}(G(W), R, \varepsilon), \bar{\mathcal{V}})$$
$$= \varrho_{\mathrm{proj}, G(W)}(\prod_{l=1}^{L} \mathcal{V}_{\mathrm{eff}}(A_l(W), R, \varepsilon)^{d_l}, \prod_{l=1}^{L} \bar{\mathcal{V}}_l^{d_l})$$
$$= \max_l \varrho_{\mathrm{proj}, A_l(W)}(\mathcal{V}_{\mathrm{eff}}(A_l(W), R, \varepsilon), \bar{\mathcal{V}}_l), \tag{78}$$

where the first equality is by Lemma 22, and the second equality is by the properties of the spectral norm: $\|\text{blockdiag}(A, B)\|_{\text{op}} = \max\{\|A\|_{\text{op}}, \|B\|_{\text{op}}\}$ and $\|A \otimes I_d\|_{\text{op}} = \|A\|_{\text{op}}$.

Taking $\varepsilon_1 = \frac{\sqrt{n}\varepsilon}{4R}$, by definition (10) on the threshold to determine effective rank, we obtain $\lambda_{r_l+1}(A_l(W)) \leq 8\varepsilon_1^2 = n\varepsilon^2/(2R^2) \leq \lambda_{r_l}(A_l(w))$, thus this particular choice satisfies the required eigenvalue condition to establish (77) with $c = 8$. Then for all layers $l = 1, \cdots, L$, given a fixed $\{r_1, \cdots, r_L\}$, by (77), we have that there exists a prior

$$\mu_{\{r_l\}_{l=1}^L} = \mu_1^{d_1} \otimes \cdots \otimes \mu_L^{d_L} = \prod_{l=1}^L (\underbrace{\mu_l \otimes \cdots \otimes \mu_l}_{d_l \text{ times}}) \tag{79}$$

over the product Grassmannian $\text{Gr}(d_0, r_1)^{d_1} \times \cdots \times \text{Gr}(d_{L-1}, r_L)^{d_L}$ such that uniformly over all $W \in B_{\mathbf{F}}(R)$ such that $r_{\text{eff}}(A_l(W), R, \varepsilon) = r_l, \forall l \in [L]$ (here $[L]$ is the notation of $\{1, 2, \cdots, L\}$), the "Grassmannian covering cost" term in (74) is bounded by

$$\log \frac{1}{\mu(\bar{\mathcal{V}} : \bar{\mathcal{V}} \text{ satisfies (71)})}$$

$$= \log \frac{1}{\mu_{\{r_l\}_{l=1}^L}(\bar{\mathcal{V}} : \varrho_{\text{proj}, G(W)}(\mathcal{V}_{\text{eff}}(G(W), R, \varepsilon), \bar{\mathcal{V}}) \leq \frac{\sqrt{n}\varepsilon}{4R} = \varepsilon_1)}$$

$$\leq \log \frac{1}{\mu_{\{r_l\}_{l=1}^L}((\cdots, \underbrace{\bar{\mathcal{V}}_l, \cdots, \bar{\mathcal{V}}_l}_{d_l \text{ times}}, \cdots) : \varrho_{\text{proj}, A_l(W)}(\mathcal{V}_{\text{eff}}(A_l(W), R, \varepsilon), \bar{\mathcal{V}}_l) \leq \varepsilon_1, \quad \forall l \in [L])}$$

$$= \sum_{l=1}^L \log \frac{1}{\mu_{\{r_l\}_{l=1}^L}(\cdots, \bar{\mathcal{V}}_l, \cdots) : \varrho_{\text{proj}, A_l(W)}(\mathcal{V}_{\text{eff}}(A_l(W), R, \varepsilon), \bar{\mathcal{V}}_l) \leq \varepsilon_1)}$$

$$\leq \sum_{l=1}^L \frac{d_{l-1}}{2} \sum_{k=1}^{r_l} \log \frac{C_1 \lambda_k(A_l(W))}{\varepsilon_1^2}$$

$$\leq \sum_{l=1}^L d_{l-1} d_{\text{eff}}(A_l(W), \sqrt{2C_1}R, \varepsilon), \tag{80}$$

where the first inequality is by restricting $\bar{\mathcal{V}}$ to the form $\prod_{l=1}^L \bar{\mathcal{V}}_l^{d_l}$ and using (78); the second equality is by the choice of the product prior (79); the second inequality is by the layer-wise covering bound (77); and the last inequality is by the choice $\varepsilon_1 = \sqrt{n}\varepsilon/(4R)$, and definition (11) of effective dimension.

Note that (80) is uniformly over all $W \in B_{\mathbf{F}}(R)$ such that $r_{\text{eff}}(A_l(W), R, \varepsilon) = r_l, \forall l \in [L]$, not uniformly over all $W \in B_{\mathbf{F}}(R)$. We would like to extend (80) to all $W \in B_{\mathbf{F}}(R)$ over uniform prior over possible integer values of $r_l$. Now assign uniform prior over $[d_{l-1}] = \{1, \cdots, d_{l-1}\}$ for $r_l$, we obtain the "universal" prior $\pi$ (as we have pursued in in our hierarchical covering argument (73)) defined by

$$\mu(\mathcal{V}) = \prod_{l=1}^L \underbrace{\text{Unif}([d_{l-1}])}_{\text{prior of } r_l} \otimes \underbrace{\mu_{\{r_k\}_{k=1}^L}}_{\text{prior over product Grassmannian in (79)}},$$

$$\pi(W) = \sum_{\mathcal{V}} \underbrace{\mu(\mathcal{V})}_{\text{prior over subspaces defined above}} \otimes \underbrace{\text{Unif}(B_2(1.58R) \cap \bar{\mathcal{V}})}_{\text{uniform prior constrained in subspace}}. \tag{81}$$

Then we have that uniformly over all $W \in B_{\mathbf{F}}(R)$,

$$\log \frac{1}{\pi(B_{\varrho_{G(W)}}(W, \sqrt{n}\varepsilon))}$$

$$\leq \log \frac{1}{\mu(\bar{\mathcal{V}} : \bar{\mathcal{V}} \text{ satisfies (71)})} + \sum_{l=1}^{L} d_l \cdot d_{\text{eff}}(A_l(W), \sqrt{5}R, \varepsilon))$$

$$\leq \sum_{l=1}^{L} \log d_{l-1} + \log \frac{1}{\mu_{\{r_k\}_{k=1}^{L}}(\bar{\mathcal{V}} : \bar{\mathcal{V}} \text{ satisfies (71)})} + \sum_{l=1}^{L} d_l \cdot d_{\text{eff}}(A_l(W), \sqrt{5}R, \varepsilon))$$

$$\leq \sum_{l=1}^{L} \log d_{l-1} + \sum_{l=1}^{L} d_{l-1} \cdot d_{\text{eff}}(A_l(W), \sqrt{2C_1}R, \varepsilon) + \sum_{l=1}^{L} d_l \cdot d_{\text{eff}}(A_l(W), \sqrt{5}R, \varepsilon)),$$

where $C_1 > 0$ is an absolute constant. Here the first inequality is by the hierarchical covering argument (74); the second inequality is by the prior construction (81); and the third inequality is by the Grassmannian covering bound (80) for fixed $\{r_k\}_{k=1}^{L}$. This shows that for NN-surrogate metric tensor $G(W)$, the pointwise dimension is bounded by the Riemannian Dimension as the following:

$$\log \frac{1}{\pi(B_{\varrho_{G(W)}}(W, \sqrt{n}\varepsilon))} \leq \sum_{l=1}^{L}(d_l + d_{l-1}) \cdot d_{\text{eff}}(A_l(W), CR, \varepsilon) + \log(d_{l-1}),$$

where $C$ is a positive absolute constant. This finishes the proof of Lemma 23 with $R$ in effective dimension being a global upper bound of $\|W\|_{\mathbf{F}}$.

$\square$

**Proof of Theorem 3:** Motivated by the "uniform pointwise convergence" principle (proposed in Xu & Zeevi (2025) and illustrated in Lemma 4), we apply a peeling argument to adapt the Riemannian Dimension to $\|W\|_{\mathbf{F}}$. Given any $R_0 \in (0, R]$, we take $R_k = 2^k R_0$ for $k = 0, 1, \cdots \log_2 \lceil R/R_0 \rceil$. Taking a uniform prior on these $R_k$, and set

$$\tilde{\pi} = \underbrace{\text{Unif}(\{R_0, \cdots, 2^{\log_2 \lceil R/R_0 \rceil} R_0\})}_{\text{prior over upper bound } \tilde{R} \text{ of } \|W\|_{\mathbf{F}}} \otimes \underbrace{\pi_{\tilde{R}}}_{\text{prior defined via (81)}},$$

where $\pi_{\tilde{R}}$ is the prior defined via (81) in the proof of Lemma 23. Then for every $W \in B_{\mathbf{F}}(R)$ where $\|W\|_{\mathbf{F}} > R_0$, denote $k(W)$ to be the integer such that $2^{k(W)} R_0 < \|W\|_{\mathbf{F}} \leq 2^{k(W)+1} R_0$, then

$$\log \frac{1}{\tilde{\pi}(B_{\varrho_{G(W)}}(W, \sqrt{n}\varepsilon))}$$

$$\leq \underbrace{\log \log_2 \lceil R/R_0 \rceil}_{\text{density of } 2^{k(W)+1} R_0} + \underbrace{\log \frac{1}{\pi_{2^{k(W)+1} R_0}(B_{\varrho_{G(W)}}(W, \sqrt{n}\varepsilon))}}_{\pi \text{ is constructed via (81), with global radius taken to be } 2^{k(W)+1} R_0}$$

$$\leq \log \log_2 \lceil R/R_0 \rceil + \sum_{l=1}^{L}((d_l + d_{l-1}) \cdot d_{\text{eff}}(A_l(W), C_1 2^{k(W)+1} R_0, \varepsilon) + \log d_{l-1})$$

$$\leq \log \log_2 \lceil R/R_0 \rceil + \sum_{l=1}^{L}((d_l + d_{l-1}) \cdot d_{\text{eff}}(A_l(W), C_1 \cdot 2\|W\|_{\mathbf{F}}, \varepsilon) + \log d_{l-1}),$$

where the first inequality is due to the product construction of $\tilde{\pi}$; the second inequality is due to Lemma 23, with $C_1 > 0$ being an absolute constant; and the last inequality uses the fact $\|W\|_{\mathbf{F}} \leq 2^{k(W)+1} R_0 \leq 2\|W\|_{\mathbf{F}}$, with $C_1 > 0$.

The above bound assumes $\|W\|_{\mathsf{F}} > R_0$. When $\|W\|_{\mathsf{F}} \leq R_0$, we directly apply Lemma 23 and obtain

$$\log \frac{1}{\tilde{\pi}(B_{\varrho_{G(W)}}(W, \sqrt{n}\varepsilon))}$$

$$\leq \underbrace{\log \log_2 \lceil R/R_0 \rceil}_{\text{density of } R_0} + \underbrace{\log \frac{1}{\pi_{R_0}(B_{\varrho_{G(W)}}(W, \sqrt{n}\varepsilon))}}_{\pi \text{ is constructed via (81), with global radius taken to be } R_0}$$

$$\leq \log \log_2 \lceil R/R_0 \rceil + \sum_{l=1}^{L} (d_l + d_{l-1}) \cdot d_{\text{eff}}(A_l(W), C_1 \cdot R_0, \varepsilon) + \log d_{l-1}).$$

Combining the two cases discussed above, we conclude that the pointwise dimension for NN-surrogate metric tensor $G(W)$ in Definition 3 is bounded by the Riemmanin Dimension

$$\log \frac{1}{\tilde{\pi}(B_{\varrho_{G(W)}}(W, \sqrt{n}\varepsilon))} \leq d_{\mathsf{R}}(W, \varepsilon)$$

$$= \sum_{l=1}^{L} (d_l + d_{l-1}) \cdot d_{\text{eff}}(A_l(W), C \max\{\|W\|_{\mathsf{F}}, R_0\}) + \log(d_{l-1} \log_2 \lceil R/R_0 \rceil),$$

where $C = 2C_1$ is a positive absolute constant.

Finally, by the sentence below (9) (which is a straightforward result from non-perturbative feature expansion for DNN (Lemma 1) and the metric domination lemma (Lemma 19)), we know that there exists a prior $\tilde{\pi}$ such that uniformly over all $W \in B_{\mathsf{F}}(R)$,

$$\log \frac{1}{\tilde{\pi}(B_{\varrho_n}(f(W, \cdot), \varepsilon))} \leq \log \frac{1}{\tilde{\pi}(B_{\varrho_{G_{\mathrm{NP}}(W)}}(W, \sqrt{n}\varepsilon))}$$

$$\leq d_{\mathsf{R}}(W, \varepsilon) = \sum_{l=1}^{L} (d_l + d_{l-1}) \cdot d_{\text{eff}}(A_l(W), C \max\{\|W\|_{\mathsf{F}}, R_0\}) + \log(d_{l-1} \log_2 \lceil R/R_0 \rceil),$$

where $A_l(W) = LM_{l \to L}^2(W, \varepsilon) \cdot F_{l-1}(W, X) F_{l-1}^\top(W, X)$ when taking $G(W)$ to be $G_{\mathrm{NP}}(W)$ defined in (9). Taking $R_0 = R/2^n$ proves Theorem 3.

$\square$

## E  ELLIPSOIDAL COVERING OF THE GRASSMANNIAN (LEMMA 3)

The central goal of this section is to prove the following result on the ellipsoidal metric entropy of the Grassmannian manifold. The definition for Gr (Grassmannian manifold), St (Stiefel parameterization manifold) are temporarily deferred to Section E.1.

**Definition 4 (Ellipsoidal Projection Metric)** *For two subspaces* $\mathcal{V}, \bar{\mathcal{V}} \in \mathrm{Gr}(d, r)$, *and a positive semidefinite matrix* $\Sigma$, *define the ellipsoidal projection metric* $\varrho_{\mathrm{proj},\Sigma}$ *by*

$$\varrho_{\mathrm{proj},\Sigma}(\mathcal{V}, \bar{\mathcal{V}}) = \|\Sigma^{\frac{1}{2}}(\mathcal{P}_{\mathcal{V}} - \mathcal{P}_{\bar{\mathcal{V}}})\|_{\mathrm{op}},$$

*where* $\mathcal{P}_{\mathcal{V}}$ *and* $\mathcal{P}_{\bar{\mathcal{V}}}$ *are orthogonal projectors to subspace* $\mathcal{V}$ *and* $\bar{\mathcal{V}}$, *respectively.*

We view orthogonal projectors as matrices (see the definition via the Stiefel parameterization in (87)), consistent with the earlier operator notation characterized by $\ell_2$–distance in Lemma 20. In the isotropic case $\Sigma = I_d$, the ellipsoidal projection metric reduces to the standard isotropic projection metric

$$\varrho_{\mathrm{proj}}(\mathcal{V}, \bar{\mathcal{V}}) = \|\mathcal{P}_{\mathcal{V}} - \mathcal{P}_{\bar{\mathcal{V}}}\|_{\mathrm{op}}.$$

We now state our main result in this section (Lemma 3 in the main paper).

Consider the Grassmannian $\mathrm{Gr}(d, r)$ and the uniform prior $\mu = \mathrm{Unif}(\mathrm{Gr}(d, r))$, then for every $\mathcal{V} \in \mathrm{Gr}(d, r)$, every $\varepsilon > 0$ and every PSD matrix $\Sigma$ with eigenvalues $\lambda_1 \geq \cdots \lambda_d \geq 0$, we have

$$\log \frac{1}{\mu(B_{\varrho_{\mathrm{proj}, \Sigma}}(\mathcal{V}, \varepsilon))} \leq \frac{r}{2} \sum_{k=1}^{d-r} \log \frac{C \max\left\{\lambda_k, \varepsilon^2\right\}}{\varepsilon^2} + \frac{d-r}{2} \sum_{k=1}^{r} \log \frac{C \max\left\{\lambda_k, \varepsilon^2\right\}}{\varepsilon^2}, \quad (82)$$

where $C > 0$ is an absolute constant.

Recall that the traditional covering number bound for the Grassmannian manifold states that

$$\left(\frac{c}{\varepsilon}\right)^{r(d-r)} \leq \mathrm{N}(\mathrm{Gr}(d, r), \varrho_{\mathrm{proj}}, \varepsilon) \leq \left(\frac{C}{\varepsilon}\right)^{r(d-r)}. \quad (83)$$

Here $\mathrm{N}(\mathcal{F}, \varrho, \varepsilon)$ is the standard covering number— the smallest size of an $\varepsilon$-net that covers $\mathcal{F}$ under the metric $\varrho$; see Definition 5 for details. In comparison, Lemma 3 is much more challenging than proving classical isotropic covering number bounds (83) because

- 1) we consider ellipsoidal metric;
- 2) we require the prior $\mu$ to be independent with $\Sigma$ and $\varepsilon$.

We need to firstly understand how such classical results are proved, and then proceed to generalized them. This suggests that deep mathematical insights are necessary for the purpose to study neural networks generalization, as we will introduce below.

**From Pure Mathematics to Machine Learning Language.** Understanding the classical proof for the Grassmannian and generalizing them to prove Lemma 3 necessitate the a deep dive in to the geometry and algebra of subspaces and Grassmannians. In fact, traditional treatments to study Grassmannian manifold often invoke advanced machinery—ranging from differential geometry (Bendokat et al., 2024) and Lie-group theory (Szarek, 1997) to algebraic geometry (Devriendt et al., 2024), and the seminal covering number proof (Szarek, 1997) is particularly stated in Lie-algebra and differential-geometry language.

Motivated by the subsequent covering number proof (Pajor, 1998) that uses relatively more elementary language, we give an exposition that is elementary and entirely self-contained, relying only on matrix-analysis and learning-theoretic techniques familiar from machine learning. In particular, every "advanced" fact—for example, the group theory of continuous symmetries traditionally handled via Lie groups—is derived by elementary means (explicit matrix parameterizations, principal-angle/cosine-sine representations, and basic spectral arguments) while preserving the high-level geometric intuition. We hope that this versatile framework—and our novel contributions (e.g., Definition 4 and Lemma 3), which are new even in a pure-mathematics setting—will establish subspaces, the Grassmannian, and their underlying algebraic structures as powerful tools for future machine learning applications.

**Effective Rank vs. Full-Spectrum Complexity.** Consider a covariance matrix $\Sigma$ with eigenvalues $\lambda_1 \geq \cdots \lambda_d \geq 0$. By Definition 4, the ellipsoidal metric satisfies

$$\varrho_{\mathrm{proj}, \Sigma}(\mathcal{V}, \bar{\mathcal{V}}) \leq \lambda_1^{\frac{1}{2}} \varrho_{\mathrm{proj}}(\mathcal{V}, \bar{\mathcal{V}}).$$

If one is willing to accept a coarser complexity scaling, then one could invoke existing Grassmannian covering results under the canonical isotropic metric (83) (taking $\mu = \mathrm{Unif}(\mathrm{Gr}(d, r))$) and obtain

$$\log \frac{1}{\mu\left(B_{\varrho_{\mathrm{proj}, \Sigma}}(\mathcal{V}, \varepsilon)\right)} \leq \log \frac{1}{\mu\left(B_{\varrho_{\mathrm{proj}}}(\mathcal{V}, \varepsilon/\sqrt{\lambda_1})\right)} \leq \left(d - r_{\mathrm{eff}}(\Sigma, R, \varepsilon)\right) r_{\mathrm{eff}}(\Sigma, R, \varepsilon) \log \frac{C\lambda_1}{\varepsilon^2}. \quad (84)$$

However, this makes the *global atlas* cost dominate the *local chart* cost, yielding a suboptimal bound than the full–spectrum effective dimension in (82). The refined analysis in this section—also simplifying and strengthening the isotropic route—establishes the correct structural principle: the global–atlas cost must be balanced by the local–chart cost. Thus, while the effective–rank bound (84) serves as a useful sanity check, the full–spectrum treatment is what delivers the sharpened complexities required for our main results.

## E.1 GRASSMANNIAN MANIFOLD, STIEFEL PARAMETERIZATION, AND ORTHOGONAL GROUPS

Fix integers $r \leq d$. Define

$$\mathrm{Gr}(d, r) := \{r\text{–dimensional linear subspaces of } \mathbb{R}^d\}$$

as the *Grassmann manifold*. Write

$$\mathrm{St}(d, r) := \{V \in \mathbb{R}^{d \times r} : V^\top V = I_r\}$$

for the *Stiefel manifold* of $r$ orthonormal columns in $\mathbb{R}^d$. $\mathrm{St}(d, r)$ is a convenient *parameterization* of that class $\mathrm{Gr}(d, r)$.

If for subspace $\mathcal{V} \in \mathrm{Gr}(d, r)$ and matrix $V \in \mathrm{St}(d, r)$ we have $\mathcal{V} = \mathrm{span}(V)$, then we say $V$ is a *parameterization matrix* of $\mathcal{V}$. Though such parameterization is not unique, the associated orthogonal projector and projection metric are both unique. Moreover, the anisometric projection we define in Definition 4 is also unique. We will prove these shortly.

Write

$$O(r) := \{Q \in \mathbb{R}^{r \times r} : Q^\top Q = QQ^\top = I_r\}$$

to be the *orthogonal group*. Optionally, we also state that (in the real setting)

$$\mathrm{Gr}(d, r) \cong O(d)/(O(r) \times O(d - r)) \cong \mathrm{Gr}(d, d - r), \tag{85}$$

where "/" denotes the *quotient* and "$\cong$" denotes a canonical *isomorphism* (indeed, a *diffeomorphism* of smooth manifolds or a *homeomorphism* of topological manifolds; see, e.g., Chapter 1.5 in (Awodey, 2010)). Moreover, $\mathrm{Gr}(d, r)$ can be regarded as a standard *algebraic variety* (Devriendt et al., 2024). We do not aim to explain these notions in detail, but merely note that:

1. The geometric properties of $\mathrm{Gr}(d, r)$ coincide with those of $\mathrm{Gr}(d, d - r)$ under this isomorphism (geometric equivalence).

2. The number of degrees of freedom of $\mathrm{Gr}(d, r)$ is

$$\underbrace{\frac{d(d - 1)}{2}}_{\dim O(d)} - \underbrace{\frac{r(r - 1)}{2}}_{\dim O(r)} - \underbrace{\frac{(d - r)(d - r - 1)}{2}}_{\dim O(d - r)} = r(d - r), \tag{86}$$

which also appears as the dimension factor in the precise covering-number bounds (83).

We now define the orthogonal projector and the projection metric on the Grassmannian manifold.

**Definition of Orthogonal Projector.** For $V \in \mathrm{St}(d, r)$ and its column-space $\mathcal{V} = \mathrm{span}(V)$, define the rank-$r$ orthogonal projector[3]

$$\mathcal{P}_{\mathcal{V}} := VV^\top \in \mathbb{R}^{d \times d}. \tag{87}$$

Then $\mathcal{P}_{\mathcal{V}}$ depends *only* on the subspace $\mathcal{V}$. Indeed, if $Q \in O(r)$ then $(VQ)(VQ)^\top = VQQ^\top V^\top = VV^\top$, so $V$ and $VQ$ represent the same subspace. Hence the map

$$\Psi : \mathrm{St}(d, r) \longrightarrow \mathrm{Gr}(d, r), \quad V \mapsto \mathrm{span}(V),$$

is an $O(r)$−*quotient*: two frames give the same subspace iff they differ by a right orthogonal factor.

**Ellipsoidal Projection Metric.** Following Definition 4, for $\mathcal{V}, \bar{\mathcal{V}} \in \mathrm{Gr}(d, r)$,

$$\varrho_{\mathrm{proj}, \Sigma}(\mathcal{V}, \bar{\mathcal{V}}) := \|\Sigma^{\frac{1}{2}}(\mathcal{P}_{\mathcal{V}} - \mathcal{P}_{\bar{\mathcal{V}}})\|_{\mathrm{op}}, \tag{88}$$

where $\mathcal{P}_{\mathcal{V}} := VV^\top$ for *any* $V$ such that $\mathrm{span}(V) = \mathcal{V}$ (similarly $\mathcal{P}_{\bar{\mathcal{V}}}$). Because $\mathcal{P}_{\mathcal{V}}$ is unique for each subspace, $\varrho_{\mathrm{proj}, \Sigma}$ is well defined (independent of the chosen $V$). The metric can be pulled back to $\mathrm{St}(d, r)$:

$$\varrho_{\mathrm{proj}, \Sigma}(V, \bar{V}) := \varrho_{\mathrm{proj}, \Sigma}(\mathrm{span}(V), \mathrm{span}(\bar{V})) = \|\Sigma^{\frac{1}{2}}(VV^\top - \bar{V}\bar{V}^\top)\|_{\mathrm{op}}. \tag{89}$$

---

[3]By elementary linear algebra, the matrix definition of the orthogonal projector $\mathcal{P}$ here coincides with the $\ell_2$−projection characterized in Lemma 20; thus the notation is consistent.

## E.2 PRINCIPAL ANGLES BETWEEN SUBSPACES

W study how metrics and angles between images $\mathcal{V}$ and $\bar{\mathcal{V}}$ affect their spectral properties. We introduce principal angles and the cosine–sine (CS) decomposition—standard tools for analyzing subspaces (see, e.g., Chapter 6.4.3 in (Golub & Van Loan, 2013)).

**Principle Angles and Cosine-Sine representation.** Let $U$ and $\bar{U}$ be two $d \times d$ orthogonal matrix, and $V$ and $\bar{V}$ be the first $r$ columns of $U$ and $\bar{U}$, respectively. We are interested in studying the metrics and angles between $r-$dimensional subspaces $\mathcal{V} = \mathrm{span}(V)$ and $\bar{\mathcal{V}} = \mathrm{span}(\bar{V})$. Formally, denote

$$U, \bar{U} \in O(d), \qquad U = \begin{bmatrix} V & V_\perp \end{bmatrix}, \ \ \bar{U} = \begin{bmatrix} \bar{V} & \bar{V}_\perp \end{bmatrix},$$

where

$$V, \bar{V} \in \mathbb{R}^{d \times r}, \quad V^\top V = I_r, \quad \bar{V}^\top \bar{V} = I_r,$$

and

$$V_\perp, \bar{V}_\perp \in \mathbb{R}^{d \times (d-r)}, \quad V_\perp^\top V_\perp = I_{d-r}, \quad \bar{V}_\perp^\top \bar{V}_\perp = I_{d-r}.$$

Since $U, \bar{U} \in O(d)$, their product $U^\top \bar{U}$ is itself orthogonal. Writing

$$U^\top \bar{U} \ = \ \begin{pmatrix} V^\top \\ V_\perp^\top \end{pmatrix} \begin{bmatrix} \bar{V} & \bar{V}_\perp \end{bmatrix} \ = \ \begin{pmatrix} V^\top \bar{V} & V^\top \bar{V}_\perp \\ V_\perp^\top \bar{V} & V_\perp^\top \bar{V}_\perp \end{pmatrix},$$

define the four blocks

$$\underbrace{C}_{r \times r} \ = \ V^\top \bar{V}, \qquad \underbrace{C_\perp}_{r \times (d-r)} \ = \ V^\top \bar{V}_\perp, \tag{90}$$

$$\underbrace{S}_{(d-r) \times r} \ = \ V_\perp^\top \bar{V}, \qquad \underbrace{S_\perp}_{(d-r) \times (d-r)} \ = \ V_\perp^\top \bar{V}_\perp. \tag{91}$$

Thus

$$U^\top \bar{U} \ = \ \begin{pmatrix} C & C_\perp \\ S & S_\perp \end{pmatrix} \ \in \ O(d).$$

Now we introduce principal angles between $\mathcal{V} = \mathrm{span}(V)$ and $\bar{\mathcal{V}} = \mathrm{span}(\bar{V})$ by writing

$$C = V^\top \bar{V} \ = \ Q_1 \, \mathrm{diag}(\cos\theta_1, \cdots, \cos\theta_r) \, W_1^\top, \quad Q_1, W_1 \in O(r), \tag{92}$$

where

$$0 \le \theta_1 \le \theta_2 \le \cdots \le \theta_r \le \pi/2$$

are called the principle angles between subspaces $\mathcal{V}$ and $\bar{\mathcal{V}}$; and where $\{\cos\theta_1, \cdots, \cos\theta_r\}$ are the singular values of $C$. Simultaneously, we have that the eigenvalues of $S, C_\perp, S_\perp$ are (notation sepc means spectrum, the set of singular values)

$$\mathrm{spec}(S) = \{\sin\theta_1, \cdots, \sin\theta_{\min\{r,d-r\}}, \ \underbrace{0, \cdots, 0}_{\max\{d-2r,0\}} \},$$

$$\mathrm{spec}(C_\perp) = \{\sin\theta_1, \cdots, \sin\theta_{\min\{r,d-r\}}, \ \underbrace{0, \cdots, 0}_{\max\{d-2r,0\}} \}$$

$$\mathrm{spec}(S_\perp) = \{\cos\theta_1, \cdots, \cos\theta_{\min\{r,d-r\}}, \ \underbrace{1, \cdots, 1}_{\max\{d-2r,0\}} \}. \tag{93}$$

The above representation in (92) and (93) are without loss of generality: if $r \le d - r$, then all the four spectrum contain all $r$ principal angles; if $r > d - r$, then only first $d - r$ principal angles $\{\theta_k\}_{k=1}^{d-r}$ can be smaller than $\pi/2$ and $\theta_k = 0$ for all $d - r + 1 \le k \le r$.

The cosine–sine representation of the eigenvalues in (92) and (93) motivates our notation $C$ and $S$ when defining block matrices in (90) and (91). This representation is an immediate consequence of the classical CS decomposition for orthogonal matrices (Paige & Wei, 1994; Golub & Van Loan, 2013), and we henceforth regard the resulting eigenvalue characterization as given.

**Projection Metric via Principal Angles.** For subspaces $\mathcal{V}$ and $\bar{\mathcal{V}}$, recall that for orthogonal projectors

$$\mathcal{P}_{\mathcal{V}} = VV^{\top}, \quad \mathcal{P}_{\bar{\mathcal{V}}} = \bar{V}\bar{V}^{\top},$$

It is known that the projection metric defined in (88) and (89) are equal to $\sin\theta_r$, sine of the largest principal angle between the two subspaces. Formally, there is the fact (see, e.g., the last equation in Section 6.4.3 in (Golub & Van Loan, 2013))

$$\varrho_{\text{proj}} = \|\mathcal{P}_{\mathcal{V}} - \mathcal{P}_{\bar{\mathcal{V}}}\|_{\text{op}} = \max_{1 \leq k \leq r} \sin\theta_k = \sin\theta_r. \tag{94}$$

Here $\theta_i$ is the $i$-th principal-angle between $\mathcal{V}$ and $\bar{\mathcal{V}}$, and the spectral norm of the difference of two projectors equals the largest of these sines.

### E.3 LOCAL CHARTS OF THE GRASSMANNIAN

In differential geometry, a *chart* is a single local coordinate map. An *atlas* is the whole collection of charts that covers the manifold. We introduce a useful atlas that consists of finite graph charts, which only rely on elementary linear algebra and avoid more advanced Lie algebra and exponential map techniques in Szarek (1997).

Choose a reference subspace $\bar{\mathcal{V}} \in \text{Gr}(d, r)$ and its parameterization matrix $\bar{V} \in \text{St}(d, r)$. Denote $X \in \mathbb{R}^{(d-r) \times r}$ to be mappings from $r-$dimensional subspace $\bar{\mathcal{V}}$ to $(d-r)-$dimensional subspace $\bar{\mathcal{V}}_{\perp}$. Every $r$–dimensional subspace close to $\bar{\mathcal{V}}$ can be written as the *graph*

$$\mathcal{V}(X) := \text{span}\left\{[\bar{V}\bar{V}_{\perp}]\begin{pmatrix} I_r \\ X \end{pmatrix}\right\}, \qquad X \in \mathbb{R}^{(d-r) \times r}, \tag{95}$$

where $\mathcal{V}(X)$ is the subspace spanned by the columns of $[\bar{V} \ \bar{V}_{\perp}]\begin{pmatrix} I_r \\ X \end{pmatrix}$ (the matrix multiplication).

Given the reference subspace $\bar{\mathcal{V}}$, define the local *graph chart* from $\mathbb{R}^{(d-r) \times r}$ to $\text{Gr}(d, r)$ by

$$\phi_{\bar{\mathcal{V}}}: X \longmapsto \mathcal{V}(X) \in \text{Gr}(d, r). \tag{96}$$

Note that for the $(d-r) \times r$ zero matrix (denoted as 0), we have $\phi_{\bar{\mathcal{V}}}(0) = \bar{\mathcal{V}}$.

**Intuition for the graph chart.** If a subspace $\mathcal{V}$ is close to $\bar{\mathcal{V}}$—specifically, $\varrho_{\text{proj}}(\mathcal{V}, \bar{\mathcal{V}}) = \sin\theta_r < 1$—then all principal angles between $\mathcal{V}$ and $\bar{\mathcal{V}}$ satisfy $\theta_i < \pi/2$. Equivalently, the orthogonal projection $\mathcal{P}_{\bar{\mathcal{V}}}$ restricted to $\mathcal{V}$ is a bijection $\mathcal{P}_{\bar{\mathcal{V}}}|\mathcal{V} : \mathcal{V} \to \bar{\mathcal{V}}$. In the orthonormal basis $[\bar{V} \ \bar{V}_{\perp}]$, this means every $v \in \mathcal{V}$ can be written uniquely as

$$v = [\bar{V}\bar{V}_{\perp}]\begin{pmatrix} \bar{v} \\ X\,\bar{v} \end{pmatrix}, \qquad \begin{pmatrix} \bar{v} \\ 0 \end{pmatrix} \in \text{span}\left\{\begin{pmatrix} I_r \\ 0 \end{pmatrix}\right\},$$

for a linear map $X \in \mathbb{R}^{(d-r) \times r}$. Thus, locally around $\bar{\mathcal{V}}$ (all principal angles $< \pi/2$), every $r-$plane admits—and is uniquely determined by—its graph parameter $X$. We call $X$ the *graph parameterization* of $\mathcal{V}(X)$ in this image. This is formalized as the following lemma.

**Lemma 24 (Local Bijection of Graph Chart)** *Fix an orthonormal decomposition $\mathbb{R}^d = \bar{\mathcal{V}} \oplus \bar{\mathcal{V}}_{\perp}$ with basis $[\bar{V} \ \bar{V}_{\perp}]$. Then every $r-$dimensional subspace $\mathcal{V}$ such that $\varrho_{\text{proj}}(\mathcal{V}, \bar{\mathcal{V}}) < 1$ (i.e., all principal angles $< \pi/2$) can be written uniquely as a graph*

$$\mathcal{V} = \phi_{\bar{\mathcal{V}}}(X) = \text{span}\left\{[\bar{V} \ \bar{V}_{\perp}]\begin{pmatrix} I_r \\ X \end{pmatrix}\right\}, \qquad X \in \mathbb{R}^{(d-r) \times r}.$$

**Proof of Lemma 24:** If $V \in \text{St}(d, r)$ spans $\mathcal{V}$, block it in the $[\bar{V} \ \bar{V}_{\perp}]$ basis: denote

$$\begin{pmatrix} A \\ B \end{pmatrix} := \begin{pmatrix} \bar{V}^{\top} \\ \bar{V}_{\perp}^{\top} \end{pmatrix} V \quad (A \in \mathbb{R}^{r \times r}, \ B \in \mathbb{R}^{(d-r) \times r}).$$

Then by the principal angle representation (92), $A = \bar{V}^{\top}V$ is invertible iff all principal angles $< \pi/2$, and choosing

$$X = B\,A^{-1}$$

leads to

$$\mathcal{V} = \mathrm{span}(V) = \mathrm{span}\left\{ [\bar{V}\bar{V}_\perp] \begin{pmatrix} A \\ B \end{pmatrix} \right\} = \mathrm{span}\left\{ [\bar{V}\bar{V}_\perp] \begin{pmatrix} I_r \\ X \end{pmatrix} \right\},$$

where the last equality is because for invertible $A$ one always have $\mathrm{span}(ZA) = \mathrm{span}(Z)$ for any matrix $Z$.

We have already shown existence. For uniqueness, assuming there are two different $X_1, X_2$ such that $\phi_{\bar{\mathcal{V}}}(X_1) = \phi_{\bar{\mathcal{V}}}(X_2)$. Because two bases of the same $r$–dimensional subspace differ by an invertible change of coordinates, so there exists an invertible $r \times r$ matrix $Y$ such that

$$[\bar{V}\bar{V}_\perp] \begin{pmatrix} I_r \\ X_1 \end{pmatrix} Y = [\bar{V}\bar{V}_\perp] \begin{pmatrix} I_r \\ X_2 \end{pmatrix},$$

which results in $Y = I_r$ and $X_1 = X_2$. Thus the parameterization $X$ of $\mathcal{V}$ is unique.

$\square$

**Sine-tangent Relationship in Graph Chart.**    We will show that there is a sine-tangent relationship between $\varrho_{\mathrm{proj}}(\mathcal{V}, \bar{\mathcal{V}})$ and $\|X\|_{\mathrm{op}}$. To be specific, we have the following lemma.

**Lemma 25 (Sine-Tangent Relationship in Graph Chart)** *Denote $\theta_r$ is the maximal principal angle between the subspaces $\mathcal{V}(X)$ and $\bar{\mathcal{V}}$, defined in (92). For the graph chart (96), we have*

$$\varrho_{\mathrm{proj}}(\mathcal{V}(X), \bar{\mathcal{V}}) = \sin\theta_r, \quad \|X\|_{\mathrm{op}} = \tan\theta_r.$$

*The above relationship immediately implies that*

$$\varrho_{\mathrm{proj}}(\mathcal{V}(X), \bar{\mathcal{V}}) = \|X\|_{\mathrm{op}} / \sqrt{1 + \|X\|_{\mathrm{op}}^2}.$$

**Proof of Lemma 25:**    Given the fact $\varrho_{\mathrm{proj}}(\mathcal{V}(X), \bar{\mathcal{V}}) = \sin\theta_r$ (which is already shown in (94)), where $\theta_r$ is the largest principal angle between the subspaces $\mathcal{V}(X)$ and the reference subspace $\bar{\mathcal{V}}$, we want to show $\|X\|_{\mathrm{op}} = \tan\theta_r$.

**Step 1: Setup and Simplification.**    The projection metric is invariant under orthogonal transformations of the ambient space $\mathbb{R}^d$. We can therefore choose a coordinate system that simplifies the calculations without loss of generality. We choose a basis such that the reference frame $\bar{V}$ and its orthogonal complement $\bar{V}_\perp$ are represented as:

$$\bar{V} = \begin{pmatrix} I_r \\ 0 \end{pmatrix} \in \mathrm{St}(d, r), \qquad \bar{V}_\perp = \begin{pmatrix} 0 \\ I_{d-r} \end{pmatrix} \in \mathrm{St}(d, d-r). \tag{97}$$

In this basis, the reference subspace is $\bar{\mathcal{V}} = \mathrm{span}(\bar{V})$. The parameterization matrix (orthonormal basis) $V(X)$ for the subspace $\mathcal{V}(X)$ simplifies to (here $(I_r + X^\top X)^{-1/2}$ normalize $V(X)$ to be an orthogonal matrix):

$$V(X) = [\bar{V}\ \bar{V}_\perp] \begin{pmatrix} I_r \\ X \end{pmatrix} (I_r + X^\top X)^{-1/2} = I_d \begin{pmatrix} I_r \\ X \end{pmatrix} (I_r + X^\top X)^{-1/2} = \begin{pmatrix} I_r \\ X \end{pmatrix} (I_r + X^\top X)^{-1/2}, \tag{98}$$

where the second equality follows from our choice of basis without loss of generality: the reference frame $\bar{V}$ and its complement $\bar{V}_\perp$ are represented as block identity matrices as in (97).

**Step 2: Projection Metric and Principal Angles.**    A fundamental result in matrix analysis, our equation (92), states that the cosines of the principal angles, $\cos\theta_i$, between two subspaces spanned by orthonormal bases $V$ and $\bar{V}$ are the singular values of $V^\top\bar{V}$. In our case, the principal angles between $\mathcal{V}(X)$ and $\bar{\mathcal{V}}$ are determined by the singular values of $V(X)^\top\bar{V}$—which are, equivalently, the singular values of $\bar{V}^\top V(X)$.

**Step 3: Calculation of** $\cos\theta_i$. Let's compute the matrix product $\bar{V}^\top V(X)$ using our simplified forms:

$$\bar{V}^\top V(X) = (I_r \quad 0)\left[\binom{I_r}{X}(I_r + X^\top X)^{-1/2}\right]$$

$$= \left((I_r \quad 0)\binom{I_r}{X}\right)(I_r + X^\top X)^{-1/2}$$

$$= I_r \cdot (I_r + X^\top X)^{-1/2}$$

$$= (I_r + X^\top X)^{-1/2}.$$

To find the singular values of this matrix, we use the Singular Value Decomposition (SVD) of $X$. Let $X = U\Sigma W^\top$, where $U \in \mathbb{R}^{(d-r)\times(d-r)}$ and $W \in \mathbb{R}^{r\times r}$ are orthogonal, and $\Sigma \in \mathbb{R}^{(d-r)\times r}$ is a rectangular diagonal matrix with the singular values $\lambda_1 \geq \lambda_2 \geq \cdots \geq 0$ on its diagonal. The spectral norm is $\|X\|_{\text{op}} = \lambda_1$.

Then, $X^\top X = (U\Sigma W^\top)^\top(U\Sigma W^\top) = W\Sigma^\top U^\top U\Sigma W^\top = W\Sigma_r^2 W^\top$, where $\Sigma_r^2$ is the $r \times r$ diagonal matrix with entries $\lambda_i^2$. So, the matrix $I_r + X^\top X = W(I_r + \Sigma_r^2)W^\top$. Its inverse square root is: $(I_r + X^\top X)^{-1/2} = W(I_r + \Sigma_r^2)^{-1/2}W^\top$.

The singular values of $\bar{V}^\top V(X)$ are the diagonal entries of $(I_r + \Sigma_r^2)^{-1/2}$, which are: $s_i = \frac{1}{\sqrt{1+\lambda_i^2}}$. These singular values are the values of $\cos\theta_i$. The largest principal angle, $\theta_r$, corresponds to the smallest cosine value. This occurs when the singular value $\lambda_i$ is largest, i.e., for $\lambda_1 = \|X\|_{\text{op}}$. Thus,

$$\cos\theta_r = \frac{1}{\sqrt{1 + \|X\|_{\text{op}}^2}}.$$

**Step 4: Deriving** $\tan\theta_r$. Using the fundamental trigonometric identity $\sin^2\theta + \cos^2\theta = 1$ and the fact that principal angles lie in $[0, \pi/2)$, we have:

$$\tan\theta_r = \|X\|_{\text{op}}.$$

We have shown that for graph charts, there is the relationship $\varrho_{\text{proj}}(\mathcal{V}(X), \bar{\mathcal{V}}) = \sin\theta_r$ and $\|X\|_{\text{op}} = \tan\theta_r$. This suggests

$$\varrho_{\text{proj}}(\mathcal{V}(X), \bar{\mathcal{V}}) = \frac{\|X\|_{\text{op}}}{\sqrt{1 + \|X\|_{\text{op}}^2}}.$$

$\square$

### E.4 GLOBAL ATLAS OF GRAPH CHARTS

For the Grassmannian $\text{Gr}(d, r)$ we have that for all $\varepsilon > 0$, we have the coarse covering number bound $\text{N}(\text{Gr}(d,r), \varrho_{\text{proj}}, \varepsilon) \leq C^{\frac{r(d-r)}{\varepsilon}}$, where $C > 0$ is an absolute constant. This is a coarse bound—its dependence is exponential in $1/\varepsilon$ (hence not rate–optimal; the optimal dependence is polynomial)—and we use it only as a preliminary supporting estimate. This coarse estimate suggests that, a finite $O(e^{r(d-r)})$ number of graph charts are sufficient to cover the entire $\text{Gr}(d,r)$ such that every subspace $\mathcal{V} \in \text{Gr}(d,r)$ is contained in the image of a graph chart with its graph parameterization $X$ satisfies $\|X\|_{\text{op}} \leq 1$. From this intuition, we have the following lemma.

**Lemma 26 (Pointwise Dimension Consequence of Finite Global Atlas)** *The uniform prior $\mu = \text{Unif}(\text{Gr}(d,r))$ satisfies that for every $\mathcal{V} \in \text{Gr}(d,r)$, every PSD matrix $\Sigma$ and every $\varepsilon > 0$,*

$$\log\frac{1}{\mu(B_{\varrho_{\text{proj}},\Sigma}(\mathcal{V},\varepsilon))} \leq C_1 r(d-r) + \sup_{X\in\mathcal{X}}\log\frac{1}{\text{Unif}(\bar{\mathcal{X}})\{X' \in \bar{\mathcal{X}} : \varrho_{\text{proj},\Sigma}(\mathcal{V}(X),\mathcal{V}(X')) \leq \varepsilon\}},$$

*where $\mathcal{X} = \{X \in \mathbb{R}^{(d-r)r} : \|X\|_{\text{op}} \leq 1\}$ and $\bar{\mathcal{X}} = \{X \in \mathbb{R}^{(d-r)r} : \|X\|_{\text{op}} \leq 2\}$ (we make $\bar{\mathcal{X}}$ slightly larger than $\mathcal{X}$ for later technical derivation), $\text{Unif}(\bar{\mathcal{X}})\{\cdot\}$ is the uniform measure over $\bar{\mathcal{X}}$, and $C_1 > 0$ is an absolute constant.*

**Proof of Lemma 26:** Proposition 6 in (Pajor, 1998) prove a coarse covering number bound

$$\mathrm{N}(\mathrm{Gr}(d, r), \varrho_{\mathrm{proj}}, \varepsilon) \leq C^{\frac{r(d-r)}{\varepsilon}}$$

where $C > 0$ is an absolute constant; this coarse estimate is exponential rather than polynomial in $\varepsilon$, so it is used only for preliminary supporting purposes. For every $\mathcal{V} \in \mathrm{Gr}(d, r)$, by the homogeneity of the Grassmannian (under the action of $O(d)$), the $\varrho_{\mathrm{proj}}$-ball $B_{\mathrm{proj}}(\mathcal{V}, \varepsilon)$ has volume independent of its center. We therefore refer to this common value as the volume of an $\varepsilon$–$\varrho_{\mathrm{proj}}$ ball, written as $\mathrm{Vol}(\varepsilon - \varrho_{\mathrm{proj}}$ ball). By the definition of covering number (see Definition 5 and the subsequent inequality for background), we have that

$$\mathrm{N}(\mathrm{Gr}(d, r), \varrho_{\mathrm{proj}}, \varepsilon) \cdot \mathrm{Vol}(\varepsilon - \varrho_{\mathrm{proj}} \text{ ball}) \geq \mathrm{Vol}(\mathrm{Gr}(d, r)),$$

then for the uniform prior $\nu = \mathrm{Unif}(\mathrm{Gr}(d, r))$, we have that for every $\bar{\mathcal{V}} \in \mathrm{Gr}(d, r)$,

$$\log \frac{1}{\nu(B_{\varrho_{\mathrm{proj}}}(\bar{\mathcal{V}}, \varepsilon))} = \log \frac{\mathrm{Vol}(\mathrm{Gr}(d, r))}{\mathrm{Vol}(\varepsilon - \varrho_{\mathrm{proj}} \text{ ball})} \leq r(d - r) \frac{\log C}{\varepsilon}.$$

Note that $\varrho_{\mathrm{proj}}$ is not the target metric; our goal is the ellipsoidal metric $\varrho_{\mathrm{proj},\Sigma}$. Taking $\varepsilon = 1/\sqrt{2}$, we obtain:

$$\log \frac{1}{\nu(B_{\varrho_{\mathrm{proj}}}(\bar{\mathcal{V}}, 1/\sqrt{2}))} \leq C_1 r(d - r), \tag{99}$$

where $C_1 > 0$ is an absolute constant. By Lemma 25, we have that inside the ball $B_{\varrho_{\mathrm{proj}}}(\bar{\mathcal{V}}, 1/\sqrt{2})$, by choosing $\bar{\mathcal{V}}$ as the reference subspace, the graph parameterization $X$ of $\mathcal{V}$ satisfies

$$\|X\|_{\mathrm{op}} \leq 1,$$

which follows from that if $\varrho_{\mathrm{proj}}(\mathcal{V}(X), \bar{\mathcal{V}}) \leq 1/\sqrt{2}$ (i.e., $\sin \theta_r \leq 1/\sqrt{2}$), we have $\|X\|_{\mathrm{op}} \leq 1$. See (95) for the definition of this graph chart parameterization; the existence and uniqueness of the parameterization $X$ is by Lemma 24 (local bijection of graph chart). Furthermore, again by Lemma 24 and Lemma 25, $\mathcal{X} = \{X \in \mathbb{R}^{(d-r)r} : \|X\|_{\mathrm{op}} \leq 1\}$ satisfies ($\cong$ means isomorphism/bijection)

$$B_{\varrho_{\mathrm{proj}}}(\bar{\mathcal{V}}, 1/\sqrt{2}) \cong \mathcal{X} \subset \bar{\mathcal{X}} \cong B_{\varrho_{\mathrm{proj}}}(\bar{\mathcal{V}}, 2/\sqrt{5}). \tag{100}$$

Let

$$\mu_{\bar{\mathcal{V}}} = \mathrm{Unif}(B_{\mathrm{proj}}(\bar{\mathcal{V}}, 2/\sqrt{5})), \quad \mu(\mathcal{V}) = \int \nu(\bar{\mathcal{V}}) \mu_{\bar{\mathcal{V}}}(\mathcal{V}) d\bar{\mathcal{V}} = \mathrm{Unif}(\mathrm{Gr}(d, r)).$$

Then we have

$$
\begin{aligned}
\log \frac{1}{\mu(B_{\varrho_{\mathrm{proj},\Sigma}}(\mathcal{V}, \varepsilon))} &= \log \frac{1}{\int \nu(\bar{\mathcal{V}}) \mu_{\bar{\mathcal{V}}}(B_{\varrho_{\mathrm{proj},\Sigma}}(\mathcal{V}, \varepsilon)) d\bar{\mathcal{V}}} \\
&= \log \frac{1}{\int \nu(\bar{\mathcal{V}}) \mu_{\bar{\mathcal{V}}}(B_{\varrho_{\mathrm{proj},\Sigma}}(\mathcal{V}, \varepsilon) \cap B_{\mathrm{proj}}(\bar{\mathcal{V}}, 2/\sqrt{5})) d\bar{\mathcal{V}}} \\
&\leq \log \frac{1}{\nu(B_{\varrho_{\mathrm{proj}}}(\mathcal{V}, 1/\sqrt{2})) \min_{\bar{\mathcal{V}} \in B_{\varrho_{\mathrm{proj}}}(\mathcal{V}, 1/\sqrt{2})} \mu_{\bar{\mathcal{V}}}(X' \in \bar{\mathcal{X}} : \varrho_{\mathrm{proj},\Sigma}(\mathcal{V}(X), \mathcal{V}(X')) \leq \varepsilon)} \\
&\leq C_1 r(d - r) + \sup_{X \in \mathcal{X}} \log \frac{1}{\mathrm{Unif}(\bar{\mathcal{X}})\{X' \in \bar{\mathcal{X}} : \varrho_{\mathrm{proj},\Sigma}(\mathcal{V}(X), \mathcal{V}(X')) \leq \varepsilon\}},
\end{aligned}
$$

where the first inequality is by restricting $\bar{\mathcal{V}}$ to $B_{\varrho_{\mathrm{proj}}}(\mathcal{V}, 1/\sqrt{2})$; and the second inequality is by (99) as well as the bijection stated in (100) and Lemma 24. Note that we use different radius here than in $\mu_{\bar{\mathcal{V}}}$ to enusre that the set $\bar{\mathcal{X}}$ for $X'$, which is inside the uniform distribution in the final bound, to be larger than the domain $\mathcal{X}$ for $X$ to take sup. This will help later technical derivation.

$\square$

### E.5 DECOMPOSITION AND LIPSCHITZ PROPERTIES INSIDE GRAPH CHART

We apply a non-perturbative analysis to the ellipsoidal projection metric.

**Lemma 27 (Non-Perturbative Decomposition of Projector Difference)** *Let $X, X' \in \mathbb{R}^{(d-r) \times r}$ be two matrices. Given any reference subspace $\bar{\mathcal{V}}$, consider the graph chart $\phi_{\bar{\mathcal{V}}} : X \mapsto \mathcal{V}(X)$ defined in (95). Then the difference between two projectors $\mathcal{P}_{\mathcal{V}(X)}$, $\mathcal{P}_{\mathcal{V}(X')}$ be decomposed as follows:*

$$\mathcal{P}_{\mathcal{V}(X)} - \mathcal{P}_{\mathcal{V}(X')}$$
$$= \mathcal{P}_{\mathcal{V}(X)_\perp} \begin{pmatrix} 0 \\ I_{d-r} \end{pmatrix} (X - X') \begin{pmatrix} I_r & 0 \end{pmatrix} \mathcal{P}_{\mathcal{V}(X')} + \mathcal{P}_{\mathcal{V}(X)} \begin{pmatrix} I_r \\ 0 \end{pmatrix} (X^\top - X'^\top) \begin{pmatrix} 0 & I_{d-r} \end{pmatrix} \mathcal{P}_{\mathcal{V}(X')_\perp}.$$

**Proof of Lemma 27:**   The projector is invariant under orthogonal transformations of the ambient space $R^d$. We can therefore choose a coordinate system that simplifies the calculations without loss of generality. By the matrix representation (98) (which, without loss of generality, uses a convenient orthogonal basis specified by (97)), we denote

$$A(X) = \begin{pmatrix} I_r \\ X \end{pmatrix}, \quad M(X) = (I_r + X^\top X)^{-1},$$

and have the following facts:

$$V(X) = A(X) M(X)^{1/2},$$
$$\mathcal{P}_{\mathcal{V}(X)} = A(X) M(X) A(X)^\top = A(X) M(X) \begin{pmatrix} I_r & X^\top \end{pmatrix} \tag{101}$$
$$\mathcal{P}_{\mathcal{V}(X)} - \mathcal{P}_{\mathcal{V}(X')} = A(X) M(X) A(X)^\top - A(X') M(X') A(X')^\top$$
$$A(X) M(X) = \mathcal{P}_{\mathcal{V}(X)} \begin{pmatrix} I_r \\ 0 \end{pmatrix} \tag{102}$$
$$A(X) M(X) X^\top = \mathcal{P}_{\mathcal{V}(X)} \begin{pmatrix} 0 \\ I_{d-r} \end{pmatrix}, \tag{103}$$

where (102) and (103) are straightforward consequences of (101).

We begin with a non-perturbative decomposition:

$$\mathcal{P}_{\mathcal{V}(X)} - \mathcal{P}_{\mathcal{V}(X')}$$
$$= A(X) M(X) A(X)^\top - A(X') M(X') A(X')^\top$$
$$= (A(X) - A(X')) M(X') A(X')^\top + A(X)(M(X) - M(X')) A(X')^\top + A(X) M(X)(A(X) - A(X'))^\top. \tag{104}$$

We continue to decompose each term non-perturbatively. First,

$$(A(X) - A(X')) M(X') A(X')^\top$$
$$= \begin{pmatrix} 0 \\ X - X' \end{pmatrix} M(X') A(X')^\top$$
$$= \begin{pmatrix} 0 \\ I_{d-r} \end{pmatrix} (X - X') M(X') A(X')^\top$$
$$= \begin{pmatrix} 0 \\ I_{d-r} \end{pmatrix} (X - X') \begin{pmatrix} I_r & 0 \end{pmatrix} \mathcal{P}_{\mathcal{V}(X')}, \tag{105}$$

where the last equality uses the fact (102) and symmetry of $\mathcal{P}_{\mathcal{V}(X)}$.

Second, because we have the non-perturbative decomposition

$$M(X) - M(X')$$
$$= (I_r + X^\top X)^{-1} \left( (I_r + X'^\top X') - (I_r + X^\top X) \right) (I_r + X'^\top X')^{-1}$$
$$= (I_r + X^\top X)^{-1} \left( X'^\top X' - X^\top X \right) (I_r + X'^\top X')^{-1}$$
$$= (I_r + X^\top X)^{-1} \left( X^\top (X' - X) + (X'^\top - X^\top) X' \right) (I_r + X'^\top X')^{-1}$$
$$= M(X) X^\top (X' - X) M(X') + M(X)(X'^\top - X^\top) X' M(X'),$$

we have

$$A(X)(M(X) - M(X'))A(X')^\top$$

$$= A(X)M(X)X^\top(X' - X)M(X')A(X')^\top + A(X)M(X)(X'^\top - X^\top)X'M(X')A(X')^\top$$

$$= -\mathcal{P}_{\mathcal{V}(X)} \begin{pmatrix} 0 \\ I_{d-r} \end{pmatrix} (X - X') \begin{pmatrix} I_r & 0 \end{pmatrix} \mathcal{P}_{\mathcal{V}(X')} - \mathcal{P}_{\mathcal{V}(X)} \begin{pmatrix} I_r \\ 0 \end{pmatrix} (X^\top - X'^\top) \begin{pmatrix} 0 & I_{d-r} \end{pmatrix} \mathcal{P}_{\mathcal{V}(X')},$$

$$(106)$$

where the last equality uses the fact (102) and the fact (103).

Third, we have

$$A(X)M(X)(A(X) - A(X'))^\top$$

$$= A(X)M(X) \begin{pmatrix} 0 & X^\top - X'^\top \end{pmatrix}$$

$$= \mathcal{P}_{\mathcal{V}(X)} \begin{pmatrix} I_r \\ 0 \end{pmatrix} (X^\top - X'^\top) \begin{pmatrix} 0 & I_{d-r} \end{pmatrix}, \tag{107}$$

where the last equality uses the fact (102).

Substituting (105), (106), (107) back into (104), we have

$$\mathcal{P}_{\mathcal{V}(X)} - \mathcal{P}_{\mathcal{V}(X')}$$

$$= \begin{pmatrix} 0 \\ I_{d-r} \end{pmatrix} (X - X') \begin{pmatrix} I_r & 0 \end{pmatrix} \mathcal{P}_{\mathcal{V}(X')}$$

$$- \mathcal{P}_{\mathcal{V}(X)} \begin{pmatrix} 0 \\ I_{d-r} \end{pmatrix} (X - X') \begin{pmatrix} I_r & 0 \end{pmatrix} \mathcal{P}_{\mathcal{V}(X')} - \mathcal{P}_{\mathcal{V}(X)} \begin{pmatrix} I_r \\ 0 \end{pmatrix} (X^\top - X'^\top) \begin{pmatrix} 0 & I_{d-r} \end{pmatrix} \mathcal{P}_{\mathcal{V}(X')}$$

$$+ \mathcal{P}_{\mathcal{V}(X)} \begin{pmatrix} I_r \\ 0 \end{pmatrix} (X^\top - X'^\top) \begin{pmatrix} 0 & I_{d-r} \end{pmatrix}$$

$$= \mathcal{P}_{\mathcal{V}(X)_\perp} \begin{pmatrix} 0 \\ I_{d-r} \end{pmatrix} (X - X') \begin{pmatrix} I_r & 0 \end{pmatrix} \mathcal{P}_{\mathcal{V}(X')} + \mathcal{P}_{\mathcal{V}(X)} \begin{pmatrix} I_r \\ 0 \end{pmatrix} (X^\top - X'^\top) \begin{pmatrix} 0 & I_{d-r} \end{pmatrix} \mathcal{P}_{\mathcal{V}(X')_\perp},$$

where the last equality uses $I_d - \mathcal{P}_{\mathcal{V}(X)} = \mathcal{P}_{\mathcal{V}(X)_\perp}$ and $I_d - \mathcal{P}_{\mathcal{V}(X')} = \mathcal{P}_{\mathcal{V}(X')_\perp}$.

$$\square$$

Building upon the non-perturbative decomposition in Lemma 27, we have the following Lipschitz property of graph chart.

**Lemma 28 (Lipschitz of Graph Chart)** *Let $X, X' \in \mathbb{R}^{(d-r) \times r}$ be two matrices. Given any reference subspace $\mathcal{V}$, consider the graph chart defined in (98). Then the ellipsoidal projection metric is Lipschitz to ellipsoidal spectral metrics as follows: for every rank-$r$ PSD $\Sigma \in \mathbb{R}^{d \times d}$,*

$$\varrho_{\mathrm{proj},\Sigma}(\mathcal{V}(X), \mathcal{V}(X'))$$

$$\leq \left\| \left( \begin{pmatrix} 0 & I_{d-r} \end{pmatrix} \mathcal{P}_{\mathcal{V}(X)_\perp}^\top \Sigma \mathcal{P}_{\mathcal{V}(X)_\perp} \begin{pmatrix} 0 \\ I_{d-r} \end{pmatrix} \right)^{\frac{1}{2}} (X - X') \right\|_{\mathrm{op}} + \left\| \left( \begin{pmatrix} I_r & 0 \end{pmatrix} \mathcal{P}_{\mathcal{V}(X)}^\top \Sigma \mathcal{P}_{\mathcal{V}(X)} \begin{pmatrix} I_r \\ 0 \end{pmatrix} \right)^{\frac{1}{2}} (X^\top - X'^\top) \right\|_{\mathrm{op}}.$$

**Proof of Lemma 28:** By Lemma 27, we have

$$\varrho_{\mathrm{proj},\Sigma}(\mathcal{V}(X), \mathcal{V}(X')) = \left\| \Sigma^{\frac{1}{2}}(\mathcal{P}_{\mathcal{V}(X)} - \mathcal{P}_{\mathcal{V}(X')}) \right\|_{\mathrm{op}}$$

$$= \left\| \Sigma^{\frac{1}{2}} \mathcal{P}_{\mathcal{V}(X)_\perp} \begin{pmatrix} 0 \\ I_{d-r} \end{pmatrix} (X - X') \begin{pmatrix} I_r & 0 \end{pmatrix} \mathcal{P}_{\mathcal{V}(X')} + \Sigma^{\frac{1}{2}} \mathcal{P}_{\mathcal{V}(X)} \begin{pmatrix} I_r \\ 0 \end{pmatrix} (X^\top - X'^\top) \begin{pmatrix} 0 & I_{d-r} \end{pmatrix} \mathcal{P}_{\mathcal{V}(X')_\perp} \right\|_{\mathrm{op}}$$

$$\leq \left\| \Sigma^{\frac{1}{2}} \mathcal{P}_{\mathcal{V}(X)_\perp} \begin{pmatrix} 0 \\ I_{d-r} \end{pmatrix} (X - X') \right\|_{\mathrm{op}} + \left\| \Sigma^{\frac{1}{2}} \mathcal{P}_{\mathcal{V}(X)} \begin{pmatrix} I_r \\ 0 \end{pmatrix} (X^\top - X'^\top) \right\|_{\mathrm{op}}$$

$$= \left\| \left( \begin{pmatrix} 0 & I_{d-r} \end{pmatrix} \mathcal{P}_{\mathcal{V}(X)_\perp}^\top \Sigma \mathcal{P}_{\mathcal{V}(X)_\perp} \begin{pmatrix} 0 \\ I_{d-r} \end{pmatrix} \right)^{\frac{1}{2}} (X - X') \right\|_{\mathrm{op}} + \left\| \left( \begin{pmatrix} I_r & 0 \end{pmatrix} \mathcal{P}_{\mathcal{V}(X)}^\top \Sigma \mathcal{P}_{\mathcal{V}(X)} \begin{pmatrix} I_r \\ 0 \end{pmatrix} \right)^{\frac{1}{2}} (X^\top - X'^\top) \right\|_{\mathrm{op}}.$$

where the inequality follows from the triangle inequality and the facts that the spectral norms of $\mathcal{P}_{\mathcal{V}(X')}$, $\mathcal{P}_{\mathcal{V}(X')_\perp}$, and the two block–identity matrices are all at most 1 (the fact that spectral norms of projectors are at most 1 can be proved via the first inequality in Lemma 21); and the last equality is because for any matrices $A$, $B$ we have

$$\|\Sigma^{\frac{1}{2}} AB\|_{\mathrm{op}} = \sqrt{\|B^\top A^\top \Sigma AB\|_{\mathrm{op}}} = \|(A^\top \Sigma A)^{\frac{1}{2}} B\|_{\mathrm{op}}.$$

$\square$

We continue to present the following lemma, which implies that the projectors and the block-identity matrices in Lemma 28 only reduces the effective dimensions of the ellipsoidal map, and does not increase the eigenvalues (up to absolute constants).

**Lemma 29 (Spectral domination under contractions)** *Let $\Sigma \succeq 0$ be a $d \times d$ PSD matrix with ordered eigenvalues $\lambda_1(\Sigma) \geq \cdots \geq \lambda_d(\Sigma)$. Let $A \in \mathbb{R}^{d \times m}$ for some $m \leq d$ and write $s := \|A\|_{\mathrm{op}}$. Denote by $\mu_1 \geq \cdots \geq \mu_m$ the eigenvalues of $A^\top \Sigma A$. Then, for every $k = 1, \ldots, m$,*

$$\mu_m \leq s^2 \lambda_m(\Sigma).$$

**Proof of Lemma 29:** By the Courant–Fischer–Weyl max-min characterization (see, e.g., (Wikipedia contributors, 2025b)), we have

$$
\begin{aligned}
\lambda_k(A^\top \Sigma A) &= \min_{\substack{S \subset \mathbb{R}^d \\ \dim S = d-k+1}} \sup\{\|A^\top \Sigma^{\frac{1}{2}} x\|_2^2 : x \in S, \ \|x\|_2 = 1\} \\
&\leq s^2 \cdot \min_{\substack{S \subset \mathbb{R}^d \\ \dim S = d-k+1}} \sup\{\|\Sigma^{1/2} x\|_2 : x \in S, \ \|x\|_2 = 1\} \\
&= s^2 \lambda_k(\Sigma).
\end{aligned}
$$

$\square$

### E.6  Proof of the Main Result

From Lemma 26, to cover $\mathrm{Gr}(d, r)$ it suffices to cover the unit ball of $(d - r) \times r$ matrices under the ellipsoidal spectral metric. We are now ready to prove Lemma 3, the main result for ellipsoidal Grassmannian covering.

**Proof of Lemma 3:** We present the proof in multiple parts.

**Part 1: Applying Lemma 26.** Define $\mathcal{X} = \{X \in \mathbb{R}^{(d-r) \times r} : \|X\|_{\mathrm{op}} \leq 1\}$ and $\bar{\mathcal{X}} = \{X \in \mathbb{R}^{(d-r) \times r} : \|X\|_{\mathrm{op}} \leq 2\}$. By Lemma 26 (Pointwise Dimension Consequence of Finite Global Atlas), for $\mu = \mathrm{Unif}(\mathrm{Gr}(d, r))$, we have that for all $\mathcal{V} \in \mathrm{Gr}(d, r)$ and all $\varepsilon > 0$,

$$\log \frac{1}{\mu(B_{\varrho_{\mathrm{proj},\Sigma}}(\mathcal{V}, \varepsilon))} \leq C_1 r(d-r) + \sup_{X \in \mathcal{X}} \log \frac{1}{\mathrm{Unif}(\bar{\mathcal{X}})\{X' \in \bar{\mathcal{X}} : \varrho_{\mathrm{proj},\Sigma}(\mathcal{V}(X), \mathcal{V}(X')) \leq \varepsilon\}}, \tag{108}$$

where $C_1 > 0$ is an absolute constant.

Define the $(d - r) \times (d - r)$ positive definite matrices $H_1(X)$ and the $r \times r$ positive definite matrix $H_2(X)$ as the following

$$H_1(X) = \begin{pmatrix} 0 & I_{d-r} \end{pmatrix} \mathcal{P}_{\mathcal{V}(X)_\perp}^\top \Sigma \mathcal{P}_{\mathcal{V}(X)_\perp} \begin{pmatrix} 0 \\ I_{d-r} \end{pmatrix},$$

$$H_2(X) = \begin{pmatrix} I_r & 0 \end{pmatrix} \mathcal{P}_{\mathcal{V}(X)}^\top \Sigma \mathcal{P}_{\mathcal{V}(X)} \begin{pmatrix} I_r \\ 0 \end{pmatrix}.$$

By Lemma 28 (Lipschitz of Graph Chart), we have that

$$\varrho_{\mathrm{proj},\Sigma}(\mathcal{V}(X), \mathcal{V}(X')) \leq \|H_1(X)^{\frac{1}{2}}(X' - X)\|_{\mathrm{op}} + \|H_2(X)^{\frac{1}{2}}(X' - X)^\top\|_{\mathrm{op}}.$$

**Part 2: Volumetric Arguments.** We analyze the log density complexity in (108) via volumetric arguments.

**A technical step: ball inclusion via thresholding** In order to compute the log density complexity with the uniform prior, one needs the operator norm ball to be included in the support of the prior. Given a PSD matrix $H \in \mathbb{R}^{m \times m}$ and an eigenvalue threshold $\alpha$, assume its eigendecomposition is $H = U \operatorname{diag}(\beta_1, \cdots, \beta_m) U^\top$, define the thresholding function $T_\alpha$ by

$$T_\alpha(H) = U \operatorname{diag}(\max\{\beta_1, \alpha\}, \cdots, \max\{\beta_m, \alpha\}) U^\top.$$

Clearly this function only increases the metric. We further define the following two ellipsoidal metrics:

$$\varrho_1^2(X, X') = \|(X' - X)^\top \bar{H}_1(X)(X' - X)\|_{\mathrm{op}}, \quad \bar{H}_1(X) = T_{\varepsilon^2}(H_1(X))$$
$$\varrho_2^2(X, X') = \|(X' - X) \bar{H}_2(X)(X - X')^\top\|_{\mathrm{op}}, \quad \bar{H}_2(X) = T_{\varepsilon^2}(H_2(X))$$

We note that the two balls $B_{\varrho_1}(X, \varepsilon)$, $B_{\varrho_2}(X, \varepsilon)$ are contained in $\bar{\mathcal{X}}$, as we have applied the thresholding function to ensure this inclusion. For example, for the first ball, from

$$X' - X = \left(\bar{H}_1(X)\right)^{-1/2} \underbrace{\left(\bar{H}_1(X)\right)^{\frac{1}{2}}(X' - X)}_{\text{spectral norm} \leq \varepsilon \text{ for } X' \in B_{\varrho_1}(X, \varepsilon)},$$

we have (by using the $\varepsilon$ estimate from the second underbraced term above, and combining it with the thresholding guarantee $\lambda_{\min}(\bar{H}_1(X)) \geq \varepsilon^2$)

$$\|X' - X\|_{\mathrm{op}} \leq \lambda_{\min}(\bar{H}_1(X))^{-1/2} \cdot \varepsilon \leq 1,$$

which resulting in $\|X'\|_{\mathrm{op}} \leq \|X' - X\|_{\mathrm{op}} + \|X\|_{\mathrm{op}} \leq 2$ and thus $B_{\varrho_1}(X, \varepsilon) \subseteq \bar{\mathcal{X}}$. Similarly, we can show $B_{\varrho_2}(X, \varepsilon) \subseteq \bar{\mathcal{X}}$. this gives us the auxiliary ball-inclusion result:

$$B_{\varrho_1 + \varrho_2}(X, \varepsilon) \subseteq B_{\varrho_1}(X, \varepsilon) \cap B_{\varrho_2}(X, \varepsilon) \subseteq B_{\varrho_1}(X, \varepsilon) \cup B_{\varrho_2}(X, \varepsilon) \subseteq \bar{\mathcal{X}}. \tag{109}$$

Now we are ready to proceed with the main part of the proof. By Lemma 28 (Lipschitz of Graph Chart) and the fact that threholding only increase the spectral norm, the ellipsoidal projection metric is bounded by $\varrho_1 + \varrho_2$, so for any $X \in \mathcal{X}$,

$$\log \frac{1}{\mathrm{Unif}(\bar{\mathcal{X}})\{X' \in \bar{\mathcal{X}} : \varrho_{\mathrm{proj}, \Sigma}(\mathcal{V}(X), \mathcal{V}(X')) \leq \varepsilon\}}$$
$$\leq \log \frac{1}{\mathrm{Unif}(\bar{\mathcal{X}})\{X' \in \bar{\mathcal{X}} : \varrho_1(X, X') + \varrho_2(X, X') \leq \varepsilon\}}$$
$$= \log \frac{1}{\mathrm{Unif}(\bar{\mathcal{X}})\{B_{\varrho_1 + \varrho_2}(X, \varepsilon)\}} \tag{110}$$
$$= \frac{\mathrm{Vol}(\bar{\mathcal{X}})}{\mathrm{Vol}(B_{\varrho_1 + \varrho_2}(X, \varepsilon))}, \tag{111}$$

where the first equality uses the ball-inclusion result (109).

**Background on covering number.** Classical volume-ratio arguments give the following results on the covering number of balls in general normed space $\mathcal{Y}$. For a $p$-dimensional normed space equipped with the metric associated to its norm $\|\cdot\|$, we denote by $B(y, R)$ the ball in $\mathcal{Y}$ centered at $y \in \mathcal{Y}$ with radius $R$, and by $\mathsf{N}(\mathcal{Z}, \|\cdot\|, \varepsilon)$ the covering number of a subset $\mathcal{Z} \subseteq \mathcal{Y}$. Formally, we give the definition of covering number as follows.

**Definition 5 (Covering numbers)** *Let $(\mathcal{Y}, \|\cdot\|)$ be a normed space and let $\mathcal{Z} \subseteq \mathcal{Y}$. For $\varepsilon > 0$, a set $\mathcal{N} \subseteq \mathcal{Z}$ is an* internal $\varepsilon$–cover *of $\mathcal{Z}$ if for every $z \in \mathcal{Z}$ there exists $y \in \mathcal{N} \subseteq \mathcal{Z}$ with $\|z - y\| \leq \varepsilon$. The (internal) covering number is*

$$\mathsf{N}(\mathcal{Z}, \|\cdot\|, \varepsilon) := \min\{m : \exists \text{ internal } \varepsilon\text{–cover of } \mathcal{Z} \text{ with size } m\}.$$

A set $\mathcal{N}_{\text{ext}} \subseteq \mathcal{Y}$ (not necessarily inside $\mathcal{Z}$) is an external $\varepsilon$–cover of $\mathcal{Z}$ if for every $z \in \mathcal{Z}$ there exists $y \in \mathcal{N}_{\text{ext}}$ with $\|z - y\| \leq \varepsilon$. The external covering number is

$$\mathsf{N}_{\text{ext}}(\mathcal{Z}, \|\cdot\|, \varepsilon) := \min\{m : \exists \text{ external } \varepsilon\text{–cover of } \mathcal{Z} \text{ with size } m\}.$$

Internal covering numbers depend only on the metric induced on $\mathcal{Z}$, while external covering numbers also depend on the ambient space $\mathcal{Y}$. Throughout the paper, "covering number" means the internal one unless otherwise stated.

We now relate the internal and external covering numbers, showing they are equivalent up to a constant factor in the radius—and thus interchangeable for our purposes.

**Lemma 30 (Properties of External Covering Number)** *For every $\varepsilon > 0$ and $\mathcal{Z} \subseteq \mathcal{Y}$,*

$$\mathsf{N}_{\text{ext}}(\mathcal{Z}, \|\cdot\|, \varepsilon) \leq \mathsf{N}(\mathcal{Z}, \|\cdot\|, \varepsilon) \leq \mathsf{N}_{\text{ext}}(\mathcal{Z}, \|\cdot\|, \varepsilon/2). \tag{112}$$

*And the external covering number enjoys monotonicity under set inclusion: if $\mathcal{Z}_1 \subseteq \mathcal{Z}_2$ then $\mathsf{N}_{\text{ext}}(\mathcal{Z}_1, \|\cdot\|, \varepsilon) \leq \mathsf{N}_{\text{ext}}(\mathcal{Z}_2, \|\cdot\|, \varepsilon)$.*

**Proof of Lemma 30:** The left inequality in (112) is immediate since any internal cover is also an external cover. For the right inequality in (112), let $\{y_1, \ldots, y_m\} \subseteq \mathcal{Y}$ be an external $(\varepsilon/2)$–cover of $\mathcal{Z}$. For each $i$, define the (possibly empty) cell $V_i := \{z \in \mathcal{Z} : \|z - y_i\| \leq \varepsilon/2\}$. By the very definition of external $(\varepsilon/2)$-cover, every $z \in \mathcal{Z}$ is within distance $\varepsilon/2$ of some $y_i$; hence

$$\bigcup_{i=1}^{m} V_i = \mathcal{Z}.$$

If $V_i \neq \emptyset$, pick a representative $z_i \in V_i$. Then for any $z \in V_i$,

$$\|z - z_i\| \leq \|z - y_i\| + \|y_i - z_i\| \leq \varepsilon/2 + \varepsilon/2 = \varepsilon,$$

so the selected $\{z_i\} \subseteq \mathcal{Z}$ form an internal $\varepsilon$–cover. Hence $\mathsf{N}(\mathcal{Z}, \|\cdot\|, \varepsilon) \leq m = \mathsf{N}_{\text{ext}}(\mathcal{Z}, \|\cdot\|, \varepsilon/2)$. Lastly, the monotonicity under set inclusion for the external covering number is a straightforward consequence of its definition.

$\square$

Proposition 4.2.10 in Vershynin (2018) (the proof is elementary and clearly holds true for general metric in a normed space) states that for $\mathcal{Z} \subseteq \mathcal{Y}$ and general metric $\|\cdot\|$, we have that for any $y \in \mathcal{Y}$,

$$\frac{\text{Vol}(\mathcal{Z})}{\text{Vol}(B(y, \varepsilon))} \leq \mathsf{N}(\mathcal{Z}, \|\cdot\|, \varepsilon) \leq \frac{\text{Vol}(\mathcal{Z} + B(y, \frac{\varepsilon}{2}))}{\text{Vol}(B(y, \frac{\varepsilon}{2}))},$$

where the set $\mathcal{A} + \mathcal{B} := \{a + b : a \in \mathcal{A}, b \in \mathcal{B}\}$. When $\mathcal{Z}$ is convex and $B(y, \varepsilon) \subseteq \mathcal{Z}$, we further have

$$\frac{\text{Vol}(\mathcal{Z})}{\text{Vol}(B(y, \varepsilon))} \leq \mathsf{N}(\mathcal{Z}, \|\cdot\|, \varepsilon) \leq \frac{\text{Vol}(\mathcal{Z} + B(y, \frac{\varepsilon}{2}))}{\text{Vol}(B(y, \frac{\varepsilon}{2}))} \leq \frac{\text{Vol}(\frac{3}{2}\mathcal{Z})}{\text{Vol}(B(y, \frac{\varepsilon}{2}))} = 3^p \frac{\text{Vol}(\mathcal{Z})}{\text{Vol}(B(y, \varepsilon))}, \tag{113}$$

where $\lambda \mathcal{A} := \{\lambda a : a \in \mathcal{A}\}$ for $\lambda > 0$. Lastly, when the normed space $\mathcal{Y}$ is $p$–dimensional, for every $\varepsilon \in (0, R]$, setting $\mathcal{Z} = B(0, R)$ turns the above inequality (113) into the optimal covering number bound

$$\left(\frac{R}{\varepsilon}\right)^p \leq \mathsf{N}(B(0, R), \|\cdot\|, \varepsilon) \leq \left(\frac{3R}{\varepsilon}\right)^p. \tag{114}$$

Note that this result is for general normed space, not only for the $\ell_2$ norm in Euclidean space (see, e.g., display (1) in Pajor (1998); see also Milman & Schechtman (1986); Pisier (1999)).

**A technical step–lifting to product space.** Consider the product space $\mathbb{R}^{(d-r) \times r} \times \mathbb{R}^{(d-r) \times r}$ (of dimension $2 \times (d-r) \times r$). Given any $(d-r) \times (d-r)$ positive definite matrix $H_1$ and $r \times r$ positive definite matrix $H_2$, define the modified spectral norm by

$$\|(X_1, X_2) - (X_1', X_2')\|_{\text{op}, H_1, H_2} := \|H_1^{\frac{1}{2}}(X_1 - X_1')\|_{\text{op}} + \|H_2^{\frac{1}{2}}(X_2^\top - X_2'^\top)\|_{\text{op}}.$$

Consider the constrained set

$$\mathcal{S} := \{(X_1, X_2) \in \mathbb{R}^{(d-r)\times r} \times \mathbb{R}^{(d-r)\times r} : X_1 = X_2\} = \{(X, X) : X \in \mathbb{R}^{(d-r)\times r}\},$$

which is a normed space with dimension $(d-r)\times r$ (isomorphic to $\mathbb{R}^{(d-r)\times r}$), equipped with the modifed spectral norm

$$\|(X,X) - (X',X')\|_{\mathrm{op}, H_1, H_2} = \|H_1^{\frac{1}{2}}(X - X')\|_{\mathrm{op}} + \|H_2^{\frac{1}{2}}(X^\top - X'^\top)\|_{\mathrm{op}}.$$

Denote $B_{\mathrm{op}, H_1, H_2}^{\mathcal{S}}((X,X), R) = \{(X', X') \in \mathcal{S} : \|(X',X') - (X,X)\|_{\mathrm{op}, H_1, H_2} \leq R\}$ (the ball constrained in $\mathcal{S}$). Because there is a bijective, distance-preserving (isometric) map between $B_{\varrho_1 + \varrho_2}(X, \varepsilon)$ and $B_{\mathrm{op}, \bar{H}_1(X), \bar{H}_2(X)}^{\mathcal{S}}((X,X), \varepsilon)$, and likewise $B_{\mathrm{op}, I_{d-r}, I_r}^{\mathcal{S}}((0,0), 4)$ and $\bar{\mathcal{X}}$ (here 0 denotes the $(d-r)\times r$ 0 matrix), we obtain

$$\frac{\mathrm{Vol}(\bar{\mathcal{X}})}{\mathrm{Vol}(B_{\varrho_1 + \varrho_2}(X, \varepsilon))} = \frac{\mathrm{Vol}(B_{\mathrm{op}, I_{d-r}, I_r}^{\mathcal{S}}((0,0), 4))}{\mathrm{Vol}(B_{\mathrm{op}, \bar{H}_1(X), \bar{H}_2(X)}^{\mathcal{S}}((X,X), \varepsilon))}, \tag{115}$$

where the volume on $\mathcal{S}$ is defined via the surface area measure. (115) is exactly the objective we need to bound in (110).

Given $\varepsilon > 0$, by the property (113) of covering number, we have that for every $X \in \mathcal{X}$ and $\varepsilon > 0$,

$$\frac{\mathrm{Vol}(B_{\mathrm{op}, I_{d-r}, I_r}^{\mathcal{S}}((0,0), 4))}{\mathrm{Vol}(B_{\mathrm{op}, \bar{H}_1(X), \bar{H}_2(X)}^{\mathcal{S}}((X,X), \varepsilon))} \leq \mathrm{N}\big(B_{\mathrm{op}, I_{d-r}, I_r}^{\mathcal{S}}((0,0), 4), \|\cdot\|_{\mathrm{op}, \bar{H}_1(X), \bar{H}_2(X)}, \varepsilon\big). \tag{116}$$

**Remark on why lifting to product space double the degree of freedom.** We now lift the $\mathcal{S}$−constrained ball $B_{\mathrm{op}, I_{d-r}, I_r}^{\mathcal{S}}((0,0), 4)$ to the product space $\bar{\mathcal{X}} \times \bar{\mathcal{X}}$, using the covering number of the lifted product space to bound the covering number of the original space, in order to obtain an upper bound on (116) and (115). This is the reason why our final bound will scale (in the isotropic case) in the order $O((d-r)r\log\frac{1}{\varepsilon^2}) = O(2(d-r)r\log\frac{1}{\varepsilon})$ rather than the classical optimal order $\Theta((d-r)r\log\frac{1}{\varepsilon})$—the lifting to product space increase the number of freedom by a multiplicative factor of 2. Nevertheless, such difference is negligible in our theory.

For every $(X_1, X_2) \in \mathbb{R}^{(d-r)\times r} \times \mathbb{R}^{(d-r)\times r}$, every $(d-r)\times(d-r)$ matrix $H_1 \succ 0$, and every $r \times r$ matrix $H_2 \succ 0$, and radius $R$, denote $B_{\mathrm{op}, H_1, H_2}((X_1, X_2), R)$ to be the unconstrained ball in $\mathbb{R}^{(d-r)\times r} \times \mathbb{R}^{(d-r)\times r}$:

$$B_{\mathrm{op}, H_1, H_2}((X_1, X_2), R) := \{(X_1', X_2') \in \mathbb{R}^{(d-r)\times r} \times \mathbb{R}^{(d-r)\times r} : \|(X_1, X_2) - (X_1' - X_2')\|_{\mathrm{op}, H_1, H_2} \leq R\}.$$

Lifting to the product space can only increase the external covering number (monotonicity under set inclusion), and the external covering number is equivalent to the internal covering number up to a constant factor in the radius. To be specific, by Lemma 30, we have

$$\mathrm{N}\big(B_{\mathrm{op}, I_{d-r}, I_r}^{\mathcal{S}}((0,0), 4), \|\cdot\|_{\mathrm{op}, \bar{H}_1(X), \bar{H}_2(X)}, \varepsilon\big)$$
$$\leq \mathrm{N}_{\mathrm{ext}}\big(B_{\mathrm{op}, I_{d-r}, I_r}^{\mathcal{S}}((0,0), 4), \|\cdot\|_{\mathrm{op}, \bar{H}_1(X), \bar{H}_2(X)}, \varepsilon/2\big)$$
$$\leq \mathrm{N}_{\mathrm{ext}}\big(B_{\mathrm{op}, I_{d-r}, I_r}((0,0), 4), \|\cdot\|_{\mathrm{op}, \bar{H}_1(X), \bar{H}_2(X)}, \varepsilon/2\big)$$
$$\leq \mathrm{N}\big(B_{\mathrm{op}, I_{d-r}, I_r}((0,0), 4), \|\cdot\|_{\mathrm{op}, \bar{H}_1(X), \bar{H}_2(X)}, \varepsilon/2\big). \tag{117}$$

For every $X \in \mathcal{X}$, the ball-inclusion argument (109) is strong enough to imply that the unconstrained ball $B_{\mathrm{op}, \bar{H}_1(X), \bar{H}_2(X)}((X,X), \varepsilon) \subseteq \mathbb{R}^{(d-r)\times r} \times \mathbb{R}^{(d-r)\times r}$ is also included in the lifted ball $B_{\mathrm{op}, I_{d-r}, I_r}((0,0), 4)$, which gives that

$$B_{\mathrm{op}, \bar{H}_1(X), \bar{H}_2(X)}((X,X), \varepsilon/2) \subset B_{\mathrm{op}, \bar{H}_1(X), \bar{H}_2(X)}((X,X), \varepsilon) \subseteq B_{\mathrm{op}, I_{d-r}, I_r}((0,0), 4).$$

This satisfies the inclusion condition required to establish (113), and we have

$$\mathrm{N}\big(B_{\mathrm{op}, I_{d-r}, I_r}((0,0), 4), \|\cdot\|_{\mathrm{op}, \bar{H}_1(X), \bar{H}_2(X)}, \varepsilon/2\big) \leq 3^{2(d-r)r} \frac{\mathrm{Vol}(B_{\mathrm{op}, I_{d-r}, I_r}((0,0), 4))}{\mathrm{Vol}(B_{\mathrm{op}, \bar{H}_1(X), \bar{H}_2(X)}((X,X), \varepsilon/2))}$$
$$= 6^{2(d-r)r} \frac{\mathrm{Vol}(B_{\mathrm{op}, I_{d-r}, I_r}((0,0), 4))}{\mathrm{Vol}(B_{\mathrm{op}, \bar{H}_1(X), \bar{H}_2(X)}((X,X), \varepsilon))}. \tag{118}$$

**Part 3: Applying Change of Variable and Calculating the Jacobian Determinant.** Applying the standard change of variables

$$Y_1 = \bar{H}_1(X)^{1/2} X_1, \qquad Y_2 = X_2 \bar{H}_2(X)^{1/2},$$

the map on vectorized variables is

$$\text{vec}(Y_1) = (I_r \otimes \bar{H}_1(X)^{1/2}) \, \text{vec}(X_1), \qquad \text{vec}(Y_2) = (\bar{H}_2(X)^{\top 1/2} \otimes I_{d-r}) \, \text{vec}(X_2),$$

and the total Jacobian is

$$J(X) = \begin{pmatrix} I_r \otimes \bar{H}_1(X)^{1/2} & 0 \\ 0 & \bar{H}_2(X)^{\top 1/2} \otimes I_{d-r}. \end{pmatrix}$$

The two block–diagonal Jacobian determinants are

$$\left| \det(I_r \otimes \bar{H}_1(X)^{1/2}) \right| = (\det \bar{H}_1(X)^{1/2})^r = \det(\bar{H}_1(X))^{r/2},$$

$$\left| \det(\bar{H}_2(X)^{\top 1/2} \otimes I_{d-r}) \right| = (\det \bar{H}_2(X)^{1/2})^{d-r} = \det(\bar{H}_2(X))^{(d-r)/2}.$$

Multiplying the two factors, the total Jacobian of the linear change of variables is

$$\det(J(X)) = \det(\bar{H}_1(X))^{r/2} \det(\bar{H}_2(X))^{(d-r)/2}.$$

(We used $\det(B^\top) = \det(B)$ and that $\bar{H}_1(X), \bar{H}_2(X) \succ 0$, so determinants are positive.) By the change of variable formula in integration (see, e.g., Wikipedia contributors (2025a)), we have

$$\text{Vol}(B_{\text{op}, \bar{H}_1(X), \bar{H}_2(X)}((X, X), \varepsilon))$$

$$= \text{Vol}(B_{\text{op}, I_{d-1}, I_r}((X, X), \varepsilon)) \, (\det(J(X)))^{-1}$$

$$= \text{Vol}(B_{\text{op}, I_{d-1}, I_r}((X, X), \varepsilon)) \prod_{k=1}^{d-r} \lambda_k(\bar{H}_1(X))^{-r/2} \prod_{k=1}^{r} \lambda_k(\bar{H}_2(X))^{-(d-r)/2},$$

which implies

$$\frac{\text{Vol}(B_{\text{op}, I_{d-r}, I_r}((0,0), 4))}{\text{Vol}(B_{\text{op}, \bar{H}_1(X), \bar{H}_2(X)}((X, X), \varepsilon))} = \prod_{k=1}^{d-r} \lambda_k(\bar{H}_1(X))^{r/2} \prod_{k=1}^{r} \lambda_k(\bar{H}_2(X))^{(d-r)/2} \frac{\text{Vol}(B_{\text{op}, I_{d-r}, I_r}((0,0), 4))}{\text{Vol}(B_{\text{op}, I_{d-r}, I_r}((X, X), \varepsilon))}.$$

$$(119)$$

**Part 4: Proving the Final Bound.** For all $X \in \mathcal{X}$ and $\varepsilon \leq 1$, we have that $B_{\text{op}, I_{d-r}, I_r}((X, X), \varepsilon) \subseteq B_{\text{op}, I_{d-r}, I_r}((0,0), 4)$ and thus by (113) and (114), we have

$$\frac{\text{Vol}(B_{\text{op}, I_{d-r}, I_r}((0,0), 4))}{\text{Vol}(B_{\text{op}, I_{d-r}, I_r}((X, X), \varepsilon))} \leq \left( \frac{12}{\varepsilon} \right)^{2(d-r)r}. \tag{120}$$

Combining the above inequality (120) with (118) and (119), we have

$$\log \text{N}(B_{\text{op}, I_{d-r}, I_r}((0,0), 4), \, \|\cdot\|_{\text{op}, \bar{H}_1(X), \bar{H}_2(X)}, \, \varepsilon/2)$$

$$\leq 2(d-r)r \log \frac{72}{\varepsilon} + \frac{r}{2} \sum_{k=1}^{d-r} \log \lambda_k(\bar{H}_1(X)) + \frac{d-r}{2} \sum_{k=1}^{r} \log \lambda_k(\bar{H}_2(X))$$

$$= \frac{r}{2} \sum_{k=1}^{d-r} \log \frac{72^2 \lambda_k(\bar{H}_1(X))}{\varepsilon^2} + \frac{d-r}{2} \sum_{k=1}^{r} \log \frac{72^2 \lambda_k(\bar{H}_2(X))}{\varepsilon^2}. \tag{121}$$

Combing the above inequality (121) with (115), (116) and (117), we have that for all $X \in \mathcal{X}$,

$$\log \frac{\text{Vol}(\bar{\mathcal{X}})}{\text{Vol}(B_{\varrho_1 + \varrho_2}(X, \varepsilon))} \leq \frac{r}{2} \sum_{k=1}^{d-r} \log \frac{72^2 \lambda_k(\bar{H}_1(X))}{\varepsilon^2} + \frac{d-r}{2} \sum_{k=1}^{r} \log \frac{72^2 \lambda_k(\bar{H}_2(X))}{\varepsilon^2}. \tag{122}$$

Finally, combine the above inequality (122) with (108) and (110), we prove that for $\mu = \text{Unif}(\text{Gr}(d, r))$, we have that for all $\mathcal{V} \in \text{Gr}(d, r)$ and all $\varepsilon > 0$,

$$
\begin{aligned}
\log \frac{1}{\mu(B_{\varrho_{\text{proj},\Sigma}}(\mathcal{V}, \varepsilon))} &\leq C_1 r(d-r) + \frac{r}{2} \sum_{k=1}^{d-r} \log \frac{72^2 \lambda_k(\bar{H}_1(X))}{\varepsilon^2} + \frac{d-r}{2} \sum_{k=1}^{r} \log \frac{72^2 \lambda_k(\bar{H}_2(X))}{\varepsilon^2} \\
&= \frac{r}{2} \sum_{k=1}^{d-r} \log \frac{C\lambda_k(\bar{H}_1(X))}{\varepsilon^2} + \frac{d-r}{2} \sum_{k=1}^{r} \log \frac{C\lambda_k(\bar{H}_2(X))}{\varepsilon^2},
\end{aligned} \tag{123}
$$

where $C > 0$ is an absolute constant.

We end the proof by applying Lemma 29 and Lemma 21: since

$$
\begin{aligned}
\lambda_k(H_1(X)) &\leq \lambda_k(\mathcal{P}_{\mathcal{V}(X)_\perp}^\top \Sigma \mathcal{P}_{\mathcal{V}(X)_\perp}) \leq \lambda_k, \quad k = 1, \cdots, d-r; \\
\lambda_k(H_2(X)) &\leq \lambda_k(\mathcal{P}_{\mathcal{V}(X)}^\top \Sigma \mathcal{P}_{\mathcal{V}(X)}) \leq \lambda_k, \quad k = 1, \cdots, r,
\end{aligned}
$$

we have

$$
\begin{aligned}
\lambda_k(\bar{H}_1(X)) &\leq \max\{\lambda_k, \varepsilon^2\}, \quad k = 1, \cdots, d-r; \\
\lambda_k(\bar{H}_2(X)) &\leq \max\{\lambda_k, \varepsilon^2\}, \quad k = 1, \cdots, r.
\end{aligned}
$$

Substituting this bound to (123), we prove that for $\mu = \text{Unif}(\text{Gr}(d, r))$, we have that for all $\mathcal{V} \in \text{Gr}(d, r)$ and all $\varepsilon > 0$,

$$
\log \frac{1}{\mu(B_{\varrho_{\text{proj},\Sigma}}(\mathcal{V}, \varepsilon))} \leq \frac{r}{2} \sum_{k=1}^{d-r} \log \frac{C \max\{\lambda_k, \varepsilon^2\}}{\varepsilon^2} + \frac{d-r}{2} \sum_{k=1}^{r} \log \frac{C \max\{\lambda_k, \varepsilon^2\}}{\varepsilon^2},
$$

where $C > 0$ is an absolute constant.

$\square$

# F   FURTHER EXPLANATIONS AND PROOFS FOR GENERALIZATION BOUNDS (SECTION 4)

## F.1   COMPARISON WITH NORM BOUNDS, VC, AND NTK

We compare our generalization bound for fully connected DNN (Theorem 4) with three established lines of work: (i) bounds based on products of spectral norms, (ii) VC–dimension–type capacity bounds, and (iii) Neural Tangent Kernel (NTK) linearizations that are valid only in an infinitesimal neighborhood of initialization. Our framework yields *exponentially* tighter rates than norm–product bounds, refines VC–type statements into hypothesis– and data–dependent guarantees, and replaces infinitesimal linearization with a finite-scale, non-perturbative analysis that holds simultaneously for every trained hypothesis. For space, we defer the recovery of representative norm bounds to Appendix F.5.1 and a broader literature review to Appendix B.

**Norm Bounds:**   Starting from the Riemannian–dimension term in Theorem 4, apply the elementary inequality
$$
\log x \leq \log(1 + x) \leq x, \quad \forall x > 0
$$
together with $\sum_{k \geq 1} \lambda_k(F_{l-1}F_{l-1}^\top) = \|F_{l-1}(W, X)\|_{\mathsf{F}}^2$, we obtain: for each layer $l$,

$$
\sum_{k \geq 1} \log \left( \frac{\lambda_k(F_{l-1}F_{l-1}^\top) \|W_l\|_{\mathsf{F}}^2 L M_{l \to L}^2(W, \varepsilon)}{n \varepsilon^2} \right) \leq \frac{\|F_{l-1}(W, X)\|_{\mathsf{F}}^2 \|W_l\|_{\mathsf{F}}^2 L M_{l \to L}^2(W, \varepsilon)}{n \varepsilon^2}.
$$

Aggregating over layers and controlling $M_{l \to L}(W, \varepsilon)$ through $\prod_{i > l} \|W_i\|_{\text{op}}$, Theorem 4 yields the following rank–free, spectrally normalized consequence: uniformly over all $W \in B_{\mathsf{F}}(R)$

$$
(\mathbb{P} - \mathbb{P}_n)\ell(f(W, x), y) \leq O\left( \frac{\beta \|W\|_{\mathsf{F}}}{n} \sqrt{\sum_{l=1}^{L} L(d_l + d_{l-1}) \prod_{i \neq l} \|W_i\|_{\text{op}}^2} \right), \tag{124}
$$

where the $O(\cdot)$ notation hides only nonessential terms (see Corollary 1 in Appendix F.5.1); moreover, since $\|X\|_{\mathsf{F}} \leq \sqrt{n} \max_{1 \leq i \leq n} \|x_i\|_2$, the bound in (124) typically scales as $n^{-1/2}$. Therefore, we illustrate that the Riemannian–dimension bound in Theorem 4 is *exponentially tighter* than (124), a representative spectral-norm bound in the style of Bartlett et al. (2017); Neyshabur et al. (2018); Golowich et al. (2020); Pinto et al. (2025); Ledent et al. (2025). Appendix F.5.1 provides the full derivation and a detailed, side-by-side comparison.

**VC Dimension:** Let $L$ be the number of layers and $P = \sum_{l=1}^{L} d_l d_{l-1}$ be the total number of weights, Bartlett et al. (2019) prove a nearly tight VC–dimension bound $\mathrm{VCdim} \leq O\left(PL \log P\right)$, supported by a lower bound $\mathrm{VCdim} \geq \Omega\left(PL \log(P/L)\right)$. This VC dimension bound is roughly equivalent to be $L \sum_{l=1}^{L} d_l d_{l-1}$.[4] Our Riemannian Dimension bound, by contrast, substantially sharpens this rate: it removes the explicit dependence on depth $L$ and replaces the crude width factor with a (layerwise) effective-rank term.

**Neural Tangent Kernel (NTK):** Our approach uses an exact, non-perturbative expansion that preserves the finite-scale geometry of deep networks, going beyond NTK's Taylor linearizations, which remain valid only in an infinitesimal neighborhood around initialization (or equivalently, in the infinite-width "lazy" regime) (Jacot et al., 2018; Arora et al., 2019). Outside this regime the NTK approximation typically breaks down, limiting its explanatory power for practical networks. From a generalization standpoint, the initialization-centric, infinitesimal view suppresses the feature learning that actually drives generalization, and thus cannot account for why modern deep networks generalize well. In contrast, our results provide a finite-scale, pointwise theory that operates directly in practical regimes and explicitly captures feature learning through the spectra of the *learned* feature matrices.

## F.2 Algorithmic Implications and Excess Risk Bound

**Pointwise Dimension as Regularization and Excess Risk Bound.** Our bounds imply a natural regularization strategy for algorithm design. Given the pointwise generalization inequality (3) (e.g., the Riemannian Dimension bound in Theorem 4), we consider a regularized ERM objective that explicitly minimizes this complexity measure:

$$\hat{f} = \arg\min_{f \in \mathcal{F}} \left\{ \mathbb{P}_n \ell(f; z) + C\sqrt{\frac{d(f) + \log(2/\delta)}{n}} \right\}. \tag{125}$$

With probability at least $1 - \delta$, its excess risk is bounded by (compared to any benchmark $f^\star \in \mathcal{F}$):

$$\mathbb{P}\ell(\hat{f}; z) - \mathbb{P}\ell(f^\star; z)$$
$$\leq \inf_{f \in \mathcal{F}} \left\{ \mathbb{P}_n \ell(f; z) + C\sqrt{\frac{d(f) + \log(2/\delta)}{n}} \right\} - \mathbb{P}\ell(f^\star; z) \tag{126}$$
$$\leq (C + \sqrt{1/2})\sqrt{\frac{d(f^\star) + \log(2/\delta)}{n}};$$

see Appendix F.4 for full proof. Thus we obtain a problem–dependent oracle bound of order $\sqrt{d(f^\star)/n}$ that adapts to the optimal hypothesis $f^\star$.

**From Explicit Regularization to Implicit Bias of Practical Algorithms.** Since modern optimizers like SGD routinely drive empirical risk to near-zero, convergence analysis alone offers limited insight into generalization. The central theoretical challenge is therefore not determining *whether* a minimum is reached, but identifying *which* of the infinite interpolating solutions the optimizer selects. Plain ERM is insufficient for this task: without constraints on pointwise dimension, an empirical risk minimizer yields no guarantee of controlled excess risk. In contrast, our RD–regularized objective (125) explicitly enforces the low-complexity structure required for the generalization bound

---

[4]The extra factor $L$ beyond parameter count in VCdim is essentially unavoidable: for nonlinear compositional models, VC/packing dimensions depend on the logarithm of a global worst-case Lipschitz constant, and in depth$-L$ networks that constant grows multiplicatively across layers, yielding an additional linear dependence on $L$.

in (126). Although this intuition is rooted in the earliest practices of deep learning, our pointwise theory rigorously articulates the underlying mathematical reasoning.

This motivates a concrete agenda for optimization in deep learning: characterize algorithms whose implicit bias drives iterates toward solutions with *low pointwise complexity*, in particular low *Riemannian Dimension (RD)*. Analogous phenomena are well documented in linear and kernel settings: gradient descent converges to max–margin (logistic loss) or minimum–norm (least squares) solutions (Soudry et al., 2018; Gunasekar et al., 2018), iterate–averaged SGD behaves like ridge regression (Neu & Rosasco, 2018), and "ridgeless" kernel regression can generalize with an optimally zero ridge parameter (Liang & Rakhlin, 2020); see Vardi (2023) for a survey. Our regularizer in (125), based on pointwise dimension and, in particular, the RD from Theorem 4, is strictly more informative than any single norm, making it a natural target for such analyses.

Empirically, we say an algorithm exhibits *Riemannian–Dimension implicit bias* if it preferentially returns solutions with small RD despite RD's large dynamic range; in Section 5 we observe that SGD indeed finds low-RD solutions.

### F.3 PROOF OF THEOREM 4 IN SECTION 4

The proof consists of two steps: 1. Obtaining the Integral Bound on Generalization Gap; and 2. Obtaining the Expression of Riemannian Dimension.

**Step 1: Obtaining the Integral Bound on Generalization Gap.** As presented in (9), we construct the metric tensor

$$G_{\mathrm{NP}}(W) := \mathrm{blockdiag}\left(\cdots, LM_{l\to L}^2(W,\varepsilon)\cdot F_{l-1}(W,X)F_{l-1}^\top(W,X)\otimes I_{d_l}, \cdots\right).$$

By Lipschitz property of the loss function we have

$$\varrho_{n,\ell}(f(W',\cdot), f(W,\cdot)) = \sqrt{\mathbb{P}_n(\ell(f(W',x),y) - \ell(f(W,x),y))^2}$$

$$\leq \beta\sqrt{\mathbb{P}_n\|f(W',x) - f(W,x)\|_2^2} = \beta\varrho_n(W',W)$$

By Lemma 1 we have the metric dominating relationship: for every $W \in B_{\mathbf{F}}(R)$,

$$\sqrt{n}\varrho_n(f(W',\cdot), f(W,\cdot)) \leq \varrho_{G_{\mathrm{NP}}(W)}(W',W), \quad \forall W' \in B_{\mathbf{F}}(R).$$

Combining the above two inequalities we have

$$\varrho_{n,\ell}(f(W',\cdot), f(W,\cdot)) \leq \frac{\beta}{\sqrt{n}}\varrho_{G_{\mathrm{NP}}(W)}(W',W), \quad \forall W' \in B_{\mathbf{F}}(R).$$

By the metric domination lemma (Lemma 19), we have the pointwise dimension bound: for every $W \in B_{\mathbf{F}}(R)$,

$$\log\frac{1}{\pi(B_{\varrho_{n,\ell}}(f(W,\cdot),\varepsilon))} \leq \log\frac{1}{\pi(B_{\varrho_{G_{\mathrm{NP}}(W)}}(W,\sqrt{n}\varepsilon/\beta))},$$

By Theorem 3 (Riemannian Dimension Bound for DNN), we have that there exists a prior $\pi$ such that uniformly over every $W \in B_{\mathbf{F}}(R)$,

$$\log\frac{1}{\pi(B_{\varrho_{n,\ell}}(f(W,\cdot),\varepsilon))} \leq \log\frac{1}{\pi(B_{\varrho_{G_{\mathrm{NP}}(W)}}(W,\sqrt{n}\varepsilon/\beta))} \leq d_{\mathrm{R}}(W,\varepsilon/\beta), \tag{127}$$

where the definition of Riemannian Dimension $d_{\mathrm{R}}$ can be found in Theorem 3. By Theorem 1, we have that there exists an absolute constant $C_1$ such that with probability at least $1 - \delta$, uniformly over all $W \in B_{\mathbf{F}}(R)$,

$$(\mathbb{P} - \mathbb{P}_n)\ell(f(W,x),y) \leq C_1\left(\inf_{\alpha\geq 0}\left\{\alpha + \frac{1}{\sqrt{n}}\int_\alpha^1\sqrt{\log\left(\frac{1}{\pi(B_{\varrho_{n,\ell}}(f(W,\cdot),\varepsilon))}\right)}d\varepsilon\right\} + \sqrt{\frac{\log\frac{\log(2n)}{\delta}}{n}}\right)$$

$$\leq C_1\left(\inf_{\alpha\geq 0}\left\{\alpha + \frac{1}{\sqrt{n}}\int_\alpha^1\sqrt{d_{\mathrm{R}}(W,\varepsilon/\beta)}d\varepsilon\right\} + \sqrt{\frac{\log\frac{\log(2n)}{\delta}}{n}}\right)$$

$$= C_1\left(\inf_{\alpha\geq 0}\left\{\alpha + \frac{\beta}{\sqrt{n}}\int_\alpha^1\sqrt{d_{\mathrm{R}}(W,\varepsilon)}d\varepsilon\right\} + \sqrt{\frac{\log\frac{\log(2n)}{\delta}}{n}}\right). \tag{128}$$

where $C_1$ is an absolute constant; the first inequality uses Theorem 1; and the second inequality uses (127). This finishes the first part of Theorem 4 (integral upper bound).

**Step 2: Obtaining the Expression of Riemannian Dimension.** It remains to express the Riemannian Dimension $d_R$ by Theorem 3 and prove the second part of Theorem 4. By Theorem 3, we have that the expression of Riemannian Dimension is

$$d_R(W, \varepsilon) = \sum_{l=1}^{L} \left( (d_l + d_{l-1}) \cdot d_{\text{eff}}(LM_{l \to L}^2(W, \varepsilon) \cdot F_{l-1}(W, X)F_{l-1}(W, X)^\top, C_2 \max\{\|W\|_{\mathsf{F}}, R/2^n\}, \varepsilon) \right.$$
$$\left. + \log(d_{l-1}n) \right), \tag{129}$$

where $R = \sup_{\mathcal{W}} \|W\|_{\mathsf{F}}$, $C_2$ is an absolute constant, and the effective dimension (defined via (11)) is

$$d_{\text{eff}}(LM_{l \to L}^2(W, \varepsilon) \cdot F_{l-1}(W, X)F_{l-1}(W, X)^\top, C_2 \max\{\|W\|_{\mathsf{F}}, R/2^n\}, \varepsilon)$$
$$= \frac{1}{2} \sum_{k=1}^{r_{\text{eff}}[W,l]} \log \frac{8C_2^2 \max\{\|W\|_{\mathsf{F}}^2, R^2/4^n\} LM_{l \to L}^2(W, \varepsilon)\lambda_k(F_{l-1}F_{l-1}^\top)}{n\varepsilon^2}, \tag{130}$$

where $F_{l-1}$ is the abbreviation of $F_{l-1}(W, X)$ and $r_{\text{eff}}[W, l]$ is the abbreviation of $r_{\text{eff}}(LM_{l \to L}^2(W, \varepsilon) \cdot F_{l-1}(W, X)F_{l-1}(W, X)^\top, C_2 \max\{\|W\|_{\mathsf{F}}, R/2^n\}, \varepsilon)$.

Combining the identities (129) and (130), we have the pointwise dimension bound

$$d_R(W, \varepsilon)$$
$$= \sum_{l=1}^{L} \left( (d_l + d_{l-1}) \sum_{k=1}^{r_{\text{eff}}[W,l]} \log \frac{8C_2^2 \lambda_k(F_{l-1}F_{l-1}^\top) \cdot \max\{\|W\|_{\mathsf{F}}^2, R^2/4^n\}LM_{l \to L}^2(W, \varepsilon)}{n\varepsilon^2} + \log(d_{l-1}n) \right)$$
$$= \sum_{l=1}^{L} \left( (d_l + d_{l-1}) \sum_{k=1}^{r_{\text{eff}}[W,l]} \log \frac{8C_2^2 \lambda_k(F_{l-1}F_{l-1}^\top)}{n\varepsilon^2} \right.$$
$$\left. + (d_l + d_{l-1})r_{\text{eff}}[W, l] \cdot \log \left( M_{l \to L}^2(W, \varepsilon)L \max\{\|W\|_{\mathsf{F}}^2, R^2/4^n\} \right) + \log(d_{l-1}n) \right) \tag{131}$$

where $F_{l-1}$ is the abbreviation of $F_{l-1}(W, X)$; $r_{\text{eff}}[W, l]$ is the abbreviation of $r_{\text{eff}}(LM_{l \to L}^2(W, \varepsilon) \cdot F_{l-1}(W, X)F_{l-1}(W, X)^\top, C_2 \max\{\|W\|_{\mathsf{F}}, R/2^n\}, \varepsilon)$; and $C_2$ is an absolute constant.

This finishes the second part of Theorem 4 (expression of Riemannian Dimension).

Combining the integral upper bound (128) and the Riemannain dimension expression (131) concludes the proof of Theorem 4.

$\square$

### F.4 Proof for Regularized ERM in Section F.2

**Lemma 31 (Excess Risk Bound for Regularized ERM)** *Assume we have high-probability pointwise generalization bound in the form of* (3)*, and the loss $\ell(f; z)$ is uniformly bounded by $[0, 1]$. Then for the regularized ERM*

$$\hat{f} = \text{argmin}_f \left\{ \mathbb{P}_n \ell(f; z) + C\sqrt{\frac{d(f) + \log(2/\delta)}{n}} \right\},$$

*we have the excess risk bound against the population risk minimizer $f^\star := \arg\min_{\mathcal{F}} \mathbb{P}\ell(f; z)$: with probability at least $1 - \delta$,*

$$\mathbb{P}\ell(\hat{f}; z) - \mathbb{P}\ell(f^\star; z) \leq \inf_{f \in \mathcal{F}} \left\{ \mathbb{P}_n \ell(f; z) + C\sqrt{\frac{d(f) + \log(2/\delta)}{n}} \right\} - \mathbb{P}\ell(f^\star; z)$$

$$\leq (C + \sqrt{1/2})\sqrt{\frac{d(f^\star) + \log(2/\delta)}{n}}.$$

**Proof of Lemma 31:** By (3), for every $\delta \in (0,1)$, take $\delta_1 = \delta_2 = \delta/2$, we have that with probability at least $1 - \delta_1 - \delta_2 = 1 - \delta$, we have

$$\mathbb{P}\ell(\hat{f}; z) \leq \inf_{f \in \mathcal{F}} \left\{ \mathbb{P}_n \ell(f; z) + C\sqrt{\frac{d(f) + \log(1/\delta_1)}{n}} \right\}$$

$$\leq \mathbb{P}_n \ell(f^\star; z) + C\sqrt{\frac{d(f^\star) + \log(1/\delta_1)}{n}}$$

$$\leq \mathbb{P}\ell(f^\star; z) + \sqrt{\frac{\log(1/\delta_2)}{2n}} + C\sqrt{\frac{d(f^\star) + \log(1/\delta_1)}{n}}$$

$$= \mathbb{P}\ell(f^\star; z) + \sqrt{\frac{\log(2/\delta)}{2n}} + C\sqrt{\frac{d(f^\star) + \log(2/\delta)}{n}}$$

$$\leq \mathbb{P}\ell(f^\star; z) + (C + \sqrt{1/2})\sqrt{\frac{d(f^\star) + \log(2/\delta)}{n}}.$$

where the first inequality uses the bound of the form (3); the second inequality uses definition of $\hat{f}$; and the third inequality is an application of the Hoeffding's inequality (Lemma 17) at $f^\star$; the equality is by $\delta_1 = \delta_2 = \delta/2$; and the last inequality follows from the monotonicity of the square root function. Thus we have that the excess risk is bounded by

$$\mathbb{P}\ell(\hat{f}; z) - \mathbb{P}\ell(f^\star; z) \leq \inf_{f \in \mathcal{F}} \left\{ \mathbb{P}_n \ell(f; z) + C\sqrt{\frac{d(f) + \log(2/\delta)}{n}} \right\} - \mathbb{P}\ell(f^\star; z)$$

$$\leq (C + \sqrt{1/2})\sqrt{\frac{d(f^\star) + \log(2/\delta)}{n}}.$$

$\square$

### F.5 Improvement over Norm Bounds in Section F.1

#### F.5.1 Exponential Improvement to a Norm Bound and Comparison

We now provide norm-constrained bound from Theorem 4 without any expression $r_{\text{eff}}$ and $d_{\text{eff}}$ in the bound. Invoking the elementary bound $\log x \leq \log(1 + x) \leq x$, the effective dimension factor in Theorem 4 can be relaxed to the dimension-independent bound

$$\sum_{k=1}^{\infty} \log\left( \frac{\lambda_k\left(F_{l-1}F_{l-1}^\top\right) \|W\|_{\mathsf{F}}^2 \, L \, M_{l\to L}^2(W, \varepsilon)}{n \, \varepsilon^2} \right) \leq \frac{\sum_{k=1}^{\infty} \lambda_k(F_{l-1}F_{l-1}^\top) \|W\|_{\mathsf{F}}^2 \, L \, M_{l\to L}^2(W, \varepsilon)}{n \, \varepsilon^2}$$

$$\leq \frac{\|F_{l-1}(W, X)\|_{\mathsf{F}}^2 \|W\|_{\mathsf{F}}^2 \, L \, M_{l\to L}^2(W, \varepsilon)}{n \, \varepsilon^2},$$

and one arrives at the following rank–free consequence.

**Corollary 1 (Norm-constrained bound)** *Theorem 4 is never worse than: uniformly over all $W \in B_{\mathsf{F}}(R)$, the generalization gap $(\mathbb{P} - \mathbb{P}_n)\,\ell\big(f(W, x), y\big)$ is bounded by*

$$O\left( \frac{\beta \sqrt{\sum_{l=1}^{L} (d_l + d_{l-1}) L \|F_{l-1}(W, X)\|_{\mathsf{F}}^2 \|W\|_{\mathsf{F}}^2 \sup_{\varepsilon > 0} M_{l\to L}^2(W, \varepsilon)}}{n} + \sqrt{\frac{\beta^2 \sum_{l=1}^{L} \log(d_{l-1} n) + \log \frac{\log(2n)}{\delta}}{n}} \right).$$

(132)

*Furthermore, (132) implies the spectrally normalized bound: uniformly over $W \in B_{\mathsf{F}}(R)$, the generalization gap $(\mathbb{P} - \mathbb{P}_n)\,\ell\big(f(W, x), y\big)$ is bounded by*

$$O\left( \frac{\beta \|X\|_{\mathsf{F}} \|W\|_{\mathsf{F}} \cdot \sqrt{\sum_{l=1}^{L} L(d_l + d_{l-1}) \prod_{i \neq l} \|W_i\|_{\text{op}}^2}}{n} + \sqrt{\frac{\beta^2 \sum_{l=1}^{L} \log(d_{l-1} n) + L \log \frac{n \log \max\{R, 2\}}{\delta}}{n}} \right).$$

(133)

*Here in both* (132) *and* (133), $O$ *hides multiplicative absolute constants and two ignorable high-order terms:* $\frac{\beta\sqrt{\sum_{l=1}^{L}(d_l+d_{l-1})d_{l-1}}}{n^{5.5}}$ *and* $\frac{\beta\sqrt{\sum_{l=1}^{L}(d_l+d_{l-1})L\|F_{l-1}\|_{\mathsf{F}}^2 R^2 \sup_{\varepsilon>0} M_{l\to L^2(W,\varepsilon)}}}{n 2^n}$; *and in* (133), $O$ *additionally hides an ignorable high-order term* $\frac{\beta\sqrt{L\|W\|_{\mathsf{F}}^2\|X\|_{\mathsf{F}}^2 \sum_{l=1}^{L}(d_l+d_{l-1})(R/\sqrt{L-1})^{L-1}}}{n\max\{R,2\}^n}$.

Note that (132) and (133), the data matrix $X$ contain $n$ features vectors so their Frobenius norms scales with $\sqrt{n}$, making the order of both bounds to be $n^{-1/2}$.

**Discussion of Corollary 1:** We proceed in three paragraphs of discussion. First, we show that the Riemannian Dimension bound in Theorem 4 is *exponentially* tighter than the spectrally normalized bound in (133). Second, we offer a metric–tensor interpretation that clarifies the source of this improvement. Finally, we position (133) relative to the most representative spectrally normalized bounds (SNB) in the existing literature.

**I: Why the improvement is exponential.** Empirically one observes

$$\|F_{l-1}\|_{\mathsf{F}} \ll \prod_{i<l}\|W_i\|_{\mathrm{op}}\|X\|_{\mathsf{F}}, \qquad M_{l\to L}(W,\varepsilon) \le \sup_{W'\in B_{\varrho_n}(W,\varepsilon)}\prod_{i>l}\|W_i'\|_{\mathrm{op}}.$$

Combining this dramatic improvement with the *already–exponential* gain that comes *solely* from the elementary inequality $\log x \le \log(1+x) \le x$ (for $x \ge 0$), we conclude that Theorem 4 is *exponentially tighter* than (133). Therefore, Theorem 4 improves on Corollary 1 by an exponential factor.

**II: Metric tensor interpretation.** For understand the improvement deeper, we highlight that the spectral norm bound (133) can be equivalently viewed as replacing the metric tensor $G_{\mathrm{NP}}$ (9) used in Theorem 4 by the diagonal metric tensor

$$G_{\mathrm{SNB}}(W) = \mathrm{blockdiag}\Big(\dots, L\sup_{W'\in B_{\mathsf{F}}(R)}\prod_{k\neq l}\|W_k'\|_{\mathrm{op}}\|X\|_{\mathsf{F}}^2 \otimes I_{d_l\times d_{l-1}}, \dots\Big),$$

which is a far coarser relaxation that completely discards the learned feature $F_l(W,X)$.

**III: Relation to existing spectrally normalized bounds.** The bound in (133) is structurally close to the classical SNB results of Bartlett et al. (2017) and Neyshabur et al. (2018); the three bounds differ only in the *global ball* used to constraint the hypothesis class.

(a) Our bound (133) controls *all* layers simultaneously via the global Frobenius norm $\|W\|_{\mathsf{F}}$, hence the factor $\|W\|_{\mathsf{F}}$ in the numerator.

(b) Neyshabur et al. (2018) bounds each layer $l$ separately by its Frobenius norm $\|W_l\|_{\mathsf{F}}$. Strengthening their argument with Dudley's entropy integral (one-shot optimization in the original paper) gives

$$(\mathbb{P}-\mathbb{P}_n)\,\ell\big(f(W,x),y\big) \le \tilde{O}\Big(\frac{\beta\,\|X\|_{\mathsf{F}}\sqrt{\sum_{l=1}^{L}L^2(d_l+d_{l-1})\|W_l\|_{\mathsf{F}}^2\prod_{i\neq l}\|W_i\|_{\mathrm{op}}^2}}{n}+\sqrt{\frac{\log\frac{1}{\delta}}{n}}\Big). \tag{134}$$

Neither (133) nor (134) strictly dominates the other, since factors of the form $(\sum_l a_l)(\sum_l b_l)$ in (133) *vs.* factors of the form $L\sum_l a_l b_l$ in (134) can swap their relative order.

(c) Bartlett et al. (2017) replaces each Frobenius norm by the $\|\cdot\|_{2,1}$ norm, obtaining the tighter

$$(\mathbb{P}-\mathbb{P}_n)\,\ell\big(f(W,x),y\big) \le \tilde{O}\Big(\frac{\beta\,\|X\|_{\mathsf{F}}\big(\sum_l\|W_l\|_{2,1}^{2/3}\sum_l\big(\prod_{i\neq l}\|W_i\|_{\mathrm{op}}\big)^{2/3}\big)^{3/2}}{n}+\sqrt{\frac{\log\frac{1}{\delta}}{n}}\Big), \tag{135}$$

which improves on (133) and (134) thanks to the sharper $2,1$ norm. Extending our Riemannian-dimension analysis to the $2,1$ norm setting is an interesting direction for future work.

**(d)** Size-independent SNB bounds (pioneered by Golowich et al. (2020)) remove all depth/width dependence at the price of a worse scaling in $n$; incorporating their technique is left for future research.

**(e)** Pinto et al. (2025) impose explicit per-layer rank constraints on the weight matrices, thereby replacing the width factors in (134) with the corresponding ranks while leaving the product of spectral norms unchanged. Their bound includes an additional $C^L$ factor, which is subsequently removed by Ledent et al. (2025). Moreover, Ledent et al. (2025) seek to bridge the spectral–norm and parameter–count regimes by leveraging the Schatten–$p$ framework of Golowich et al. (2020), which interpolates between the product-of-spectral-norm regime ($p \to \infty$) and layerwise low-rank scalings ($p \to 0$). In the extreme $p \to 0$ limit, a representative consequence (Theorem E.8 of Ledent et al., 2025) yields

$$(\mathbb{P} - \mathbb{P}n)\ell\big(f(W,x),y\big) \leq \widetilde{O}\left( \frac{\sup_i \|x_i\|_2^2}{\sqrt{n}} \sqrt{\sum_{l=1}^{L} L(d_l + d_{l-1}) \operatorname{rank}(W_l)} \right).$$

Notably, the explicit dependence on the *ranks of the weight matrices*—rather than on spectrum-aware or *feature*-rank quantities—renders this result structurally similar to VC-dimension bounds (indeed, the proof proceeds via uniform covering numbers, and packing/VC dimensions for matrices are known to adapt to explicit rank constraints (Srebro et al., 2004)). As the authors acknowledge, this is a principal limitation: empirical evidence suggests that deep networks exhibit low rank in their *features* rather than their *weights*, a phenomenon this bound does not capture.

In any case, (133) is a representative SNB bound, and the key message in this subsection is that our Riemannian-Dimension result in Theorem 4 is *exponentially* sharper than (133).

### F.5.2 Proof of Corollary 1

The bound in Theorem 4 (or (128) in its proof) can be further upper bounded by the following form

$$(\mathbb{P} - \mathbb{P}_n)\ell(f(W,x),y) \leq C_1 \left( \frac{1}{\sqrt{n}} \int_0^1 \sqrt{d_R(W,\varepsilon)}d\varepsilon + \sqrt{\frac{\log \frac{\log(2n)}{\delta}}{n}} \right), \tag{136}$$

where the integral

$$\int_0^1 \sqrt{d_R(W,\varepsilon)}d\varepsilon = \inf_{\alpha \geq 0} \left( \int_0^\alpha \sqrt{d_R(W,\varepsilon)}d\varepsilon + \int_\alpha^1 \sqrt{d_R(W,\varepsilon)}d\varepsilon \right).$$

Building on this inequality, we structure the proof in four steps.

**Step 1: Bounding the Dominating Integral.** As we will take $\alpha$ to be very small so that the $\int_0^\alpha \sqrt{d_R(W,\varepsilon)}$ will be not exceed the order of $\int_\alpha^1 \sqrt{d_R(W,\varepsilon)}$, we firstly prove $\int_\alpha^1 \sqrt{d_R(W,\varepsilon)}d\varepsilon$. By the basic inequality $\log x \leq \log(1+x) \leq x$ for $x > 0$, we have

$$\sum_{k=1}^{r_{\mathrm{eff}}[W,l]} \log \left( \frac{8C_2^2 \lambda_k(F_{l-1}F_{l-1}^\top) \cdot \max\{\|W\|_{\mathsf{F}}^2, R^2/4^n\} LM_{l \to L}^2(W,\varepsilon)}{n\varepsilon^2} \right)$$

$$\leq \sum_{k=1}^{r_{\mathrm{eff}}[W,l]} \frac{8C_2^2 \lambda_k(F_{l-1}F_{l-1}^\top) \cdot \max\{\|W\|_{\mathsf{F}}^2, R^2/4^n\} LM_{l \to L}^2(W,\varepsilon)}{n\varepsilon^2}$$

$$\leq \sum_{k=1}^{d_{l-1}} \frac{8C_2^2 \lambda_k(F_{l-1}F_{l-1}^\top) \cdot \max\{\|W\|_{\mathsf{F}}^2, R^2/4^n\} LM_{l \to L}^2(W,\varepsilon)}{n\varepsilon^2}$$

$$= \frac{8C_2^2 \|F_{l-1}\|_{\mathsf{F}}^2 \{\|W\|_{\mathsf{F}}^2, R^2/4^n\} LM_{l \to L}^2(W,\varepsilon)}{n\varepsilon^2}, \tag{137}$$

where $F_{l-1}$ is the abbreviation of $F_{l-1}(W,X)$; $r_{\mathrm{eff}}[W,l]$ is the abbreviation of $r_{\mathrm{eff}}(LM_{l \to L}^2(W,\varepsilon) \cdot F_{l-1}(W,X)F_{l-1}(W,X)^\top, C_2 \max\{\|W\|_{\mathsf{F}}, R/2^n\}, \varepsilon)$; and $C_2$ is a positive absolute constant. Here

the second inequality uses the definition that $r_{\text{eff}}[W, l]$ as the effective rank of a $d_{l-1} \times d_{l-1}$ matrix, is no larger than the matrix width $d_{l-1}$; the first equality is because

$$\sum_{k=1}^{d_{l-1}} \lambda_k(F_{l-1}F_{l-1}^\top) = \text{Tr}(F_{l-1}F_{l-1}^\top) = \|F_{l-1}\|_{\mathbf{F}}^2, \tag{138}$$

a well-known property of the Frobenius norm (the squared Frobenius norm $\|F_{l-1}\|_{\mathbf{F}}^2$ equals trace of $F_{l-1}F_{l-1}^\top$). By (137) and Theorem 4 we have the Riemannian Dimension upper bound

$$d_{\mathbf{R}}(W, \varepsilon) \le 8C_2^2 \sum_{l=1}^{L} (d_l + d_{l-1}) \frac{\|F_{l-1}\|_{\mathbf{F}}^2 \max\{\|W\|_{\mathbf{F}}^2, R^2/4^n\} L M_{l \to L}^2(W, \varepsilon)}{n\varepsilon^2} + \sum_{l=1}^{L} \log(d_{l-1}n), \tag{139}$$

where $C_2$ is a positive absolute constant.

Taking (139) to the integral $\int_\alpha^1 \sqrt{d_{\mathbf{R}}(W, \varepsilon)} d\varepsilon$, we have

$$\int_\alpha^1 \sqrt{d_{\mathbf{R}}(W, \varepsilon)} d\varepsilon$$

$$\le 2\sqrt{2} C_2 \int_\alpha^1 \sqrt{\sum_{l=1}^{L} (d_l + d_{l-1}) \frac{\|F_{l-1}\|_{\mathbf{F}}^2 \max\{\|W\|_{\mathbf{F}}^2, R^2/4^n\} L M_{l \to L}^2(W, \varepsilon)}{n\varepsilon^2}} d\varepsilon + (1 - \alpha)\sqrt{\sum_{l=1}^{L} \log(d_{l-1}n)}$$

$$\le C_3 \sqrt{\frac{\sum_{l=1}^{L}(d_l + d_{l-1}) L \|F_{l-1}\|_{\mathbf{F}}^2 \max\{\|W\|_{\mathbf{F}}^2, R^2/4^n\} \sup_{\varepsilon > 0} M_{l \to L}^2(W, \varepsilon)}{n}} \log\frac{1}{\alpha} + (1 - \alpha)\sqrt{\sum_{l=1}^{L} \log(d_{l-1}n)},$$

where $C_3 > 0$ is an absolute constant.

**Step 2: Bounding the Rest Integral.** We then prove $\int_0^\alpha \sqrt{d_{\mathbf{R}}(W, \varepsilon)} d\varepsilon$. Again, by the basic inequality $\log x \le \log(1 + x) \le x$ for $x > 0$, we have

$$\sum_{k=1}^{r_{\text{eff}}[W, l]} \log\left(\frac{8C_2^2 \lambda_k(F_{l-1}F_{l-1}^\top) \cdot \max\{\|W\|_F^2, R^2/4^n\} L M_{l \to L}^2(W, \varepsilon)}{n\varepsilon^2}\right)$$

$$\le \sum_{k=1}^{d_{l-1}} \log\left(\frac{8C_2^2 \lambda_k(F_{l-1}F_{l-1}^\top) \cdot \max\{\|W\|_F^2, R^2/4^n\} L M_{l \to L}^2(W, \varepsilon)}{n\varepsilon^2}\right)$$

$$= \sum_{k=1}^{d_{l-1}} \log\left(\frac{8C_2^2 \lambda_k(F_{l-1}F_{l-1}^\top) \cdot \max\{\|W\|_F^2, R^2/4^n\} L M_{l \to L}^2(W, \varepsilon)}{n\alpha^2}\right) + d_{l-1} \log\frac{\alpha^2}{\varepsilon^2}$$

$$\le \frac{8C_2^2 \sum_{k=1}^{d_{l-1}} \lambda_k(F_{l-1}F_{l-1}^\top) \cdot \max\{\|W\|_F^2, R^2/4^n\} L M_{l \to L}^2(W, \varepsilon)}{n\alpha^2} + d_{l-1} \log\frac{\alpha^2}{\varepsilon^2}$$

$$= \frac{8C_2^2 \|F_{l-1}(W, X)\|_{\mathbf{F}}^2 \cdot \max\{\|W\|_F^2, R^2/4^n\} L M_{l \to L}^2(W, \varepsilon)}{n\alpha^2} + d_{l-1} \log\frac{\alpha^2}{\varepsilon^2}. \tag{140}$$

Taking (140) to the integral $\int_0^\alpha \sqrt{d_{\mathbf{R}}(W, \varepsilon)} d\varepsilon$, we have

$$\int_0^\alpha \sqrt{d_{\mathbf{R}}(W, \varepsilon)} d\varepsilon$$

$$\le 2\sqrt{2} C_2 \int_0^\alpha \sqrt{\sum_{l=1}^{L} (d_l + d_{l-1}) \frac{\|F_{l-1}\|_{\mathbf{F}}^2 \max\{\|W\|_{\mathbf{F}}^2, R^2/4^n\} L M_{l \to L}^2(W, \varepsilon)}{n\alpha^2}} d\varepsilon + \int_0^\alpha \sqrt{\sum_{l=1}^{L} (d_l + d_{l-1}) d_{l-1} \log\frac{\alpha^2}{\varepsilon^2}} d\varepsilon$$

$$\le C_4 \left(\sqrt{\frac{\sum_{l=1}^{L}(d_l + d_{l-1}) \|F_{l-1}\|_{\mathbf{F}}^2 \max\{\|W\|_{\mathbf{F}}^2, R^2/4^n\} L \sup_{\varepsilon > 0} M_{l \to L}^2(W, \varepsilon)}{n}} + \alpha\sqrt{\sum_{l=1}^{L}(d_l + d_{l-1}) d_{l-1}}\right),$$

where the second inequality holds by calculating the integral $\int_0^\alpha \sqrt{\log\left(\frac{\alpha^2}{\varepsilon^2}\right)}d\varepsilon = \alpha\sqrt{\frac{\pi}{2}}$, and $C_4 > 0$ is an absolute constant. Taking $\alpha = \frac{1}{n^5}$, the high-order term $\alpha\sqrt{\sum_{l=1}^L (d_l + d_{l-1})d_{l-1}}$ will be $\frac{\sqrt{\sum_{l=1}^L (d_l + d_{l-1})d_{l-1}}}{n^5}$ and is ignorable.

**Step 3: Combing the Two Integrals.** Combining Step 1 and Step 2, we get the full Riemannian Dimension integral upper bound

$$\frac{1}{\sqrt{n}}\int_0^\infty \sqrt{d_{\mathbf{R}}(W,\varepsilon)}d\varepsilon \leq O\left(\frac{\sqrt{\sum_{l=1}^L (d_l + d_{l-1})L\|F_{l-1}\|_{\mathbf{F}}^2\|W\|_{\mathbf{F}}^2 \sup_{\varepsilon>0} M_{l\to L}^2(W,\varepsilon)}}{n} + \sqrt{\frac{\sum_{l=1}^L \log(d_{l-1}n)}{n}}\right),$$

where $O$ hides multiplicative absolute constants and two ignorable high-order terms: $\frac{\sqrt{\sum_{l=1}^L (d_l+d_{l-1})d_{l-1}}}{n^{5.5}}$ and $\frac{\sqrt{\sum_{l=1}^L (d_l+d_{l-1})L\|F_{l-1}\|_{\mathbf{F}}^2 R^2 \sup_{\varepsilon>0} M_{l\to L}^2(W,\varepsilon)}}{n2^n}$.

Put this bound into Theorem 4 (or (136)), we have with probability at least $1 - \delta$, uniformly over all $W \in B_{\mathbf{F}}(R)$,

$$(\mathbb{P} - \mathbb{P}_n)\ell(f(W,x),y)$$
$$\leq O\left(\frac{\beta\sqrt{\sum_{l=1}^L (d_l + d_{l-1})L\|F_{l-1}\|_{\mathbf{F}}^2\|W\|_{\mathbf{F}}^2 \sup_{\varepsilon>0} M_{l\to L}^2(W,\varepsilon)}}{n} + \sqrt{\frac{\beta^2\sum_{l=1}^L \log(d_{l-1}n) + \log\frac{\log(2n)}{\delta}}{n}}\right),$$
(141)

where $O$ hides multiplicative absolute constants and two ignorable high-order terms: $\frac{\beta\sqrt{\sum_{l=1}^L (d_l+d_{l-1})d_{l-1}}}{n^{5.5}}$ and $\frac{\beta\sqrt{\sum_{l=1}^L (d_l+d_{l-1})L\|F_{l-1}\|_{\mathbf{F}}^2 R^2 \sup_{\varepsilon>0} M_{l\to L}^2(W,\varepsilon)}}{n2^n}$. Note that here $F_{l-1}(W;X) \in R^{d_{l-1}\times n}$ contains $n$ features vectors in dimension $d_{l-1}$ so its Frobenius norm $\|F_{l-1}\|_{\mathbf{F}}$ scales with $\sqrt{n}$ with respect to sample size; and $\sup_{\varepsilon>0} M_{l\to L}(W,\varepsilon)$ is the "one-point" Lipschitz constant at $W$ in the sense that

$$\|F_L(F_l(W',X),\{W_i'\}_{i=l+1}^L) - F_L(F_l(W,X),\{W_i'\}_{i=l+1}^L)\|_{\mathbf{F}}$$
$$\leq \left(\sup_\varepsilon M_{l\to L}(W,\varepsilon)\right)\|F_l(W',X) - F_l(W,X)\|_{\mathbf{F}}, \quad \forall W' \in B_{\mathbf{F}}(R).$$

This concludes the first generalization bound in Corollary 1.

**Step 4: Prove the Second Generalization Bound.** Now we continue to show that the bound in Corollary 1 is strictly better than the spectrally normalized bound. To see this, as we presented under Corollary 1, we have

$$\|F_{l-1}(W,X)\|_{\mathbf{F}}$$
$$=\|\sigma_{l-1}(W_{l-1}\cdots W_2\sigma_1(W_1 X))\|_{\mathbf{F}}$$
$$\leq \prod_{i<l} \|W_i\|_{\mathrm{op}} \cdot \|X\|_{\mathbf{F}},$$
(142)

by the property of spectral norm ($\|AB\|_{\mathbf{F}} \leq \|A\|_{\mathrm{op}}\|B\|_{\mathbf{F}}$), and the fact that all activation functions are $1-$Lipschitz in column.

In the meanwhile, we know that

$$\left(\sup_\varepsilon M_{l\to L}(W,\varepsilon)\right) \leq \sup_\varepsilon \prod_{i>l} \|W_i'\|_{\mathrm{op}},$$

again by the property of spectral norm ($\|AB\|_{\mathbf{F}} \leq \|A\|_{\mathrm{op}}\|B\|_{\mathbf{F}}$) and the fact that all activation functions are $1-$Lipschitz in column. This results in

$$\sup_\varepsilon \prod_{i>l} \|W_i'\|_{\mathrm{op}} \leq \sup_{W\in B_{\mathbf{F}}(R)} \prod_{i>l} \|W_i\|_{\mathrm{op}}.$$
(143)

Combining (142) and (143) together with (141), we have that for any $\delta \in (0,1)$, with probability at least $1 - \delta$, uniformly over every $W \in B_{\mathbf{F}}(R)$, we have

$$(\mathbb{P} - \mathbb{P}_n)\ell(f(W,x),y)$$

$$\leq O\left(\frac{\beta\sqrt{L\|W\|_{\mathbf{F}}^2\|X\|_{\mathbf{F}}^2 \cdot \sum_{l=1}^L (d_l + d_{l-1}) \prod_{i<l}\|W_i\|_{\mathrm{op}}^2 \sup_{W \in B_{\mathbf{F}}(R)} \prod_{i>l}\|W_i\|_{\mathrm{op}}^2}}{n}\right.$$

$$\left.+\sqrt{\frac{\beta^2 \sum_{l=1}^L \log(d_{l-1}n) + \log\frac{\log(2n)}{\delta}}{n}}\right), \tag{144}$$

where $O$ hides multiplicative absolute constants and two ignorable high-order terms: $\frac{\beta\sqrt{\sum_{l=1}^L (d_l+d_{l-1})d_{l-1}}}{n^{5.5}}$ and $\frac{\beta\sqrt{\sum_{l=1}^L (d_l+d_{l-1})L\|F_{l-1}\|_{\mathbf{F}}^2 R^2 \sup_{\varepsilon>0} M_{l\to L}^2(W,\varepsilon)}}{n2^n}$.

The next step is to use a multi-dimensional extension of the "uniform pointwise convergence" principle (resulting in pointwise generalization bound (35) in this paper) to give a conversion from the uniform convergence to the pointwise convergence. Denote the functional $T_l : B_{\mathbf{F}}(R) \to (0, R_l]$ is defined by

$$T_l(W) = \prod_{i \neq l}\|W_i\|_{\mathrm{op}}^2.$$

Since $\sum_{i \neq l}\|W_i\|_{\mathbf{F}}^2 \leq \|W\|_{\mathbf{F}}^2 \leq R^2$, we have $T_l(W) = \prod_{i \neq l}\|W_i\|_{\mathrm{op}}^2 \leq (R/\sqrt{L-1})^{2(L-1)}$ according to the AM-GM inequality. The bound in (144) implies that for any $l = 1, \cdots, L$, $\forall t_l \in (0, (R/\sqrt{L-1})^{2(L-1)})]$, with probability at least $1 - \delta$,

$$\sup_{W:T_l(W)\leq t_l,\forall l\in[L]} (\mathbb{P} - \mathbb{P}_n)\ell(f(W,x),y)$$

$$\leq O\left(\frac{\beta\sqrt{L\|W\|_{\mathbf{F}}^2\|X\|_{\mathbf{F}}^2 \cdot \sum_{l=1}^L (d_l + d_{l-1})t_l}}{n} + \sqrt{\frac{\beta^2 \sum_{l=1}^L \log(d_{l-1}n) + \log\frac{\log(2n)}{\delta}}{n}}\right). \tag{145}$$

With the smallest radius $r_0$ chosen to be $r_0 = (R/\sqrt{L-1})^{2(L-1)}/\max\{R,2\}^n$, and a grid of size $(\log_2(2\max_{W,l}\{T_l(W)\}/r_0))^k$ (partition each coordinate into $\log_2(2\max_{W,\ell}\{T_l(W)\}/r_0)$ dyadic scales, we can prove that for any $\delta \in (0,1)$, with probability at least $1 - \delta$, uniformly over every $W \in B_{\mathbf{F}}(R)$,

$$(\mathbb{P} - \mathbb{P}_n)\ell(f(W,x),y)$$

$$\leq O\left(\frac{\beta\sqrt{L\|W\|_{\mathbf{F}}^2\|X\|_{\mathbf{F}}^2 \sum_{l=1}^L (d_l + d_{l-1})\max\{4T_l^2(W), \frac{(R/\sqrt{L-1})^{2L-2}}{\max\{R,2\}^{2n}}\}}}{n}\right.$$

$$\left.+\sqrt{\frac{\beta^2 \sum_{l=1}^L \log(d_{l-1}n) + L\log\frac{n\log\max\{R,2\}}{\delta}}{n}}\right)$$

$$= O\left(\frac{\beta\sqrt{L\|W\|_{\mathbf{F}}^2\|X\|_{\mathbf{F}}^2 \cdot \sum_{l=1}^L (d_l + d_{l-1})\prod_{i\neq l}\|W_i\|_{\mathrm{op}}^2}}{n} + \sqrt{\frac{\beta^2 \sum_{l=1}^L \log(d_{l-1}n) + L\log\frac{n\log\max\{R,2\}}{\delta}}{n}}\right), \tag{146}$$

where $O$ hides multiplicative absolute constants and three ignorable high-order terms: $\frac{\beta\sqrt{\sum_{l=1}^L (d_l+d_{l-1})d_{l-1}}}{n^{5.5}}$, $\frac{\beta\sqrt{\sum_{l=1}^L (d_l+d_{l-1})L\|F_{l-1}\|_{\mathbf{F}}^2 R^2 \sup_{\varepsilon>0} M_{l\to L}^2(W,\varepsilon)}}{n2^n}$ and $\frac{\beta\sqrt{L\|W\|_{\mathbf{F}}^2\|X\|_{\mathbf{F}}^2 \sum_{l=1}^L (d_l+d_{l-1})}(R/\sqrt{L-1})^{L-1}}{n\max\{R,2\}^n}$. The proof of this multi-dimensional "uniform pointwise convergence" is essentially the same peeling argument as in Lemma 4, with the only change that we use multi-dimensional grid; alternatively, this can be proved by applying Lemma 4 for $k$ times, where at each step we remove one dimension functional and divided confidence by $\log_2(2R/r_0)$. We omit the repetitive proof details.

Now we see from (142) and (143) that the derived norm-constraint bound (141) implies the spectrally normalized bound (146). This completes the proof.

$\square$

