# OpenReview forum: "Pointwise Generalization in Deep Neural Networks"
_ICLR.cc/2026/Conference — ICLR 2026 Conference Withdrawn Submission_

### Official Review · Reviewer_9dbr · 2025-10-27

**Soundness:** 1
**Presentation:** 1
**Contribution:** 1
**Rating:** 2
**Confidence:** 4

**Summary:**

This paper gives new generalization bounds to explain the good generalization properties of deep neural networks, a long-standing topic in learning theory. The proposed bounds are called ``pointwise'', meaning that they apply for every hypothesis individually and are also data-dependent. The main technical elements are based on an application of PAC-Bayesian bounds with well-chosen probability distributions. This allows the authors to introduce an effective dimension, which is related to a fractal dimension notion of a data-dependent prior. In the case of fully-connected deep neural networks to a notion of Riemannian dimension, based, in particular, on the feature Gram matrices. Empirical studies support the theoretical findings.

**Strengths:**

- Understanding the generalization error of modern deep neural networks is an important topic.
- The introduced notions of effective dimensions might have interest on their own.

**Weaknesses:**

In my opinion, there are three main weaknesses.

 *1. Poor literature review*: The paper pretends to address the long-standing question of generalization and explicitly states "theory has not kept pace". Despite this, there is an important lack of literature review on the rich literature review. Most of the classical references are missed and the introduction barely contains citations. For instance, algorithmic stability or information-theoretic bounds are never mentioned. PAC-Bayes bound are mentioned but with almost no reference except (Dziugaite and Roy, 2017). NTK theory is mentioned but with no reference. The same remark can be made or uniform covers, VC dimension, and product of norms. Moreover, similar notions of intrinsic dimensions already appear in machine learning, but with no reference, see [1] for instance.
 In conclusion, the authors claim that theory has not kept pace on generalization without acknowledging the rich literature on this question.

*2. The proofs seem to contain critical mistakes*, such as:
 - Proof of Lemma 4: The proof is based on classical PAC-Bayes bounds but they cannot be applied here. Indeed, Lemma 9 is only true for a prior distribution $\pi$ that is independent of the data. It can be seen in the proof of thm 2.1 of [Alquier, 2024] (cited for lemma 9): the critical step is to apply Fubini's theorem to switch the prior and the data distribution, it is only possible if \pi does not depend on the data.
 - Proof of thm 1: First, at line 1030, the notation does not make sense because there is an expectation on z outside the sup, while z is already the integrand of $\mathbb{P} - \mathbb{P}_n$ according to your notation. Even if this is fixed, I am not convinced by the symmetrisation argument at line 1040 because the inside of the supremum is not symmetric in z and z'.

*3. Empirical section:* at line 81 and in the contributions section, the empirical validation section is clearly treated as though it was part of the main paper and not of the appendix. In my opinion, an effort should be made to compress the main paper so that a proper empirical section can be included in the main text.

Regarding, the proofs, please correct me if I am wrong.

Finally, here are some more minor issues:
 - Line 69: a effective -> an effective
 - The notation at equation 3 is confusing to me because $z$ because $z$ is the integrand on the left-hand side, so it should not appear in my opinion. Maybe something like $\mathbb{P}(\ell(f, \cdot))$ would be better.
 - Line 133 - 134, is a word missing in the sentence?
 - Line 901 - 902: too much space between so and on
 - Equation 15, I think it should be said that the uniform distribution is meant with respect to $\pi$
 - I don't think the notion of prior distribution has been properly defined.
 - Proof of Theorem 1: sometimes the $\otimes$ is missing in the products between distributions, is it a typo?


[1] "Fractal Structure and Generalization Properties of Stochastic Optimization Algorithms" Cameo et al., 2021.

**Questions:**

- Line 167, it is claimed that the bounds covers a rich class of models. To my understanding, if the point wise dimension grows as $d(f) \varepsilon^{-2}$, it means that the fractal dimension is at most 2, can you comment further on how this is restrictive?
- To my knowledge, eq. (14) is a step to prove eq. (13), why not go directly go to eq. (14)?

---

> ### Author Response · Authors · 2025-12-03
>
> **Dear Reviewer 9dbr**, we sincerely appreciate your invaluable and constructive comments, which are very helpful to improve the paper.
>
> **R1:  Poor literature review.**
>
> **A**: Thank you for pointing this out. We have added a dedicated *Related Work* section (Appendix B). There we now review both theoretical perspectives (algorithmic stability, PAC-Bayes, uniform covers, VC dimension, product of norms, NTK theory, and fractal dimensions) and empirical lines of work (intrinsic dimensions, neural collapse, dynamic NTK variants, etc.) on neural networks. We also discuss pointwise and non-perturbative foundations, connections to differential geometry and Lie algebras, and feature compression in deep models for vision and language (including information-theoretic viewpoints).
>
> **R2: The proofs seem to contain critical mistakes.**
>
> **A**: Thank you for carefully checking the proofs. You are correct that our previous argument for Theorem 1 had a local flaw related to the treatment of the prior, which should be data-independent. In the revision, we re-derive Theorem 1 (the pointwise generalization bound) using a new method and rewrite Section 2 accordingly. The issue is localized to Section 2 and is orthogonal to the main structural contributions in Sections 3–4, which already use a data-independent prior and remain unchanged. We have carefully re-checked every step; the proofs are now self-contained and mathematically rigorous, and all main claims of the paper continue to hold (with the prior now explicitly data-independent). We kindly invite the Reviewer to re-examine the revised version and we would very much welcome further discussion.
>
> **R3: Empirical section.**
>
> **A**: Thank you for this suggestion. We have reorganized the empirical section.
>
> **R4: Finally, here are some more minor issues.**
>
> **A**: Thank you for these detailed comments. We have carefully addressed all of the listed minor issues in the revised manuscript (typos, notation, and presentation).
>
> **Q1: Line 167, it is claimed that the bounds covers a rich class of models. To my understanding, if the point wise dimension grows as $d(f)\varepsilon^{-2}$, it means that the fractal dimension is at most 2, can you comment further on how this is restrictive?**
>
> **A**: Thank you for this question. In the revised version our generic-chaining bound (Theorem 1) states, uniformly for every $f\in\mathcal F$,
> \begin{align*}(\mathbb{P} -\mathbb{P}\_n) \ell(f;z)\leq C\left(\inf\_{\alpha>0}\left\\{\alpha+ \frac{1}{\sqrt{n}}\int\_{\alpha}^{1} \sqrt{\log \frac{1}{\pi(B\_{\varrho\_{n,\ell}}(f,\varepsilon))}}d\varepsilon\right\\}+\sqrt{\frac{\log\frac{\log (2n)}{\delta}}{n}}\right).
> \end{align*}
> If the point wise dimension grows as $d(f)\varepsilon^{-2}$,
> then
> \begin{align*}
>    \log \frac{1}{\pi(B_{\varrho_{n,\ell}}(f,\varepsilon))} = O (\sqrt{\frac{d(f)}{n}}).
> \end{align*}
> and the bound becomes
> \begin{align*}
>     (\mathbb{P} -\mathbb{P}_n) \ell(f;z)&\leq C\left(\inf\_{\alpha>0}\left\\{\alpha+ \frac{1}{\sqrt{n}}\int\_{\alpha}^{1} \sqrt{\log \frac{1}{\pi(B\_{\varrho\_{n,\ell}}(f,\varepsilon))}}d\varepsilon\right\\}+\sqrt{\frac{\log\frac{\log (2n)}{\delta}}{n}}\right)\\\\
>     & = C\left(\inf\_{\alpha>0}\left\\{\alpha+ \sqrt{\frac{d(f)}{n}} \log \frac{1}{\alpha}\right\\}+\sqrt{\frac{\log\frac{\log (2n)}{\delta}}{n}}\right)\\\\
>     &= O\left(\sqrt{\frac{d(f)}{n}}\right),
> \end{align*}  where $O$ hides logarithmic factors.
>
> Thus, the bound adapts to this quantity $d(f)$ and yields a non-trivial $\sqrt{d(f)/n}$-type generalization rate. In particular, it applies to a broad family of models, with their complexity captured in a pointwise way by $d(f)$. Here $d(f)$ can vary across functions and need not be tied to the ambient parameter dimension, so the rate remains informative for many over-parameterized models.
>
>
> **Q2: To my knowledge, eq. (14) is a step to prove eq. (13), why not go directly go to eq. (14)?**
>
> **A**: Thanks for this review. We have revised this proof.

---

### Official Review · Reviewer_1RYM · 2025-10-29

**Soundness:** 2
**Presentation:** 2
**Contribution:** 3
**Rating:** 6
**Confidence:** 3

**Summary:**

This paper aims to sharp understand the generalization of neural network. This is done by developing pointwise, spectrum-aware PAC-Bayesian generalization bounds. Their prior is then constructed considering the low-rank features in FFNs. The new bounds reveal that generalization can be captured by finite-scale Riemannian dimension, which also empirically better capture generalization under overparameterization.

**Strengths:**

- This paper provides deeper understanding by connecting generalization with finite-scale Riemannian dimension, which is then connected to effective dimensions of FFNs. The resulting bounds provide better indicator of generalization compared to existing results and reflects many important generalization behaviours under overparameterization.
- This paper contributes many advancement in PAC-Bayesian theories, eg, single-hypothesis bounds. The bounds are then refined with consideration on exploiting implicit biases like spectral properties of hidden features.
- This paper involves multiple techniques of novelty, including loss symmetrization, non-perturbative expansion, etc. It also gives new results in pure mathematics.

**Weaknesses:**

- Weight and data dependence of $\pi$: The **Key Challenge** paragraph has emphasized that $\pi$ cannot rely on $W$. However, the hierarchical construction of $\pi$ involves subspaces subspaces, whose dimension is the effective rank of $G(W)$. It relies not only on weight $W$ but also on data $X$ or $z$.
- Evidence on tightness is not direct: The bound seems easy to compute and its most complicated part, ie, the Riemannian dimension, has been computed. So why not compute the entire bound.
- Presentation and typos:
  - Theorem 1 emphasizes prior's data dependence. However, in latter construction of $\pi$, lots of efforts have been put to make $\pi$ independent of $W$ and thus data $z$ (please correct me if my understanding is wrong), making this data dependence useless. Also I found the proof incomplete  for the data dependence of Theorem 1 (See my Question 1 below). I believe it is better to remove the claimed data dependence.
  - Eq.(7) has incomplete parentheses.
  - Unspecified matrix norm in Eq.(9).

**Questions:**

- Questions on proof details: In the proof of Theorem 1, Line 1040 bounds original losses using symmetrized losses. This step relies on the fact that $\\ell(f ; z′) − \\ell(f ; z)$ has a symmetric distribution. But as far as I am concerned (please correct me if I was wrong), this step not only involves this symmetrically distributed term, but also the $\\log \\frac{1}{\\pi(\\dots)}$ term where $\\pi$ depends on $z$ only and does not have a symmetric distribution. Could the authors provide more details on how this term is handled in this complicated mixture with $\\sup_f$?
    - I did some calculation and the symmetric distribution seems not sufficient: Let's assume a simplification with $n=1$ so all $\\mathbb{P}\_n$ notations disappear. Then this step can be abstractly seen as whether

        $$
            \\begin{aligned}
                \\mathbb{E}_{z, z'}  \\sup_f \\ell(f; z') - \\ell(f; z) - b(f; z)
                &\\overset{?}{=} \\mathbb{E}_{\xi} \\mathbb{E}_{z, z'}  \\sup_f \xi (\\ell(f; z') - \\ell(f; z)) - b(f; z)\\\\
                &=\\frac{1}{2} \\left(\\mathbb{E}_{z, z'}  \\sup_f \\ell(f; z') - \\ell(f; z) + b(f; z) \\right) + \frac{1}{2} \\left(\\mathbb{E}_{z, z'}  \\sup_f \\ell(f; z) - \\ell(f; z') - b(f; z) \\right)\\\\
                &=\\frac{1}{2} \\left(\\mathbb{E}_{z, z'}  \\sup_f \ell(f; z') - \\ell(f; z) - b(f; z) \\right) + \\frac{1}{2} \\left(\mathbb{E}_{z, z'}  \\sup_f \\ell(f; z') - \\ell(f; z) - b(f; z') \\right),
            \\end{aligned}
        $$

        where $b(\\cdot, \\cdot) \\ge 0$ abstracts the pointwise dimension term.
        It is equivalent to whether

        $$
            \\begin{aligned}
                \\mathbb{E}_{z, z'}  \\sup_f \\ell(f; z') - \\ell(f; z) - b(f; \\red{z})
                &\\overset{?}{=} \\mathbb{E}_{z, z'}  \\sup_f \\ell(f; z') - \\ell(f; z) - b(f; \\red{z'})
            \\end{aligned}
        $$

        There are counter-examples for this statement:

        $$
            z, z' \\in \\{-1, +1\\}, f \\in \\{-1, +1\},\\\\
            \\ell(f; z) = f z + 42, b(f; z) = \\begin{cases}
                0 & z = f\\\\
                42 & z \\neq f
            \\end{cases}
        $$

        Then we have

        $$
            \\begin{aligned}
                \\sup_f \\ell(f; z') - \\ell(f; z) - b(f; z)
                =& \\sup_f f \\times (z' - z) - 42 \\cdot \\mathbb{I}[z \neq f]\\\\
                =& z(z' - z) \\quad\\quad (\\text{$f$ must equal $z$}),
            \\end{aligned}
        $$

        whose expectation is $\\mathbb{E}[-z^2] = -1$.
        On the other hand, we have

        $$
            \\begin{aligned}
                \\sup_f \\ell(f; z') - \\ell(f; z) - b(f; z)
                =& \\sup_f f \\times (z' - z) - 42 \\cdot \\mathbb{I}[z' \neq f]\\\\
                =& z'(z' - z) \\quad\\quad (\\text{$f$ must equal $z'$}),
            \\end{aligned}
        $$

        whose expectation is $\\mathbb{E}[z'^2] = 1$.
        The two sides do not equal and the statement is generally untrue.
        Therefore, the symmetrical distribution of the loss difference term alone seems not enough and I kindly ask whether the step relies on more conditions or the symmetrical distribution is applied in some special way?
    - The conclusion of Theorem 1 also seems a bit unexpected because the data-dependent prior seems comes at little cost. For example, in Theorem 2.4 from Alquier (2024), the data-dependent choice of $\\lambda$ costs a penalty of $\\log \\text{card } \\Lambda$. In contrast, Theorem 1 does not contain any similar penalty. It seems all extra terms compared to Lemma 9 comes from either making the hypothesis deterministic ($\\epsilon$) or from standard procedures (I skimmed the proof and my finding may be imprecise, so please correct me; but at least they do not depend on $\\pi$). As a result, it seems one can arrive at extremely small generalization bounds at least for any deterministic learner: Assume $f = A(z)$ is the output of the deterministic learner. Then we can fix a small $\\epsilon \\sim \\sqrt{1/n}$ and then construct $\\pi$ to exactly cover the ball around $f=A(z)$. Since $f$ only depends on $z$, the loss-induced metric only depends on $z$ as well, and the radius $\epsilon$ of the ball is fixed, such $\\pi$ is a well-defined data dependent prior. However, this $\\pi$ makes the pointwise dimension term $0$ *without* any penalty. In this case, one would have a $\\tilde{O}(1/\\sqrt{n})$ rate, regardless of architecture, optimizer or data properties.

---

> ### Author Response · Authors · 2025-12-03
>
> **Dear Reviewer 1RYM**, we sincerely appreciate your invaluable and constructive comments, which are very helpful to improve the paper.
>
> **R1: Weight and data dependence of $\pi$: The Key Challenge paragraph has emphasized that $\pi$ cannot rely on $W$. However, the hierarchical construction of $\pi$ involves subspaces subspaces, whose dimension is the effective rank of $G(W)$. It relies not only on weight $W$ but also on data $X$ or $z$.**
>
> **A**: Thanks for this review. We indeed used a cover (a uniform prior) on the effective rank, which leads to the term $\sum_{l=1}^L\log d_{l-1}$ in the upper bound. Thus, the prior $\pi$ itself is *completely independent* of both the weights $W$ and the data. For clarity, we first describe the construction in the single-layer case. The general $L$-layer setting is obtained by applying the same construction independently to each layer and taking a product measure, which is enabled by the layer-wise decomposable structure of neural networks (a consequence of our non-perturbative analysis).
>
> 1. Data-independent prior on ranks.
>
>   We first fix a probability measure on the discrete ranks, before seeing any data.
> Let
> \\[
> r  \in \\{1,\dots,d \\}
> \\]
> We choose a fixed prior $\rho(r)$ on this finite set (in the paper we use the uniform distribution).
>
> 2. Conditional data-independent prior on the Grassmannian given $r$.
>
> Given a rank $r$, we put a (conditional) prior $\mu_r$
> on the Grassmannian $Gr(p,r)$:
> \\[
> \mu_r = \mathrm{Unif}(Gr(p,r)).
> \\]
> This measure is fixed in advance and does not depend on the data.
>
> 3. Local priors on each subspace.
>
> For every subspace $\bar{\mathcal{V}}\in\mathrm{Gr}(p,r)$ we define a “local” prior
>   \\[
>    \pi_{\bar{\mathcal{V}} ,r} = \mathrm{Unif}\bigl(B_2(1.58R)\cap \bar{\mathcal{V}}\bigr)
>   \\]
>    on the weight space restricted to $\bar{\mathcal{V}}$.
>
>
> 4. Mixture prior $\pi$ is a universal, data-independent prior.
>
>  The global prior used in the proof of the pointwise dimension bound
> is:
> \\[
> \pi(W)
>   = \sum_{r} \rho(r)
>     \int_{Gr(p,r)} \pi_{\bar{\mathcal{V}},r}(W)\\,\mu_r(d \bar{\mathcal{V}}),
> \\]
> All of $\rho$, $\mu_r$,  $\pi_{\bar{\mathcal{V}},r}$ are fixed before seeing the data and do not depend on \\(W,X\\), or \\(z\\); hence $\pi$ is a valid data-independent prior.
>
> 5. From the mixture prior to the pointwise dimension bound.
>
> Plugging the mixture prior into the ball ($B:=B_{\varrho_{G(W)}}(W,\sqrt{n}\varepsilon)$), we have
>    \\[
>    \pi(B)
>    = \sum_{r} \rho(r)\int_{\mathrm{Gr}(p,r)} \pi_{\bar{\mathcal{V}} ,r}(B) \mu_r(d \bar{\mathcal{V}})
>    \ge   \rho(r)
>    \int_{\mathcal G_r(W,\varepsilon)} \pi_{\bar{\mathcal{V}},r}(B)\mu_r(d \bar{\mathcal{V}}),
>    \\]
>    where $\mathcal G_r(W,\varepsilon)$ denotes the subset of “good” subspaces that are close to $\mathcal{V}\_{\mathrm{eff}}(G(W),R,\varepsilon)$: $\mathcal{G}\_r(W,\varepsilon) = \\{  \bar{\mathcal{V}} \in \mathrm{Gr}(p,r):\\|G(W)^{\frac{1}{2}}(\mathcal P_{\mathcal V_{eff}}-\mathcal P_{\bar{\mathcal{V}}})\\|\_{op}\leq \frac{\sqrt{n} \varepsilon}{4R} \\}$. For subspaces $\bar{\mathcal{V}}$ that are not close to $\mathcal{V}\_{\mathrm{eff}}$, they are simply not used in the estimate. Taking $-\log$ and using the inequality above yields
>    \begin{align*}
>    \log\frac{1}{\pi(B)}
>    \le  \underbrace{ \log d }\_{\text{rank cost}}+
>    \underbrace{\log\frac{1}{\mu\_r(\mathcal{G}\_r(W,\varepsilon))}}\_{\text{global Grassmannian cost}}+   \underbrace{\sup\_{\bar{\mathcal{V}}\in\mathcal{G}\_r(W,\varepsilon)}
>    \log\frac{1}{\pi\_{\bar{\mathcal{V}} ,r}(B)}}\_{\text{local chart cost}}.
>    \end{align*}
> The “rank cost’’ term $\log d $ comes from the discrete prior $\rho(r)$. The local chart cost term is bounded by the effective rank $d_{\mathrm{eff}}(G(W),\sqrt{5}R,\varepsilon)$ via Lemma 2. The global Grassmannian cost term depends only on the Grassmannian geometry and the fixed measure $\mu_r$; Lemma 3 provides an explicit upper bound. Together these three contributions give the final pointwise dimension bound.
>
>
> Hence, the resulting prior is fully data-independent, thanks to the construction of $\rho$, $\mu_r$ and $ \pi_{\bar{\mathcal{V}},r}$, and the use of the $-\log$ structure in our analysis. In the revised version, we have also added a schematic figure that visually illustrates this hierarchical construction of $\pi$ and its data-independence. We are grateful for this comment, which prompted us to clarify the construction. In our view, identifying such a data-independent prior $\pi$ is the crucial step that
> makes the neural-network generalization bound possible.

---

> ### Author Response · Authors · 2025-12-03
>
> **R2: Evidence on tightness is not direct: The bound seems easy to compute and its most complicated part, ie, the Riemannian dimension, has been computed. So why not compute the entire bound.**
>
> **A**: Thank you for raising this point.
>
> **(a) Why a numerically tight evaluation is hard.** A numerically tight evaluation of our bound for neural networks is to use Theorem 1 directly. However, as with many abstract bounds (PAC–Bayes, mutual-information, generic chaining), Theorem~1 (although tight) is not computationally tractable on its own; practical use requires adapting it to the function class at hand and introducing suitable relaxations. If we write $d(f,\varepsilon)=\log \frac{1}{\pi(B_{\varrho_{n,\ell}}(f,\varepsilon))}$, then a brute-force Monte Carlo estimator with i.i.d. draws $f'\sim\pi$ needs on the order of $e^{d(f,\varepsilon)}$ just to obtain even a single hit $f'\in B_{\varrho_{n,\ell}}(f,\varepsilon)$ with constant probability. Even moderately large values of $d$ are prohibitive in practice (e.g., $d=100$ already entails $\sim e^{100}$ draws).  This computational barrier helps explain why earlier PAC–Bayes works adopt global linearizations of $f$ in the parameters (typically by placing Gaussian priors and posteriors over the parameter space) to obtain closed-form objectives [Hinton and Van Camp, 1993, Dziugaite and Roy, 2017]. Moreover, the bound alone does not identify the distinctive nonlinearity and representation learning of deep networks [Wilson, 2025]. The central contribution of our paper is to pair explicit structural principles of deep nets with the pointwise-dimension foundation, yielding  generalization bounds that are both theoretically sharp and practically computable and interpretable.
> We therefore focus on the Riemannian dimension itself as the key quantity that captures the complexity of neural networks.
>
> **(b) Our aim: mechanism and guidance.**
> The primary role of a useful theory should be: (i) to reveal governing regularities and (ii) to guide practice.
>
> *(i) Revealing regularities.*
>      Deep networks exhibit the striking empirical fact that larger models often generalize better, underpinning the rise of large language models and empirical scaling laws. Our results uncover a theoretical mechanism behind this phenomenon: the proposed Riemannian / effective dimension (RD) decreases as width and depth increase, and this decrease tracks the test error. This explains why over-parameterized networks can generalize better instead of overfitting.
>
> *(ii) Guiding practice.*
>      The theory yields actionable guidance: algorithmic design and architecture
> design. 1) In the “Implicit Bias and Algorithmic Implications” section, we show that adding an RD-based penalty to the training objective improves test performance; 2) Architectures that compress RD more effectively generalize better, motivating geometry-aware architecture designs.
>
>    We thus highlight that the main value of our bound and experiments is that they
>    (a) explain why larger models can generalize better, and
>    (b) predict and steer algorithmic and architectural choices.
>
>
> **R3:Presentation and typos: (1) Theorem 1 emphasizes prior's data dependence. However, in latter construction of
> $\pi$, lots of efforts have been put to make $\pi$
>  independent of $W$
>  and thus data $z$
>  (please correct me if my understanding is wrong), making this data dependence useless. Also I found the proof incomplete for the data dependence of Theorem 1 (See my Question 1 below). I believe it is better to remove the claimed data dependence.
> (2) Eq.(7) has incomplete parentheses.
> (3) Unspecified matrix norm in Eq.(9).**
>
> **A**: Thank you for these careful observations.
> - Thank you for carefully checking the proofs. You are correct that our previous argument for Theorem 1 had a local flaw related to the treatment of the prior, which should be data-independent. In the revision, we re-derive Theorem 1 (the pointwise generalization bound) using a new method and rewrite Section 2 accordingly. The issue is localized to Section 2 and is orthogonal to the main structural contributions in Sections 3–4, which already use a data-independent prior and remain unchanged. We have carefully re-checked every step; the proofs are now self-contained and mathematically rigorous, and all main claims of the paper continue to hold (with the prior now explicitly data-independent). We kindly invite the Reviewer to re-examine the revised version and we would very much welcome further discussion.
> - The parentheses in Eq. (7) should be removed; we have corrected it.
> - In Eq.~(9) the matrix norm is the spectral norm; we now state this explicitly.
>
> **Q: Questions on proof details and Theorem 1.**
>
> **A.** Please see the revised paper, where we present a new route to proving Theorem~1 and address these issues in detail.

---

### Official Review · Reviewer_63MQ · 2025-10-31

**Soundness:** 3
**Presentation:** 2
**Contribution:** 3
**Rating:** 4
**Confidence:** 3

**Summary:**

The authors consider the generalization of fully connected networks at the trained parameters. The proposed generalization bound depends on the Riemannian dimension, which is based on the spectral properties of the feature representations. The proposed complexity measure is tighter compared to related approaches, and the Riemannian dimension exhibits appealing properties that can potentially enhance and explain generalization.

**Strengths:**

- Addresses an important and timely question about understanding the behavior of neural networks.
- The analysis and technical content appear rigorous and sound. However, I have not checked the theoretical derivations in detail.
- The pointwise dimension and the non-perturbative expansion are interesting contributions.
- The experiments support the theory, showing that the proposed complexity measure is smaller than related metrics.

**Weaknesses:**

- I believe that the clarity and accessibility could be improved. While this is a technical theoretical work and simplifying further is challenging, the text is somewhat hard to follow. Providing clearer intuition and explanations for the theoretical results would help.
- The paper is dense, as the generalization bound depends on both probabilistic and differential geometric concepts. The exposition of the latter (Sec. 3.2) could benefit from illustrations.
- Some terms may be well-known in the literature, but I believe, should still be defined in the paper to make it self-contained. See questions below for examples.
- The relevance of certain sections is not directly clear. See questions below.
- From the experiments, while the Riemannian dimension appears smaller than related metrics, it does not seem to provide a non-vacuous generalization bound. I may be missing something.

**Questions:**

1. In Eq. 4, the denominator represents the integral of the prior over a ball centered at $f$?
2. How can a data-dependent prior be considered, and how is it related to Theorem 4?
3. The discussion after Theorem 2 is somewhat unclear, and similarly, the relevance of Sec. 2.2. As regards, Sec. 4.2 is not immediately obvious if the Riemannian dimension is implicitly regularized during training or if the idea is to consider it as an explicit regularizer.
4. In the experiments, even if the bounds are smaller than the compared approaches, they still appear rather vacuous. Am I missing something?
5. The considered architecture does not include bias. Would the analysis change significantly if biases were included?

---

> ### Author Response · Authors · 2025-12-03
>
> **Dear Reviewer 63MQ**, we sincerely appreciate your invaluable and constructive comments, which are very helpful to improve the paper.
>
> **Reviewer’s consolidated request (covers W1–W4):
> The paper would benefit from better clarity/accessibility, more intuitions, brief definitions of key terms, and a clearer role/relevance of several sections.**
>
> **A**:
> Thank you for this review. We have thoroughly revised the paper to address all of these points: clarity and accessibility, precise definitions, relevance of individual sections, and proof hygiene. We have also added an illustration for Section 3.2 to provide clearer intuition. Please see the revised version for details.
>
>
>
> **R2: From the experiments, while the Riemannian dimension appears smaller than related metrics, it does not seem to provide a non-vacuous generalization bound. I may be missing something.**
>
> **A**:
> Thank you for this comment.
>
> **(a) Why a numerically tight evaluation is hard.** A numerically tight evaluation of our bound for neural networks is to use Theorem 1 directly. However, as with many abstract bounds (PAC–Bayes, mutual-information, generic chaining), Theorem~1 (although tight) is not computationally tractable on its own; practical use requires adapting it to the function class at hand and introducing suitable relaxations. If we write $d(f,\varepsilon)=\log \frac{1}{\pi(B_{\varrho_{n,\ell}}(f,\varepsilon))}$, then a brute-force Monte Carlo estimator with i.i.d. draws $f'\sim\pi$ needs on the order of $e^{d(f,\varepsilon)}$ just to obtain even a single hit $f'\in B_{\varrho_{n,\ell}}(f,\varepsilon)$ with constant probability. Even moderately large values of $d$ are prohibitive in practice (e.g., $d=100$ already entails $\sim e^{100}$ draws).  This computational barrier helps explain why earlier PAC–Bayes works adopt global linearizations of $f$ in the parameters (typically by placing Gaussian priors and posteriors over the parameter space) to obtain closed-form objectives [Hinton and Van Camp, 1993, Dziugaite and Roy, 2017]. Moreover, the bound alone does not identify the distinctive nonlinearity and representation learning of deep networks [Wilson, 2025]. The central contribution of our paper is to pair explicit structural principles of deep nets with the pointwise-dimension foundation, yielding  generalization bounds that are both theoretically sharp and practically computable and interpretable.
>
> **(b) Our aim: mechanism and guidance.**
> More importantly, we believe that the primary role of a useful theory is not to chase the smallest possible numerical upper bound—which is in any case highly sensitive to surrogate choices and worst-case constants (e.g., Lipschitz surrogates)—but to: (i) to reveal governing regularities and (ii) to guide practice.
>
> *(i) Revealing regularities.*
> Deep networks display the striking empirical fact that larger models often generalize better, which underpins the rise of large language models and empirical scaling laws. Our results uncover a theoretical mechanism behind this: the proposed Riemannian/effective dimension (RD) decreases as width/depth increase; it explains why over-parameterized neural networks can exhibit better generalization.
>
> *(ii) Guiding practice.* The theory yields actionable guidance: algorithmic design and architecture design. 1) In the “Implicit Bias and Algorithmic Implications” section, we show that adding an RD-based penalty to the training objective improves test performance; 2) Architectures that compress RD more effectively generalize better, motivating geometry-aware architecture designs.
>
>  We thus highlight that the main value of our bound and experiments is that they
>    (a) explain why larger models can generalize better, and
>    (b) predict and steer algorithmic and architectural choices.
>
> **Q1: In Eq. 4, the denominator represents the integral of the prior over a ball centered at $f$?**
>
> **A**:
> Thanks for this review.  Yes, the denominator is the prior mass of the $\varepsilon$-ball $B_\varrho(f,\varepsilon)=\\{f'\in\mathcal F:\varrho(f,f')\le\varepsilon\\}$ centered at $f$, i.e., $\pi(B_\varrho(f,\varepsilon))$.

---

> > ### Author Response · Authors · 2025-12-03
> >
> > **Q2: How can a data-dependent prior be considered, and how is it related to Theorem 4?**
> >
> > **A**:
> > Thanks for this review. The claim on data-dependent prior is wrong, and we have removed this claim in the revised version. The proof of Theorem~4 does *not* require a data-dependent prior; it goes through entirely with the data-independent prior we construct.
> >
> > **Q3: The discussion after Theorem 2 is somewhat unclear, and similarly, the relevance of Sec. 2.2. As regards, Sec. 4.2 is not immediately obvious if the Riemannian dimension is implicitly regularized during training or if the idea is to consider it as an explicit regularizer.**
> >
> > **A**:
> > Thanks for this review.
> >
> > (i) Discussion after Theorem 2.  We have revised Section 2.1 thoroughly; please see the revised paper.
> >
> > (ii) Section 2.2 (purpose and relevance). This section addresses two conceptual questions.
> >
> > **What is the nature of generalization?** In contrast to the geometric dimension (studying pointwise dimension at the infinitesimal scale $\varepsilon \to 0$), Theorem 1 indicates that generalization is governed by pointwise dimension at the finite scale $\varepsilon \ge \alpha$; this contrast suggests that the nature of generalization is *Finite-Scale Dimension*.
> >
> > **Why can a learning problem generalize?** At finite scale, the pointwise dimension decreases as $\varepsilon$ grows; therefore a problem may have a large ambient/asymptotic dimension yet a small finite-scale dimension, yielding small generalization error. This gives the base intuition for why over-parameterized models can still generalize well.
> >
> >
> >
> > (iii) Section 4.2 (implicit vs. explicit regularization). Section 4.2 illustrates regularization from both explicit and implicit perspectives.
> >
> > **Explicit perspective:** If we *explicitly* add an RD-based penalty to the training objective, the resulting algorithm tends to achieve better test performance.
> >
> > **Implicit perspective:** If an optimization algorithm *implicitly* compresses the Riemannian dimension during training, it also improves generalization. Our experiments show that SGD tends to compress RD over epochs, providing evidence of such implicit bias.
> >
> > **Q4: In the experiments, even if the bounds are smaller than the compared approaches, they still appear rather vacuous. Am I missing something?**
> >
> > **A**:
> > Thanks for this review. Please see the response to *R2*.
> >
> > **Q5: The considered architecture does not include bias. Would the analysis change significantly if biases were included?**
> >
> > **A**:
> > Thank you for raising this. Including biases does not change the analysis in any essential way. For a batch with column-wise samples,
> > let $X\in\mathbb{R}^{d\times n}$, $W\in\mathbb{R}^{k\times d}$, $b\in\mathbb{R}^{k}$.
> > The affine map is
> > \\[
> > Y \\;=\\; W X \\;+\\; b\\,\mathbf{1}_n^{\top}\ \in\mathbb{R}^{k\times n};
> > \\]
> > augment input and weights by
> > \\[
> > X' \\;=\\; \begin{bmatrix} X \\\\ \mathbf{1}_n^{\top} \end{bmatrix} \in \mathbb{R}^{(d+1)\times n},
> > \qquad
> > \widetilde W \\;=\\; \big[\\, W \ \ b \\,\big] \in \mathbb{R}^{k\times (d+1)},
> > \\]
> > then the bias is absorbed linearly:
> > \\[
> > \\, Y \\;=\\; \widetilde W\\,X' \\,
> > \quad\Longleftrightarrow\quad
> > Wx_i + b \\;=\\; \widetilde W \begin{bmatrix} x_i \\\\ 1 \end{bmatrix}\quad(\text{per-sample}).
> > \\]
> > Under this standard augmentation, all proofs go through unchanged. Regarding the bound, our Riemannian dimension depends on feature-Gram structures: adding a constant channel corresponds to at most a rank-$1$ update. For uncentered Grams it typically sharpens the spectrum and thus does not increase (and often slightly decreases) the effective rank. Thus our rates and qualitative trends remain valid when biases are included.

---

### Author Response · Authors · 2025-12-03
**Global Response (Cont'd)**

**Writing and presentation.** We have carefully revised the entire paper and addressed the issues pointed out by the reviewers. We rechecked all proofs and performed a thorough pass for typos, notation inconsistencies, and exposition; we believe the current version is substantially clearer and technically sound.

**We kindly invite the reviewers and area chairs to examine the revised manuscript and remain confident in the significance of the contributions. We would be very grateful for any further comments or suggestions.**

---

### Author Response · Authors · 2025-12-03
**Global Response**

**Dear PCs, SACs, ACs, and Reviewers,**

We would first like to sincerely thank you for the time and care you have devoted to handling and reviewing our submission, as well as for the many constructive suggestions that have helped us improve the paper.

We believe this work makes a substantive step toward a principled theory of the nonlinear, feature-learning regime in deep neural networks and toward a reshaped statistical foundation for representation learning. We understand that the initial reviews may not fully capture the scope of this contribution. This is in part because (i) the original submission contained an inaccurate description of the PAC–Bayesian prior in Section 2 and some imprecise phrasing that could obscure the main ideas, and (ii) the breadth of new results meant that we did not sufficiently foreground the core message. The misstatement concerning the PAC–Bayesian prior in Section 2 is orthogonal to our main technical innovations, which lie in the localization framework for pointwise generalization and the highly non-trivial hierarchical covering paradigm and finite-scale structural analysis of deep networks.


In the revision, we have corrected this issue and streamlined the exposition to make the main contributions explicit. As a result, several specific critiques no longer apply to the revised manuscript. At the same time, some of the most substantive aspects—our pointwise, finite-scale, structural analysis of deep networks (going beyond uniform, class-level analyses of linear/kernel models)—were only lightly touched on in the initial discussion. The core novelty remains intact and central to our contribution. We respectfully invite reassessment against the revised version, which we believe provides a substantial advance toward a rigorous generalization theory for the nonlinear, feature-learning regime.

**Clarification about the prior (non-central issue).** The reviewers are correct that, in a PAC–Bayesian analysis, the prior should be data-independent. In the original submission, we described the prior in Section 2 as “data-dependent”; this was intended as a non-essential strengthening in the expository discussion and was **not** used as an assumption in any subsequent results. In the revision, we have removed this unnecessary remark and rewritten Section 2 and its associated proofs so that the prior is explicitly data-independent throughout. All our main results, including the priors constructed in Sections 3–4 for the neural-network generalization bounds, already used a data-independent prior and remain unchanged. The correction is therefore orthogonal to the central contribution of the pointwise localization framework.

**Revised Section 2 and pointwise framework.** We have rewritten the exposition and proofs of the pointwise generalization framework. The revised theory (i) extends PAC–Bayes to deterministic hypotheses via localization to metric balls in an uncountable hypothesis class; (ii) upgrades uniform-convergence and generic-chaining analyses to pointwise generalization with hypothesis-dependent complexity; and (iii) provides tight tools for generalization, matched by a worst-case lower bound. The proofs are now streamlined and fully rigorous, with clearer intuition.

As requested by the reviewers, we also expanded the Related Work to clarify how our notion fundamentally differs from prior “intrinsic dimension” concepts and how it unifies and strengthens PAC–Bayes, generic chaining, and mutual-information bounds.

**Main contributions beyond Section 2.**
While Section 2 provides a new foundation for pointwise generalization, it represents only one part of the paper. The manuscript also develops:

- **Hierarchical covering paradigm.**	We replace uniform covering numbers with a pointwise, finite-scale dimension, and develop a local-chart–to–global-atlas approach to characterize this dimension for deep neural networks. This overcomes long-standing barriers and enables a principled analysis of nonlinear statistical models.

- **Mathematical tools, structural analysis, and empirical validation.** On the mathematical side, we develop new finite-scale structural tools for analyzing neural-network mappings, including ellipsoidal coverings on Grassmannian subspaces. On the empirical side, we provide experiments that quantitatively corroborate the theory’s predictions.

Taken together, these advances address a central challenge in deep learning theory and move toward a principled, pointwise generalization theory for the nonlinear, feature-learning regime. Our bounds are, to our knowledge, the first to express the complexity of a deep network in terms of a pointwise, finite-scale dimension of the learned feature representation, rather than solely in terms of weight-space compression. We have clarified these points in the revised Introduction and expanded Related Work, with the hope that the contribution can serve as a useful reference point for future work.

---

### Note · Authors · 2025-12-17

I have read and agree with the venue's withdrawal policy on behalf of myself and my co-authors.